# QSpace – An Open-Source Tensor Library for Abelian and non-Abelian Symmetries

Andreas Weichselbaum$^\star$

Department of Condensed Matter Physics and Materials Science,
Brookhaven National Laboratory, Upton, NY 11973-5000, USA
$^\star$ weichselbaum@bnl.gov

July 3, 2024

## Abstract

This is the documentation for the tensor library `QSpace` (v4.0), a toolbox to exploit 'quantum symmetry spaces' in tensor network states in the quantum many-body context. `QSpace` permits arbitrary combinations of symmetries including the abelian symmetries $\mathbb{Z}_n$ and $U(1)$, as well as all non-abelian symmetries based on the semisimple classical Lie algebras: $A_n$, $B_n$, $C_n$, and $D_n$, or respectively, the special unitary group $SU(n)$, the odd orthogonal group $SO(2n+1)$, the symplectic group $Sp(2n)$, and the even orthogonal group $SO(2n)$. The code (`C++` embedded via the MEX interface into Matlab) is available **open-source as of `QSpace` v4.0 on bitbucket** [1] under the Apache 2.0 license.

`QSpace` is designed as a bottom-up approach for non-abelian symmetries. It starts from the defining representation and the respective Lie algebra. By explicitly computing and tabulating generalized Clebsch-Gordan coefficient tensors, `QSpace` is versatile in its operations across all symmetries. At the level of an application, much of the symmetry-related details are hidden within the `QSpace` `C++` core libraries. Hence when developing tensor network algorithms with `QSpace`, these can be coded (nearly) as if there are no symmetries at all, despite being able to fully exploit general non-abelian symmetries.

# 1 Introduction

The treatment of tensor network states in numerical simulations requires a set of standardized elementary routines: ways to initialize tensors, Kronecker products to build tensor product spaces, contractions to compute matrix elements or expectation values, etc. Together these routines form a tensor library that is designed to provide a user-friendly environment. While the individual tensor operations are straightforward linear algebra operations per se, the complexity of a tensor library increases significantly if one intends to exploit symmetries. Yet symmetries are important, both for the sake of numerical efficiency, but also to gain detailed symmetry-resolved physical insights into the fabric of particular tensor network state realizations. By now, several mature open-source tensor libraries are available. Yet the overwhelming majority of these only permit one to exploit abelian symmetries. This includes, for example, iTensor [2,3], Uni10 [4], ALPS [5], or TeNPy [6]. Implementations of non-abelian symmetries in tensor network libraries are scarce, with TensorKit [7] a notable recent exception.

QSpace also started from plain abelian symmetries originally in version 1 (v1). Yet with v2, QSpace implemented non-abelian symmetries [8]. It was designed as a bottom-up approach, based on the realization [9] that, as a generalization of the Wigner-Eckart theorem [10–14], Clebsch-Gordan coefficients of an arbitrary-rank symmetric tensor can be *factorized* in the form of a tensor product with reduced matrix elements. QSpace explicitly computes and tabulates Clebsch-Gordan coefficients for standard tensor product decomposition (rank-3), as well as higher-rank Clebsch-Gordan tensors (CGTs) obtained via contractions. It is in this sense that

QSpace is considered a *bottom-up* approach: by explicitly computing and utilizing CGTs, the tensor representation is known in full numerical detail which then enables general tensor operations in a self-contained fashion. Specifically, QSpace does not rely on or use $6j$ symbols [9, 15], bearing in mind that these are known analytically for SU(2) only, where these are simple numbers. For general non-abelian symmetry, however, the $6j$ symbols become rank-4 tensors in outer multiplicity ('4M symbols') which introduces significant complications.

Tensor network simulations typically deal with lattice models comprised of sites whose state space is considered small (microscopic). By iteratively adding site after site, ever larger many-body Hilbert spaces can be built. Eventually, these are truncated in tensor network simulations in a controlled fashion based on the underlying entanglement structure. This iterative build-up of a Hilbert space precisely also underlies the design principle of QSpace for non-abelian symmetries: one starts from small local units where all symmetry-related aspects can be generated, and thus are known in a simple, transparent way. With these initial building blocks one can 'play lego' then. One can build ever larger, more complex tensor network structures. From the QSpace perspective, this iterative nature automatically generates the respective multiplet spaces *as they occur on demand* at each step, based on well-defined elementary steps such as tensor product decomposition, contractions, etc. Newly generated symmetry related data is tabulated and stored much like in a database (the RC_STORE in QSpace). QSpace thus automatically adapts to what is needed in the simulations that are run, bearing in mind that it is impossible to store all symmetry-related data for continuous non-abelian symmetries, since that data set is infinite.

## 1.1  Version history

The QSpace tensor library has a long-standing history that started around 2006. Since its inception, it has already been thoroughly scrutinized, debugged, and optimized. A distinctive feature of QSpace has been to exploit arbitrary symmetries from the very beginning for the sake of numerical efficiency. The initial motivation of QSpace was at the interface between the density matrix renormalization group (DMRG [16, 17]) and the numerical renormalization group (NRG [13, 18]). Yet since it was designed as a general tensor library from the very beginning, QSpace is also equally applicable in other tensor network algorithms. QSpace went open-source as of version 4 [1].

- QSpace v1.* (2006-2011) started as a tensor library for arbitrary sets of U(1) abelian symmetries, with applications in the realm of NRG (fdm-NRG [19–21]), DMRG [22–24], and their crossover [25, 26].

- QSpace v2.* (2012-2015) newly introduced general continuous non-abelian symmetries to its capability [8, 27, 28], then including the special unitary group SU($N$) and the symplectic group Sp($2N$) where $N \geq 2$ is dealt with as a parameter.

- QSpace v3.* (2015-2022) completed the set of semi-simple symmetries by also implementing the special orthogonal groups SO($N$), both for even and odd $N$, noting that these represent different Lie algebras $D_n$ and $B_n$ with $N = 2n$ or $N = 2n + 1$, respectively. QSpace v3 also included significant performance upgrades, with full tabulation of generalized Clebsch-Gordan tensors (CGTs) as well as their contractions via the introduction of X-symbols [15].

- QSpace v4 (since 2022) finally was geared towards open source in response to frequent requests from the community. QSpace v4 is mostly the state of QSpace v3.2 (last version in v3.*), yet cleaned up in the sense of having completed and merged still open development branches. It went open-source in 2022 as a git repository on bitbucket [1] under the Apache 2.0 license.

## 1.2 Target audience and additional literature

This documentation of QSpace assumes basic familiarity with tensor network states for corre-
lated quantum many-body states. This includes general tensor network operations and strate-
gies, as well as their graphical description in the form of tensor network diagrams. Many ex-
cellent introductory papers, tutorials, and reviews already exist in the literature in this regard,
e.g., see [6, 29–31] or the lecture notes in Ref. [32] amongst many others. This documenta-
tion does not intend to replicate these. Instead, it focuses on the implementation of general
symmetries in tensor network states, with a strong emphasis on the distinguishing feature of
QSpace, namely non-abelian symmetries. This documentation builds on the earlier publica-
tions on QSpace in Refs. [8, 15] which already also put an emphasis on a more self-contained
pedagogical presentation (e.g., see the extensive appendices in [8] including App. A on *Non-
abelian symmetries 101*). As such, these are highly recommended additional literature that
complement this documentation.

## 1.3 Format conventions

The format for the `coding syntax` (in this font and color coding) used throughout this docu-
mentation is in intuitive compact Matlab semantics. The main reason is that this also reflects
the QSpace environment that the C++ QSpace core routines are embedded into (the many
cross-references within this documentation via hyperlinks are formatted in blue as shown;
non-hyperlinked terms or highlighted text are differentiated in a somewhat lighter color). A
detailed discussion of the structure of the vast low-level C++ code of the core routines, on the
other hand, is beyond the scope of this documentation. The Matlab syntax used here is basic
and concerns, for example, access to fields in structures, like `X.field`, cell arrays `Q{i}`, the
representation of `'strings'`, or matrix notation such as `M(i,:)` or `M(:,j)` for row *i* or column
*j*, etc. The color format is changed when required for the sake of differentiation or emphasis
of Matlab-specific QSpace topics, e.g., like the general (C++) QSpace implementation vs. its
Matlab QSpace, i.e., @QSpace counterpart. To visually indicate when the latter terms are hy-
perlinked, the color changes to a darker shade. Optional arguments to functions are usually
bracketed as in `[...]`. At the level of the operating system, (environmental) variables like `v`
are also referenced as `$v` or `$(v)` as in `bash` or `Makefile` semantic.

In this documentation, snippets of Matlab code using QSpace will be displayed based on
the `minted` latex package that also enables syntax highlighting,

```
1  >> fprintf(1,'\n   Hello world from Matlab prompt!');  % some comment
2  >> [a,b]=helloworld('some','input',pi);                % helloworld MEX example
```

Here `>>` indicates the Matlab command line prompt. The second line, for example, is a simple
MEX test routine, that assigns `a='some'`, `b='input'`, etc. As with the above example, QSpace
always assumes Matlab strings in single quotes (`'...'`). Textual output from Matlab com-
mands such as the above is displayed without any further syntax highlighting in the format

```
3     Hello world from Matlab prompt!
4     Hello world from MEX! (.mexmaci64: nargout=2, nargin=3)
```

The line numbers to the left are included for ease of reference in subsequent discussions.
These output displays may include further commands shown with a leading Matlab prompt,
together with their respective output. The output is frequently adapted for the purpose of this
documentation, e.g., by skipping empty or trivial lines such as `'ans = '` indicating the answer
to, i.e., result of a command.

## 1.4   Outline

This documentation makes use of a broad range of semantics and definitions. Essential terminology and concepts are introduced in detail in Sec. 2. This is complemented by Apps. A.1 and A.2 which provide glossaries on frequently used acronyms and general terminology for tensor networks and symmetries, respectively. These short glossaries are grouped by their meaning, rather than alphabetically, such that quickly glossing through them from top to bottom should be meaningful. The bare-bones data structure of QSpace tensors is discussed in App. A.3.

The general QSpace approach to implementing physical model systems is summarized in Sec. 3. Because QSpace examples need QSpace tensors that need to be generated first, more detailed examples for Secs. 2 and 3 are collected into Sec. 4 for clarity, with cross-references as relevant. Section 5 provides a compilation of more involved examples, intended as simple tutorials already geared toward applications. The general part of this documentation concludes in Sec. 6 which also comments on feedback and support.

System requirements are detailed in App. B.1. Download, compilation, and QSpace environment are discussed in Apps. B.2 and B.3. Appendix C comments on state-of-the-art NRG and DMRG applications that are already also present in QSpace, but whose detailed documentation is beyond the scope of this documentation. App. D comments on detailed usage information (like 'man' pages) and further documentation in the QSpace repository. Appendix E provides a listing of the most relevant compiled binary QSpace MEX routines. Appendix F finally gives an overview of the Matlab-intrinsic environment that QSpace is embedded into.

## 2   General QSpace Approach and Conventions

The simple main idea underlying QSpace is that for any tensor $X$ with an arbitrary but fixed number $r$ of legs or indices, referred to as (tensor) rank $r$, the presence of a global symmetry leads to a block decomposition of the tensor. Then for each block with well-defined symmetry sectors $q$ on all its legs, the symmetry-related aspects factorize [8,9],

$$X = \bigoplus_q \|X\|_q \otimes C_q \tag{1}$$

which is generalized and discussed in significantly more detail in Sec. 2.11 still. Here $\|X\|_q$ represents the reduced matrix element tensors (RMTs), and $C_q$ the generalized Clebsch-Gordan coefficient tensors (CGTs). Both mimic the structure of the original tensor $X$, and thus, in particular, are also of the same rank $r$. A scalar rank-2 tensor, like a Hamiltonian, becomes block diagonal. The listing of such non-zero blocks is denoted by the direct sum ($\oplus$) over symmetry block configurations permitted by symmetry. These are labeled by $q$ here, which is assumed collective over all legs. Effectively, a QSpace tensor consists of a list of RMTs $\|X\|_q$ with references to CGTs $C_q$. QSpace takes care of all the symmetry-related aspects and the required bookkeeping. In this sense, the tensor X is also referred to as QSpace X.

The decomposition in (1) is also applicable to abelian symmetries. This is useful from the coding point of view for the sake of a shared consistent data structure when having abelian and non-abelian symmetries present at the same time. Fusing state spaces with abelian symmetries can also be cast into the framework of CGTs, albeit trivially so by taking $C_q = 1$ when the set of labels $q$ collected over all legs is permissible from a fusion point of view, and $C_q = 0$ otherwise. These trivial factors can be created on the spot, of course, with no need to tabulate.

In QSpace, the CGTs $C_q$ for non-abelian symmetries are explicitly constructed on demand once and for all, and maintained in a database for later reference. In the presence of multiple symmetries, the CGTs in the decomposition in (1) further factorize, $C_q = \bigotimes_s C_{q_s}^{(s)}$, with $C_{q_s}^{(s)}$ the CGT for symmetry $s$. The symmetry labels for any symmetry sector on a particular leg then

also become the collection of labels across all $C_{q_s}^{(s)}$, i.e., $q \equiv \{q_s\}$ . All bookkeeping for this additional layered structure is automatically taken care of by QSpace.

The remainder of this section summarizes frequently used terminology, conventions, and acronyms that are extensively used with QSpace. For more elementary tensor-network and symmetry-related definitions, it is advised to briefly also glance over Apps. A.1 and A.2 before reading on.

## 2.1 QSpace stores tensors (not state spaces)

QSpace is a tensor library. Therefore it represents tensors (blobs in a pictorial representation), but not state spaces per se (individual isolated legs or lines in tensor networks, i.e., indices). QSpace operates on tensor objects that have indices (legs) attached. From the point of view of symmetries then, e.g., when incoming charge needs to be preserved into outgoing charge, this necessitates at least two indices. In this sense, QSpace necessarily represents tensors of rank $r \geq 2$. There are only two trivial exceptions that permit rank $r < 2$. These are:

**QSpace tensor with rank $r < 2$**

- rank 0: when computing expectation values or overlaps, the resulting object is fully contracted, i.e., has no more open legs. Hence this represents a tensor of rank $r = 0$. For an example, see (40) where a tensor is fully contracted with itself.

- rank 1: the only rank-1 tensor that is permitted from a symmetry point of view is a tensor with a single leg in the vacuum symmetry sector i.e., with symmetry labels $q = 0$ (if there is no charge coming in, there is no need for it to leave). For an example, see (41). A rank-1 tensor can occur naturally out of contractions, e.g., when fully contracting a tensor of rank $r$ on all its $r$ indices with a tensor of rank $r + 1$. On general grounds, the result of such a contraction can only have a non-zero contribution in the $q = 0$ symmetry sector on the single remaining open leg.

**Referencing state spaces**   To specify a particular state space in QSpace, this needs to be done by referring to respective legs on an existing tensor (e.g., see `getIdentity`). Since objects are referenced when handed over to routines (both Matlab and MEX routines alike), this is efficient in the sense that tensors do not need to be copied for this purpose [cf. Matlab's just-in-time (JIT) concept where copies are only generated once an object needs to be altered]. Nevertheless, if desired, the minimal setting that gets closest to storing a state space on leg $l$ of QSpace X of some rank $r \geq l$, is by defining the identity tensor on that leg in compact diagonal form: `diag(getIdentity(X,l))`. This also inherits the dimensionality of each symmetry sector, as well as the index tag (`itag`) for that leg.

## 2.2 Composite index for state spaces

To implement symmetries, any state space associated with tensor indices needs to be organized into symmetry sectors. When exploiting non-abelian symmetries, any state $s$ must be identifiable by answering the following questions: (1) what symmetry sector does it belong to? Within that symmetry sector, (2) to which multiplet does it belong? And in that symmetry multiplet, (3) what state does it represent? Hence for the description of any state space $s$ in the presence of non-abelian symmetries, this naturally acquires a composite index structure that contains the answers to the three questions above [8],

$$|s\rangle \equiv |qn; q_z\rangle \ . \tag{2}$$

This addresses the above questions in that $q$ denotes (1) the combined set of symmetry labels that specify the symmetry sector for all symmetries included. The index $n$ then specifies (2) the multiplet, and $q_z$ spans (3) the internal structure of an individual multiplet in that symmetry sector $q$. To emphasize that the symmetry labels can refer to any symmetry or combinations thereof, QSpace generally uses the letter $q$ for symmetry labels, as in QSpace for quantum symmetry spaces. For the particular case of a single SU(2) symmetry, for example, $(q, q_z) \equiv (2S, 2S_z)$ reflects the spin quantum numbers.

The composite index structure in (2) can also be trivially applied to abelian symmetries. Since each abelian 'multiplet' has only one state, this also represents the 'maximum weight' state. Hence the $q_z$ label may be promoted to the multiplet label, i.e., $q \equiv q_z$ and $q_z$ rendered obsolete then. For the case of all-abelian symmetries, the redundant $q_z$ labels together with their respective trivial CGTs can be skipped altogether. In QSpace this explicitly happens for the case of all-U(1) symmetries which results in an empty X.info.cgr field in the QSpace data structure for QSpace X (this reflects the original setting as of QSpace v1).

Within the QSpace data structure [see Eq. (35) and subsequent lines for an explicit example], the symmetry labels $q$ in Eq. (2) for a QSpace tensor X are stored as rows in the matrices X.Q{l} where $l = 1, \ldots, r$ indexes the leg of the tensor. There is a one-to-one correspondence between the rows X.Q{l}(i,:) with the respective RMT X.data{i} which describes the i'th non-zero block in QSpace X. The index $n$ in the decomposition (2) is not explicitly stored per se, but is implicit as the index within the blocks of the RMTs stored in the cell array X.data, e.g., when taking some matrix element X.data{i}(...,$n_l$,...) with respect to leg $l$ [for an example, see discussion of output to (48)]. Similarly, $q_z$ in Eq. (2) corresponds to an implicit index within the CGTs that also needs to deal with *inner multiplicity* [8]. The CGTs are tabulated and stored in RC_STORE/CStore since fixed by symmetry. Hence an actual QSpace tensor X only stores the relevant *reduced* data based on the multiplet structure $(q, n)$. Concerning the symmetry related aspects $(q_z)$ it suffices to only *reference* the underlying Clebsch-Gordan data, referred to as Clebsch-Gordan references (CGRs) and stored in X.info.cgr. Overall, this switches the description of a tensor from a state-based to a multiplet-based setting.

## 2.3   Direction of legs for tensors

All lines in a tensor network diagram are directed, i.e, carry an arrow (e.g., see Fig. 2; [15]). This binary scheme of whether an index is 'incoming' (+) or 'outgoing' (−) from a tensor derives from, and thus is equivalent to the respective binary concepts of raised vs. lowered indices, or contra- vs. co-variant indices, but also bra- vs. ket-states. This binary concept is intrinsic to tensors. For this reason, the direction of legs is meticulously enforced and tracked for all tensors in QSpace.

The semantical preference of the notion of 'direction' is mainly due to pictorial representations where directed lines are easier to visualize. Besides, this also gives a simple pictorial representation of the Einstein summation convention: a raised index that is summed over with a paired-up lowered index ensures a well-defined direction of the line that depicts that contracted index. This directed line necessarily leaves from one tensor and and enters another. From the perspective of a tensor, the related notation of using +/− for directions of its legs, as also used in the code this way, is motivated from the point of view of symmetries: for example, for U(1) charge conservation, the total combined incoming charge must equal the total outgoing charge, i.e., $\sum_{\text{in}} q_{\text{in}} = \sum_{\text{out}} q_{\text{out}}$ which may be trivially rewritten as $+\sum_{\text{in}} q_{\text{in}} - \sum_{\text{out}} q_{\text{out}} = 0$. This motivates that from the point of view of a tensor, an incoming (outgoing) index adds (removes) charge, and hence is associated with a + (−), respectively.

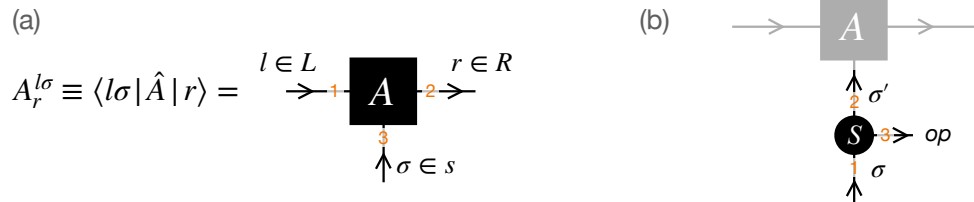

Figure 1: (a) Fusion of the local state space $\sigma_{(i)}$ by a tensor $A_i$ as in an MPS. Skipping the index $i$ for readability, the left state space $l \in L$, and the local state space $\sigma \in s$ are fused into the state space $r \in R$ to the right of site $i$. Irrespective of leg directions on $l$ and $r$, A-tensors use an LRs index order convention as indicated by the orange labels. (b) (Irreducible) operator $S$ like a spin operator as in Eq. (10) acting on local state space $\sigma, \sigma' \in s$. By convention, operators use $(\sigma, \sigma', \text{op})$ index order convention as indicated by the orange labels. The *operator index* (third leg) may be skipped and thus absent for scalar operators where this represents a trivial singleton dimension.

### 2.3.1 Convention: Incoming leg to tensor is written as superscript index

Consider an A-tensor that fuses a local state space $s_{(i)}$, e.g., with a matrix product state (MPS) $|A\rangle$ and local state spaces $\sigma \in s_{(i)}$ for some site $i$ in mind [Fig. 1(a)]. For simplicity, the index $i$ is skipped as well as any other indices present on $A$ for the argument here. The local identity operator is given by

$$\hat{\mathbb{1}} = \sum_{\sigma \in s} \underbrace{|\sigma\rangle}_{\equiv |s_\sigma\rangle} \underbrace{\langle\sigma|}_{\equiv \langle s^\sigma|} \equiv |s_\sigma\rangle\langle s^\sigma| \equiv |s_\sigma\rangle\!\!\longrightarrow\!\!\boxed{\delta^\sigma_\sigma}\!\!\longrightarrow\!\!\langle s^{\sigma'}| \tag{3}$$

with implicit summation over pairs of raised and lowered indices if not explicitly specified (cf. Einstein summation convention). The above starts with the convention that (i) ket-states have lowered indices, and hence are considered covariant. Therefore bra-states are contra-variant and written with a raised index. Applying the identity in Eq. (3) to $\hat{A}$, one obtains

$$\hat{A} = \sum_\sigma |s_\sigma\rangle \underbrace{\langle s^\sigma|\hat{A}}_{\equiv A^\sigma} \equiv A^\sigma |\sigma\rangle \ . \tag{4}$$

In a numerical context, one always has to use 'matrix elements' $A^\sigma$ that are obtained in a particular basis. By convention, such tensor coefficient spaces are written without hats. Now also with graphical depictions in mind, raised or lowered indices will be assigned directions. Intuitively from the perspective of a tensor, by convention, (ii) the index of the local state space $\sigma$ is considered 'incoming' to the tensor $A$. Hence in QSpace an *incoming* index to a tensor coefficient space [as in $A^\sigma \equiv \langle s^\sigma|\hat{A}$ in Eq. (4)] corresponds to a projection onto a bra state and is written as a raised index. The identity in Eq. (3) may thus be depicted graphically as shown at its r.h.s., which also shows that a ket-state in itself is considered outgoing. Eventually, via contractions, a lowered index needs to be paired with a raised index, or equivalently, a ket- with a bra-state, or a co- with a contra-variant index, or an outgoing with an incoming leg.

The incoming index in $A^\sigma$ in Eq. (4) is contracted with the outgoing index of the ket state $|\sigma\rangle$ [see also right term in Eq. (3)]. As such, this represents the 'operator' $\hat{A}$ with no open index, thus written with a hat. It is more of an abstract object that is independent of a particular basis. In a numerical context, however, only matrix elements occur, such as $A^\sigma = \langle s^\sigma|\hat{A}$. This way, the index $\sigma$ as in in $A^\sigma$ becomes an open, i.e., uncontracted index. Similarly, the identity operator $\hat{\mathbb{1}}$ as written in Eq. (3), has no open indices. However, when expressed in a particular basis,

the matrix elements are $\langle s^\sigma | \hat{\mathbb{1}} | s_{\sigma'} \rangle = \delta^\sigma_{\sigma'}$. This 'strips off' the ket and bra from the r.h.s. in Eq. (3), resulting in $\rightarrow\!\!\bullet\!\!\rightarrow$ $\delta^\sigma_{\sigma'}$ with open indices $\sigma$ and $\sigma'$.

Now when applying an operator $\hat{S}$ on the site that is fused by the tensor $\hat{A}$, the matrix element corresponds to the projection onto the local state $\sigma$ [cf. Fig. 1(b)],

$$\langle \sigma | \hat{S} \hat{A} = \sum_{\sigma'} \underbrace{\langle s^\sigma | \hat{S} | s_{\sigma'} \rangle}_{\equiv S^\sigma_{\sigma'}} \underbrace{\langle s^{\sigma'} | \hat{A} \rangle}_{\equiv A^{\sigma'}} \equiv S^\sigma_{\sigma'} A^{\sigma'} , \tag{5}$$

having inserted the identity as in Eq. (3). Since by convention the index $\sigma'$ is *incoming* to $A$, it has to leave and thus be *outgoing* for the tensor $S^\sigma_{\sigma'}$ via the projection to the ket-state $|s_{\sigma'}\rangle$. Therefore $S^\sigma_{\sigma'} \equiv \langle s^\sigma | \hat{S} | s_{\sigma'} \rangle$ has bra-state $\sigma$ as incoming, and the ket-state $\sigma'$ as outgoing. The order of the bra indices relative to the ket indices is irrelevant, in that it can be trivially changed. For physical operators, nevertheless, by standard convention, bra indices are typically listed before ket indices. Hence their index order convention is $\langle 1 | \hat{S} | 2 \rangle \equiv S^1{}_2 \equiv S^1_2$, and therefore

$$S^\sigma_{\sigma'} \equiv S^\sigma{}_{\sigma'} \equiv \langle s^\sigma | \hat{S} | s_{\sigma'} \rangle . \tag{6}$$

When following the direction of the indices in a pictorial representation, Eq. (5) 'flows' from the left towards the right (upwards in Fig. 2). This is reverse to how one would read such an expression, e.g., as part of an expectation value from a 'time-ordered' perspective, namely right to left (downwards in Fig. 2): one starts with the ket-state $|A\rangle$ at the right, and then applies particular operators in a well-defined sequence, and in this sense 'time order'. This proceeds towards the left until it meets a bra state at the very left for the case of a matrix element or expectation value.

### 2.3.2 Leg directions in `QSpace`

The concept of raised or lowered indices applies to every level of a `QSpace` tensor. It applies to the whole tensor, but also all RMTs and CGRs individually. The CGRs are references to sorted CGTs (these are tabulated with sorted $q$-labels to avoid proliferation of entries). Bearing in mind that the relative index order of raised relative to the lowered indices is irrelevant, the raised and lowered $q$-labels in CGTs can be simply grouped without paying attention to their relative order. For example, irreducible operators (irops) have CGTs of the form $C^{q'}_{q q_{\text{op}}} \equiv \langle q' | C_{q_{\text{op}}} | q \rangle$. For the case of a scalar operator with a singleton operator dimension having $q_{\text{op}} = 0$, this can be reduced to $C^q_q$. Standard CGCs have $C^{q_1 q_2}_{q_3}$ as they fuse $(q_1, q_2) \to q_3$. 1$j$ symbols have CGTs $C^{q\bar{q}}$ or their conjugate $C_{q\bar{q}}$ which fuse $(q, \bar{q}) \to 0$ or vice versa, with $\bar{q}$ the dual irreducible representation (irep) to $q$. Again since the relative index order of raised relative to the lowered indices does not matter, this documentation also uses the alternative notation

$$(q_1 \ldots q_l q^*_{l+1} \ldots q^*_r) \equiv (q_1 \ldots q_l | q_{l+1} \ldots q_r) \equiv (q_1, \ldots, q_l; q_{l+1}, \ldots, q_r) \tag{7}$$

for a rank-$r$ CGT $C^{q_1 \ldots q_l}_{q_{l+1} \ldots q_r}$, where the asterisk $*$ on the l.h.s. explicitly denotes an outgoing index (see also tensor conjugation). By listing all $l$ incoming legs first, followed by the remaining $r-l$ outgoing legs with $l \in [0, r]$, this permits the notation on the r.h.s. This index ordering is adopted in the `RC_STORE`/CStore to avoid a proliferation of CGTs. On top of this simple grouping of indices, the sorted CGTs in the `RC_STORE` also have the $q$-labels non-trivially sorted in a lexicographical manner within the incoming as well as the outgoing indices.

QSpace permits an arbitrary permutation of legs of a tensor, with an example shown in Eq. (43). For this the information on in- or outgoing cannot be simply grouped as in Eq. (7), but needs to be specified explicitly with each leg. It is stored with the `QSpace` tensor as a

whole with the tensor's `itags`. By convention, a trailing `'*'` in the string `X.info.itags{`$l$`}` indicates an outgoing index for leg $l$. If there is no trailing `'*'`, the index is considered ingoing instead. `QSpace` insists that directions are specified with the `itags` globally for a tensor, i.e., the field `X.info.itags` must be set, at the very minimum containing the conjugate flags for every leg. Having the directions specified with the `itags` is important for the purely abelian setting, since in that case `X.info.cgr` is not required and can be set empty. In the presence of non-abelian symmetries, however, the leg directions specified with the `itags` is redundant since this information can be derived from each individual CGR. The $q$-directions specified with the `itags` via trailing asterisks for the tensor overall are thus fixed for a given tensor and must be preserved when modifying its `itags`. To ensure this, `QSpace` provides the function setitags(), with an example shown in Eq. (51).

The q-direction for CGTs as referenced by CGRs are collected and encoded as plain strings `qdir='`$\eta_1 \eta_2 \ldots \eta_r$`'`, with $\eta_i \in \{+, -\}$. For example, see `X.info.cgr(i,j).qdir` for a QSpace `X`. The $q$-directions may also be derived in string form from the `itags` of a `QSpace` tensor via `qdir=getqdir(X,'-s')`. For example, this yields the strings

$$
\begin{array}{ll}
\texttt{'+-'} & \text{for scalar operators} \\
\texttt{'+--'} & \text{for irops} \\
\texttt{'+-+'} & \text{for } A\text{-tensors, etc.}
\end{array}
\tag{8}
$$

`QSpace` also uses this string notation to organize much of the folder structure for each non-abelian symmetry in the `RC_STORE`. Having sorted CGTs there, this groups all `+`'s to the front.

## 2.4 Conventions in pictorial representations

Pictorial representations are instrumental in describing tensor network approaches and algorithms [17, 30, 33]. For example, Fig. 2 shows a tensor network diagram for an expectation value of some product of interactions $\hat{S}_i^\dagger \cdot \hat{S}_j$. Based on this, general conventions in graphical depictions of tensor networks with `QSpace` are as follows [8, 15]:

- Tensors are denoted as some blob (box, circle, etc.). Each tensor has as many legs attached as it has indices, the number of which defines its rank. The legs of tensors represent starting or end points of lines.

- Every line in a tensor network carries an arrow: from the point of view of a tensor, arrows specify incoming/outgoing legs or, equivalently, raised/lowered indices. A contracted line has no open ends: it starts from some tensor, and terminates in another (possibly also the same) tensor. Such a contracted line represents a summed-over index where, by construction, a lowered index leaving some tensor needs to terminate as a raised index with some other tensor (cf. Einstein summation convention).

- The tensor network diagram as in Fig. 2 is read top to bottom: starting from the ket-state $|A\rangle$ at the top, a set of operators, gates, or interactions are applied (round objects $S_i^\dagger \cdot S_j$). The order of the operators applied is relevant, up to trivial shifts along the vertical line they act upon, as long as they do not cross with other operators acting on the same line (state space). The diagram is terminated eventually by a bra-state $\langle A|$ at the bottom. Overall, this results in the desired matrix elements or expectation value.

- By way of reading the diagram, this suggests an implicit 'temporal' order top to bottom (big green arrow to the left), as defined by the particular order of operators applied onto $|A\rangle$. With the intuitive convention that the local state space enters a ket state, however, the 'temporal' direction in Fig. 2 is opposite to the arrows shown with the lines that are associated with the local state space $\sigma_i$.

- When taking a ket-state $|A\rangle$, one implies its *coefficients* $\langle s^\sigma | A \rangle$ with index $\sigma$. That index represents a projection onto a *bra*-state [cf. Eq. (4)]. This reflects the general situation in numerical simulations: tensors are always cast into a particular basis. By attaching legs to a tensor in a tensor network diagram, one implies that it is cast into a coefficient space in some arbitrary but fixed basis that may be given a label (cf. itags). By convention the ket-state like $|\sigma\rangle$ represents an outgoing index. Therefore the tensor coefficients projected onto a bra-state are reverse and thus ingoing.

- When a tensor is shown as a filled (empty) object such as a box, circle, etc., its conjugate tensor is shown as a mirrored empty (filled) shape, respectively, for visual differentiation (e.g., see $A^*$, $S^*$ or $S^\dagger$ in Fig. 2).

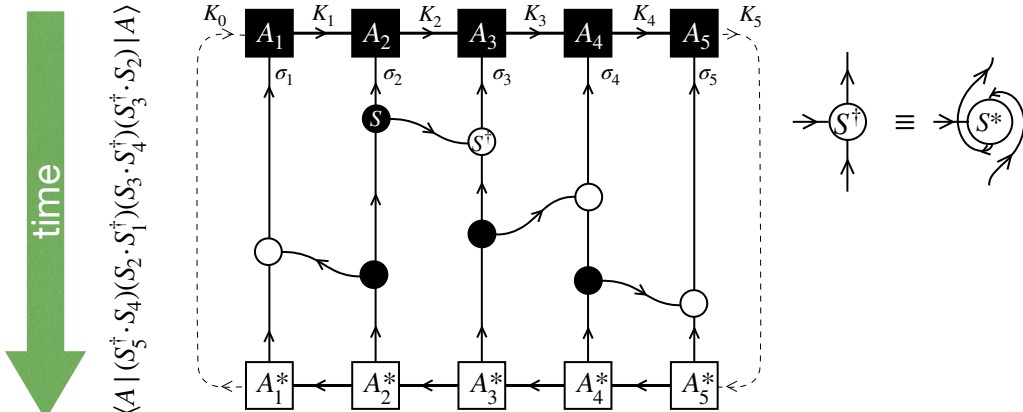

Figure 2: Conventions in tensor network diagrams based on an MPS-based example for the expectation value of some product of interactions $\hat{S}_i^\dagger \cdot \hat{S}_j$ (pairs of connected circles) like for Heisenberg spins. The corresponding mathematical expression is specified to the left and vertically aligned with the various terms in the pictorial representation. The definition of the operator $S^\dagger$ is depicted to the right in terms of the conjugate tensor $S^*$. It is the mirror image of $S$ reflected horizontally and leg directions reverted, such that the operator index enters $S^*$ (and thus also $S^\dagger$) from the left, but exits $S$ from the right (see also Fig. 9 for a similar context). The state is given by $|A\rangle \equiv \prod_i^L A_i$ (ket-state described by the MPS at the top) operating on the local state space $\sigma_i$ for sites $i = 1, \ldots, L$, having $L = 5$. Bond states, e.g., up to and including site $i$ are denoted as $K_i$ (like 'kept' many-body states up to iteration $i$ in the presence of truncation). Here $K_0$ is assumed the left vacuum state with trivial bond dimension $D_0 = 1$. Similarly, $K_5$ describes the full MPS $|A\rangle$. Being a single many-body state, hence also $D_5 = 1$. The outer contractions (dashed lines) are thus trivial. The direction of 'time' (green arrow) indicates the order of operators applied to the initial ket-state $|A\rangle$ at the top.

## 2.5 Tensor conjugation (denoted by '*')

The standard concept of *Hermitian conjugate*, or equivalently *conjugate transpose*, is well-defined only for rank-2 tensors, i.e., matrices or operators. For an irreducible operator $S$, for example, its Hermitian conjugate $S^\dagger = \text{permute}(S, \text{'}21*\text{'})$ [1] consists of transposition on

---

[1] The potentially missing 3 in the permutation '21*' → '213*' is implied in QSpace for trailing indices that keep their position preserved, i.e., act like an identity permutation. This way, S may be a rank-2 or rank-3 operator.

the first two indices (`'21'`), and conjugation (the trailing `'*'`). For a general tensor, however, transposition in itself is not well-defined, as it generalizes to a non-unique permutation that is chosen depending on the context. The situation is different, though, in pictorial representations where the generalization of the transpose to arbitrary-rank tensors is well-defined in that one takes the mirror image of the tensor (cf. $A_i^*$ or $S^*$ in Fig. 2; [17, 30, 34]). Nevertheless, this leaves the freedom to rotate the tensor as a whole in the pictorial setting. But then this holds for any tensor, conjugate or not. More specifically, tensor conjugation thus proceeds as follows [15]:

(1) draw the *mirror image* of the original tensor in pictorial representations

(2) complex conjugate the array entries (relevant for complex RMTs only)

(3) revert arrows on all legs (i.e., swap raised/lowered indices)

(4) index spaces remain the *same*, such that symmetry labels are also left intact.

Point (4) is added for emphasis only here concerning symmetry labels. For better visual differentiation of conjugate tensors concerning point (1), one may furthermore switch from filled tensor objects to empty (outlined) ones, or vice versa (see conventions in Sec. 2.4, or $A_i^*$ or $S^*$ in Fig. 2 as explicit examples).

Point (1) only applies to pictorial representations but not to the practical numerical context. Because the permutation underlying point (1) is not well-defined, tensor conjugation in QSpace adheres to points (2-4) only. QSpace preserves the index order in tensor conjugations unless specified otherwise by the user (see Sec. 4.2 for examples). Point (1) reflects the generalization of the transposition of matrices. Effectively, this fully reverts the order of indices. But for tensors of arbitrary rank, the starting point of what is considered the first index is not well-defined unless one adheres to some arbitrary but fixed convention. For example, there are many different ways in which the mirror plane can be drawn. In Fig. 2, for example, the $A_i^*$ (bottom) are mirrored vs. their respective $A_i$ (top) by a horizontal plane, whereas for the interactions, the operator $S^*$ is mirrored by a vertical plane w.r.t. their paired up $S$ operator (see definition of $S^\dagger$ to the right of Fig. 2).

Due to (3), tensor conjugation (or 'Hermitian conjugate') where present must always be explicitly specified and included in QSpace, even if the matrix elements themselves in the RMTs are not complex (point 2). This is important since reverting directions is equivalent to raising all lowered indices and vice versa. This is an intrinsic concept of tensors that needs to be respected for overall consistency. From the symmetry point of view, directions on legs frequently also indicate types of orthonormalization. Hence by QSpace insisting on well-defined directions of legs throughout the tensor network, point (3) also helps to avoid errors in the coding of tensor network algorithms.

When conjugating a tensor, the action in steps (3-4) will also be referred to as 'conjugating' indices or legs. For a given leg and symmetry sector $q$, this is denoted by $q \to q^*$. It needs to be differentiated from the dual representation denoted by $\bar{q}$. The latter only comes into the picture when reverting directions of individual legs by applying, i.e., contracting $1j$ symbols. This then not only reverts the direction of the affected leg, but also maps symmetry labels to their duals, $q \to \bar{q}$. Hypothetically, if one were to revert all arrows explicitly after tensor conjugation back to their original direction by applying, i.e., contracting the respective $1j$ symbols onto every leg, the resulting (QSpace) tensor is generally different from the original one. Aside from having flipped symmetry labels to their duals, for the case of complex RMTs the reduced matrix elements would still also remain complex conjugated, bearing in mind that $1j$ symbols only have non-trivial structure within the CGTs, but behave like an identity within the RMTs.

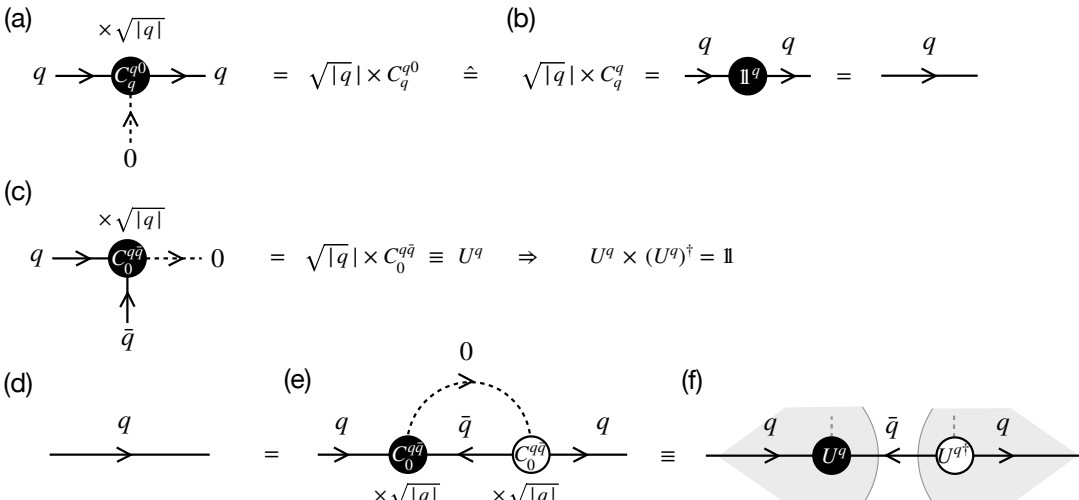

Figure 3: Identity and $1j$ symbols – (a) Fusing a multiplet $q$ with a scalar (singleton 0) trivially leads to the same multiplet $q$. Hence the underlying CGT must be the identity $\mathbb{1}^q/\sqrt{|q|}$ up to the CGT (ortho)normalization convention. Since the singleton index for the scalar irep (0) can be skipped, the same holds for rank-2 CGTs as in (b). The identity $\sqrt{|q|}\, C_q^q = \mathbb{1}^q$ can be trivially inserted into or removed from any directed line. (c) Construction of the $1j$ symbol $U^q$ ('one $q \to j$' symbol [35,36]) from a standard rank-3 CGC $C_0^{q\bar{q}}$ up to normalization involving the dual representation $\bar{q}$. The construction in (d-f) shows that $1j$ symbols represent orthogonal matrices (unitaries).

## 2.6 $1j$ symbols (individual ireps)

In the literature of SU(2), $3j$ symbols refer to standard CGCs $(j_1 j_2 | j_3)$ that fuse angular momenta $(j_1, j_2) \to j_3$. Within QSpace, symmetry labels are generally written as $j \to q$ to emphasize that these may represent labels for general (non-)abelian symmetries. Hence a standard CGC is written as rank-3 CGT $C_{q_3}^{q_1 q_2}$. Now for the case that two multiplets $q_1$ and $q_2$ fuse into the scalar irep $q_3 = 0$ [Fig. 3(c)], the respective CGT has a singleton dimension as third index that may be skipped. This case is possible if and only if $q_2$ is the dual representation to $q_1$, i.e., $q \equiv q_1 = \bar{q}_2$. This results in the CGT $C^{q\bar{q}} \equiv C_0^{q\bar{q}}$ which exists and is unique for every irep $q$, since no outer multiplicity can occur for CGTs of rank $\leq 2$, the CGTs $C^{q\bar{q}}$ are unique up to the standard CGC sign convention. In QSpace, these particular rank-2 CGTs $C^{q\bar{q}}$ are automatically computed and stored for every new irep $q$ that is generated in a tensor product decomposition for non-abelian symmetries without necessarily performing the full decomposition of $(q, \bar{q})$ (see Ref. [15] for details). For abelian symmetries, the $1j$ symbols can be trivially generated on the fly with no need to store.

Importantly, every $C^{q\bar{q}}$ exactly also represents an orthogonal transformation up to a normalization factor [Fig. 3(c)],

$$U^q \equiv \sqrt{|q|}\, C^{q\bar{q}} \qquad (1j \text{ symbol}) , \qquad (9)$$

where $|q|$ represents the state space dimension of irep $q$, such that $U^q U^{q\dagger} = U^{q\dagger} U^q = \mathbb{1}$ (bearing in mind that with all CGTs real, the dagger is equivalent to a simple transpose). These $U^q$ can be used to revert arrows on lines in a tensor network state, as sketched in Fig. 3(d-f) [15] from the symmetry perspective of CGTs [in practice, reverting arrows typically also affects the RMTs, e.g., when orthonormalizing state spaces in the process of shifting the orthogonality

center (OC), etc.]. By inserting the identity $UU^\dagger$ into a fully contracted line for irep $q$ in between, say, tensors $A$ and $B$, splitting this product, and contracting $U$ and $U^\dagger$ onto $A$ and $B$, respectively [indicated by the shaded gray arrows in the background of Fig. 3(f)], this results in a (i) reverted arrow on that line, and also on the respective legs of the tensors $A$ and $B$. Furthermore, by construction, this simultaneously (ii) switches the symmetry sector on that line to the dual representation, $q \to \bar{q}$. Because of its usefulness, $U^q$ is given a name. Since it only depends on a single symmetry label $q$ [or 'one $j$' symbol for SU(2)] with $\bar{q}$ simply inferred, it is called a $1j$ symbol with reference to the literature of SU(2) [15, 35, 36].

**Caveat self-dual ireps**   In the construction of the $1j$ symbol as rank-2 tensors, the singleton dimension in the scalar representation 0 [black dashed line in Fig. 3(e,f)] was skipped. This results in a caveat, in that the truncated 'stem' of this singleton dimension [remnant of the black dashed line in Fig. 3(f)] needs to emerge on the *same* side for both $U^q$ and $U^{q\dagger}$, here emerging at the top. Alternatively, one could have closed the black dashed line in Fig. 3(e) at the bottom.

More specifically, the caveat concerns self-dual ireps here, $q = \bar{q}$, like all ireps in SU(2). In that case the $1j$ symbol $U^q$ becomes indistinguishable from $U^{\bar{q}}$ in terms of symmetry labels and $q$-directions, despite that they may differ by a sign [15], since $U^q \equiv C^{q\bar{q}} = C^{qq}$, yet $U^{\bar{q}} \equiv C^{\bar{q}q} \equiv (C^{q\bar{q}})^T = (C^{qq})^T = \pm U^q$. The last sign is negative, for example, for all half-integer spins for SU(2). Hence when inserting $U^q U^{q\dagger} = \mathbb{1}$ into a contracted line of a tensor network for the sake of reverting its arrow, proper care must be taken that the *same* object is used twice, once with and once without the dagger. Otherwise, this can give rise to erroneous signs. As a safeguard in that regard, trailing primes are introduced with itags to mark dual state spaces, as in $C^{q\bar{q}}$ even if $q = \bar{q}$.

## 2.7   Identity operator vs. $1j$ tensor and marker for dual state space

The $1j$ symbol was introduced above for a particular multiplet $q$ of some symmetry. Yet it can be easily generalized to cover full state spaces as implemented with getIdentity. This will also be referred to as the $1j$ symbol or $1j$ tensor, nevertheless, based on its origin. The $1j$ tensor is closely related to the identity operator [cf. Fig. 3]. Hence both of them refer to the state space associated with a particular leg of a tensor and are also obtained by the same QSpace routine getIdentity. See Sec. 4.3 for detailed examples. To start with, both objects are of rank 2. However, while the identity operator is a plain scalar operator, the $1j$ tensor is not an operator per se, as it has both of its indices incoming (or both of them outgoing for its conjugate). Therefore it is also not necessarily block-diagonal, but rather anti-block-diagonal with respect to its symmetry sectors.

The identity operator $E$ as obtained by getIdentity adheres to the index order convention of operators as in Fig. 1(b). By having a scalar rank-2 operator here, $E$ has one in- and one out-going index. The $1j$ tensor is obtained by simply adding the option '-0' ('fuse into $q = 0$') to the same call to getIdentity otherwise. By contrast to the identity operator, this returns a unitary $1j$-tensor $U$ on the same state space, yet with both indices ingoing. It is real and therefore satisfies $UU^T = UU^\dagger = \mathbb{1}$. The QSpace $U$ returned by getIdentity has the same RMTs as the identity operator. However, instead of identity CGTs, it references the respective $1j$ CGTs for the given state space, properly normalized via the respective CGRs. In this sense, the $1j$-tensor $U$ defined over an entire state space of a leg of a tensor is also frequently referred to as a $1j$ symbol itself.

**Marking itag of dual state space**   When calling U=getIdentity(A,l,'-0') this returns the full state space as present in QSpace A on leg $l$ (first index in $U$) with its dual on the second

index in $U$. To safeguard against the above caveat, the itag of the dual state space on the second index is marked with a trailing prime ('). After all, the dual state space is *different* from the original state space, and hence its itag also should be differentiated. The trailing marker character serves as a toggle: reverting the arrow twice returns to the original itag. Therefore adding a prime to an existing trailing prime, annihilates both of them, i.e., foo'' → foo. So if by the history of an algorithm, the input space already has a mark on its itag, then that mark is removed for the dual space on the second index. Detailed examples are given and discussed in Sec. 4.3.

## 2.8  Generating scalar (interacting) Hamiltonian terms from irops

A Hamiltonian represents a scalar operator. Hence from the point of view of symmetries, all terms constituting a Hamiltonian term must be scalars themselves. With this in mind, general two-body interaction terms in a Hamiltonian can always be written as or built from scalar bilinears $\hat{S}_i^\dagger \cdot \hat{T}_j$ acting on sites $i$ and $j$, also permitting $i = j$. Typically, $\hat{S} = \hat{T}$, with pictorial examples shown in Fig. 2, and explicit QSpace contractions to describe such pairwise interactions provided in Eq. (42) or Sec. 5.1, e.g., Eqs. (48)–(50) together with Fig. 9.

From a symmetry point of view, the bilinear $\hat{S}_i^\dagger \cdot \hat{T}_j$ requires (i) that the operators $\hat{S}$ and $\hat{T}$ are irops, or more generally operators. In the case of a pair of irops, (ii) these must transform according to the *same* operator irep $q_{\text{op}}$, as this is summed over (contracted) in the dot-product $\hat{S}_i^\dagger \cdot \hat{T}_j$. In addition, for the contraction to be valid, the index must be outgoing from one irop, and incoming to the other irop. Therefore the general natural structure includes the conjugate tensor, i.e., always makes explicitly use of the dagger, as in $\hat{S}_i^\dagger \cdot \hat{T}_j$. The dagger may equally well be written with the second operator, instead. For the example, consider the SU(2) spin operator written as the irop that transforms like an $S = 1$ multiplet

$$
S \;=\; \begin{pmatrix} -\frac{1}{\sqrt{2}} S_+ \\ S_z \\ \frac{1}{\sqrt{2}} S_- \end{pmatrix}
\tag{10}
$$

where the minus sign with the first component originates from the application of raising or lowering operators [8]. The inverse square root factors ensure the simple dot-product structure for isotropic spin interactions, having $\hat{S}_i^\dagger \cdot \hat{S}_j = \frac{1}{2}(\hat{S}_{i+}\hat{S}_{j-}+\text{H.c.})+\hat{S}_{iz}\hat{S}_{jz} = \hat{S}_{ix}\hat{S}_{jx}+\hat{S}_{iy}\hat{S}_{jy}+\hat{S}_{iz}\hat{S}_{jz}$. The components of the spin operator in Eq. (10) are non-hermitian, and thus do require the dagger with $\hat{S}_i^\dagger \cdot \hat{S}_j$. For general operators, if the scalar result out of $\hat{S}_i^\dagger \cdot \hat{T}_j$ is non-hermitian [e.g., as is the case for fermionic hopping described by $S \to F$ with $F$ some set of fermionic annihilation operators; see Eq. (50) and subsequent discussion], then the Hermitian conjugate $(\hat{S}_i^\dagger \cdot \hat{T}_j)^\dagger = \hat{T}_j^\dagger \cdot \hat{S}_i$ needs to be added if part of a Hamiltonian. Hence a typical Hamiltonian can be written as a sum of scalar operator contributions,

$$
\hat{H} = \sum_{i\alpha} \varepsilon_i^\alpha \hat{n}_{\alpha i} + \sum_{ij,\alpha} V_{ij}^\alpha \, \hat{S}_{\alpha i}^\dagger \cdot \hat{T}_{\alpha j} \;\; [+\text{ H.c.}]
\tag{11}
$$

which may be further extended also to include products of terms such as the above. Here $\hat{n}_\alpha \equiv \hat{c}_\alpha^\dagger \cdot \hat{c}_\alpha$ describes some set of local scalar operators, and $\hat{S}_\alpha$ and $\hat{T}_\alpha$ some set of local irops, typically having $\hat{S} = \hat{T}$ for a particular interaction. The Hermitian conjugate is required for all interactions whose operator irep $q$ is not self-dual. For example, it is not required for SU(2) spin interactions $\hat{S}_i^\dagger \cdot \hat{S}_j$. It is required for fermionic hopping, though, when using U(1) charge symmetry, since the annihilation operators $F$ transform like charge $q = -1$ (they reduce charge by 1), whose dual are creation operators with $\bar{q} = +1$.

Taking the irop $S$, its daggered version $S^\dagger$ could be explicitly constructed, in principle: the tensor $S$ would have to be conjugated which reverts all legs, including the irop index. By

subsequently applying a $1j$ symbol $U$ onto the irop index 3 and permuting indices 1 and 2, this brings the irop $\tilde{S} \equiv US^\dagger$ back to canonical QSpace form for irops, with the crucial difference, that the irop now transforms according to $\bar{q}$. In practice, such a transformation to obtain a canonical form of $S^\dagger$ by using $1j$ symbols etc. is never required when computing $\hat{S}_i^\dagger \cdot \hat{T}_j$ [e.g., see Fig. 9 and corresponding text]. Much to the contrary, it would be rather impractical and prone to errors. Having both, $S$ and $\tilde{S} \equiv S^\dagger$ in canonical irop form, the irop index is outgoing in both tensors. This would prevent a simple contraction as in $\tilde{S} \cdot S$ on the irop index, because leg directions are incompatible. Considering $\tilde{S}^\dagger \cdot S$, instead, this would be contractible in principle for self-dual $q_{\mathrm{op}}$, such as for the SU(2) spin operator, but wrong in most cases, in the sense that it gives a different operator $\tilde{S}^\dagger \cdot S = (SU^\dagger) \cdot S \neq S^\dagger \cdot S$.

Hence there shall never be a need to explicitly construct $\tilde{S} \equiv S^\dagger$ as canonical irop based on $1j$ symbols, etc., when computing interaction terms in a Hamiltonian. What is solely required is to contract the conjugate tensor of $S_i$. This can be indicated on the fly as an option when calling `contract` itself while ensuring that the 'transposed' indices are correctly contracted. The latter is also dictated by the directions of the legs [e.g., see Fig. 9 and corresponding text]. Hence much of this is taken care of automatically when using auto-contraction based on `itags`.

## 2.9 Compact symmetry labels for non-abelian symmetries

The motivation for a 'compact' notation for symmetry labels ($q$-labels) of non-abelian symmetries is mainly for readability purposes, e.g., to have well-aligned displayed tables, but also a means for a simple standardized notation in text. The compact notation also reflects the situation in practice in the code, where an $n$-tupel of labels is just a vector of numbers without any white space or separators. The contact notation permits a lean, concise specification of symmetry labels. It is specific to non-abelian symmetries since ireps for any abelian symmetry are always described by a single symmetry label only (see symmetry rank). Therefore plain signed integer notation is used for abelian symmetries. This is also the only setting where negative symmetry labels $q < 0$ can occur.

**QSpace label convention for SU(2):** $q = 2S$    The concept of compact symmetry labels for non-abelian symmetries relies on symmetry labels that are (i) non-negative and (ii) integer. This is the case for all simple Lie algebras, except for the half-integer spins in SU(2) for historical reasons. The latter 'exception', however, is only by convention. Hence QSpace adopts the alternative labeling convention for SU(2) which is fully aligned with the labeling scheme for SU($N > 2$) ireps more generally, namely by choosing the SU(2) multiplet label $q = 2S \in \mathbb{N}_0$. This is consistent with the symmetry labels for SU($N$) in general, in that they specify Young tableaus [e.g., see Eq. (12)]: an SU(2) spin $S$ has a Young tableau of a single row with $q = 2S$ boxes. This convention for SU(2) also permits the usage of plain integer labels for symmetry labels, and thus avoids, e.g., the need to print half-integers. Half-integer spins $S$ then map to odd integers for $q = 2S$.

As a corollary, if an SU(2) symmetry gets broken down to U(1), the symmetry labels become $q = 2S_z \in \{-2S, -2S + 1, \ldots, 2S\}$, thus also ensuring integer labels in the U(1) context for the case of half-integer spins. For example, a spin $S = 1/2$ then acquires the U(1) spin labels $q_S \equiv 2S_z \in \{-1, 1\}$.

**QSpace label convention for U(1) charge**    From a practical point of view, it is desirable in tensor network simulations to keep symmetry labels around $q \sim 0$. This way the low-energy symmetry sector remains the same with increasing system size also towards the thermodynamic limit. This avoids a 'running' symmetry label that keeps growing proportional to block size. For counting particle number (charge), it is therefore desirable to count particles relative

to the average filling, i.e., $q \equiv n - n_0$ with $n_0$ the average filling per site for the entire system, typically half-filling. For example, for a single spinful fermionic level, the total occupation relative to half-filling $n_0 = 1$ is given by the sectors $q_C \equiv n - n_0 \in \{-1, 0, 1\}$, corresponding to empty, half-filled, and fully occupied, respectively. This reflects the charge symmetry labels as returned by getLocalSpace.

For the description of an odd number of flavors, however, such as a single spinless fermionic level, the prescription above with a focus on 'relative to half-filling' would result in half-integer labels for charge. Hence in the case of an odd number of flavors per site, where $n_0$ for half-filling becomes a half-integer, QSpace adopts the convention $q_C \to 2(n - n_0)$, which includes a factor of 2 for the sake of having integer symmetry labels for charge. For a single spinless fermionic level, therefore getLocalSpace returns $q_C \in \{-1, 1\}$. As for the case of SU(2) labels above, this convention of applying a factor 2 is mainly for convenience and readability, yet also for consistency across symmetries. For example, SU(2) particle/hole symmetry always takes the viewpoint of particle number relative to half-filling. In this sense, when breaking down an SU(2) particle/hole symmetry to U(1) charge, $q_C \to 2(n - n_0)$ is consistent with the QSpace convention $q_S \to 2S_z$ for spin.

On general grounds, U(1) labels may undergo an arbitrary linear map $n \to q(n) = bn - a$ with $a, b \in \mathbb{N}$. Thus if the desired target value for the average filling $n_0 = a/b > 0$ is rational, this ensures that the integer charge labels $q \in \mathbb{Z}$ remain centered around $q \sim 0$ when increasing block or system size. Since this is a rather specialized setting, though, it is not implemented in getLocalSpace as is. Nevertheless, one may tweak its output in this regard.

**Compact alpha-numeric labeling scheme**  By QSpace convention, all symmetry labels for non-abelian symmetries are compact non-negative integers. Therefore any set of $q$-labels $q = (q_1, q_2, \ldots, q_\mathfrak{r})$ with $q_i \in \mathbb{N}_0$ for a particular symmetry of rank $\mathfrak{r}$ is a meaningful irep. As already pointed out above, for the example of SU($N$), these $q$-labels directly specify the *Young tableau* in the sense that $q_i \geq 0$ specifies the offset (i.e., difference) of boxes from rows $i$ to $i + 1$,

$$(q_1, q_2, \ldots, q_\mathfrak{r}) \quad \equiv$$ 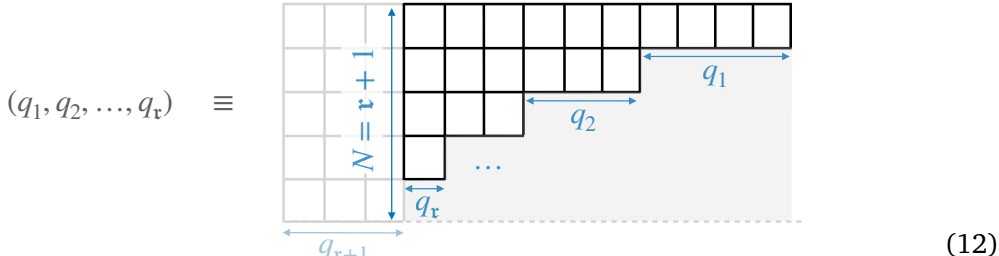

$$\tag{12}$$

This represents the Dynkin symmetry labels for SU($N$) ireps. For a rank-$\mathfrak{r}$ symmetry, there are at most $\mathfrak{r}$ rows since any $q_{\mathfrak{r}+1}$ of completely filled columns, i.e., with $\mathfrak{r} + 1$ boxes, if present, can be split off as a scalar factor in a product state like fashion: a filled column represents the largest set of $N$ particles that can be brought into a completely antisymmetrized state which represents a singlet state. Hence from an irep point of view, the label $q_{\mathfrak{r}+1=N}$ is irrelevant and can be skipped. This is indicated in Eq. (12) by fading out the leftmost $q_{\mathfrak{r}+1}$ filled columns.

When exploiting non-abelian symmetries, one usually assumes that there is no spontaneous symmetry breaking on physical grounds.[2] Therefore assuming a 'non-magnetic' state,

---

[2]If a non-abelian symmetry of a Hamiltonian is broken spontaneously in its low-energy regime, non-abelian multiplets become macroscopically large. In this case, however, there is no gain in exploiting non-abelian symmetries. Rather, the symmetry should be reduced, e.g., to an abelian U(1) symmetry in the simulation itself for the sake of numerical efficiency based on the underlying physics. As a corollary, the tendency of a system to generate ever larger multiplets during initialization of a tensor network wave function, e.g., via iterative diagonalization in an NRG-like fashion even for a uniform system, is a strong indicator of a spontaneously symmetry-broken ground state.

the symmetry labels $q_i$ do not grow macroscopically large with system size, but are rather spread around $q_i \geq 0$. Then if the values for $q_i \in \mathbb{N}_0$ do not grow macroscopically large, QSpace adopts the following extension to the hexadecimal format for the sake of a compact notation: the alphabetic characters `A-Z` are used do describe the range $q_i \in [10, 35]$. If the operating system supports a case-sensitive file system, the range is further extended to $q_i \in [36, 61]$ which maps to the lower-case alphabet `a-z`, respectively (the reference to a file system comes into play here since $q$-labels are also inserted into file names within the file-based database `RC_STORE`). For example then, for SU(2) the irep `q='Y'` $\equiv 34$ corresponds to a multiplet with spin $S = 34/2 = 17$. Therefore if a given set of $q$-labels satisfies $q_i \in \mathbb{N}_0 \leq q_1^{\max}$ for all $i = 1, \ldots, \mathfrak{r}$ with

$$q_1^{\max} = \begin{cases} 61 & (= 9 + 2 \times 26) \;\; \text{for a fully case-sensitive file system, e.g., Linux} \\ 35 & (= 9 + 1 \times 26) \;\; \text{otherwise, like MacOS} \end{cases} \tag{13}$$

the above compact extended hexadecimal notation is adopted for this particular set of symmetry labels in QSpace.

The range in Eq. (13) covers the vast majority of tensor network simulations. For example for SU(2), typical tensor network ground state calculations in the singlet symmetry sector $q = 0$ stay within $q \lesssim 10$ i.e., $S \lesssim 5$. This also generalizes to general non-abelian symmetries, typically having $q_i \lesssim 10$, yet with significantly more possible combinations due to the presence of $i = 1, \ldots, \mathfrak{r}$ symmetry labels. Hence, the compact symmetry labels overwhelmingly stay within the 'numeric' range $0 \ldots 9$ for all $q_i$, yet permit to exceed this range significantly within the compact notation by including alphabetic characters. Overall this motivates the compact alpha-numeric notation of symmetry labels

$$q \;\; \equiv \;\; (q_1, q_2, \ldots, q_{\mathfrak{r}}) \equiv (q_1 q_2 \ldots q_{\mathfrak{r}}) \equiv \text{'} q_1 q_2 \ldots q_{\mathfrak{r}} \text{'} \tag{14}$$

for a particular irep $q$, with all spacing (commas) in between the $q_i$ removed, and written with or without brackets, like simple strings. For the case that $q_i > q_1^{\max}$, a more extended display with regular integers is chosen with the $q_i$ separated by spaces.

For example, the scalar representation is given by $q_{\text{scalar}} = (0 \ldots 0_{\mathfrak{r}}) \equiv 0$, which is also conveniently denoted by a single 0 in this documentation for any symmetry for readability. The defining representation is given by $q_{\text{def}} = (10 \ldots 0_{\mathfrak{r}})$. For example, for SU($N$),

$$q_{\text{def}} \;\; = \;\; \begin{cases} (1) & \text{equivalent to } S = q/2 = 1/2 \;\; \text{SU(2)} \\ (10) & \equiv (1, 0) \qquad\qquad\qquad\qquad\;\; \text{SU(3)} \\ (100) & \equiv (1, 0, 0) \qquad\qquad\qquad\;\;\; \text{SU(4)} \\ \ldots & \qquad \ldots \qquad\qquad\qquad\qquad\;\;\; \text{etc.} \end{cases} \tag{15}$$

The dual representation for SU($N$) simply flips the order of labels in the standardized scheme above, i.e., $\bar{q} = (q_{\mathfrak{r}} q_{\mathfrak{r}-1} \ldots q_1)$.

## 2.10 Outer multiplicity

Outer multiplicity (OM) is ubiquitous when dealing with non-abelian symmetries in tensor network algorithms. It adds significant overhead in the book-keeping when dealing with symmetries, yet is fully taken care of by QSpace in its MEX core routines. The discussion here therefore serves as background information only. While it is important to be aware of on general grounds when interpreting QSpace tensors, the actual book-keeping is hidden from, and in this sense of no further concern for the typical QSpace user.

To be specific, consider a rank-4 tensor having SU(2) symmetry with two incoming legs and two outgoing legs which, for simplicity, all carry the same spin $S$. This is described by $(S, S | S, S)$ or, equivalently, the CGT $C_{qq}^{qq}$ with $q = 2S$ in QSpace [cf. Fig. 4]. Now since $(S, S)$ can be fused

into $M = 2S + 1$ intermediate spins $S_i = 0, 1, \ldots, 2S$, each of these can again be 'unfused' symmetrically into the outgoing indices. Hence one may contract $(SS|S_i) *_i (SS|S_i)^* \equiv (SS|SS)_i$ on the intermediate index for any fixed $S_i$ (this represents a projector onto the intermediate multiplet $S_i$ of dimension $2S_i + 1$). Then $i \to \mu = 1, \ldots, M$ takes the role of an outer multiplicity index: for given CGT $C_{qq}^{qq}$ with fixed ireps on all legs, there exist *multiple* orthogonal CGT's permissible by symmetry. Hence the CGT $C_{qq}^{qq,\mu}$ naturally acquires an additional index, the OM index $\mu$ which spans the OM space of dimension $M$. The CGT $C_{qq}^{qq}$ is unique then, up to an arbitrary orthogonal rotation in OM space [this generalizes the sign convention of standard rank-3 CGCs for SU(2)]. The choice of basis within this OM space in QSpace is discussed with the CGT (ortho)normalization convention. Overall then, one has $M > 1$ in the *presence* of OM, $M \leq 1$ in the *absence* of OM, and $M = 0$ if a particular CGT does not exist at all, i.e., is not permissible from a symmetry point of view.

For abelian symmetries, one always has $M \leq 1$, i.e., outer multiplicity does not occur. It is clear from the above example, however, that OM is already ubiquitous even for SU(2) for tensors of rank $r \geq 4$. For general non-abelian symmetries of symmetry rank $\mathfrak{r} > 1$, OM already occurs at the fundamental level of standard CGCs, i.e, CGTs of rank $r = 3$ [8, 37, 38] (there can never be OM for $r \leq 2$, like scalar operators).

**OM index is specific to tensor (not particular leg)**    Standard CGC decomposition $C_{q_3}^{q_1 q_2}$ may tempt one to assign the OM index $\mu = 1, \ldots, M$ with the fused multiplet $q_3$, as there are $M$ of these occurring in the decomposition. However, by applying $1j$ symbols, any other raised index in $C_{q_3}^{q_1 q_2}$ can be lowered, while raising $q_3$. While this changes symmetry labels to their duals, this nevertheless also demonstrates that, by construction, precisely the same multiplicity will occur in all these other instances, too. Therefore the OM index may be associated with any of the legs, depending on the context. In this sense, the OM index belongs to a CGT as a whole, and not to any particular leg. This disregard for the direction of the legs is also reflected in the CGT norm convention in Eq. (20).

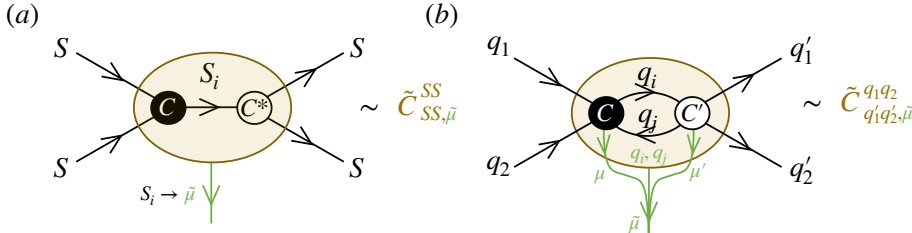

Figure 4: Outer multiplicity (OM; figure adapted from [15]) based on the simple specific example of a rank-4 CGT of SU(2) in panel (a) where the CGT contraction $(SS|S_i) *_i (SS|S_i)^* \equiv (SS|SS)_i$ on the intermediate index for fixed intermediate multiplet $S_i \in 0, \ldots, 2S$ gives rise to outer multiplicity for the rank-4 CGT $\tilde{C}_{qq,S_i \to \tilde{\mu}}^{qq}$ with $q = 2S$ and $\tilde{\mu} = 1, \ldots, M$ having $M = 2S + 1$. (b) More generally, contracting two CGTs $C$ and $C'$ on two shared indices with fixed intermediate symmetry sectors $q_i$ and $q_j$. Each of the CGTs $C$ and $C'$ can have OM on their own, for general non-abelian symmetries already so at the level of rank-3 CGTs as in (a). The combined multiplicity is therefore determined by obtaining the resulting CGTs $\tilde{C}$ for all combinations $(\mu, \mu', q_i, q_j)$. If multiple intermediate indices are contracted simultaneously, like $q_i$ and $q_j$ above, then the plain contractions $\tilde{C}(\mu, \mu', q_i, q_j)$ are generally not orthogonal and typically overcomplete. Hence these need to be orthonormalized into an arbitrary but fixed combined OM basis indexed by $\tilde{\mu}$ [cf. Eq. (21)].

**OM can become arbitrarily large** For non-abelian symmetries, OM typically grows exponentially with the number of legs, i.e., the tensor rank [15]. Yet even at the level of standard rank-3 CGCs, OM can become arbitrarily large. For example, for SU(3) the CGT $C_q^{qq}$ with $q = (nn)$ with $n \in \mathbb{N}_0$ has an outer multiplicity $M = n + 1$. That is, fusing two SU(3) multiplets $(nn)$ results in $n+1$ distinct multiplets with the *same* fused multiplet label $(nn)$ amongst others, i.e., $(nn) \otimes (nn) = (00) \oplus (nn)^{n+1} \oplus \ldots$, where the exponent indicates OM. The multiplet $(nn)$ has dimension $|q| = (n+1)^3$, and is self-dual, such that the scalar $(00)$ also occurs in the decomposition above. To be specific, fusing two ireps in the adjoint representation $(11)$, i.e., $n = 1$ above, one obtains the full decomposition $(11) \otimes (11) = (00) \oplus (03) \oplus (11)^2 \oplus (22) \oplus (30)$, this gives the adjoint $(11)$ *twice*. In terms of multiplet dimensionality this reads $\mathbf{8} \otimes \mathbf{8} = \mathbf{1} + \mathbf{10} + (2 \times \mathbf{8}) + \mathbf{27} + \mathbf{10}$, respectively, which adds up to the expected total of $8^2 = 64$ states. Based on the unboundedness of OM even for rank-3 CGTs, there appears to be no simple way to fix a canonical OM basis, e.g., w.r.t. to permutations of legs, etc., since that symmetry is necessarily finite given the finite number of legs. To the best of our knowledge, there is no superior systematic way to deal with arbitrarily large OM for general non-abelian symmetries. Hence a bottom-up constructive approach is adopted in QSpace that is based on demand, and thus arbitrary but fixed, at the expense of a history dependent RC_STORE as discussed with Sec. 2.12.

## 2.11 General tensor decomposition

The general decomposition of a QSpace X in the presence of symmetries was already briefly touched upon in the introduction [cf. Eq. (1)] in the absence of OM. In the presence of OM, this becomes [8, 15, 39, 40],

$$X = \bigoplus_q \left[ \|X\|_q^\mu \otimes \left( w_\mu^{\mu'} C_{q\mu'} \right) \right], \tag{16}$$

where the OM indices $\mu$ and $\mu'$ are implicitly summed over pairwise in a plain sum ($\sum_{\mu\mu'}$). With this, the OM indices are fully contracted in Eq. (16) as visualized in Fig. 5). In practice, however, the CGTs $C_{q\mu'}$ are referenced, and the summation over multiplicity is postponed depending on the context. For pairwise contractions, for example, this summation is translated to generating X-symbols [15] from the CGTs and contracting these onto the RMTs $\|X\|_q^\mu$. That is, the tensor product in Eq. (16) across RMTs and CGTs is never explicitly expanded per se, as this would defeat the purpose of exploiting symmetries.

The leading direct sum ($\oplus$) over $q$ in Eq. (16) implies block structure, which translates to the record index i in the QSpace tensor structure. Via the Wigner Eckart theorem one obtains the tensor product ($\otimes$) of the RMTs $\|X\|_{q\mu}$ with the CGTs $C_{q\mu}$ (assuming $w = \mathbb{1}$ for simplicity). The RMTs and CGTs can be dealt with separately, and hence are stored separately. The RMTs can be chosen freely depending on the physics or the algorithm, whereas the CGTs are fixed by symmetry and hence referenced via the CGRs. The CGRs thus store the matrices $w_\mu^{\mu'}$, while the RMTs $\|X\|_q^\mu$ carry trailing open OM indices $\mu$ for $M > 1$.

**$w$-matrix** An important element in the tensor decomposition (16) is the direct link across the tensor product structure of the RMT and CGT via a plain regular sum on the OM indices. It simply states that for every OM instance there can be a different RMT (the additional space within the RMT due to OM only needs to be explicitly allocated when fusing state spaces [8] which is taken care of by the getIdentity routine). When summing over the OM index denoted in green in Eq. (16), one may insert an additional matrix, referred to as the $w$-matrix, which lives directly on the OM link that connects the RMT to the CGT [cf. Fig. 5]. The $w$-matrix can always be absorbed, i.e., contracted onto the RMT by carrying out the summation over

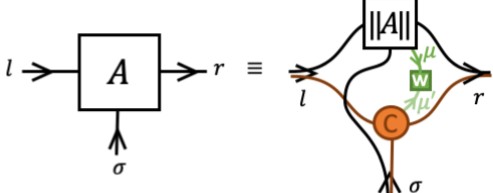

Figure 5: Pictorial representation of tensor decomposition in Eq. (16) for the case of an *A-tensor* for fixed symmetry sectors and hence a particular contribution $q$ for simplicity (figure adapted from [15]). The bare indices $(l, r, \sigma)$ in the absence of symmetries become composite indices [cf. Eq. (2)]. The RMTs $\|A\|_q$ are stored with a QSpace tensor, whereas the CGTs $C_q$ are only referenced via CGRs which point into the RC_STORE database. The CGRs also store the matrices $w_q$ with the tensor.

$\mu$.[3] However, there is practical value in keeping the $w$-matrix nevertheless: it permits one to efficiently absorb rotations in OM space, e.g., as they occur out of permutations of the legs of the tensor, but also to absorb scale factors. For this reason the $w$-matrix is stored with each QSpace tensor via the CGR. It is a real matrix with the default normalization convention

$$w^\dagger w = \mathbb{1} \,, \tag{17}$$

typically taking the first non-zero matrix element in $w$ to be positive. When $w$ is contracted onto the RMT, it can be simply replaced by $w \to \mathbb{1}$ afterwards. By the convention that the superscript index is the first index, i.e., $w_\mu^{\mu'} \equiv w^{\mu'}{}_\mu$, the index $\mu'$ specifies its rows. Then by Eq. (16) the row index of $w$ links to the CGT, whereas the column index links to the RMT. Typically, the OM dimension on the RMT is the same as that of the CGT, but it may also be smaller. At any given time, the maximal OM dimension is set by the CGT (OM may grow dynamically during simulations, until the full OM is exhausted). Yet the RMT of a given tensor may only relate to a particular subset of OM directions, hence may have lower OM. For this reason, $w$ can have fewer columns than rows, i.e., it is a matrix of dimensions $M \times M'$ with $M' \le M$. If $M' < M$, $ww^\dagger \ne \mathbb{1}$, whereas Eq. (17) holds in any case.

There are only two exceptions to the normalization convention in Eq. (17). They are intended for the sake of readability on the Matlab interface only (by contrast, the QSpace core routines always adhere to Eq. (17) internally). In these cases the normalization in Eq. (17) is only proportional to the identity, $w^\dagger w = |w|^2 \mathbb{1}$. For QSpace records $q$ where $|w_q| \ne 1$, these normalization factors are printed to the right when displaying a QSpace tensor (see display in Fig. 7 for a specific example):

- rank-2 tensors (e.g., scalar operators): for the sake of the argument, consider an identity operator $X$. In this case, for the sake of readability and also to avoid confusion, one would like the RMTs themselves to be identities, i.e., $\|X_q\| = \mathbb{1}$ an identity operator acting within symmetry sector $q$. But the CGTs have fixed CGT normalization convention themselves. In particular, for a scalar operator like the identity, they are trivially given by $C_q^q = \frac{1}{\sqrt{|q|}} \mathbb{1}^{|q|}$ with $|q|$ the multiplet dimension. Therefore to reconcile the desired behavior of the RMTs with the CGT normalization convention, the remaining scale factor can be absorbed at the level of the CGR with the $w$ matrix by setting

$$w_q = \sqrt{|q|} \,. \tag{18}$$

---

[3]The $w$-matrix cannot be absorbed into the CGT since CGTs are defined once and for all. Besides, the data format differs: CGTs are stored in higher precision, whereas the RMTs, same as the $w$-matrices, have plain double precision arithmetic.

This cancels the normalization factor of the CGT, thus resulting in a combined identity operator, $wC = \mathbb{1}$. Since there is no OM for rank-2 tensors, the $w$ in Eq. (18) is always a plain number. Aside from identity operators, the altered normalization convention on the $w$-matrix for the case of rank-2 tensors intuitively shows the 'actual' value of matrix elements in the RMTs for scalar operators such as Hamiltonians or density matrices.

- rank-3 tensors with only one in- or outgoing index: similar to the previous case for rank-2 consider here for the sake of the argument the identity $A$-tensor [8] that describes the fusion of two incoming state spaces without truncation into their combined state space. Intuitively, one would also like to see parts of the sliced-up identity matrix in the RMTs here too. That is, the entries in the RMTs should be either 0 or 1, as this indicates a map of input multiplets to fused multiplets. For this, however, the rank-3 CGTs would have to be normalized like standard CGCs, thus desiring $\sqrt{|q_3|}C_{q_3}^{q_1 q_2}$. Again this can be achieved from the CGR point of view by setting

$$w_q = \sqrt{|q_3|}\, \mathbb{1} \,. \tag{19}$$

Now an $A$-tensor is a rank-3 tensor with one outgoing index only. Conversely, irops are rank-3 tensors, with only one incoming index. In this sense, for generality, the normalization in Eq. (19) is adopted for all rank-3 tensors that have one incoming or one outgoing index only. The irep on this leg then defines the dimensional factor in Eq. (19). In the absence of OM for these rank-3 tensors [such as for SU(2) throughout], $w$ is again just a plain number. Otherwise, the $\mathbb{1}$ in Eq. (19) is the identity matrix in the OM space of the underlying $C_{q_3}^{q_1 q_2}$ or $C_{q_1 q_2}^{q_3}$.

**Avoiding high-rank tensors**    The block decomposition via $q$ in Eq. (16) only includes permissible symmetry combinations for the given number of legs. However, it is important to bear in mind in the context of tensor networks that this number of possible combinations $q$ (or equivalently, the number of records in a QSpace) can become quickly very large (exponentially so) if tensors have many open legs, i.e., are high-rank.

This proliferation of non-zero blocks for large-rank tensors is already present for plain abelian symmetries. Consider, for example, a tensor of even rank $r$, with $r/2$ incoming and $r/2$ outgoing indices and where for the sake of the argument, all legs have the same state space, say, with $n_s$ symmetry sectors each. Then *all* $n_s^{r/2}$ symmetry combinations of the $r/2$ incoming legs are permissible since they can again be 'unfused' into the outgoing legs. Therefore the number of blocks (RMTs) grows exponentially with the rank. In practice, there will be typically many more combinations still that satisfy $\sum_{\text{in}} q_{\text{in}} = \sum_{\text{out}} q_{\text{out}}$. The proliferation of non-zero blocks with tensor rank eventually holds generally. It is also irrespective of the direction of legs since these can be easily flipped by contracting $1j$ symbols which does not affect the number of non-zero blocks.

For non-abelian symmetries, the situation is further exacerbated: the fusion of multiplets typically gives rise to several combined ireps, which thus further increases the number of internal fusion channels and hence non-zero blocks (e.g., hundreds of thousands for rank-6 tensors [34]). Furthermore, when building larger-rank tensors by contraction, also the OM of the contracted tensors themselves will grow exponentially with increasing tensor rank [15]. If then one tries, for example, to contract two such large-rank tensors, this quickly translates into extremely long lists of elementary pairwise block contractions (billions when contracting rank-6 tensors [34]). While not necessarily memory-consuming, this may take far too long.

Hence reducing the tensor rank of large-rank tensors becomes mandatory. This can be achieved in two ways: (i) Simply fusing legs which in itself has a significant positive impact on the overall performance [34]. Alternatively, if the situation permits, one may also (ii)

split up large-rank tensors into a tensor network of several lower-rank tensors such as tensor trains [41]. This has strong links to *data compression* which is of central importance for tensor network algorithms such as DMRG/MPS [17, 42, 43] even in the absence of any symmetry.

In principle, approach (i) could be carried out automatically, e.g., by fusing all incoming as well as all outgoing indices into an effective matrix in the RMTs. However, such an ad hoc procedure may not always be efficient, e.g., for cases where there is a strong asymmetry between the number of incoming vs. outgoing legs, besides that this would also makes permutations considerably more involved. It is also not well-suited for the frequent situation where multiple legs with mixed directions are contracted simultaneously. Hence as of QSpace v4, an automatic recast of the (internal) representation of high-rank tensors to an effective lower-rank format is not implemented. Rather the fusion of legs is left up to the user who then can decide more sensitively depending on the algorithmic context. Similarly also the tensor decomposition (ii) strongly depends on the context, and thus is left for the user to introduce where relevant.

## 2.12  CGT orthonormalization convention

All CGTs in QSpace adhere to the Frobenius-like normalization convention,

$$\|C_q\|^2 \equiv \text{tr}(C_q^\dagger C_q) \equiv \text{tr}(C^q C_q) = \mathbb{1}^{(M)} \, , \tag{20}$$

where the tensor trace is to be interpreted such that it fully contracts $C_q$ with the conjugate of itself on all legs, with only the OM index left open if present. The tensor $C_q$ denotes a general CGT with symmetry labels $q$ combined over all legs with certain directions which are of no further interest for the argument here. For simplicity, this collective $q$ is written as a subscript here, where nevertheless its conjugate is symbolically written as $C^q$ with raised $q$.

The norm in Eq. (20) is invariant under an arbitrary orthogonal transformation on any of the tensor's legs. As this includes reverting directions of legs based on $1j$ symbols, the normalization in Eq. (20) is therefore *independent* of the directions of the legs (while also switching to dual ireps). By not tying the normalization to a particular leg as in CGCs, this bears significant practical advantages fully analogous to Wigner-$3j$ symbols.

In the presence of outer multiplicity, CGTs acquire an additional OM index $\mu = 1, \ldots, M$. By convention, this is taken as trailing and thus slowest index at position $r + 1$ with $r$ the rank of the tensor,[4] thus having $C_q \rightarrow C_{q\mu}$. In the presence of OM then, i.e., $M > 1$, the r.h.s. of Eq. (20) becomes the identity matrix $\mathbb{1}^{(M)}$ in OM space. With this, the individual OM components are fully orthonormalized [15],

$$\text{Tr}\big(C_{q\mu}^\dagger C_{q\mu'}\big) \equiv \text{Tr}\big(C^{q\mu} C_{q\mu'}\big) = \delta_{\mu'}^\mu \, . \tag{21}$$

As the above shows, CGTs are unique up to an arbitrary but fixed orthogonal transformation in OM space. By contrast, the basis within the state space of any leg is defined and thus fixed with the respective multiplet decomposition, which itself may have to deal with inner multiplicity [8]. In the absence of OM [such as in standard CGCs for SU(2)], the normalization convention in Eq. (20) reduces to a sign-convention. This sign convention is still adopted in any case in the standard fashion of CGCs, namely that the first non-zero entry in the CGT $C_{q\mu}$ for any $\mu$ is positive.

Aside from the sign-convention on the basis vectors in OM space, this still leaves considerable arbitrariness in the presence of OM, i.e., $M > 1$. This arbitrariness is fixed within QSpace by its operational procedure: CGTs are constructed once and for all with iteratively increasing $M'$ *as they occur* (first come first serve) and then stored in the RC_STORE/CStore [15], until

---

[4]QSpace uses generalized column-major storage for all tensors, throughout, RMTs as well as CGTs. While RMTs are stored as full tensors, CGTs are stored in sparse format generalized to arbitrary rank tensors [8]. In either case, listing the OM index as trailing index implies that data is blocked in storage w.r.t. the OM index since slowest.

eventually at $M' = M$ the full OM space is exhausted, and thus complete. This procedure is motivated by the fact, that for larger-rank tensors, depending on the type of contractions performed, the full dimension $M$ in OM space may never be fully explored, and thus is not required. In this sense, QSpace does not preemptively generate the complete OM space for any CGT $C_q$ the first time it occurs. This is different for standard rank-3 CGCs, which are derived from an explicit numerical tensor product decomposition. By construction, their OM space is always complete from the first time they are encountered. For other rank-3 CGTs that derive out of contractions or higher-rank tensors, the OM space is built as it occurs out of contractions. If a rank-3 CGT was first generated via a contraction, and later on is completed via a tensor product decomposition, the existing CGT always has preference to ensure consistency within the existing RC_STORE. Any additional OM space derived from the tensor product decomposition is properly projected and renormalized.

**History dependent RC_STORE**    By the operational procedure above, namely the deliberate decision to not generate the full OM space in a unique, well-defined manner for any CGT encountered, the RC_STORE becomes history dependent. As such, the RC_STORE cannot be switched across different tensor network simulations. This makes access to a global centrally maintained RC_STORE essential. By meticulously applying and checking IDs together with high-resolution time stamps with CGTs [15], QSpace insists on a particular RC_STORE database within a given tensor network simulation to ensure overall consistency.

**Norm of a QSpace tensor**    As a corollary, the norm of a QSpace $X$ is defined by the full tensor trace of $X$ with the conjugate of itself, i.e., the Frobenius norm

$$\|X\|^2 \equiv \text{norm(X)}\text{^}2 \equiv \text{tr}(X^\dagger X) \,, \tag{22}$$

With the CGTs normalized as in Eq. (21), when contracted with themselves, they result in a simple identity matrix in OM space. Then if the $w$ matrices on the OM links also follow their own orthonormalzation convention, $w^\dagger w = \mathbb{1}$ [Eq. (17)], all that remains for obtaining the norm of a QSpace tensor is to add up the square of the Frobenius norm for all RMTs (for the case that $w^\dagger w \neq \mathbb{1}$ [e.g., see Eqs. (18) or (19)], this needs to be contracted (or multiplied if a simple number $\propto \mathbb{1}$) onto the RMTs when computing their norm squared). The normalization conventions therefore considerably simplify obtaining the norm of a QSpace tensor. In any case, the norm can be simply obtained using the class routine norm which automatically takes care of all the above.

**Representation of OM in QSpace tensors**    The tensor decomposition in Eq. (16) has an RMT $\|X\|_q^\mu$ for every symmetry configuration $q$, i.e., with symmetry labels fixed for every leg. The combined set of labels for all legs as represented by $q$ thus is unique in the direct sum ($\oplus$). Consequently, also for a QSpace $X$ the combined, i.e catenated symmetry labels Q=X.Q; Q=[Q{:}]; (cf. QSpace data structure and also subref) are *unique* for any QSpace $X$. That is, QSpace records as of QSpace v4 have unique symmetry labels for each record (this is in contrast to v2 [8] or v3 which had OM split up as multiple records in a QSpace tensor with 'degenerate' $q$-labels). In the presence of outer multiplicity, i.e., $M > 1$, the OM index $\mu$ is carried as an additional open trailing index in the RMTs (single index fused over the combined OM for all non-abelian symmetries present). Hence it occurs at index position $r + 1$ with $r$ the rank of the tensor. In the QSpace display, it is indicated as @M with the RMT dimensions [see Eq. (51) and subsequent display for an example]. This OM index eventually is connected via the intermediary $w$-matrix to the referenced CGTs which then also carry trailing OM indices.

## 2.13  CGT data storage and precision

All CGTs are real, highly sparse, and computed starting from better than quad-precision by linking QSpace with the GMP/MPFR multi-precision library [44, 45] during MEX compilation. The CGTs are also stored in a higher precision format in RC_STORE/CStore. The major reason for higher precision is the iterative nature of QSpace [15] that mimics the iterative nature of any tensor network simulation: starting from the defining representation, an initial set of smaller ireps is generated which usually more than suffices for the description of a physical lattice site. Starting from this local state space (cf. getLocalSpace), ever more complex tensors can be built by adding sites to the physical Hilbert space and thus to the tensor network. The process of tensor product decomposition is fully numerical [8], and hence subject to numerical precision error. The generalized raising/lowering operators (in the literature also referred to as *simple roots* in the root space of a Lie algebra, [8, 46]) are also computed fully numerically within QSpace [cf. RC_STORE/RStore] subsequent to a tensor product decompositions via their projection into the space of the newly encountered ireps [the generalized raising/lowering operators could be generated directly, e.g., based on Gelfand-Tsetlin pattern calculus for SU($N$) [38]; but with general non-abelian symmetries also beyond SU($N$) in mind, this approach is not adopted in QSpace]. The numerical approach in QSpace also automatically ensures consistency w.r.t. *inner multiplicity* (which describes the situation ubiquitous in rank $\mathfrak{r} > 1$ symmetries, namely where within a given multiplet the same $q_z$-labels can occur multiple times [8, 37, 38]). Therefore, by the design choice of QSpace as a true bottom-up approach for non-abelian symmetries, any inaccuracy at an earlier stage is inherited to any later stage. In this sense, the accuracy of CGTs deteriorates as the RC_STORE becomes ever larger. Empirically, however, by starting better than quad precision (about 40 digits), all CGTs obtained in the history of QSpace so far have been accurate at least up to 24 digits by monitoring numerical noise. In this sense all CGTs are exact within double precision accuracy $10^{-16}$. This is also important when exploiting the sparseness of CGTs since for that purpose, one needs to be able to distinguish small CGT coefficients reliably from numerical noise [for example, for SU(4) CGT coefficients as small as $10^{-9}$ have been encountered].

   The sparse CGT data is stored in GMP/MPFR format. The MPFR library does not support binary storage per se. The numerical data needs to be converted to int8 strings in a specified base [44, 45]. In QSpace, for compactness, this numerical base is chosen as 2*3*5 = 30 [8] (for comparison, hexadecimal has base 16). The reason for this choice is that it permits the exact representation of fractions based on the prime numbers 2, 3, and 5, which thus includes binary and decimal fractions. Within the mat-binary format, these strings are automatically compressed. So eventually, the storage overhead of the string representation is expected to be minor.

**Sparse CGTs of arbitrary rank**   All CGTs are stored in sparse format. For rank $r > 2$, the tensors are reshaped by grouping leading vs. trailing indices, such that the resulting 'matrix' is closest to square. Thereafter the sparse data can be stored as a standard sparse matrix. By also storing the actual dimensions for all legs of the original tensor, this permits one to restore the entire sparse tensor of arbitrary rank by returning to the original rank and recomputing sparse index positions.

## 2.14  Order of legs (index order conventions)

QSpace supports arbitrary order of legs in a tensor. When permuting the index order (legs) of a QSpace X, this applies the relevant permutation to each of the RMTs X.data{i}, and permutes the symmetry labels for the legs in X.Q{:}, as well as the itags. The general index order is also supported by the CGRs stored with a QSpace.

**Index order and CGRs**   The CGRs store references to sorted CGTs in the `RC_STORE`. These CGTs have sorted $q$-labels to avoid the proliferation of entries. The required permutation (and possible conjugation) to match the current index order of a QSpace `X` is auto-determined and stored with the CGRs *internally* in the MEX core routines. However, when handed over to the user space in Matlab, this permutation is applied to the CGRs in `X.info.cgr`, and hence is not directly visible. An arbitrary permutation of the legs on a QSpace `X` is translated then to the respective permutation relative to the underlying sorted CGT. The order of raised indices relative to the lowered indices is irrelevant and can be trivially permuted. Subtleties arise, however, for QSpace records with the same $q$-labels and directions on multiple legs, referred to as degenerate $q$-labels, as this may lead to rotations in OM space (cf. 2M symbols). In any case, all of this is taken care of in the QSpace core routine `permute[QS]` (MEX routine `permuteQS` as called by the Matlab class wrapper routine permute).

**Compact notation for permutations (with optional conjugation)**   The order of legs can be changed using permutations. A permutation of length $n \leq r$ permutes the first $n$ legs (or all legs for $n = r$). In this sense, the specified permutation can be shorter than the rank of the tensor. At the same time, one can also conjugate the tensor as a whole by specifying a trailing `'*'` with the permutation. For permutations of length $n \leq 9$, which is nearly always the case, QSpace accepts a compact string notation for specifying permutations. For example, `'21*'` denotes a transposition of the first two legs, i.e., the permutation `p=[2 1]` together with overall conjugation `'*'`. In QSpace syntax, for example, the Hermitian conjugate of an irop $S$ can be obtained then via `permute(S,'21*')` which is equivalent to the non-compact semantic `permute(S,[2 1],'conj')`.

**Compact notation for index sets (with optional conjugation)**   Specifying a subset of legs permits the same compact notation as the permutations of legs above, including an optional trailing conjugation flag `'*'`. This is relevant, for example, for contractions when several indices are contracted simultaneously. For example, in Eq. (37) `'13*'` specifies the simultaneous contraction of the preceding tensor on indices 1 and 3 while also taking its conjugate.

**Index order for operators and leg directions**   An irreducible operator (irop) consists of an irreducible set of operators that transform like a multiplet with symmetry operations applied via a commutator. The set of operators thus acquires an additional index (referred to as irop index) which, by convention, is always listed last, i.e., third. Therefore, the natural representation of an operator in the presence of symmetries is a rank-3 tensor [cf]. Fig. 1(b)]. If the operator transforms like a scalar, i.e., with symmetry labels $q = 0$ on the irop index, this index represents a singleton dimension which can be skipped. Thus the irop for scalar operators, such as a Hamiltonian term or density matrix, can be reduced to rank 2.

The action of a particular operator is characterized by the symmetry labels on its irop index. Consider, for example, the matrix elements of an annihilation operator, e.g., $\langle s|\hat{c}|s'\rangle$. Ignoring spin for simplicity, this operator *reduces* charge $q$ by 1, i.e., $q_s = q_{s'} - 1$. Based on this intuitive notion, namely that the action of the operator is to *lower* the charge, this implies the symmetry label of the irop $q_{\mathrm{op}} = -1$. For this to hold, however, the irop index must be grouped with the ket state, and therefore is also outgoing. Together with the index order convention $S_{23}^1 \equiv \langle 1|S_{[3]}|2\rangle$ for an operator $S$, the leg directions of an irop are thus +--.

Not all rank-3 tensors with leg directions +-- are necessarily operators that act within a particular state space. Therefore to explicitly declare an arbitrary QSpace tensor `X` of rank 3 with leg directions +-- an irop or operator more generally, QSpace recommends (and in certain applications insists on) the additional flag `'operator'` with rank-3 operators (see field `X.info.otype` in Sec. A.3).

**Index order convention with *A*-tensors (MPS)** The *A*-tensors as with an MPS adhere to the LRs index order convention [cf. Fig. 1(a)], having $\langle l | A^{[\sigma]} | r \rangle$ where $l \in L$, $r \in R$ refer to left and right block states, respectively, with $\sigma \in s$ the local state space. When L→R orthonormalized, $A_r^{l\sigma}$ has leg directions +-+, when L←R orthonormalized, $A_l^{r\sigma}$ has leg directions -++, and when at the orthogonality center (OC), $A^{lr\sigma}$ has leg directions +++ (all in).

**Index order out of contractions** The contraction of a pair of tensors, symbolically written as $A * B$, always preserves the index order of the non-contracted indices. The ones of the first tensor ($A$) are collected first, followed by the non-contracted indices of the second tensor ($B$). Hence the index order out of contractions is fixed by the order of arguments handed over to `contract[QS]`.

## 2.15  Fully contracted CGT networks and $M$ symbols

When projecting a CGT network onto a particular OM basis of the resulting CGT, this corresponds to the evaluation of a fully contracted tensor network. It is relevant, for example, for pairwise contractions of tensors [cf. X-symbols further below]. With this in mind, consider then such a fully contracted tensor network of $n$ CGTs. Each participating CGT may carry an OM index $\mu$ that needs to be kept open since it links to an RMT [cf. Eq. (16)] which is outside the discussion here. In this sense, a fully contracted tensor network of $n$ CGTs has $n$ open OM indices, and thus is referred to as an $n$M symbol with $n \in \mathbb{N}$. The $q$-labels for all contracted legs are considered additional metadata.

**1M and 2M symbols** The normalization convention in Eq. (21) represents a 2M symbol $\delta_{\mu'}^{\mu}$. However, since the multiplicity indices always refer to the OM space of the same CGT, in the spirit of $1j$ symbols, one may also refer to this as a 1M symbol. It is always simply a square identity matrix in OM space. More non-trivially, in the presence of degenerate $q$-labels on multiple legs with the same directions, a given CGT may be contractable with a permuted version of itself. This can result in an orthogonal matrix and thus an actual 2M symbol.

**3M symbols** When contracting a pair of CGTs and projecting the result onto the target CGT, this corresponds to a 3M symbol. In the specific context of tensor contractions, this is equivalently referred to as X-symbols. Note that by construction $(n = 3)$M symbols with $n$ an odd number requires at least some non-rank-3 CGTs in its network, i.e., the presence of non-CGCs.

**4M $= 6j$ symbols, 6M $= 9j$ symbols, etc** In the context of SU(2), a fully contracted CGC network consists of CGTs of rank 3 only. For this to be fully contractable, the total number $n$ of CGCs must be necessarily even. It has a total of $3n/2 \equiv 3\tilde{n}$ contracted legs, each carrying a multiplet label. Using $j$ with reference to SU(2), therefore a $(3\tilde{n})$-$j$ symbol is a fully contracted network of $n = 2\tilde{n}$ CGCs. For example, a $6j$ symbol ($\tilde{n} = 2$) contracts $n = 4$ CGCs, and thus represents a 4M symbol. Similarly, a $9j$ symbol ($\tilde{n} = 3$) represents a 6M symbol, etc. In general, a $(3\tilde{n})$-$j$ symbol is a $(n = 2\tilde{n})$ M symbol, with all $j$-labels on the contracted legs considered additional metadata [15]. Therefore fully contracted networks of rank-3 CGTs correspond to $n$M symbols with $n$ an even number.

For the simple case of SU(2) where CGTs for rank $r \leq 3$ do not have OM, $(3\tilde{n})$-$j$ symbols are plain numbers. However, for a more general non-abelian symmetry of symmetry rank $\mathfrak{r} > 1$ like SU($N > 2$), for example, the $6j$ symbols are rank-4 tensors with open OM indices $\mu$ on all its legs, hence 4M symbols. Since there is no simple canonical basis in OM space, besides that the OM space even for CGCs (rank 3) can become arbitrarily large already for SU(3), this makes it considerably more cumbersome to generalize $6j$ symbols, etc., to arbitrary non-abelian symmetries. Therefore analytical results for $6j$ symbols are only available for SU(2),

but not for rank $\mathfrak{r} > 1$ symmetries. For this very reason, QSpace does not use $6j$ symbols at all. It rather uses 3M symbols (X-symbols) instead.

## 2.16 Pairwise tensor contractions and X-symbols

When contracting a set of tensors, in practice, this is always carried out as a sequence of pairwise contractions. For a particular pairwise contraction then, respective contractions occur in parallel at the level of the RMTs, but also at the level of the CGTs for each non-abelian symmetry included in the tensor representation [8].

The contraction of a pair of CGTs $C * D$ on a subset of indices results in another CGT $E$ [cf. Fig. 6]. By projecting the outcome of $C * D$ onto $E$ (while completing the OM space of $E$ whenever relevant), the fully contracted $\text{tr}(E^*(CD))$ corresponds to a 3M symbol [Fig. 6(b)]. In the context of pairwise contractions ('X'), QSpace refers to these as X-symbols [15].

In practice, the CGTs $C$ and $D$ in Fig. 6(a) as well as $E$ in Fig. 6(c) are referenced in QSpace tensors and stored in the RC_STORE once and for all once generated. Hence all that is required eventually for a pairwise contraction of QSpace tensors are the X-symbols in Fig. 6(b,c). These are contracted onto the OM indices of the input pair of RMTs [15], and in this sense *fuse* their OM indices.

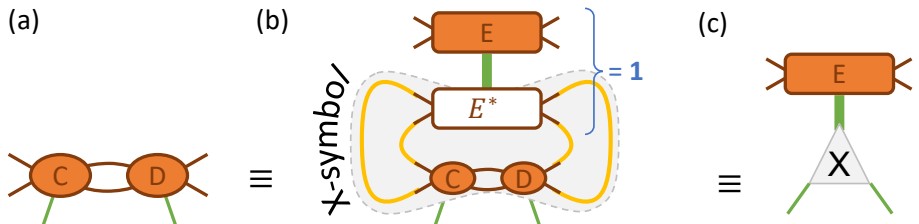

Figure 6: X-symbols (figure adapted from [15]) – When contracting a pair of CGTs $C$ and $D$ on some subset of legs [panel (a)] with OM indices indicated in green, this gives rise to a new CGT $(CD) \equiv C * D$. The $q$-labels and $q$-directions are assumed arbitrary but fixed for all non-green lines, but not shown for readability. The resulting contracted tensor is expected to have a single OM index, hence the two green lines in (a) need to be fused, as indicated in the remainder of the panels. If a CGT $E$ with the same rank and $q$-labels as $(CD)$ already exists in the RC_STORE (up to sorting) the contraction in (a) is projected into the existing OM space of CGT $E$ by inserting the identity $\mathbb{1} = E^* \cdot E$, with the sum in the product performed over the OM space of $E$ [thick green line in (b)]. If the CGT $E$ does not yet exist or is incomplete, the additional OM components are generated via Schmidt decomposition out of $(CD)$. The lower gray-shaded area in (b) now encompasses a fully contracted network for three CGTs, i.e., a 3M symbol. For the explicit purpose of pairwise contractions, these are referred to as X-symbols.

## 2.17 Index tags (itags)

QSpace supports user-defined names on tensor legs with strings up to a length of 8 characters. Since they represent tags assigned to indices, they are referred to as itags. The usage of itags makes QSpace tensors significantly more user-friendly. For one, they allow one to identify individual QSpace tensors more easily in a simulation. Hence the itags are also shown when displaying a QSpace tensor. Moreover, QSpace supports auto-contraction based on itags, in that a matching pair of itags with opposite directions (i.e., one in- and one outgoing) are auto-contracted unless indicated otherwise. See help on contractQS for more on this.

Within the QSpace core routines, itags are converted to unsigned long integers, thus setting the size limit of 8 bytes. If an itag is shorter than 8 characters, it is padded by zeros. For a QSpace tensor X of rank $r$, its itags are stored in the mandatory field X.info.itags{$l$} for leg $l = 1, \ldots, r$.

Itags are expected to be printable, mostly alpha-numeric characters in ASCII code range 33-126, excluding itag separators [,;|] (these characters are sometimes used to separate multiple itags combined into a single string). It is recommended also to avoid the trailing marker characters ['*] (see below) *within* an itag string to avoid confusion. The above ASCII range is case-sensitive and excludes white space. The sign bit for each character is exploited in QSpace core routines for other internal purposes, such as flagging itags that shall not be contracted. With this then, for example, the following itags are all different from each other: 's1', 's01', 'S01', 's:1', 's-1', etc.

**Trailing marker characters in itags** The itags are also used to store the direction of legs for a given QSpace tensor: a trailing 'conjugate' character '*' indicates outgoing legs, whereas the absence of a trailing '*' indicates incoming. These directions must be maintained and present even if the itags are empty otherwise. In case of inconsistency of the leg directions specified with the itags of a QSpace tensor as compared to the $q$-directions in the CGRs (if present), QSpace will throw an error.

Moreover, QSpace uses trailing primes to mark dual state spaces. Reverting the direction of a leg changes the interpretation of the associated block state space. Therefore an automated modification of the itag is natural. It is useful for better interpretability, but also as a safety measure that guards against contracting wrong indices that would carry the same itag otherwise (see also $1j$ symbols).

In summary, QSpace thus assigns special meaning to the following trailing marker characters in itags:

- A trailing asterisk (*) indicates a conjugate, i.e., outgoing index. Internally within the C++ core routines, this is split off as $q$-directions for an entire tensor. Therefore they do not count towards the 8-char limit of an itag.

- A trailing prime (') indicates dual state space which occurs when a leg direction is reverted. The prime is considered an integral part of an itag, and hence counts towards the 8-character length limit. For this reason, a trailing prime also occurs *before* a trailing conjugate flag '*' if present.

Both of the trailing marker characters above act as a toggle. Since the dual of the dual of a state space is the original state space, marking an already marked itag removes the trailing prime. Similarly, taking the conjugate of an already conjugate index reverts its direction twice, and therefore results in the original direction of that leg. Hence in this case, a trailing conjugate character '*' is removed.

*Remark on notation* – In this documentation, an itag 'foo' with a trailing prime is frequently printed as 'foo''. The outer single quotes '...' denote a string, and differ in font from the prime ('), for the sake of the discussion. They are considered the same, though, on the coding level. Specifically, with Matlab syntax in mind, 'foo'' ≡ 'foo''' ≡ "foo'", yet noting that as of QSpace v4, Matlab strings are always assumed to use single quotes.

**Color coding** When using QSpace in interactive Matlab terminal mode, trailing marker characters in itags are replaced by color coding for the sake of readability: outgoing indices become gray-ish (while skipping the trailing * character), and itags marked by the trailing prime ('), acquire a color other than the default text color of the terminal (darker green as of QSpace

v4, thus replacing the trailing prime; see Sec. 4.3 for examples). This way, for example, within an MPS, bond itags for *A*-tensors associated with the left block (left relative to the OC) acquire a different color as compared to the itags for *A* tensors associated with the right block.

The color coding of itags, of course, applies to non-empty itags only (i.e., itags that contain actual printable string after removing trailing marker characters). If all itags are empty, then the +/- notation for *q*-direction is used as strings when displaying itags (e.g., see Fig. 7) with color coding applied. When only some itags are empty, no color substitution is performed, with all trailing markers shown as is. By default, no color coding is applied for non-interactive batch or cluster jobs where log output is expected to be redirected to log files. The behavior can be changed via the environmental variable QS_LOG_COLOR.

## 3 QSpace Essentials

### 3.1 Starting point: getLocalSpace()

The starting point of any tensor network setup in QSpace is getLocalSpace. This permits one to define a lattice site via a set of operators that act within that state space. However, because these operators are expressed as matrix elements in a particular symmetry eigenbasis of the site, this implicitly also represents its full state space, e.g., via the identity operator. The way getLocalSpace operates is that it builds the local state space in Fock space without symmetries first, since everything can be constructed there including the symmetry operators for all requested symmetries. Based on this, getLocalSpace rotates and groups the basis into symmetry multiplet spaces, and then 'compresses' the data via compactQS based on the Wigner-Eckart theorem to obtain the RMTs while splitting off the CGCs as tensor product factors [cf. Eq. (16)].

The overall argument structure of getLocalSpace() is as follows (for more detailed usage information, see help getLocalSpace):

$$[\text{op1},\ldots,\text{opN},\text{Isym}] = \text{getLocalSpace(type [, symmetries] [, options])} \qquad (23)$$

The input specifies the type of a site, together with its symmetries, and further options. It returns a set of representative operators op1,...,opN for the specified site (therefore N depends on the input), together with an info structure Isym. The type of a site is a string that can be any of the following:

| | |
|---|---|
| 'FermionS' | spinful fermions |
| 'Fermion' | spinless fermions |
| 'Spin' | spin models |
| etc. | other experimental setups of spin models |

The argument symmetries consists of a comma-separated catenated string that specifies the type and order of symmetries to use.

**Spinfull fermions**  For spinful fermionic models, i.e., `type='FermionS'`, the following symmetry entries are supported,

| | |
|---|---|
| `'Acharge'` | abelian total charge, i.e., U(1) |
| `'ZNcharge'` | abelian total charge, modulo $N \geq 2 \in \mathbb{N}$, e.g., `'Z2charge'` for $Z_2$ |
| `'SU2charge'` | SU(2) total particle-hole |
| `'Aspin'` | abelian total spin, i.e, U(1) |
| `'SU2spin'` | SU(2) total spin ($S$) |
| `'SU2spinJ'` | SU(2) total spin ($J = L + S$) |
| `'AspinJ'` | U(1) total spin $(J = L + S)_z$ |
| `'SUNchannel'` | SU($N$) channel symmetry |
| `'SONchannel'` | SO($N$) channel symmetry |
| `'SpNchannel'` | Sp($N$) symmetry with $N$ even for $N/2$ channels combined with SU(2) particle-hole charge symmetry |

The order in which these symmetry labels are specified also determines the order of symmetries used, and hence the order of the symmetry labels for any state space. The last three symmetries only apply in the presence of multiple channels (or flavors). The number of channels can be specified via the optional argument `'NC'`, or directly with the symmetry itself. For example, `'SUNchannel'` with `'NC',3` is equivalent to `'SU3channel'`. For the case of `NC` > 1 channels, abelian (non-abelian) charge in all channels individually can be requested via `'Acharge(:)'` or `'SU2charge(:)'`, respectively.

**Spinless fermions**  Spinless fermionic models, i.e., `type='Fermion'`, support the symmetries `'Acharge'` and `'SUNchannel'` similar to the spinful case above. A subtlety concerns the case of an odd number of channels `NC`. There the particle number relative to half-filling becomes half-integer for the state space of a single site or, more generally, an odd number of sites. To avoid this (also bearing in mind that QSpace insists on integer symmetry labels), a factor 2 is applied to the charge symmetry labels. See Sec. 2.9 for more on this.

The fermionic models require that fermionic signs are explicitly taken care of in the applications via the fermionic parity operator Z. That is, *graded tensors* [47,48] are not implemented in QSpace yet which eventually may allow one to take care of fermionic signs in an automated fashion. Bosonic degrees of freedom with a finite local state space may be implemented as tweaks of the U(1) fermionic models while stripping any fermionic signs based on the parity operator Z. Bosonic degrees of freedom with infinite state space, such as harmonic oscillators, need to be truncated. This can be done efficiently, e.g., via a shifted optimal boson basis [49].

**Spin models**  Plain spin models such as SU(2) spin-$S$ Heisenberg models can be setup with `type='Spin'`. This typically only allows a single symmetry that is specified as an additional argument, with the default being `'SU2'` (see Sec. 3.1.2 below for examples). To obtain other more complex spin models based on several symmetries, the fermionic models may be tweaked e.g., via projections (see Sec. 5.3.1 for an example).

**Special case: No symmetries**  QSpace also permits simulations in the absence of any symmetry. Yet since this is a rather atypical case when using QSpace, this proceeds via a tweak. For example, no symmetries can be implemented via an auxiliary trivial abelian symmetry such as U(1), where only the scalar symmetry label $q = 0$ of the vacuum is present on any leg. With this, every tensor contains a single QSpace record only (thus a single full block as RMT). An example is provided with the spin setting in line (30) below, which returns objects of the above type. As shown in Sec. 4.4, for spin $S = 1/2$, `S(i).data{1}` contains the Pauli matrices

$[\sigma_z, \frac{1}{\sqrt{2}}\sigma_-, -\frac{1}{\sqrt{2}}\sigma_+]$ for $i = 1, 2, 3$, respectively. The order and normalization of operators can always be double-checked by inspection.

### 3.1.1  Example: Single fermionic site

The following call to `getLocalSpace` returns QSpace operators for a spinful fermionic site with a single level (i.e., $d = 4$ states)

$$[F,Z,S,IS] = getLocalSpace('FermionS','Acharge,SU2spin'); \qquad (24)$$

The detailed structure of the returned QSpace tensors as displayed within Matlab is discussed by the example of F in Sec. 3.2 (e.g., see Fig. 7) and also with (31). The operators returned by (24) are the fermionic annihilation operators F (irop with two components, one for spin up, and one for spin down), the fermionic parity operator Z, and the total spin operator S (hence an irop with three components).

The last argument IS returned by (24) is an info structure that contains additional detailed information concerning the chosen symmetry setup. As seen in (32), it contains the identity operator `IS.E`, or `IS.sym` as a more verbose description of the symmetries as specified with the input to `getLocalSpace`. This permits one to look up the order of symmetries later e.g., to reconfirm whether the charge or spin symmetry is listed first, etc., since the symmetries specified with a QSpace tensor, such as A or SU2, no longer contain that information. Other fields in IS also contain low-level information relevant to the internal workings of `getLocalSpace` which may be safely ignored.

The symmetries of a model hold throughout a tensor network simulation. In this spirit, the call to `getLocalSpace` defines the symmetry once and for all during the setup of a model. Hence symmetries can only be switched during the setup for different simulations. For example, for the same local state space of a single fermionic site, it may be changed then across the following settings depending on the symmetries of the model Hamiltonian based on the underlying model parameters:

| `'Acharge,Aspin'` | $U(1)_{\text{charge}} \otimes U(1)_{\text{spin}}$ | $d^* = 4 = d$ | (fully abelian) |
| `'Acharge,SU2spin'` | $U(1)_{\text{charge}} \otimes SU(2)_{\text{spin}}$ | $d^* = 3$ | |
| `'SU2charge,Aspin'` | $SU(2)_{\text{charge}} \otimes U(1)_{\text{spin}}$ | $d^* = 3$ | |
| `'SU2charge,SU2spin'` | $SU(2)_{\text{charge}} \otimes SU(2)_{\text{spin}}$ | $d^* = 2$ | (fully non-abelian) |

All describe the same local state space of $d = 4$ states. With increasing symmetry from top to bottom, however, the local state space dimension is effectively reduced from $d^* = 4 \rightarrow 2$ multiplets. For comparison, the more involved example for NC=3 channels with symplectic symmetry,

$$[F,Z,S,IS]=getLocalSpace('FermionS','SU2spin,SpNchannel','NC',3);$$

contains 4*NC=12 components in the irop F (creation and annihilation operators for each spin and channel). The total spin irop S still has three components. The scalar parity operator Z shares much of the structure of an identity matrix except that its diagonal entries are $\pm 1$. As such it also reflects the full state space of a site. By contrast, the spin operator only acts within partially occupied sectors since the fully occupied or empty states are singlets with spin matrix elements all-zero. Yet zero blocks are not stored. In the present example of NC=3 spinful levels, the largest non-abelian symmetry is given by the symplectic Sp(2*NC=6) symmetry. Within this setting, the local space dimension $d = 4^3 = 64$ states is reduced to an effective dimension of a mere $d^* = 4$ multiplets!

### 3.1.2 Example: Spin models

For spin-models, i.e., `type='Spin'` in Eq. (23), the argument `symmetries` is only allowed to contain a single symmetry, assuming SU(2) if not specified. The options can specify the particular irep to use, assuming the defining irep, by default. For example, the following two lines are equivalent for SU(2),

$$[\text{S,IS}] = \text{getLocalSpace('Spin',S);} \tag{25}$$

$$[\text{S,IS}] = \text{getLocalSpace('Spin','SU2',2*S);} \tag{26}$$

These return the spin operator for a spin $S$ multiplet in the local state space with $q$-label $q = 2S$. The usage in line (26) permits to switch to other non-abelian symmetries. For example,

$$[\text{S,IS}] = \text{getLocalSpace('Spin','SU3','02');} \tag{27}$$

returns the 'spin' operator [synonymous with the irop in the adjoint representation] for SU(3), with the local state space in the 6-dimensional representation `q='02'`. Here the input supports both, compact string (`'02'`) and non-compact numeric notation (`[0 2]`).

**Switch to abelian or no symmetry**    In the case of SU(2), the usage in line (25) also accepts the following switches to abelian symmetries,

$$[\text{S,IS}] = \text{getLocalSpace('Spin',S,'-A');} \tag{28}$$

$$[\text{S,IS}] = \text{getLocalSpace('Spin',S,'--Z2');} \tag{29}$$

$$[\text{S,IS}] = \text{getLocalSpace('Spin',S,'--nosym');} \tag{30}$$

where the option `'-A'` returns the spin operator in abelian U(1) symmetries (equivalent to $Z_\infty$), `'--Z2'` returns it with $Z_2$ symmetry, and `'--nosym'` with no symmetry at all (effectively implemented as trivial 'U(1)' with $q = 0$ only). In all cases, the returned spin operator `S` has three entries corresponding to $\text{S} = [\hat{S}_z, \frac{1}{\sqrt{2}}\hat{S}_-, -\frac{1}{\sqrt{2}}\hat{S}_+]$. See Sec. 4.4 for a more detailed discussion.

**Normalization of spin operator**    The spin operator `S` is a specific irop that transforms in the adjoint representation. It has as many components as there are generators in the Lie algebra. The spin operators out of `getLocalSpace` [e.g., Eq. (27)] adhere to standard normalization conventions in the literature. This can be easily double-checked, e.g., via `normQS()`. For example, for an SU(2) spin in the defining irep, $\|S\|^2 \equiv \text{tr}(S^\dagger \cdot S) = (3/4)\,\text{tr}(\mathbb{1}^{(2)}) = 3/2$. Therefore it must hold for the respective QSpace spin operator that `norm(S)^2 = 3/2`, seen as $\|S\| = \sqrt{3/2} = 1.225$ in line 17 below.

     The normalization and sign of the spin operator for all ireps are entirely determined by the definition of the spin operator in the defining representation. After all, the spin operators in any other irep can be constructed by combining defining ireps and then computing the matrix elements of $S_\text{tot}$ in the combined Hilbert space. This construction can be used to reach any other irep. With this the normalization *and sign* of $S_\text{tot}$ is fixed by specifying these for the defining irep. Hence one needs to be careful when manually constructing spin operators directly for an irep other than the defining representation, e.g., based on existing CGTs.

     Based on the above constructive approach, the spin operator is also well-defined in the presence of OM for larger ireps: while the spin operator is unique in the defining irep (this particular case never has OM), OM can arise for larger local multiplets. For example, consider the SU(3) spin operator in the local irep `q='11'` also being in the adjoint representation. This has $M = 2$. Yet from the constructive approach above, there is also only *one* well-defined spin operator in this setting, since it can be expressed as $S_\text{tot}$ in the tensor product space of three defining ireps (while there will be two ireps in `'11'` in this construction, both replicate the *same* spin operators, since for both the RMT links the same way to their shared CGR).

### 3.1.3    Beyond models implemented in `getLocalSpace`

The implemented settings for fermionic or spin models cover a wide range of applications. Yet, in practice, situations will arise where the desired symmetry setting is not yet implemented in `getLocalSpace`. In these cases, it is often sufficient to tweak the QSpace output of `getLocalSpace` that captures the closest most relevant symmetry setting. For this, however, it is essential that one fully understands in detail the general QSpace data structure. See Sec. 5.3 for detailed examples. As a last resort, one may contact QSpace support for help. Typical strategies of tweaking the output of `getLocalSpace` may include:

- appending additional symmetries [see `QSpace/addSymmetry.m`].
- projecting fermionic models to a fixed charge sector, thus effectively obtaining a spin model. If the charge label (relative to half-filling) is non-zero, the respective symmetry labels (columns in `X.Q`) may be set to zero because they are all the same due to the projection.
- converting abelian symmetries, like U(1) labels to $Z_N$ labels by changing the symmetry name (cf. `X.info.qtype`) and converting the symmetry labels in `X.Q`) accordingly.
- the absence of any symmetries was already discussed above.

## 3.2   QSpace display

A summary of the content of a QSpace tensor is displayed by simply typing its name at the Matlab command line prompt followed by enter (technically speaking, this is realized by overloading the display function in `@QSpace`).

For example, Fig. 7 displays the *A*-tensor as generated in (47), with detailed additional explanations. The symmetry setting is a combination of U(1) charge with SU(2) spin, with the order fixed by the call to `getLocalSpace` in (31). The display consists of a header info section, followed by a listing of the symmetry blocks. The pair of header lines show global tensor information, including `itags` for all legs and their directions. In case of `itags` that are empty up to trailing markers, the *q*-directions are shown in the +/- format as this enables color coding. See also applications for more examples. The subsequent listing of symmetry blocks includes symmetry labels and dimensional information of the underlying RMTs and CGRs. Note that trailing singleton dimensions are skipped for RMTs beyond rank-2 following Matlab convention for dense arrays. By contrast, trailing singleton dimensions are displayed for CGTs, as a reminder that CGTs are stored in generalized sparse format which always specifies the full CGT dimensions.

The square-root dimensional factors to the right in Fig. 7 are shown only if the *w*-matrices provided with the CGRs have anomalous normalization, i.e., $w^\dagger w = d\mathbb{1}$ with $d \neq 1$ (combined as a product across all symmetries). In this case "$\{\sqrt{d}\}$" is included in the display as a reminder of the altered normalization affecting the RMTs. From Eqs. (18) and (19), it is generally expected that $d$ reflects some multiplet dimensions, and therefore is an integer. This motivates the notation with the surd ($\sqrt{\ }$) for the sake of readability and compactness.

## 4   Collection of Examples

QSpace uses a compact syntax for tensor network commands with usability and readability in mind. While this compact notation may be self-explanatory to some extent, it takes some time to digest and get used to. Hence it is discussed in detail with the examples in this documentation. This section provides a combined collection of examples referenced earlier in

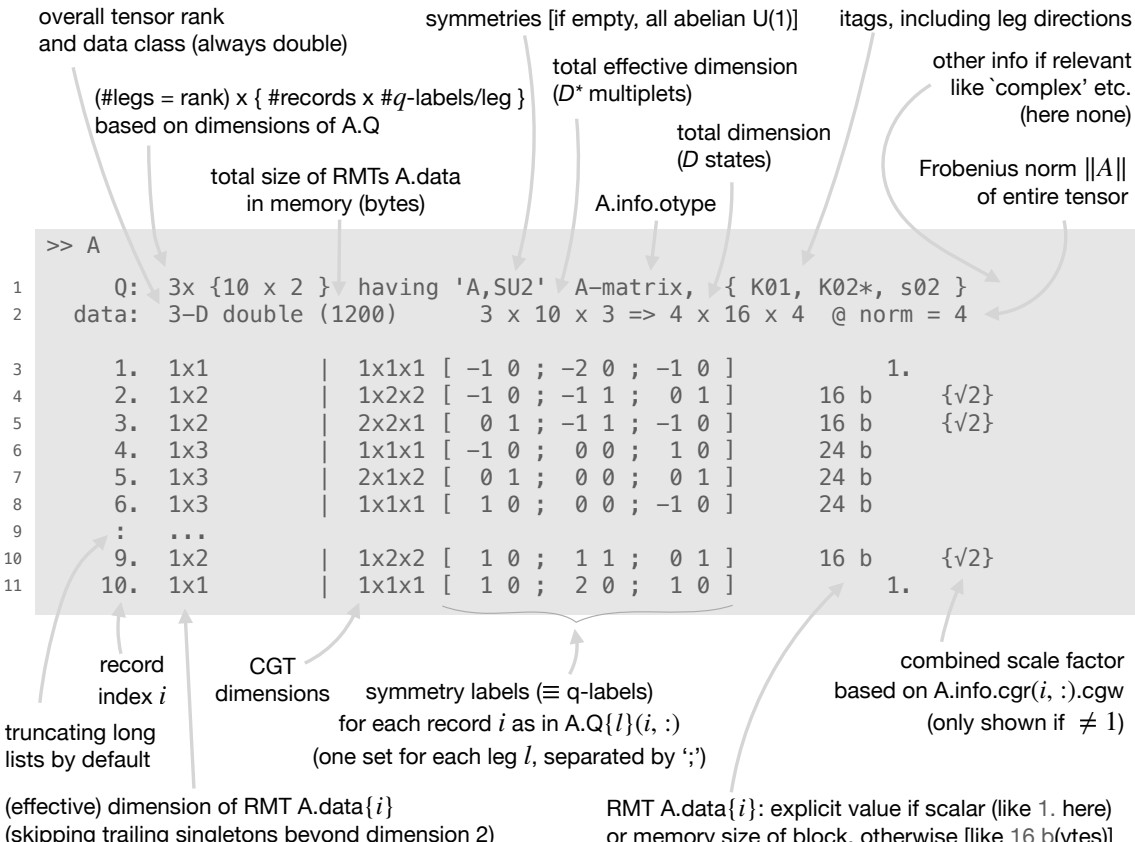

overall tensor rank
and data class (always double)

(#legs = rank) x { #records x #*q*-labels/leg }
based on dimensions of A.Q

total size of RMTs A.data
in memory (bytes)

symmetries [if empty, all abelian U(1)]

total effective dimension
(*D\** multiplets)

total dimension
(*D* states)

A.info.otype

itags, including leg directions

other info if relevant
like `complex' etc.
(here none)

Frobenius norm $\|A\|$
of entire tensor

```
    >> A
1       Q:   3x {10 x 2 }  having 'A,SU2'  A-matrix,  { K01, K02*, s02 }
2     data:  3-D double (1200)       3 x 10 x 3 => 4 x 16 x 4  @ norm = 4

3       1.   1x1       |  1x1x1 [ -1 0 ; -2 0 ; -1 0 ]           1.
4       2.   1x2       |  1x2x2 [ -1 0 ; -1 1 ;  0 1 ]      16 b      {√2}
5       3.   1x2       |  2x2x1 [  0 1 ; -1 1 ; -1 0 ]      16 b      {√2}
6       4.   1x3       |  1x1x1 [ -1 0 ;  0 0 ;  1 0 ]      24 b
7       5.   1x3       |  2x1x2 [  0 1 ;  0 0 ;  0 1 ]      24 b
8       6.   1x3       |  1x1x1 [  1 0 ;  0 0 ; -1 0 ]      24 b
9       :    ...
10      9.   1x2       |  1x2x2 [  1 0 ;  1 1 ;  0 1 ]      16 b      {√2}
11     10.   1x1       |  1x1x1 [  1 0 ;  2 0 ;  1 0 ]           1.
```

record
index $i$

CGT
dimensions

symmetry labels (≡ q-labels)
for each record $i$ as in A.Q{$l$}($i$, :)
(one set for each leg $l$, separated by ';')

combined scale factor
based on A.info.cgr($i$, :).cgw
(only shown if ≠ 1)

truncating long
lists by default

(effective) dimension of RMT A.data{$i$}
(skipping trailing singletons beyond dimension 2)

RMT A.data{$i$}: explicit value if scalar (like 1. here)
or memory size of block, otherwise [like 16 b(ytes)]

Figure 7: Typical QSpace display, e.g., as printed when typing the name of a QSpace tensor on the Matlab prompt (>>) in the user interface. This display summarizes the tensor's data structure into a more readable form. The example here is based on the *A*-tensor as obtained in (47). It fuses the state spaces of two spinful fermionic sites with $D = 4 \times 4 \to 16$ states [4 x 16 x 4 in line 2 after permutation to standard LRs index order; cf. (47)]. The symmetry setting of U(1) charge (abelian 'A') with SU(2) spin is shown as 'A,SU2' in line 1. Hence all multiplets carry two symmetry labels $q = (q_1\ q_2)$, as reflected in the dimensions of Q in line 1, and the columns of $q$-labels listed for each leg. The charge $q_1$ represents particle number relative to half-filling and hence can become negative, whereas $q_2 = 2S \geq 0$ reflects the SU(2) spin multiplet. Due to the presence of SU(2) spin, switching to the multiplet-based description reduced the Hilbert space to an effective dimension $D^* = 3 \otimes 3 \to 10$ multiplets (3 x 10 x 3 in line 2). Since each fused multiplet carries unique symmetry labels here, this also translates to a total of 10 QSpace records. The record index is shown to the left for reference. The RMTs A.data{i} represent reduced matrix elements. Hence they are tensors with *effective* dimensions as shown in the second column (for the RMTs only, this adheres to Matlab conventions that trailing singleton dimensions beyond two dimensions are skipped). The CGT dimensions are displayed in the third column (shown including trailing singletons for non-abelian symmetries only; if a QSpace has all-abelian symmetries, the CGT dimensions are skipped entirely). Long listings are truncated, by default, as in line 9 by ':   ...' for the skipped records 7-8. For a full listing of all QSpace records, one needs to explicitly call the underlying function display(A,'-f'), since this is the only way to specify options (like '-f' for full display). The combined scale factor out of the CGRs [cf. Eqs. (18) and (19)] is shown to the right. Since it always represents the square root of some (product of) multiplet dimensions $d \in \mathbb{N}$, it is printed for the sake of readability only for those records where $d \neq 1$ as $\sqrt{d}$.

this documentation. Essential syntax-highlighted Matlab code is shown in a gray box. This is followed by a detailed discussion of output objects and sanity checks on their structure. The source and output lines are given line numbers to the left for ease of reference.

To provide QSpace examples, the first step is defining a set of exemplary QSpace tensors. Hence to be specific, the examples below use the QSpace operators that act within the local state space of a single spinful fermionic site with U(1) charge and SU(2) spin symmetry,

```
1  >> [F,Z,S,IS]=getLocalSpace('FermionS','Acharge,SU2spin');                    (31)
```

The local state space $s \in \{|0\rangle, |\uparrow\rangle, |\downarrow\rangle, |\uparrow\downarrow\rangle\}$ represents empty, singly up/down, and doubly occupied states, respectively (in contrast to earlier, e.g., Fig. 1, in what follows $\sigma \in \{\uparrow, \downarrow\} \equiv \{1, 2\}$ specifies the spin flavor index that underlies the local many-body state space $s$). By not specifying the number of flavors/channels above, by default, this assumes a single flavor (for examples with NC=3 flavors, see further below). The above call to getLocalSpace thus returns the fermionic annihilation operators $\hat{F} \equiv (\hat{f}_\uparrow, \hat{f}_\downarrow)^T$ in irop index order convention,

```
2  F =
3       Q:  3x { 2 x 2 }  having 'A,SU2'  operator,  { +-- }
4     data:  3-D double (224)      2 x 2 x 1 => 3 x 3 x 2  @ norm = 2
5
6       1.  1x1        |  1x2x2 [ -1 0 ;  0 1 ; -1 1 ]     -1.41421
7       2.  1x1        |  2x1x2 [  0 1 ;  1 0 ; -1 1 ]         -1.  {√2}
```

the fermionic parity operator $\hat{Z} = (-1)^{\hat{n}}$, with $\hat{n}$ the particle number operator [cf. Eq. (37)],

```
8  Z =
9       Q:  2x { 3 x 2 }  having 'A,SU2'  { +- }
10    data:  2-D double (336)      3 x 3 => 4 x 4  @ norm = 2
11
12      1.  1x1        |    1x1 [ -1 0 ; -1 0 ]            1.
13      2.  1x1        |    2x2 [  0 1 ;  0 1 ]           -1.  {√2}
14      3.  1x1        |    1x1 [  1 0 ;  1 0 ]            1.
```

the spin irop as in Eq. (10) with $\hat{S}_a = \sum_{\sigma\sigma'} \hat{f}_\sigma^\dagger S_a^{\sigma\sigma'} \hat{f}_\sigma$ and $S_\pm \equiv S_x \pm iS_y$,

```
15  S =
16       Q:  3x { 1 x 2 }  having 'A,SU2'  operator,  { +-- }
17    data:  3-D double (112)      1 x 1 x 1 => 2 x 2 x 3  @ norm = 1.225
18
19      1.  1x1        |  2x2x3 [  0 1 ;  0 1 ;  0 2 ]   -0.866025 {√2}
```

and the structure IS with additional information,

```
20  IS =                                                                          (32)
21       NC: 1
22      sym: 'Acharge,SU2spin'
23        E: [1x1 QSpace]
24      ...
```

The latter contains the identity operator (IS.E) as a reference for the complete local state space, a reminder of the number of flavors (I.NC), as well as more verbose specification of the symmetries used (IS.sym) for later identification. The remaining fields in IS are mostly for debugging purposes, partly also duplicate data such as the parity Z, and hence can be safely ignored. The above QSpace displays for F, Z, and S are simply obtained by typing the respective name followed by enter, e.g., >> F, where the leading '>>' indicates the Matlab prompt. The generated output conveniently summarizes the content of the specified QSpace

tensor (technically, by overloading the `display` routine). The printed output is discussed in detail with Fig. 7.

In the case of the fermionic annihilation operator `F`, line 3 identifies the symmetries used in their specified order [`A` for abelian U(1), and `SU2` for SU(2) spin]. It also shows that `F` is an `operator` (see `info.otype`) with standard index order `+--`. The symmetry labels in lines 6–7 consist of two numbers $q = [q_1 \; q_2]$ specified for each leg and separated by a semicolon. The first label $q_1$ encodes the particle number (filling) relative to half-filling. Here $q_1 \in [-1, 0, 1]$ on the first two legs corresponds to $[0, 1, 2]$ particles occupying the fermionic level. The second label specifies the SU(2) symmetry labels, given by the integers $q_2 = 2S$. The irop `F` transforms in the labels of the third leg, namely $q^{\mathrm{op}} = $ `[-1 1]`: that is, the operator `F` reduces particle number by one (hence $q_1 = $ `-1`), and represents a spinor of two components (with $q_2 = 2S = 1$ it transforms like a spin $S = 1/2$ representing up and down spin). All of this is as expected for the set of fermionic annihilation operators $\hat{F} \equiv (\hat{f}_\uparrow, \hat{f}_\downarrow)^T$. Since the irop index represents a single multiplet (line 4), the operator $q$-labels are the same for all records. Hence, the operator is irreducible, i.e., it is an irop. For comparison, a pair-annihilation operator $\hat{f}_\uparrow \hat{f}_\downarrow$ would have $q_1^{\mathrm{op}} = -2$. Therefore by simply looking at the action of an operator based on its irop symmetry labels allows one to (roughly) identify an operator.

By inspection, the first record for `F` (line 6) describes the matrix elements $\langle 0|\hat{f}_\sigma|\sigma\rangle$ from singly occupied to empty, with the initial state on leg 2 and the final state on leg 1 (ket and bra state for matrix element, respectively). The corresponding CGT has the printed symmetry labels $(0|1, 1)$, bearing in mind $q = 2S = 1$ which happens to correspond to a $1j$ symbol for the first record. It is of size `1x2x2`. Conversely, the second record in line 7 describes the matrix elements from doubly to singly occupied state, $\langle -\sigma|\hat{f}_\sigma|\uparrow\downarrow\rangle$ where $-\sigma$ denotes the spin opposite to $\sigma \in \{\uparrow, \downarrow\}$. Its CGT hast the $q$-labels $(1|0, 1)$ and dimensions `2x1x2`. Therefore the CGT for the second record corresponds to an identity matrix, when ignoring the singleton on the second index. The respective reduced matrix elements (RMTs) for the operator `F` turn out to be plain numbers, thus of size `1x1[x1]` (left column), skipping the last trailing singleton since beyond rank 2). Their value is directly displayed to the right. The signs (similar to the sign in line 19 for `S`) are determined by the sign convention of the underlying CGT for the irop. They are of no further concern, otherwise. What is important, though, is that the operator `F` is defined once and for all, so that the *same* object is used for all sites that are identical.

The way `getLocalSpace` proceeds to obtain the `QSpace` representations above, is to start from the Fock space description. For a single spinless fermionic level, in the basis $(0,1)$ for empty and occupied, the annihilation operator $c$, fermionic parity operator $z$, and identity operator $e$ are given by

$$c = \begin{pmatrix} 0 & 1 \\ 0 & 0 \end{pmatrix}, \qquad z = [c, c^\dagger] = \begin{pmatrix} 1 & 0 \\ 0 & -1 \end{pmatrix}, \qquad e = \{c, c^\dagger\} = \begin{pmatrix} 1 & 0 \\ 0 & 1 \end{pmatrix} \tag{33}$$

with the sole non-zero matrix element of the annihilation operator being $\langle 0|\hat{c}|1\rangle = 1$. Hence when combining two fermionic levels with different spin, for example, the matrix representation for the fermionic annihilation operators can be written by the $4 \times 4$ matrices (assuming the first factor in the tensor product to be the fast index, consistent with column-major),

$$f_\uparrow = c \otimes e = \begin{pmatrix} 0 & 1 & 0 & 0 \\ 0 & 0 & 0 & 0 \\ 0 & 0 & 0 & 1 \\ 0 & 0 & 0 & 0 \end{pmatrix}, \qquad f_\downarrow = z \otimes c = \begin{pmatrix} 0 & 0 & 1 & 0 \\ 0 & 0 & 0 & -1 \\ 0 & 0 & 0 & 0 \\ 0 & 0 & 0 & 0 \end{pmatrix} \tag{34}$$

having assumed the fermionic order $|\uparrow\downarrow\rangle \equiv \hat{c}_\uparrow^\dagger \hat{c}_\downarrow^\dagger|\rangle$ thus already leading to the fermionic sign via $z$ for $f_\downarrow$ above. It is important for the fermionic anticommutator relations which must be

already built-in in the above matrix elements, e.g.,

$$\{f_\uparrow, f_\downarrow\} \;=\; (c \otimes e)(z \otimes c) + (z \otimes c)(c \otimes e) = \{c, z\} \otimes c = 0.$$

since $cz = -zc$ by the very property of $c$ that it changes the particle number by one. Combining the annihilation operators into the set $F = (f_\uparrow, f_\downarrow)^T$, then for example

$$N = F^\dagger \cdot F \;=\; f_\uparrow^\dagger f_\uparrow + f_\downarrow^\dagger f_\downarrow \;=\; (c^\dagger c \otimes e) + (c^\dagger c \otimes z^\dagger z) \;=\; (n \otimes e) + (e \otimes n) \;\equiv\; N_\uparrow + N_\downarrow$$

as expected, with $n = c^\dagger c$ the occupation operator. QSpace takes the combined set $\mathtt{F=(f_\uparrow, f_\downarrow)^T}$ above, organizes the state space based on symmetry operators (Class/@SymOp), and then 'compresses' it via compactQS. This diligently factorizes the CGTs as present in the RC_STORE, with the remainder being in the RMTs.

**Explicit data structure**     By typing `F` at the Matlab prompt, this invokes @QSpace/display to conveniently summarize the specified QSpace data, e.g., as shown for `F` in lines 2–7. The bare QSpace data structure as in App. A.3 is obtained by stripping the tensor `F` of its QSpace class assignment via

```
25  >> struct(F)                                                                   (35)

26     struct with fields:
27
28         Q: {[2x2 double]  [2x2 double]  [2x2 double]}
29      data: {2x1 cell}
30      info: [1x1 struct]
```

which is considerably less readable and informative as compared to lines 2–7 above, despite referring to the same bare data. The two RMTs $\mathtt{i} = 1, 2$ in lines 6–7 are stored in `F.data{i}`

```
31  >> F.data'
32
33     1x2 cell array
34
35       {[-1.4142]}    {[-1.0000]}
```

with the respective symmetry labels stored in `F.Q{l}{i,:}` for leg $l$,

```
36  >> F.Q{1}
37        -1      0
38         0      1
39
40  >> F.Q{2}
41         0      1
42         1      0
43
44  >> F.Q{3}
45        -1      1
46        -1      1
```

These are displayed as records `i` with lines 6–7 above[5]. The info structure

```
47  >> F.info
48      qtype: 'A,SU2'
49      otype: 'operator'
50      itags: {''  '*'  '*'}
51      ctime: 0
52        cgr: [2x2 struct]
```

---

[5]As a Matlab technicality, note that `F.Q{:}` would not work here to display all entries; rather one would have to use an intermediate assignment, as in `Q=F.Q; Q{:}`.

is described in more detail with the data structure in App. A.3. It contains symmetry-related content, such as the symmetry type `F.info.qtype` or the CGRs in `cgr`. The field `itags` specifies the `itag` for each leg. Hence the number of entries has to match `numel(F.Q)`, or equivalently the rank of the tensor.

The field `info.cgr` is another structure array with one column per symmetry and as many rows as there are records in the QSpace. In case of the operator `F`, for example, the entry `F.info.cgr(1,2)` contains the CGR for the QSpace record $i = 1$ and symmetry 2, i.e., the SU(2) spin symmetry as per the present setup in (31),

```
53  >> F.info.cgr(1,2)
54
55     type: 'SU2'                       % symmetry type
56     qset: [0 1 1]                     % symmetry labels (q-labels, here q = 2S)
57     qdir: '+--'                       % q-directions
58      cid: [1.50e+09 1.50e+09 690057 0 8]  % RC_STORE specific IDs incl. time stamps
59     size: [1 2 2]                     % CGT size (dimensions, uint64 data type)
60      nnz: 2                           % number of non-zero entries in CGT
61      cgw: 1                           % w-matrix
62      cgt: [1x0 double]                % auxiliary internal field for trace
```

The symmetry labels `qset` in line 56 are those from the referenced CGT and reflect the ones in `X.Q{l}(1,2)` for leg $l$ as listed with the QSpace display of this record $i = 1$. in line 6 earlier. The dimensions in line 59 are also displayed there.

By contrast, the CGRs `X.info.cgr(:,1)` for the leading abelian U(1) symmetry are all trivial. The underlying CGTs are $C = 1$., bearing in mind that only non-zero CGTs are present in a QSpace tensor, i.e., only ones that are permissible from a symmetry point of view. For example, the trivial CGR from the first record reads

```
63  >> F.info.cgr(1,1)
64
65     type: 'A'
66     qset: [-1 0 -1]
67     qdir: '+--'
68      cid: [0 0 0 0 1]
69     size: [1 1 1]
70      nnz: 0
71      cgw: 1
72      cgt: [1x0 double]
```

It reflects the plain number 1. by having `size = 1`, `cgw = 1`, `nnz` is ignored, no internal IDs `cid`, etc. The symmetry labels `qset = [-1 0 -1]` are again consistent with those in `X.Q{l}(1,1)` for leg $l$ as displayed in line 6 earlier.

**State space from operators**   The annihilation operator `F` is non-hermitian and contains only off-diagonal symmetry blocks. Its full dimensions are shown as `3 x 3 x 2` in line 4 which only references 3 out of the total of 4 states with either of the first two indices. The reason is that an annihilation operator destroys an empty ket state $|0\rangle$, yet also has no matrix elements that include the fully occupied bra state $\langle\uparrow\downarrow|$. However their union of symmetry sectors as obtained by `getIdentity` when called on an operator like `F` here,

```
73  >> E=getIdentity(F)                                                      (36)
74      Q:  2x { 3 x 2 }  having 'A,SU2'  { +- }
75    data:  2-D double (336)       3 x 3 => 4 x 4  @ norm = 2
76
77     1.  1x1        |    1x1 [ -1 0 ; -1 0 ]              1.
```

```
78        2.  1x1         |    2x2 [  0 1 ;  0 1 ]           1.  {√2}
79        3.  1x1         |    1x1 [  1 0 ;  1 0 ]           1.
```

recovers the full `4 x 4` dimensional local state space (line `75`) of the single spinful fermionic level under consideration. By not having specified an index with the operator `F` in line `73`, the default behavior of `getIdentity` assumes an `operator` as input with standard index order, and hence takes the union of the state spaces of the first two indices of `F`. The fact that the first two indices of the annihilation operator must span the full state space of a site is also apparent from the realization that its fermionic anticommutator relations must yield the identity [for an explicit sanity check in this regard, see (38)].

Figure 8: Contraction $n = F^\dagger \cdot F$ to obtain the occupation operator in some (local) state space $s$, having spin flavor $\sigma \in \{\uparrow, \downarrow\}$. (b) The contraction in (37) takes the conjugate tensor of `F` reverts all arrows and also reflects the tensor, here vertically as indicated by the gray line and rotating arrow. Hence the index order in 1 and 2 is already flipped up/down with $F^*$. By contracting indices `'13*'` on the first, i.e., lower tensor, the `1` takes the transpose, thus completing the Hermitian conjugate. The remaining open indices are collected (c), thus giving rise to the new index order in (d) for the occupation operator. Uncontracted, and thus open indices are collected in the order of the input QSpace tensors to `contract`. Therefore the chosen order of input arguments in the command Eq. (37) already returns the standard index order for operators, as indicated by the orange labels in (d).

**Occupation number operator**   The operators returned by `getLocalSpace` in (31) are considered elementary for the description of the specified site. Any other local operator can be constructed from these, usually by simple one-liners. By construction, these operators are then already consistent with the chosen symmetries. For example, the local occupation number operator is obtained by

```
80  >> nloc = contract(F,'13*',F,'13')                                          (37)
```

```
81      Q:  2x { 2 x 2 }  having 'A,SU2'  { +- }
82   data:  2-D double (224)      2 x 2 => 3 x 3  @ norm = √6
83
84      1.  1x1         |    2x2 [  0 1 ;  0 1 ]           1.  {√2}
85      2.  1x1         |    1x1 [  1 0 ;  1 0 ]           2.
```

with a pictorial representation presented in Fig. 8. In addition to the contraction of index `1` from the matrix product, the sum over spin corresponds to the simultaneous contraction of the irop index `3` as specified by the strings `'13*'` or `'13'` in compact notation for the first and second operator, respectively. The above display for `nloc` shows two block entries corresponding to RMTs with `1.` or `2.` particles as printed to the right in lines `84` and `85`, respectively. These values reflect `nloc.data{i}`. There is no block record for `0.` particles, as this would represent a zero-block (more precisely, since block-diagonal zero blocks are usually kept in

rank-2 operators, in the present case this block does not arise out of the contraction in the first place). Note that for the value `1.` to appear in line 84, the normalization of the *w*-matrix had been tweaked, as indicated by the $\{\sqrt{2}\}$ factor to the very right [cf. discussion with Eq. (18)].

The definition of `nloc` permits one to relate its trace $\text{tr}(n_{\text{loc}}) = \text{tr}(F^\dagger \cdot F) = \|F\|^2$ with the norm of $\|F\| = 2$ in line 4. This can be quickly double-checked as follows,

```
86    >> [ trace(nloc), norm(F)^2 ]
87
88        4.0000    4.0000
```

The trace above includes a factor of 2 for the sector `q=[0 1]` in line 84, since by representing a spin half ($2S = 1$), this contains two states. Hence $\text{tr}(n_{\text{loc}}) = 0 + 2 \times 1 + 2 = 4$. The fermionic anti-commutator relations can be checked by

```
89    >> E=acomm(F,F')/2         %  = ½ Σ_σ {f̂_σ, f̂†_σ} ≡ ½(F·F† + F†·F) = 𝟙          (38)
90        Q:   2x { 3 x 2 }  having 'A,SU2'  { +- }
91     data:  2-D double (336)      3 x 3 => 4 x 4  @ norm = 2
92
93        1.   1x1        |    1x1 [ -1 0 ; -1 0 ]           1.
94        2.   1x1        |    2x2 [  0 1 ;  0 1 ]           1.  {√2}
95        3.   1x1        |    1x1 [  1 0 ;  1 0 ]           1.
96
97    >> isIdentityQS(E)
98
99        1
```

This needs to sum over $\sigma \in \{\uparrow, \downarrow\}$ since F exists as an irop, where one cannot single out a particular component $\sigma$, as this would break the symmetry. Line 97 checks whether E represents an identity operator (solely within the symmetry space present in the QSpace E, i.e., without knowledge of any potentially 'absent' zero blocks; this is no issue here, though). The call to the QSpace routine `acomm()` above for the anti-commutator is equivalent to performing the contraction in (37) plus its respective counterpart to complete the anti-commutator [see comment with (38)],

```
100   >> E_ = ( contract(F,'13*',F,'13' ) + ...
101          contract(F,'23', F,'23*') ) / 2;
102   >> norm(E-E_)
103
104        0
105
106   >> E==E_
107
108        1
```

Lines 102 and 106 verify that E and E_ are the same tensors, indeed (line 102 is somewhat more flexible as compared to the overload in `@QSpace/eq.m`, as one may use the former to accept differences on the order of double precision noise in a general context). Similar to the fermionic occupation operator in Eq. (37), one can construct the local Casimir operator from the spin operator,

```
109   >> S2=contract(S,'13*',S,'13')          %  Ŝ² = Ŝ†·Ŝ = ¾ 𝟙          (39)
110        Q:   2x { 1 x 2 }  having 'A,SU2'  { +- }
111     data:  2-D double (112)      1 x 1 => 2 x 2  @ norm = 1.061
112
113        1.   1x1        |    2x2 [  0 1 ;  0 1 ]           0.75  {√2}
```

Since the spin operator (10) is non-hermitian, the dagger is relevant. The above display shows a single entry for the half-filled symmetry sector `[0 1]` only, which being spin-half ($2S = 1$), is the only one with a non-zero Casimir, indeed, having the value `0.75` shown to the right. All other symmetry sectors, namely empty and doubly occupied, have $S^2 = 0$. Thus being zero-blocks, these are absent. As a matter of fact, these were already absent in the spin operator `S` that entered the contraction. The Casimir is thus already also reflected in the RMTs of the spin operator, e.g., having the RMT $\sqrt{3/4} = 0.866025$ in line 19 up to a sign. The norm $\|S^2\|$ is given by $\frac{3}{4}\| \mathbb{1}^{(2)}\| = \sqrt{9/8} = 1.061$ as seen in line 111. This is in contrast to $\|S\| = \sqrt{3/4}\,\|\mathbb{1}^{(2)}\| = \sqrt{3/2} = 1.225$ in line 17 above.

Based on the set of `QSpace` operators introduced above, the following proceeds with explicit examples for the main text earlier. This starts with examples for Sec. 2.1.

## 4.1 QSpace tensors of rank $r < 2$

**Rank-0 tensor**    A tensor of rank $r = 0$ has no open legs, and hence can only represent a scalar number. It naturally occurs when all indices have been contracted, e.g., when computing expectation values or overlaps. For example, computing the Frobenius norm of a tensor translates to contracting a tensor fully with itself,

$$\|\hat{f}\|^2 \;\equiv\; \mathrm{tr}(\hat{f}^{\dagger}\cdot\hat{f}) = \sum_{\sigma}\mathrm{tr}(\hat{f}^{\dagger}_{\sigma}\cdot\hat{f}_{\sigma})$$

the result has no more open legs. In `QSpace`, this becomes

```
114  >> X=contract(F,'123*',F,'123')                                    (40)

115      Q:  []  having 'A,SU2'  double scalar
116   data:  { 4 }
117
118  >> getscalar(X)
119
120      4.0000
```

Therefore it has empty $q$-labels `X.Q`, and only a single RMT `X.data{1}` [cf. App. A.3] with value `X.data{1}` $= \|F\|^2 = 4$ (cf. line 4). For rank-0 tensors, the `QSpace` display switches to the simplified format in lines 115–116 above. The scalar value can be extracted by `getscalar` (line 118) which also performs consistency checks (e.g., if `X` does not represent a scalar, it throws an error).

**Rank-1 tensor**    The only permissible rank-1 tensor is a tensor that carries the symmetry sector of the vacuum state, namely $q = 0$. For a general non-empty `QSpace` `X` of rank $r \geq 2$, the rank-1 tensor representing the vacuum state can be obtained by

```
121  >> V=getvac(F,'-1d')                                               (41)

122      Q:  1x { 1 x 2 }  having 'A,SU2'  { + }
123   data:  1-D double (112)      1 x 1  @ norm = 1
124
125      1.  1           |      1 [  0 0 ]              1.
```

With the option `'-1d'`, this returns a vector of length $d = 1$ (line 125). Rank-1 tensors in $q = 0$ with dimension $d > 1$ can occur naturally out of contractions, e.g., when fully contracting a tensor of rank $r$ on all its $r$ indices with a tensor of rank $r + 1$. Without the `'-1d'` option, `getvac` in Eq. (41) returns a rank-2 tensor that represents the identity operator in the vacuum state, and thus is of dimension $1 \times 1$.

## 4.2 Permutations and index order

QSpace tensors support an arbitrary index order that can be permuted as desired. While some tensors such as operators or *A*-tensors assume certain index order conventions for coding purposes, these are relevant for applications only, but not for QSpace tensors per se. For the sake of the argument, consider an operator that describes fermionic hopping from site s02 and s01, i.e., $\hat{f}_1^\dagger \cdot \hat{f}_2 = \sum_\sigma \hat{f}_{1\sigma}^\dagger \cdot \hat{f}_{2\sigma}$ with $\sigma \in \{\uparrow, \downarrow\}$,

```
126  >> F2=contract(F,'-op:s01','*', Z*F,'-op:s02', [2 3 1 4])                    (42)

127        Q:   4x { 4 x 2 }  having 'A,SU2'  { s01, s02, s01*, s02* }
128     data:   4-D double (448)       2 x 2 x 2 x 2 => 3 x 3 x 3 x 3  @ norm = √8
129
130        1.  1x1          | 2x1x1x2 [  0 1 ; -1 0 ; -1 0 ;  0 1 ]     1.41421
131        2.  1x1          | 2x2x1x1 [  0 1 ;  0 1 ; -1 0 ;  1 0 ]    -1.41421
132        3.  1x1          | 1x1x2x2 [  1 0 ; -1 0 ;  0 1 ;  0 1 ]    -1.41421
133        4.  1x1          | 1x2x2x1 [  1 0 ;  0 1 ;  0 1 ;  1 0 ]    -1.41421
```

written as rank-4 tensor here by only contracting the operator index via the dot product in the operators, thus having $\langle s_1 s_2 | \hat{f}_1^\dagger \cdot \hat{f}_2 | s_1' s_2' \rangle$. By specifying a conjugate flag '*' with the first operator F, this takes its Hermitian conjugate. The option '-op:s01' assigns operator itags for site 's01' on the fly. Specifically, this assigns the temporary itags {'s01','s01*','op*'} to the preceding input operator [cf. Fig. 1(b)]. Similarly, the second operator Z*F gets assigned the operators itags for site 's02' (because in a simulation typically more than 10 sites are included, QSpace frequently uses a 2-digit format for site or bond indices in itags, like 's01' and 's02' above for the sake of aligned displays).

Fermionic signs are applied by contracting the parity operator $Z \equiv (-1)^{\hat{n}}$ from the left onto the operator F acting on site 2. This assumes a fermionic order where site 1 comes before site 2 when building a Fock state space, e.g., $\hat{f}_2^\dagger \hat{f}_1^\dagger | \rangle$ similar to Eq. (34). Hence the matrix elements for $\hat{f}_1$ acquire fermionic signs depending on the state of site 2 which thus gets the parity operator applied (see also appendix in [50] for more discussion on fermionic signs). Since both, Z and F, are operators, one can simply use the operator * for matrix multiplication in Z*F.[6] The default operator itag assigned above by the option '-op:..' is 'op'. In the present example, this is the only matching itag in the auto-contraction. Therefore the contraction results in a rank-4 tensor that explicitly acts on the two separate sites s01 and s02.

Command (42) also includes a trailing permutation that is applied to the overall result. Therefore it is equivalent to

```
134  >> F2=contract(F,'-op:s01','*',Z*F,'-op:s02');
135  >> F2=permute(F2,'2314')    % may also write permutation as [2 3 1 4]      (43)
```

The contraction in line 134 collects the uncontracted legs of the first operator conj(F) [note the '*' flag in the contraction], followed by the uncontracted ones from the second operator, maintaining their original order. The bra and ket indices of the rank-4 operator F2 out of the contraction can therefore be properly grouped by the permutation [2 3 1 4] (line 135, or trailing option in line 126) with the effect $(s_1^*, s_1, s_2, s_2^*) \to (s_1, s_2, s_1^* s_2^*)$. This finalizes the transpose for the first operator, which was conjugated by the flag '*' in the contraction. With this, the itags of the final output for F2 have $q$-directions ++-- as seen from the trailing conjugate flags '*' in the itags in line 127.

---

[6]The matrix multiplication operator * is overloaded via @QSpace/mtimes.m to perform a QSpace contraction; this also deals with potential rank-3 operators as input, like F here. As a safety measure, this requires the explicit marking of the QSpace as an 'operator' via the flag F.info.otype='operator'. Here this is automatically set via getLocalSpace, e.g., see line 3 above.

The permutation in line 135 accepts compact string notation aside from the plain numerical representation. By contrast, line 126 insists on a numerical representation of the permutation to differentiate it from other string options meant for contractions. Line 135 accepts any valid permutation to be applied to F2, including an optional trailing conjugation flag `'*'`. For example, the following swaps bra with ket indices and also applies overall tensor conjugation, i.e., obtains the Hermitian conjugate $\langle s_1 s_2 | \hat{f}_2^\dagger \cdot \hat{f}_1 | s_1' s_2' \rangle$ for given rank-4 tensor,

```
136  >> F2_ = permute(F2,'3412*')
137
138      Q:  4x { 4 x 2 }  having 'A,SU2'  { s01, s02, s01*, s02* }
139   data:  4-D double (448)      2 x 2 x 2 x 2 => 3 x 3 x 3 x 3  @ norm = √8
140
141      1.  1x1           | 1x2x2x1 [ -1 0 ;  0 1 ;  0 1 ; -1 0 ]      1.41421
142      2.  1x1           | 1x1x2x2 [ -1 0 ;  1 0 ;  0 1 ;  0 1 ]     -1.41421
143      3.  1x1           | 2x2x1x1 [  0 1 ;  0 1 ;  1 0 ; -1 0 ]     -1.41421
144      4.  1x1           | 2x1x1x2 [  0 1 ;  1 0 ;  1 0 ;  0 1 ]     -1.41421
```

While the itags of the resulting F2_ are identical to the ones from F2, their content is different, as can be checked by norm(F2-F2_) which results in 4.

### 4.3  $1j$ tensor and trailing marker characters

The identity operator in Eq. (36) is a regular operator with one in- and one out-going index. Instead of the identity, one may request the $1j$ tensor by the same call to getIdentity, yet adding the option `'-0'` ('fusing to $q = 0$'),

```
145  >> U=getIdentity(F,'-0')                                              (44)
146      Q:  2x { 3 x 2 }  having 'A,SU2'  { ++ }
147   data:  2-D double (336)      3 x 3 => 4 x 4  @ norm = 2
148
149      1.  1x1           |   1x1 [ -1 0 ;  1 0 ]          1.
150      2.  1x1           |   2x2 [  0 1 ;  0 1 ]          1.   {√2}
151      3.  1x1           |   1x1 [  1 0 ; -1 0 ]          1.
```

This shows a colored itag in line 146 when using the Matlab terminal mode, for readability. The actual `'U.info.itags'` are {'',''''}, with the first itag empty, and the second itag just a prime ('). Since no string delimiters are shown with itags in the QSpace display, empty strings would not display at all. Hence, in case of empty strings, the $q$-directions +/- are displayed, instead. In the presence of a trailing prime marker character, the itag is colored as above. The color coding is understood by terminals only, but not the Matlab desktop environment. For the latter by default, line 146 above would display as

```
152      Q:  2x { 3 x 2 }  having 'A,SU2'  { ++' }
```

instead, which shows the marker character with the second itag, ++' ≡ {+,+'}. The same can also be enforced by setting `setenv QS_LOG_COLOR 0` in the environment, e.g., on the Matlab prompt itself. If getIdentity is called referencing a tensor leg that carries an itag, then that itag is inherited,

```
153  >> E=setitags(getIdentity(F),{'s01','s01'});
154  >> U=getIdentity(E,'-0')
155
156      Q:  2x { 3 x 2 }  having 'A,SU2'  { s01, s01 }
157      ...
```

again with colored output as in Matlab terminal mode. The remainder (...) is the same as in lines 147–151. The actual itags U.info.itags are {'s01', 's01''}, as displayed in the Matlab desktop, where line 156 would read

```
158        Q:  2x { 3 x 2 }  having 'A,SU2'  { s01, s01' }
```

If one were to take the conjugate tensor of $U$, this results in all outgoing legs, with the second leg still also marked, which in terminal mode would translate to the color coding for the sake of readability,

```
159   >> conj(U)
160        ...
161        Q:  2x { 3 x 2 }  having 'A,SU2'  { s02, s02 }
162        ...
```

where the an outgoing itag is printed in gray, and if marked, the original brighter green in line 156 becomes a darker green. In the Matlab desktop, the itags are fully spelled out

```
163   >> conj(U)
164        ...
165        Q:  2x { 3 x 2 }  having 'A,SU2'  { s02*, s02'* }
166        ...
```

The trailing marker toggle inserted by getIdentity with the option '-0' may be disabled by using the option '-z', instead (z as in zero in '-0' for historical reasons; not recommended),

```
167   >> Uz=getIdentity(E,'-z')
168
169        Q:  2x { 3 x 2 }  having 'A,SU2'  { s01, s01 }
170    data:  2-D double (336)      3 x 3 => 4 x 4  @ norm = 2
171
172        1.  1x1         |    1x1 [ -1 0 ;  1 0 ]         1.
173        2.  1x1         |    2x2 [  0 1 ;  0 1 ]         1.  {√2}
174        3.  1x1         |    1x1 [  1 0 ; -1 0 ]         1.
```

In this case, the itags (line 169) are the same for both legs and become indistinguishable from the ones for the transpose of the tensor. However, taking the transpose of Uz,

```
175   >> Uz_=permute(Uz,'21')
176
177        Q:  2x { 3 x 2 }  having 'A,SU2'  { s01, s01 }
178    data:  2-D double (336)      3 x 3 => 4 x 4  @ norm = 2
179
180        1.  1x1         |    1x1 [  1 0 ; -1 0 ]         1.
181        2.  1x1         |    2x2 [  0 1 ;  0 1 ]         1.  {-√2}
182        3.  1x1         |    1x1 [ -1 0 ;  1 0 ]         1.
```

give rise to a minus sign with the second entry (with the permutation absorbed into the *w*-matrix, and hence showing up with the curly bracket to the very right). Therefore Uz is clearly different from Uz_, having norm(Uz-Uz_) = $\sqrt{8}$, even though indistinguishable, say, pictorially when including itags. Both are unitaries, though, and therefore can act as $1j$ symbols. However, mixing them up when reverting directions of legs, e.g., by inserting Uz * Uz_$^\dagger \neq \mathbb{1}$ into a line in a tensor network, this is likely to give wrong results because of sign errors, even though the contraction is possible on principle grounds. To avoid such sources of errors that may be difficult to spot, the trailing prime as a marker character was introduced with QSpace v4. After all, the dual state space is different from its original one, even if the considered state space may happen to be self-dual in the sense that it maps onto itself as a whole, as in the present example. Trying to contract a marked with a non-marked itag then, will result in an error because such itags are considered different.

## 4.4 Spin-half spin operators

The spin operators for a single spin $S = 1/2$ are obtained as in Eq. (28),

```
1  >> [S,IS] = getLocalSpace('Spin',1/2,'-A');
```
(45)

resulting in

```
2  >> S  % equivalent to display(S)
3
4     1.     (U1)   { +-- }        3D double      0.7071      224  2x2x1
5     2.     (U1)   { +-- }        3D double      0.7071      112  1x1x1        operator
6     3.     (U1)   { +-- }        3D double     -0.7071      112  1x1x1        operator
```

Since S is a QSpace array of more than two entries, the QSpace display switches to a list view with one-liners for each entry in S that further summarize the header lines in the detailed display for each S(i) with $i = 1, 2, 3$. To see the full QSpace display of all entries in the QSpace array S, one needs to call the underlying display routine together with the option '-v' that stands for more 'verbose',

```
7  >> display(S,'-v')
8
9  S(1) =
10      Q:  3x { 2 x 1 }  abelian U(1)  { +-- }
11    data:  3-D double (224)      2 x 2 x 1  @ norm = 0.7071
12
13      1. 1x1     [  1 ;  1 ;  0 ]         0.5
14      2. 1x1     [ -1 ; -1 ;  0 ]        -0.5
15
16  S(2) =
17      Q:  3x { 1 x 1 }  abelian U(1)  operator,  { +-- }
18    data:  3-D double (112)      1 x 1 x 1  @ norm = 0.7071
19
20      1. 1x1     [ -1 ;  1 ; -2 ]     0.707107
21
22  S(3) =
23      Q:  3x { 1 x 1 }  abelian U(1)  operator,  { +-- }
24    data:  3-D double (112)      1 x 1 x 1  @ norm = 0.7071
25
26      1. 1x1     [  1 ; -1 ;  2 ]    -0.707107
```

By inspection, this shows that the returned spin operator S has three entries corresponding to

$$\text{S} = [\hat{S}_z, \tfrac{1}{\sqrt{2}}\hat{S}_-, -\tfrac{1}{\sqrt{2}}\hat{S}_+] \, .$$
(46)

For algorithmic convenience, this keeps $\hat{S}_z$ to the front by permuting $\hat{S}_+$ to the end w.r.t. Eq. (10), such that S(1) is either $\hat{S}_z$ or the full spin irop in case of SU(2) spin. The spin operator S above is derived from the representation as $S = 1$ irop in Eq. (10), here corresponding to symmetry labels $S_z \in \{0, -1, +1\}$ or $q_{op} = 2S \in \{0, -2, +2\}$ for the irop $q$-label with the third index in lines 13f, 20, and 26, respectively [hats are used with the operators in (46) to differentiate them from the symmetry labels $|S{=}1; S_z \in -1, 0, 1\rangle$ here for the $S = 1$ multiplet underlying the irop]. This explains the normalization and signs in the components of S up to the permutation of the entries. Note that the inverse square root factors with the lowering and raising operator lead to the same norm $|\text{S(i)}| = \frac{1}{\sqrt{2}} = 0.7071$ ($i = 2, 3$) as compared to the $S_z$ operator ($i = 1$). As such, they also allow for a convenient $S^\dagger \cdot S$ contraction for isotropic spin interactions.

Having U(1) spin, all three operators in `S` are distinguishable by their irop symmetry labels. Therefore the three spin operators may be combined into a single spin operator by simply 'adding' them,

```
27  >> Sop=sum(S)
28
29        Q:   3x { 4 x 1 }  abelian U(1)  { +-- }
30     data:   3-D double (448)      2 x 2 x 3  @ norm = 1.225
31
32        1.   1x1      [ -1 ; -1 ;  0 ]          -0.5
33        2.   1x1      [ -1 ;  1 ; -2 ]      0.707107
34        3.   1x1      [  1 ; -1 ;  2 ]     -0.707107
35        4.   1x1      [  1 ;  1 ;  0 ]           0.5
```

Because all irop symmetry sectors in `S` are different, addition translates into a direct sum ⊕ here. This simply *catenates* all `QSpace` records into a single list, since the entries are still differentiated by their U(1) irop labels, thus preserving the RMT block structure. The resulting `Sop` is no longer irreducible from a symmetry perspective, though. Therefore in contrast to the individual entries `S(i)`, the combined `Sop` no longer qualifies as an 'irop', since it has multiple operator symmetry labels listed with its third index. It is still a perfectly well-defined operator, nevertheless. The norm of the spin operator `Sop`,

$$\|S\|^2 \;\equiv\; \mathrm{tr}(S^\dagger \cdot S) = \sum_{i=1,2,3} \mathrm{tr}(S_i^\dagger S_i) = \sum_{a=x,y,z} \mathrm{tr}(S_a^\dagger S_a) = 3 \times (2\tfrac{1}{2^2}) \;=\; \tfrac{3}{2}$$

agrees with line 30, having $|S| = \sqrt{3/2} = 1.225$. The combined spin operator `Sop` also permits one to easily obtain the Casimir $\hat{S}^2 \equiv \hat{S}^\dagger \cdot \hat{S}$,

```
36  >> S2=contract(Sop,'13*',Sop,'13')
37
38        Q:   2x { 2 x 1 }  abelian U(1)  { +- }
39     data:   2-D double (224)      2 x 2  @ norm = 1.061
40
41        1.   1x1  [ -1 ; -1 ]        0.75
42        2.   1x1  [  1 ;  1 ]        0.75
```

resulting in $\tfrac{3}{4}\mathbb{1}$ as expected for the underlying spin-half. The contraction on the third index above implements the dot product in $\hat{S}^\dagger \cdot \hat{S}$.

**Switch to $Z_2$ symmetry** The option `'-A'` in the setup (45) requested abelian U(1) symmetry. By specifying `'--Z2'`, instead, this switches to a $Z_2$ symmetry representation [cf. Eq. (29)],

```
43  >> [S,IS] = getLocalSpace('Spin',1/2,'--Z2')
44  S(1) =
45        Q:   3x { 2 x 1 }  having 'Z2'  operator,  { +-- }
46     data:   3-D double (224)      2 x 2 x 1  @ norm = 0.7071
47
48        1.   1x1      [ 1 ;  1 ;  0 ]          -0.5
49        2.   1x1      [ 0 ;  0 ;  0 ]           0.5
50
51  S(2) =
52        Q:   3x { 1 x 1 }  having 'Z2'  operator,  { +-- }
53     data:   3-D double (112)      1 x 1 x 1  @ norm = 0.7071
54
55        1.   1x1      [ 0 ;  1 ;  1 ]      0.707107
```

```
56
57  S(3) =
58       Q:  3x { 1 x 1 }  having 'Z2'  operator,  { +-- }
59     data:  3-D double (112)      1 x 1 x 1  @ norm = 0.7071
60
61       1.  1x1      [ 1 ;  0 ;  1 ]    -0.707107
```

By comparison to the earlier U(1) spin operator, the setup of the model changed from $q = 2S$ for U(1) spin to a $Z_2$ setting with $q \in \{0, 1\} \equiv \{\uparrow, \downarrow\}$, as seen from $S_z$ in lines 44ff. The spin operator $S(1) = S_z$ carries $q_{op} = 0$, whereas the operators $\hat{S}_+$ and $\hat{S}_-$ become indistinguishable in terms of irop symmetry labels, both having $q_{op} = 1$ due to the $Z_2$ symmetry. However, when including the symmetry labels for all legs, all entries for the spin operator above are still distinguishable from a symmetry sector point of view. Hence the earlier U(1) constructions in lines 27 and 36 still also work.

**Switch to no symmetry**    By switching the symmetry option in (45) from `'-A'` to `'--nosym'` ('no symmetries'), this effectively turns off symmetries [cf. Eq. (30)],

```
62  >> [S,IS] = getLocalSpace('Spin',1/2,'--nosym')
```

```
63  S(1) =
64       Q:  2x { 1 x 1 }  having 'A'  { +- }
65     data:  2-D double (136)      2 x 2  @ norm = 0.7071
66
67       1.  2x2 [  0 ;  0 ]      32 b
68
69  S(2) =
70       Q:  2x { 1 x 1 }  having 'A'  { +- }
71     data:  2-D double (136)      2 x 2  @ norm = 0.7071
72
73       1.  2x2 [  0 ;  0 ]      32 b
74
75  S(3) =
76       Q:  2x { 1 x 1 }  having 'A'  { +- }
77     data:  2-D double (136)      2 x 2  @ norm = 0.7071
78
79       1.  2x2 [  0 ;  0 ]      32 b
```

This is a tweak to abelian U(1), seen as `'A'` in the header lines, that only uses symmetry labels $q = 0$. This trivial setting is equivalent to having no symmetry at all. The RMTs are $2 \times 2$ matrices,

```
80  >> S(1).data{1}
81
82      0.5000         0
83          0    -0.5000
84
85  >> S(2).data{1}
86
87          0         0
88      0.7071         0
89
90  >> S(3).data{1}
91
92          0    -0.7071
93          0         0
```

now reflecting in detail the spin operators in terms of the Pauli matrices, $[\frac{1}{2}\sigma_z, \frac{1}{\sqrt{2}}\sigma_-, -\frac{1}{\sqrt{2}}\sigma_+]$, respectively. In the present case of no symmetry, there are no symmetry labels that can describe the action of the operators. Hence all operators are rank-2 scalars. This way, all spin operators become indistinguishable in terms of irop $q$-label. These spin operators can no longer be combined into a single spin operator as in line 27, since the RMTs are no longer distinguishable by symmetry labels. Hence sum(S) would no longer perform the direct sum ($\oplus$) by catenating the QSpace records, but actually add the RMTs which is not meaningful.

# 5  Simple Tutorials and Applications

This section extends on the simple examples in Sec. 4 which supplemented the earlier sections in the main text. Here the focus shifts more toward tensor network applications. For this purpose, it should be noted that the public QSpace repository already contains significantly more than the bare tensor library documented here, e.g., see the listing of (MEX) applications in App. C. The full documentation of this is left for the future. Nevertheless, the standard help explaining the purpose and usage of any function is available as part of the repository. Yet their discussion and the description of the related setup scripts are beyond the scope of this documentation, which focuses on QSpace as a tensor library. Still, the interested user may find it rewarding to explore. Since fdm-NRG [19] has been one of the first applications of QSpace, it is discussed in more detail in Sec. 5.6.

Sanity checks for tensors that have a simple interpretation and structure are also frequently included. These are helpful in practice quite generally, as they ensure that one has a good understanding of all the operators and tensors that one is dealing with. Much of the same spirit also underlies the earlier examples given, and QSpace as a whole. There are many consistency and plausibility checks internally in the QSpace MEX core routines and in the Matlab environment. At negligible numerical overhead, these assertions are safeguards to ensure overall consistency.

## 5.1  Iterative build-up of many-body state spaces: Two fermionic sites

An elementary step to building many-body state spaces is the iterative addition of a new site. The example here shows how to start this within QSpace by adding a second site. To be specific, suppose one wants to describe two interacting spinful fermionic sites with respective local state spaces $\sigma_i$ and site index $i = 1, 2$ in a combined state space. The description of a single site again starts with (31) above. With this, one can begin to take iterative tensor products of that state space with copies of itself to build a many-body basis.

Reminding oneself that QSpace is a tensor library that encodes tensors only, state spaces are specified by pointing to particular legs of a given tensor. Hence, when expanding a Hilbert space via a tensor product, one needs to pick operators that represent the complete state space, i.e., by having non-zero blocks in every symmetry sector of the target state space. Natural choices are the identity stored with IS.E [cf. Eq. (31)], or the parity Z. When building a tensor product of state spaces, the routine getIdentity takes two QSpaces as input, both representing a state space of their own (here by having identical sites, the operator Z is listed twice),

```
1  A=getIdentity(Z,Z,[1 3 2]);
2  A=setitags(A,{'K01','K02','s02'});                                            (47)
```

In line 1, one may use Z $\rightarrow$ IS.E, instead. One even could have used F instead of Z, since if no explicit leg index is provided with the input, getIdentity assumes an operator, and hence combines the state spaces from the first two legs [same argument as with Eq. (36) above].

Either way, A encodes the tensor-product fusion of the state spaces of two sites. The optional last argument [1 3 2] in line 1 specifies a permutation to be applied to the returned output *A*-tensor. By default, the output of getIdentity in line 1 has an index order that lists the two input spaces first (in the order specified), and lists the combined state space last (third index). By permuting this order as specified in line 1, this returns the rank-3 *A*-tensor with index order that adheres to the LRs index order convention [cf. Fig. 1]. As an implementational detail, the permutation in line 1 requires a numeric format, since a (compact) string representation such as '132' would be interpreted as an itag for the fused index, instead.

From an MPS perspective, the above tensor product of sites $\sigma_1$ and $\sigma_2$ may represent the start of a physical chain. Virtual bond indices specify the many-body state spaces that are *kept* from one iteration to the next. Hence $\sigma_1 \to K_1 \equiv$ 'K01' may be considered the state space that is kept from iteration 1 (cf. Fig. 2). QSpace, and hence also this documentation, frequently uses a 2-digit format for site or bond indices for the sake of aligning display output, bearing in mind that typically more than 10 sites are present in a simulation. Hence 'K01' instead of, e.g., 'K1'. By adding site $\sigma_2 \to$ 's02', one arrives at the combined state space 'K02'. The itags in line 2 are thus chosen as in Fig. 2 for $A_2$ at the start of an MPS. With this, the *A*-tensor returned with line 2 reads (see Fig. 7 for detailed explanations of this very example)

```
3   >> A
4       Q:  3x {10 x 2 }  having 'A,SU2'  A-matrix,  { K01, K02*, s02 }
5     data:  3-D double (1200)       3 x 10 x 3 => 4 x 16 x 4  @ norm = 4
6
7       1.   1x1        |  1x1x1 [ -1 0 ; -2 0 ; -1 0 ]           1.
8       2.   1x2        |  1x2x2 [ -1 0 ; -1 1 ;  0 1 ]       16 b       {√2}
9       3.   1x2        |  2x2x1 [  0 1 ; -1 1 ; -1 0 ]       16 b       {√2}
10      4.   1x3        |  1x1x1 [ -1 0 ;  0 0 ;  1 0 ]       24 b
11      5.   1x3        |  2x1x2 [  0 1 ;  0 0 ;  0 1 ]       24 b
12      6.   1x3        |  1x1x1 [  1 0 ;  0 0 ; -1 0 ]       24 b
13      7.   1x1        |  2x3x2 [  0 1 ;  0 2 ;  0 1 ]           1.  {√3}
14      8.   1x2        |  2x2x1 [  0 1 ;  1 1 ;  1 0 ]       16 b       {√2}
15      9.   1x2        |  1x2x2 [  1 0 ;  1 1 ;  0 1 ]       16 b       {√2}
16     10.   1x1        |  1x1x1 [  1 0 ;  2 0 ;  1 0 ]           1.
```

The itags are shown in the header line 4. With K02 the outgoing index, it has the trailing conjugate flag '*'. Since *q*-directions need to be preserved, when setting itags via setitags there is no need to specify the trailing asterisk with K02 in line 2 (if trailing asterisks had been specified, these would be ignored).

The norm in line 5 shows $\|A\| = 4$, i.e., $\|A\|^2 = \mathrm{tr}(\mathbb{1}) = 16$, which is consistent with the full Hilbert space dimension reflected in the dimensions [4 16 4] of *A* in the same line. The square root factors to the very right, as always, are just reminders of the tweaked normalization with the *w*-matrix [cf. Eq. (19)], so that the reduced matrix elements read 1., as one may expect for a normalized mapping of state space, represented as a sliced-up *identity* matrix in multiplet space. For example, in line 13, the factor $\sqrt{3}$ [which is due to having the fused multiplet $q = 2S = 2$ of dimension 3] is split off into the *w*-matrix, such that the RMT can read 1. When the RMT has dimensions $> 1$, this shows the size of that RMT in bytes instead of its value. For example, for the 1x2[x1] RMT in line 8, 2 * (8 bytes for double) = 16 b. The trailing singleton dimensions in the RMT on index 3 are not shown (second column).

With the 2-site Hilbert space described by *A*, one can now proceed to compute matrix elements in the combined many-body state space. Consider, for example, the spin interaction $S_{12} \equiv A^\dagger(S_1^\dagger \cdot S_2)A$ when cast into the basis *A* [see Fig. 10(d) or also Fig. 9 with $(Z)F \to S$ for pictorial representations]

```
17   S12 = contract(A,'!2*',{S,'-op:K01','*',{A,S,'-op:^s'}})
```
(48)

```
18        Q:  2x { 2 x 2 }  having 'A,SU2'  { K02, K02* }
19     data:  2-D double (288)      4 x 4 => 6 x 6  @ norm = 0.866
20
21        1.  3x3        |    1x1 [  0 0 ;  0 0 ]      72 b
22        2.  1x1        |    3x3 [  0 2 ;  0 2 ]         0.25  {√3}
23
24  >> S12.data{1}
25
26            0          0         0
27            0     -0.7500        0
28            0          0         0
```

Before explaining the compact semantics of the contraction in Eq. (48), the output in QSpace S12 is examined. As expected for the operator $S_1^\dagger \cdot S_2$, there are only matrix elements at half-filling in the above display for S12, having $q_1 = 0$ for the first symmetry label. More restrictive still, non-zero matrix elements can only derive from the half-filled space on either site. The respective tensor product of two spin-halfs gives one singlet ($q_2 = 0$) and one triplet state ($q_2 = 2S = 2$). The triplet state is represented by the second record in line 22 which has $\langle S_1^\dagger \cdot S_2 \rangle = 0.25$ as expected. The singlet across both sites is buried within the first record in line 21 where it appears as the second entry (line 27) and carries the value $\langle S_1^\dagger \cdot S_2 \rangle = -0.7500$, as expected for the singlet. In the global half-filled sector, there are two other states in the scalar sector q=[0 0], namely the combination of an empty at site 1 with completely filled at site 2, or vice versa. Their expectation value $\langle S_1^\dagger \cdot S_2 \rangle = 0$, thus corresponding to the other two zero entries along the diagonal in lines 26 and 28. Because they are part of the same global symmetry sector, they also need to appear here within the same RMT indexed by $n$ as in Eq. (2). This results in the RMT of dimension $3 \times 3$ in lines 26–28. Since the Hamiltonian preserves spin symmetry, the matrix elements of S12 must already be in a diagonal representation, as seen above, indeed.

**Contraction semantics**  The nested contraction in Eq. (48) consists of three levels, based on the pairwise contraction pattern fixed by adding brackets to $S_{12} = A^\dagger (S_1^\dagger \cdot (S_2 A))$,

$$S_{12} \;=\; \text{contract}(A, \texttt{'!2*'}, \{S, \texttt{'-op:K01'}, \texttt{'*'}, \underbrace{\{A, S, \texttt{'-op:\^s'}\}}_{\equiv C_2}\}) \tag{49}$$

$$\underbrace{\phantom{\text{contract}(A, \texttt{'!2*'}, \{S, \texttt{'-op:K01'}, \texttt{'*'},\{A, S, \texttt{'-op:\^s'}\}}}_{\equiv C_1}$$

$$\underbrace{\phantom{\text{contract}(A, \texttt{'!2*'}, \{S, \texttt{'-op:K01'}, \texttt{'*'},\{A, S, \texttt{'-op:\^s'}\}\}}}_{\equiv C_0}$$

[see Fig. 10(d) or also Fig. 9 with $(Z)F \to S$ for pictorial representations]. The underlying MEX routine contractQS supports nested cell structures based on pairwise contractions, also referred to as a contraction pattern. With this, one may contract any arbitrary number of tensors in a single call to contract (the Matlab wrapper routine contractQS). For any included pairwise contraction, the two involved tensors can be either an existing QSpace specified as argument, or another cell {...} thus adding another recursive 'level' to the contraction. The latter indicates a nested contraction that must be carried out first, so its result can be used. If further cells are encountered, contractQS proceeds recursively through the nested calls as they are encountered. Eventually, at the deepest level of any set of nested calls must be two QSpace objects to be contracted [like with $C_2$ in Eq. (49)] which then is carried out first. Each QSpace input with empty itags may have an option that assigns itags on the fly, typically via '-op:..' as above. Each QSpace or cell also accepts an option that specifies what indices (not) to contract and whether to take the conjugate. This option is specified via a compact string with the format [[!]cidx][*] where terms in square brackets are optional. The term cidx represents a compact string that explicitly specifies which indices to contract (or *not*

to contract in the presence of the leading `'!'`). If `cidx` is empty, an implicit auto-contract searches for all matching indices based on their `itags`. The trailing `*` takes the conjugate of the preceding QSpace or cell. For an explanation of all options available for contractions, see `help contractQS`, or equivalently, `contractQS -h`.

The nested cell structure, as provided by the input to `contractQS`, fully determines the order of contractions. Specifically, QSpace does not optimize or restructure the order of contractions. The optimal order of contractions is thus left for the user to determine. In any contraction like $T_{12} \equiv T_1 * T_2$, the non-contracted, i.e., kept indices of the first tensor $T_1$ are collected *before* the ones in the paired up $T_2$, while maintaining their original order. This holds for any pairwise construction in the nested structure provided to `contractQS`. This way, the structure of the input to `contractQS` also fully determines the index order of the final result.

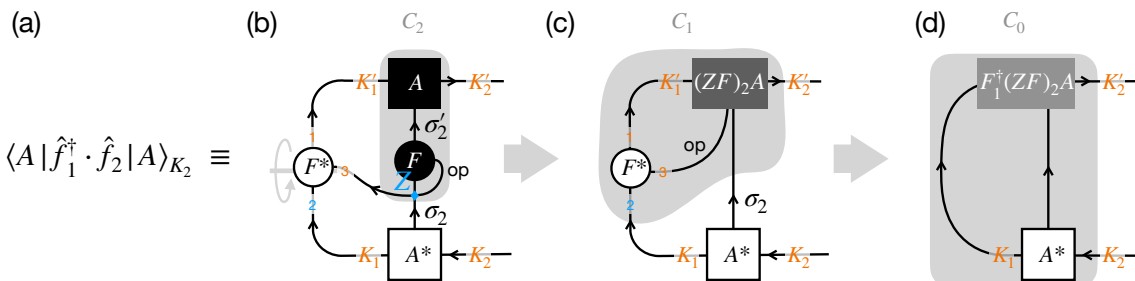

Figure 9: Computing the matrix elements for the fermionic hopping $\hat{f}_1^\dagger \cdot \hat{f}_2$ in the combined basis $K_2$ corresponding to the nested cell contraction in (50). The contraction includes four tensors, as shown in (b). By taking the conjugate of the tensor $F_1$, i.e., $F$ acting on $K_1 \equiv \sigma_1$, similar to Fig. 8 again the up/down reflected mirror image is drawn for $F^*$. This is emphasized by the gray line with a rotating arrow and the reverted index order 1 and 2. Auto-contraction based on `itags` automatically links the indices correctly to take the transpose for the Hermitian conjugate $F_1^\dagger$ overall. Since $F_2$, i.e., $F$ applied to $\sigma_2$, acts on $|A\rangle$ first [panel (b)], the operator line `op` leads to a crossing of the line $\sigma_2$ *below* $F_2$, indicated by the blue symbol. Because the operator line `op` carries an odd charge parity, this crossing gives rise to a fermionic swap gate [33]. Here this translates into the parity operator $Z$ (blue) applied to $\sigma_2$ at the location of the blue symbol. This is equivalent to taking `F` $\to$ `ZF` for site 2. If spin operators $S_1^\dagger \cdot S_2$ had been considered instead of the fermionic operators $\hat{f}_1^\dagger \cdot \hat{f}_2$ here, the precise location of the line crossing (above or below $F_2 \to S_2$) would be of no further importance because the spin operators carry an even charge parity on `op` (hence they also commute, $S_1^\dagger \cdot S_2 = S_2 \cdot S_1^\dagger$). The nested cell contraction in (50) gives the sequence of pairwise contraction indicated in panels (b-d). The contractions $C_2$, $C_1$, and $C_0$ highlighted by the gray shaded area are with reference to Eq. (49). The last contraction $C_0$ in panel (d) gives the resulting scalar operator in the state space $K_2$.

Returning to the example in Eq. (49), the innermost cell $C_2$ occurs at nested *contraction level* 2. It is contracted first. Since `S` does not have any `itags` assigned yet (after all, it can be applied to any site), they are assigned *on the fly* by the provided 'hint' whose semantics is along the lines of a simplified *regular expression*: by specifying `'-op:..'`, the tensor `S` is assumed to be an operator, i.e., of rank 2 or 3, that acts on a particular leg on the *other* paired up tensor in the contraction (here `A` in $C_2$). By looking for an `itag` that *starts with* `s` (denoted by `^s`; see C++ `std::regex` for more on this syntax), in the present case, this is sufficient to uniquely identify leg `s02` on `A`. In this sense, the syntax in the contraction internally assigns on the fly the `itags` {`'s02'`,`'s02*'`,`'op'`} to the operator `S` for this particular contraction. Based

on these `itags` then, QSpace auto-contracts this spin operator onto `s02` of `A`. The temporary `itags` in `S` that are not contracted, are inherited by and kept with the temporary tensor $C_2$.

Given $C_2$ as a temporary object in memory, the next cell to be contracted is $C_1$, at nested contraction level 1. Analogous to $C_2$, this now contracts `S` onto `K01`. More precisely, because of the presence of the `'*'` option, this applies the conjugate, and hence effectively the Hermitian conjugate and thus the dagger of `S` onto `K01`. The subsequent auto-contraction in $C_1$ also contracts the assigned default operator `itag` `'op'` for both instances of `'S'`. Overall, therefore this contracts $S_1^\dagger \cdot S_2$.

The last contraction $C_0$ at *base level* 0 uses the temporary QSpace tensor $C_1$. The result out of $C_0$ is returned and assigned to `S12` (technically, $C_0$ is already generated in the memory space of the output `S12` to avoid copying objects). For this last contraction, however, all indices can be paired up, and hence auto-contraction would contract them all. To avoid this, QSpace introduces the 'not' semantics `'!...'`. For example, here by specifying `'!2..'` after the first operator, this indicates that index 2 of that operator (here referring to the combined state space `K02`) *shall not* be contracted and thus left open, even though the `itags` could be paired up, in principle. Finally, the trailing `*` indicates also to take the conjugate tensor of `A`. This corresponds to bra states, and therefore concludes the calculation of the matrix elements in the combined state space `K02` of sites 1 and 2.

Completely analogous to the spin interaction above, one can also compute the matrix elements for the fermionic hopping across sites 1 and 2, $T_{12} \equiv f_1^\dagger \cdot f_2 \rightarrow \mathsf{F}_1^* \cdot (\mathsf{ZF})_2$,

```
29  >> T12 = contract(A,'!2*',{F,'-op:K01','*',{A,(Z*F),'-op:^s'}});                    (50)
```

This contraction is the same as in (48), except for the operators acting on the two sites, substituting $(S,S) \rightarrow (F, Z*F)$. Here by having fermionic operators, one needs to incorporate fermionic signs [33, 50]. As a short remnant from a Jordan Wigner string for the nearest-neighbor hopping here, the fermionic parity operator `Z` is applied onto the second `F` operator. In terms of fermionic order, this adopts the convention that the first site acts on the vacuum state *first*, followed by the second site, etc. Hence similar to (42), as $f_1$ needs to be pulled past any $f_2^\dagger$ when computing matrix elements in Fock space, it leaves a trail of parity operators (here to be applied onto $f_2$ from the left). In the pictorial representation of Fig. 9, the crossing [33, 50] of the operator line `op` with $\sigma_2$ is precisely the location where the parity gate `Z` needs to be applied if the local operator has a fermionic character. This results in $S \rightarrow ZF$ for the operator acting on site $\sigma_2$. Adding the Hermitian conjugate to $T_{12}$, one obtains a typical Hamiltonian term

```
30  >> H12 = T12 + T12'
31
32      Q:   2x { 3 x 2 }  having 'A,SU2'  { K02, K02* }
33    data:  2-D double (448)      7 x 7 => 11 x 11  @ norm = 4
34
35      1.  2x2        |   2x2 [ -1 1 ; -1 1 ]      32 b      {√2}
36      2.  3x3        |   1x1 [  0 0 ;  0 0 ]      72 b
37      3.  2x2        |   2x2 [  1 1 ;  1 1 ]      32 b      {√2}
```

It is hermitian, as can be explicitly checked by computing `norm(H12-H12')`, which yields `0`. When looking at the overall dimensions of `H12`, one realizes a subtlety: the combined Hilbert space for two spinful fermionic levels has a dimension of 16, yet `H12` shows a dimension of `11 x 11` in line 33. Apparently, 5 out of 16 states went 'missing'. The only way this can occur is that these belong to symmetry sectors of `H12` that represent zero blocks, i.e., have Frobenius norm below or comparable to numerical double precision noise (individual states cannot be missing within otherwise present symmetry sectors). For the efficiency of a tensor library, such zero-blocks are not stored by default. They are kept nevertheless, however, in QSpace

for scalar operators (cf. `skipZerosQS`). With diagonal zero blocks absent here, nevertheless, these sectors did not arise out of the contraction itself. Indeed, this is a consequence of the symmetry sectors present in the input tensors of the contraction.

From the one-particle picture, there are two spin-degenerate single-particle levels for the Hamiltonian `H12` above at energies $\varepsilon_\pm = \pm 1$. When building the many-body state space for this, the ground state represents the doubly-filled level $\varepsilon_-$ at energy $E_0 = -2$. It corresponds to a half-filled state and hence belongs to the symmetry sector $q = $ [0 0] (line 36). Adding or removing particles then gives the many-body excitation spectrum $E \in \{-2, -1, 0, 1, 2\}$ (e.g., see first column in lines 64–73 below). Here the completely empty or filled sectors ([±2 0], both 1-dimensional) are missing in `H12`, since indeed, these have hopping matrix elements equal to zero. By comparing to the symmetry sectors on `A` in lines 7–16 [or equivalently, but more readable, by comparing to the output of `E2` in line 38 below], one finds that the remaining symmetry sector that is missing is ([0 2]). One also quickly realizes why: this is the triplet state at half-filling (having $S = 2/2 = 1$). Indeed, with an up-spin on each site, spin-preserving hopping is impossible. This adds up to the total of $1 + 1 + 3 = 5$ 'missing' states, or correspondingly, 3 symmetry sectors. They may be added to `H12` by an infinitesimal (with the value zero also permitted) to also include all-zero eigenvalue blocks. Since `H12` has non-diagonal RMTs, eigenvalue decomposition yields,

```
38  >> E2=getIdentity(A,2);
39     [U12,E12,I12] = eig( H12 + 0*E2 );
```

Here `eig` is a wrapper to the MEX routine `eigQS`. Line 39 adds the identity operator of the full state space on leg 2 in `A`, i.e., `K02`, with weight zero. Being a scalar operator, the resulting diagonal zero-blocks are kept. The reference to the *A*-tensor ensures a complete state space by construction. The output of the above can be inspected, as usual, by simply typing the object's name,

```
40  >> U12
41      Q: 2x { 6 x 2 }  having 'A,SU2'  { K02, K02* }
42    data: 2-D double (784)      10 x 10 => 16 x 16  @ norm = 4
43
44      1.  1x1        |    1x1 [ -2 0 ; -2 0 ]          1.
45      2.  2x2        |    2x2 [ -1 1 ; -1 1 ]       32 b     {√2}
46      3.  3x3        |    1x1 [  0 0 ;  0 0 ]       72 b
47      4.  1x1        |    3x3 [  0 2 ;  0 2 ]          1.   {√3}
48      5.  2x2        |    2x2 [  1 1 ;  1 1 ]       32 b     {√2}
49      6.  1x1        |    1x1 [  2 0 ;  2 0 ]          1.
50
51  >> E12
52      Q: 2x { 6 x 2 }  having 'A,SU2'  { K02, K02* }
53    data: 2-D double (704)      6 x 10 => 10 x 16  @ norm = 4
54
55      1.  1x1        |    1x1 [ -2 0 ; -2 0 ]          0.
56      2.  1x2        |    2x2 [ -1 1 ; -1 1 ]       16 b     {√2}
57      3.  1x3        |    1x1 [  0 0 ;  0 0 ]       24 b
58      4.  1x1        |    3x3 [  0 2 ;  0 2 ]          0.   {√3}
59      5.  1x2        |    2x2 [  1 1 ;  1 1 ]       16 b     {√2}
60      6.  1x1        |    1x1 [  2 0 ;  2 0 ]          0.
61
62  >> I12.ee
63
64     -2.0000     1.0000
65     -1.0000     2.0000
```

```
66    -1.0000    2.0000
67     0.0000    1.0000
68     0.0000    1.0000
69     0.0000    3.0000
70     0.0000    1.0000
71     1.0000    2.0000
72     1.0000    2.0000
73     2.0000    1.0000
```

The columns of `U12` encode the eigenstates that bring `H12` into diagonal form `E12`. The latter is stored in compact diagonal format (storing the diagonal only for each RMT as a row vector, as seen by the RMT dimensions in the second column). The compact diagonal format can be expanded to a regular diagonal matrix via `diag(E12)`. As seen from `E12`, there are three symmetry sectors that have zero blocks (all of them containing a single multiplet, which thus prints the `0.` to the right).

The full eigenspectrum is provided in plain numeric format with the info structure `IS` returned as third argument. The first column in `I12.ee` in lines 64–73 shows the many-body spectrum $E \in \{-2, -1, 0, 1, 2\}$ as discussed above. Each row represents a particular multiplet. The combined multiplet dimension over all symmetries, i.e., the eigenenergy's degeneracy is specified in the second column. In the absence of non-abelian symmetries, the second column is omitted since in that case it would contain only trivial `1`'s. In the example here, the second column adds up to 16. This confirms the full Hilbert space dimension.

### 5.1.1 Changing local state space or symmetries

In the above discussion, the overall symmetry was specified exactly once, namely in Eq. (31) which defines the type of 'site' to use. Since symmetries are global, all symmetries to be included need to be defined there once and for all. If one were to switch to a different symmetry combination, line (31) is the only one that needs to be changed for the entire subsequent discussion. For example, one may use the same fermionic model, yet with `NC=1` $\rightarrow$ 3 fermionic channels (flavors),

```
1  [F,Z,S,IS]=getLocalSpace('FermionS','Acharge,SU2spin,SUNchannel','NC',3);
2  [F,Z,S,IS]=getLocalSpace('FermionS','Acharge,SU2spin,SU3channel');    % equivalent
```

The local state space of a site now consists of $d_{\text{loc}} = 4^3 = 64$ states which can be reduced to an effective $d_{\text{loc}}^* = 10$ multiplets, e.g., as seen by inspecting the identity operator [cf. (32)],

```
3  >> IS.E
4
5     Q:  2x {10 x 4 }  having 'A,SU2,SU3'  { +- }
6   data:  2-D double (1120)      10 x 10 => 64 x 64  @ norm = 8
7
8      1.  1x1      |    1x1    1x1 [ -3 0 00 ; -3 0 00 ]        1.
9      2.  1x1      |    2x2    3x3 [ -2 1 10 ; -2 1 10 ]        1. {√6}
10     3.  1x1      |    1x1    6x6 [ -1 0 20 ; -1 0 20 ]        1. {√6}
11     4.  1x1      |    3x3    3x3 [ -1 2 01 ; -1 2 01 ]        1. {√9}
12     5.  1x1      |    2x2    8x8 [  0 1 11 ;  0 1 11 ]        1. {√16}
13     6.  1x1      |    1x1    6x6 [  1 0 02 ;  1 0 02 ]        1. {√6}
14     7.  1x1      |    4x4    1x1 [  0 3 00 ;  0 3 00 ]        1. {√4}
15     8.  1x1      |    3x3    3x3 [  1 2 10 ;  1 2 10 ]        1. {√9}
16     9.  1x1      |    2x2    3x3 [  2 1 01 ;  2 1 01 ]        1. {√6}
17    10.  1x1      |    1x1    1x1 [  3 0 00 ;  3 0 00 ]        1.
```

Here every symmetry sector on each of the `2` legs in the total of `10` records carries `4` symmetry labels $q \equiv (q_1, q_2, q_3, q_4) \equiv (q_1\ q_2\ q_3 q_4)$, as also indicated with the dimensions `2x {10 x 4}` in line `5`. Based on the order of symmetries requested with `getLocalSpace`, the first symmetry label describes the filling relative to half-filling. Given `NC=3` flavors, this has the range $q_1 \in \{-3, -2, \dots, 3\}$ from empty to completely filled, respectively. The second symmetry label describes SU(2) spin, $q_2 = 2S$, with the largest spin multiplet given by $q_2 = 3$ (line `14`), i.e., spin $S = 3/2$. It derives from the spin-half for each half-filled level, and hence has combined symmetry labels [0 3 00]. The last two symmetry labels ($q_3 q_4$) describe the SU(3) channel or flavor symmetry. The largest SU(3) multiplet above $(11) \equiv \mathbf{8}$ in line `12` contains 8 states (hence referred to as octet; for SU(3) this is also the adjoint representation). Having combined symmetry labels [0 1 11], again this describes states at half-filling. With `NC=3` being odd, half-filling necessarily has a non-zero spin, here $2S = 1$. Hence the combined multiplet dimension in [0 1 11] is $2 \times 8 = 16$ states (see CGT dimensions in line `12`).

The `QSpace` displays as the ones above are just for information. Their understanding requires a rudimentary understanding of the symmetries used, including their labeling structure. The display is generated automatically by `QSpace`. Looking more closely at the individual entries permits simple consistency checks like the ones above. This ensures that one has a good understanding of the tensors under consideration.

Once a single site is defined with the symmetries employed, one can proceed identically to the earlier case of a single flavor to build a many-body state space. For example, two sites can be combined identically as in (47)

```
18  >> A=getIdentity(Z,Z,[1 3 2]);
19  >> A=setitags(A,{'K01','K02','s02'})                                          (51)
```

In the present case, however, this results in

```
20  A =
21
22     Q:  3x {258 x 4 }  having 'A,SU2,SU3'  A-matrix,  { K01, K02*, s02 }
23  data:  3-D double (38.08k)      10 x 260 x 10 => 64 x 4,096 x 64  @ norm = 64
24
25     1.  1x1      |    1x1x1  1x1x1 [ -3 0 00 ; -6 0 00 ; -3 0 00 ]       1.
26     2.  1x2      |    1x2x2  1x3x3 [ -3 0 00 ; -5 1 10 ; -2 1 10 ]   16 b    {√6}
27     3.  1x2      |    2x2x1  3x3x1 [ -2 1 10 ; -5 1 10 ; -3 0 00 ]   16 b    {√6}
28     4.  1x1      |    2x1x2  3x3x3 [ -2 1 10 ; -4 0 01 ; -2 1 10 ]       1. {√3}
29     5.  1x3      |    1x1x1  1x6x6 [ -3 0 00 ; -4 0 20 ; -1 0 20 ]   24 b    {√6}
30     :   ...
31   117.  1x8x1 @2  | 2x1x2x1 8x8x8x2 [  0 1 11 ;  0 0 11 ;  0 1 11 ] 128 b   {√8}
32   138.  1x12x1 @2 | 2x3x2x1 8x8x8x2 [  0 1 11 ;  0 2 11 ;  0 1 11 ] 192 b   {√24}
33     :   ...
34   257.  1x2      |    1x2x2  1x3x3 [  3 0 00 ;  5 1 01 ;  2 1 01 ]   16 b    {√6}
35   258.  1x1      |    1x1x1  1x1x1 [  3 0 00 ;  6 0 00 ;  3 0 00 ]       1.
```

As compared to the *A*-tensor for `NC=1` in (47) and its output following line `3`, the fusion of two sites here leads to considerably more entries (258 vs. 10 earlier). By default, the listing is truncated, strongly so in the present case, skipping records 6–256 in lines `30`–`33`, except for two largest entries, either in size or outer multiplicity. Lines `31`–`32` show records 117 and 138, both of which are entries with outer multiplicity $M = 2$, denoted via "`@2`" with the dimensions of the RMTs. This outer multiplicity is also reflected in the trailing dimensions of the respective CGTs [e.g., having a trailing "`x2`" for the CGTs for SU(3), as in `8x8x8x2`, while having no OM with the CGTs for SU(2), hence showing a trailing "`x1`" with their dimensions, as in `2x1x2x1`]. To display all records in `A`, one can type `display(A,'-f')`. The overall tensor dimensions in the header (line `23`) are as expected, fusing $64 \times 64 \to 4,096$ states. In terms of multiplets,

this fuses $10 \times 10 \to 260$ multiplets, total which is about an order of magnitude smaller. From the header in line 23 still, the norm is given by $\|A\| = \sqrt{4,096} = 64$ also reflecting the fused state space dimension. While from the display of the identity for a single site in lines 3–17 all input multiplets have unique symmetry labels showing an RMT size of `1x ...`, record 138 (line 32) here reports 12 multiplets in the fused sector [0 2 11]. This combines a spin $S = 1$ with an SU(3) multiplet (11), such that these multiplets contain $3 \times 8 = 24$ states each (cf. also *w*-factor to the very right).

The other commands earlier with `NC=1`, such as the spin interaction or fermionic hopping, can be used here also for `NC=3` *identically* without any change. Hence with the symmetries only specified at the very beginning of a setup with `getLocalSpace`, the actual tensor network commands can be implemented the same way irrespective of the symmetry setting, whether abelian or non-abelian symmetries were used, whether multiple symmetries are used in parallel, or just individual ones, or no symmetry at all. Hence once the local state space with the desired symmetries is defined, the subsequent code can proceed nearly as if there had been no symmetries at all. This allows one to focus on tensor network algorithms without having to worry much about symmetries, while fully exploiting complex symmetry settings, nevertheless.

Further increasing the symmetry for `NC=3` in line 1 above to symplectic (e.g., by having particle/hole symmetry on a bipartite lattice as well as SU(NC) channel symmetry [8]),

```
36    [F,Z,S,IS]=getLocalSpace('FermionS','SU2spin,SpNchannel','NC',3);
37    [F,Z,S,IS]=getLocalSpace('FermionS','SU2spin,Sp6channel');    % equivalent    (52)
```

As seen from the fermionic parity (or equally from the identity in `IS.E`),

```
38   Z =
39        Q:  2x { 4 x 4 }  having 'SU2,Sp6'  { +- }
40      data:  2-D double (448)      4 x 4 => 64 x 64  @ norm = 8
41
42        1.  1x1      |    4x4    1x1 [ 3 000 ; 3 000 ]        -1.  {√4}
43        2.  1x1      |    3x3    6x6 [ 2 100 ; 2 100 ]         1.  {√18}
44        3.  1x1      |    2x2  14x14 [ 1 010 ; 1 010 ]        -1.  {√28}
45        4.  1x1      |    1x1  14x14 [ 0 001 ; 0 001 ]         1.  {√14}
```

in the symplectic case the local state space of $d = 64$ states has been reduced to an effective $d^* = 4$ multiplets. There are still a total of four symmetry labels here to identify a multiplet, $q = (q_1\, q_2 q_3 q_4)$, but the split up changed. The first label $q_1 = 2S$ now specifies the SU(2) spin multiplet. Since Sp(6) is a rank $\mathfrak{r} = 3$ symmetry, it carries the remaining three labels $(q_2 q_3 q_4)$. The scalar `000` $\equiv$ **1** is shown with line 42, the defining `100` $\equiv$ **6** in line 43. The largest Sp(6) muliplet occurs with line 45, namely `001` $\equiv$ **14**. Note that Sp(6), like the leading spin SU(2), is self-dual. The annihilation operator

```
46   F =
47        Q:  3x { 6 x 4 }  having 'SU2,Sp6'  operator, { +-- }
48      data:  3-D double (672)      4 x 4 x 1 => 64 x 64 x 12  @ norm = √384
49
50        1.  1x1      |  3x4x2   6x1x6 [ 2 100 ; 3 000 ; 1 100 ]      1.15470  {√18}
51        2.  1x1      |  4x3x2   1x6x6 [ 3 000 ; 2 100 ; 1 100 ]      2.44949  {√4}
52        3.  1x1      |  2x3x2  14x6x6 [ 1 010 ; 2 100 ; 1 100 ]     -1.73205  {√28}
53        4.  1x1      |  3x2x2  6x14x6 [ 2 100 ; 1 010 ; 1 100 ]     -2.16025  {√18}
54        5.  1x1      |  1x2x2 14x14x6 [ 0 001 ; 1 010 ; 1 100 ]     -2.44949  {√14}
55        6.  1x1      |  2x1x2 14x14x6 [ 1 010 ; 0 001 ; 1 100 ]      1.73205  {√28}
```

now has become an irop with (`NC=3`) $\times\, 4 = 12$ components (line 48) that transforms like a single multiplet in the symmetry sector `[1 100]`, consistently also of dimension $2 \times 6 = 12$.

Each of these components contributes equally to the norm $\|F\|^2 = \text{tr}(F^\dagger F) = 12 * 2^{6-1} = 384$ (line 48). The SU(2) spin operator acts trivially in the symplectic sector having `000`,

```
56  S =
57        Q:  3x { 3 x 4 }  having 'SU2,Sp6'  operator,  { +-- }
58      data:  3-D double (336)      3 x 3 x 1 => 50 x 50 x 3  @ norm = √72
59
60        1.  1x1        |  4x4x3   1x1x1 [ 3 000 ; 3 000 ; 2 000 ]     1.936491  {√4}
61        2.  1x1        |  3x3x3   6x6x1 [ 2 100 ; 2 100 ; 2 000 ]    -1.414214  {√18}
62        3.  1x1        |  2x2x3 14x14x1 [ 1 010 ; 1 010 ; 2 000 ]    -0.866025  {√28}
```

The spin sectors $S = 0$ are absent since these correspond to zero blocks. The norm of the individual RMTs in lines 60–62 already reflect the Casimir for the respective local spin, having values $(\sqrt{\frac{15}{4}}, -\sqrt{2}, \sqrt{\frac{3}{4}})$, and thus $\pm\sqrt{S(S+1)}$ for $q_1 = 2S \in \{3, 2, 1\}$. This emphasizes the importance of the relative weight across the RMTs. The relative signs are also equally important.

The tensor product space of two sites can proceed the same way as in the initial setting in (47). The *A*-tensor now becomes

```
63  A =
64        Q:  3x {61 x 4 }  having 'SU2,Sp6'  A-matrix,  { K01, K02*, s02 }
65      data:  3-D double (7.828k)      4 x 61 x 4 => 64 x 4,096 x 64  @ norm = 64
66
67        1.  1x4        |  1x1x1   14x1x14 [ 0 001 ; 0 000 ; 0 001 ]     32 b
68        2.  1x4        |  2x1x2   14x1x14 [ 1 010 ; 0 000 ; 1 010 ]     32 b
69        3.  1x4        |  3x1x3    6x1x6 [ 2 100 ; 0 000 ; 2 100 ]     32 b
70        4.  1x4        |  4x1x4    1x1x1 [ 3 000 ; 0 000 ; 3 000 ]     32 b
71        5.  1x1        |  1x1x1 14x84x14 [ 0 001 ; 0 002 ; 0 001 ]            1.  {√84}
72        6.  1x2        |  2x1x2 14x14x14 [ 1 010 ; 0 010 ; 1 010 ]     16 b     {√14}
73        :   ...
74       35.  1x6        |  3x3x3   6x14x6 [ 2 100 ; 2 010 ; 2 100 ]     48 b     {√42}
75       36.  1x6        |  4x3x2 1x14x14 [ 3 000 ; 2 010 ; 1 010 ]     48 b     {√42}
76        :   ...
77       60.  1x2        |  4x6x3    1x6x6 [ 3 000 ; 5 100 ; 2 100 ]     16 b     {√36}
78       61.  1x1        |  4x7x4    1x1x1 [ 3 000 ; 6 000 ; 3 000 ]            1.  {√7}
```

This reduces the full Hilbert space for the two sites from $D = 64^2 = 4096$ states to an effective $D^* = 61$ multiplets across 23 symmetry sectors [the latter can be seen from `getIdentity(A,2)`]. The matrix elements in (48) or (50) can be computed as previously without any change.

## 5.2  Swap operator for two spins $S = 1$

The following proceeds similar to the previous two-site examples [e.g., see (47)], yet switches from a fermionic state space for a site to a spin $S = 1$ in SU(2),

```
1    [S,IS]=getLocalSpace('Spin','SU2',2);   % 2S=2 => S=1
2    A=getIdentity(S,S,[1 3 2]);             % may also use S → IS.E instead
3    A=setitags(A,{'K01','K02','s02'});

4  >> A
5
6        Q:  3x { 3 x 1 }  having 'SU2'  A-matrix,  { K01, K02*, s02 }
7      data:  3-D double (336)      1 x 3 x 1 => 3 x 9 x 3  @ norm = 3
8
9        1.  1x1        |  3x1x3 [ 2 ; 0 ; 2 ]          1.
10       2.  1x1        |  3x3x3 [ 2 ; 2 ; 2 ]          1.  {√3}
11       3.  1x1        |  3x5x3 [ 2 ; 4 ; 2 ]          1.  {√5}
```

The state space of two sites as described by A results in a total spin $S \in \{0, 2, 4\}/2 = \{0, 1, 2\}$, as expected. The following then computes the 'interaction' $X$ that describes a plain swap of the state space of two sites, also known as the permutation operator. This operator $X$ is Hermitian with eigenvalues $\pm 1$ since $X^2 = 1$. Its matrix elements can be obtained as follows,

```
12   A_= untag(A);                    % required for the next line
13   X = contract(A_,'13*',A_,'31');  % swap operator
```

```
14   >> X
15      Q:  2x { 3 x 1 }  having 'SU2'  { +- }
16    data:  2-D double (336)      3 x 3 => 9 x 9  @ norm = 3
17
18      1.  1x1        |    1x1 [ 0 ; 0 ]          1.
19      2.  1x1        |    3x3 [ 2 ; 2 ]         -1.   {√3}
20      3.  1x1        |    5x5 [ 4 ; 4 ]          1.   {√5}
```

This swap operator $X$ is already diagonal with eigenvalues $\pm 1$ as shown with the RMTs. The itags in the contracted local indices were temporarily removed (untag in line 12), since the swap operator contracts legs with otherwise *different* itags [see Fig. 10(a)]. Leaving the itags in place would result in an error due to itag mismatch [cf. Fig. 10(c)]. The contraction in line 13 uses compact index notation. It contracts legs 1 and 3, i.e., $K_1 \equiv \sigma_1$ and $\sigma_2$ of the conjugate of the first tensor with legs 3 and 1, i.e., $\sigma_2$ and $\sigma_1$ and thus reverse order, on the second tensor (tensor conjugation leaves the order of legs intact).

Now an interaction of two spin-$S$ sites can be decomposed into powers $(S_1^\dagger \cdot S_2)^n$ of the spin-spin interaction, which forms a (non-orthonormalized) operator basis. Clearly, by only

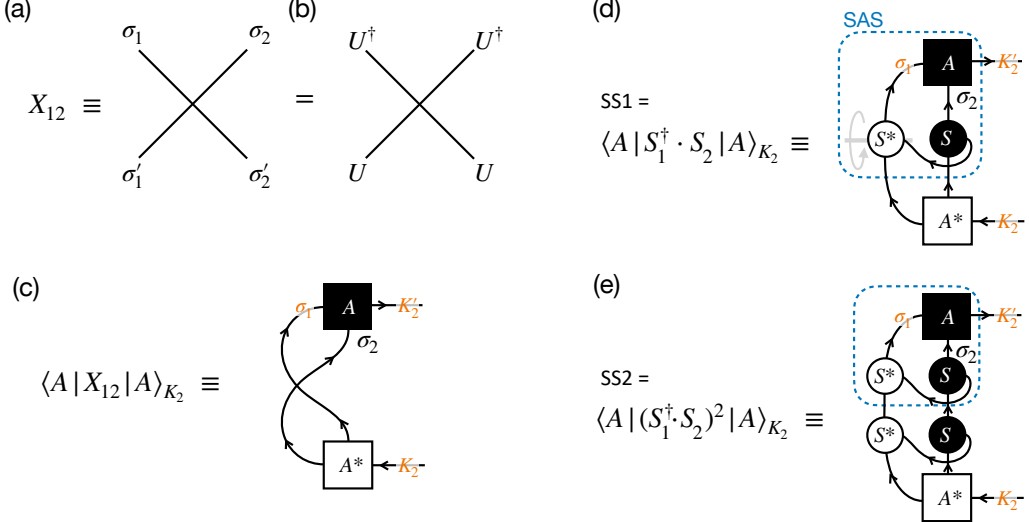

Figure 10: (a) The swap operator $X_{12}$ acting on two sites $i \in \{1, 2\}$ with respective local state spaces $\sigma_i$. (b) The swap operator is invariant under arbitrary unitary transformations on the local state spaces since $UU^\dagger = 1$ annihilate each other along the diagonal lines. (c) The matrix elements of the swap operator expressed in the combined state space as described by an $A$-tensor. (d-e) The matrix elements of $(S_1^\dagger \cdot S_2)^n$ for $n = 1$ and 2, respectively. The contraction in (d) is the same as in Fig. 9, except that spin operators are applied here. Similarly, also the conjugate operator for the first site is drawn as the vertical mirror image as indicated by the gray circular arrow and line. The intermediate object in the contraction indicated by the blue dashed box in (d), referred to as SAS in the text (line 21), can be reused in (e).

having $2S+1$ RMTs, each of which is given by a single number, one reaches a complete basis for $n \leq 2S$. Hence powers $n \geq 2S+1$ can be expressed in terms of smaller powers. For example, for spin-half sites the identity and the spin operator already exhaust the complete set of local operators, since $2^2 = 1 + 3$. For the swap operator here for $S = 1$ sites, this implies

$$X = \sum_{n=0}^{2} a_n \, (S_1^\dagger \cdot S_2)^n \, . \tag{53}$$

with coefficients $a_n$ to be determined. The operator for $n = 0$ refers to the identity which has the same QSpace structure as displayed with X in line 14, except that it has all +1's in the right column for the RMTs. The case $n = 1$ gives the Heisenberg interaction SS1 [cf. Fig. 10(d)],

```
21  >> SAS = contract(S,'-op:K01','*',{A,S,'-op:^s'});
22  >> SS1 = contract(A,'!2*',SAS)
23
24      Q:  2x { 3 x 1 }  having 'SU2'  { K02, K02* }
25    data:  2-D double (336)      3 x 3 => 9 x 9  @ norm = √12
26
27      1.  1x1          |    1x1 [ 0 ; 0 ]           -2.
28      2.  1x1          |    3x3 [ 2 ; 2 ]           -1.  {√3}
29      3.  1x1          |    5x5 [ 4 ; 4 ]            1.  {√5}
```

and $n = 2$ the quadrupolar term SS2 [cf. Fig. 10(e)],

```
30  >> SS2 = contract(A,'!2*',{S,'-op:K01','*',{SAS,S,'-op:^s'}})
31
32      Q:  2x { 3 x 1 }  having 'SU2'  { K02, K02* }
33    data:  2-D double (336)      3 x 3 => 9 x 9  @ norm = √24
34
35      1.  1x1          |    1x1 [ 0 ; 0 ]            4.
36      2.  1x1          |    3x3 [ 2 ; 2 ]            1.  {√3}
37      3.  1x1          |    5x5 [ 4 ; 4 ]            1.  {√5}
```

Now by collecting the RMTs for $n = 0, 1, 2$, as well as for X in lines 18–20 into columns, one can determine the coefficients $a_n$ in Eq. (53),

$$\begin{pmatrix} 1 \\ -1 \\ 1 \end{pmatrix} \stackrel{!}{=} \begin{pmatrix} 1 & -2 & 4 \\ 1 & -1 & 1 \\ 1 & 1 & 1 \end{pmatrix} \begin{pmatrix} a_0 \\ a_1 \\ a_2 \end{pmatrix} \quad \Rightarrow \quad \begin{pmatrix} a_0 \\ a_1 \\ a_2 \end{pmatrix} = \begin{pmatrix} -1 \\ 1 \\ 1 \end{pmatrix} \tag{54}$$

and therefore

$$X = -1 + S_1^\dagger \cdot S_2 + (S_1^\dagger \cdot S_2)^2 \, . \tag{55}$$

A simple countercheck confirms the coefficients in Eq. (55), indeed,

```
38  >> E2 = getIdentity(A,2);
39  >> X_ = -E2 + SS1 + SS2
40
41      Q:  2x { 3 x 1 }  having 'SU2'  { +- }
42    data:  2-D double (336)      3 x 3 => 9 x 9  @ norm = 3
43
44      1.  1x1          |    1x1 [ 0 ; 0 ]            1.
45      2.  1x1          |    3x3 [ 2 ; 2 ]           -1.  {√3}
46      3.  1x1          |    5x5 [ 4 ; 4 ]            1.  {√5}
```

with norm(X_-X) returning numerical double precision noise $< 10^{-14}$. This not being exactly zero is due to the square root factors shown to the right. For this reason, X==X_ returns false. The QSpace routine sameas(X,X_) based on norm(X_-X) returns true.

**Physical interpretation** By using a simple swap operator on local state spaces of dimension $d_{loc} = 3$, an arbitrary 3-dimensional unitary $U$ can be inserted as $UU^\dagger$ into any of the two lines in Fig. 10(a). This takes the interpretation of transforming the local state space for each site. However, with $X_{12} \to (U \otimes U)X_{12}(U^\dagger \otimes U^\dagger) = (UU^\dagger \otimes 1)(1 \otimes UU^\dagger)X_{12} = X_{12}$ being identical to $X$ [most easily seen by considering the pictorial representation in Fig. 10(a)], this implies that the swap operation in the above Hamiltonian actually has an enlarged symmetry, namely SU($d_{loc}$). For the case of the spin $S = 1$ here, this is the well-known SU(3) symmetric point of the bilinear-biquadratic Heisenberg model for $S = 1$ at the phase boundary of the Haldane phase [51, 52].

## 5.3 Tweaking local state space

The local state space of a site is set up via a call to getLocalSpace, which has many standard symmetry setups already implemented. Nevertheless, situations that may go beyond what is already provided by getLocalSpace can arise. The typical way to proceed then is to tweak its output starting from a symmetry setup that may be considered closest to the target setup.

To be specific, consider for example SU(2) spin-halfs with an additional two symmetric orbital flavors, i.e., $SU(2)_{spin} \otimes SU(2)_{orbital}$. This does not exist ready-made in getLocalSpace. Two ways to deal with this situation are discussed below.

### 5.3.1 Project fermionic setup to spin model and complete operator basis

Starting from a spinfull fermionic model with NC=2 orbital flavors

```
47    [F,Z,S,IS]=getLocalSpace('FermionS','Acharge,SU2spin,SU2channel');                    (56)
```

the state space can be inspected by Z or IS.E,

```
1   >> E=IS.E
2
3        Q:  2x { 6 x 3 }  having 'A,SU2,SU2'  { +- }
4     data:  2-D double (672)       6 x 6 => 16 x 16  @ norm = 4
5
6        1.  1x1        |    1x1    1x1 [ -2 0 0 ; -2 0 0 ]            1.
7        2.  1x1        |    2x2    2x2 [ -1 1 1 ; -1 1 1 ]            1.  {√4}
8        3.  1x1        |    1x1    3x3 [  0 0 2 ;  0 0 2 ]            1.  {√3}
9        4.  1x1        |    3x3    1x1 [  0 2 0 ;  0 2 0 ]            1.  {√3}
10       5.  1x1        |    2x2    2x2 [  1 1 1 ;  1 1 1 ]            1.  {√4}
11       6.  1x1        |    1x1    1x1 [  2 0 0 ;  2 0 0 ]            1.
```

Looking for the spin $S = 1/2$ sector at a filling of $n = 1$ particle on the site, this corresponds to $q_1 = -1$ relative to half-filling. This contains a single multiplet in line 7 which is thus the one of interest. It can be projected by just picking this record from the identity using getsub,

```
12  >> q0=-1; i=find(Z.Q{1}(:,1)==q0);  % finds record i=2. (line 7 above)
13  >> E=getsub(E,i)                     % reduces E to record(s) i
14
15       Q:  2x { 1 x 3 }  having 'A,SU2,SU2'  { +- }
16    data:  2-D double (112)       1 x 1 => 4 x 4  @ norm = 2
17
18       1.  1x1        |    2x2    2x2 [ -1 1 1 ; -1 1 1 ]            1.  {√4}
```

Since now only a trivial single abelian symmetry sector is present in terms of charge, it can be skipped (removed) as follows,

```
19  >> E=rmAbelian(E,1)
20
21  E =
22      Q:  2x { 1 x 2 }  having 'SU2,SU2'  { +- }
23   data:  2-D double (112)      1 x 1 => 4 x 4  @ norm = 2
24
25      1.  1x1        |    2x2    2x2 [ 1 1 ; 1 1 ]              1.  {√4}
26
27  >> clear F Z
```

The syntax in line 19 removes the symmetry at position 1 from E, i.e., the abelian U(1) charge. Projecting to a single charge sector renders the fermionic operators irrelevant (line 27). The spin operator, finally, can be projected the same way as the identity above,

```
28  >> S=getsub(find(S.Q{1}(:,1)==q0));   % having q0=-1 from line 12 above
29  >> S=rmAbelian(S,1)
30
31  S =
32      Q:  3x { 1 x 2 }  having 'SU2,SU2'  operator,  { +-- }
33   data:  3-D double (112)      1 x 1 x 1 => 4 x 4 x 3  @ norm = √3
34
35      1.  1x1        | 2x2x3  2x2x1 [ 1 1 ; 1 1 ; 2 0 ]    -0.866025  {√4}
```

which transforms like a $S = q/2 = 1$ irop. This is the same spin operator as for a single fermionic flavor [cf. (31), and subsequent display in lines 15ff] that acts trivially, i.e., like an identity in the orbital space.

The pseudo-spin operator in the orbital space can be obtained from a complete operator basis for the local state space. For its construction, one needs to obtain the tensor product space in E with the dual of its state space as obtained by the $1j$ symbol in line 36,

```
36  >> U=getIdentity(E,'-0');
37  >> X=getIdentity(U,1,U,2);
38  >> X=contract(U,'!1*',X,[2 1 3])
39
40      Q:  3x { 4 x 2 }  having 'SU2,SU2'  { +-- }
41   data:  3-D double (448)      1 x 1 x 4 => 4 x 4 x 16  @ norm = 4
42
43      1.  1x1        | 2x2x1  2x2x1 [ 1 1 ; 1 1 ; 0 0 ]        0.5  {√4}
44      2.  1x1        | 2x2x1  2x2x3 [ 1 1 ; 1 1 ; 0 2 ]   0.866025  {√4}
45      3.  1x1        | 2x2x3  2x2x1 [ 1 1 ; 1 1 ; 2 0 ]   0.866025  {√4}
46      4.  1x1        | 2x2x3  2x2x3 [ 1 1 ; 1 1 ; 2 2 ]        1.5  {√4}
```

The tensor product in line 37 derives from the fact that the full operator space of a state space of dimension $d$ is described by $d^2 \equiv d \otimes d$ operators. However, the incoming dual state space, i.e., primed index in X still needs to be reverted by contracting the same $1j$ symbol in line 38, so that X represents a standard operator [cf. Fig. 1]. The trailing permutation [2 1 3] restores the operator index order +--, as seen in line 40. Having a dimension of 1x1x4 in terms of multiplets (line 41), the object X is not an irop, though. Rather, it contains a (direct) sum of four irops that span the complete operator space of a single site. This consists of the four records in the above display, each of which represents an irop with a unique operator $q$-label:

- Record 1. (line 43) transforms like a scalar $q_{\text{op}} = [0\ 0]$. Since this is the only operator of this type and bearing in mind that X represents a complete operator basis, it must be the identity operator up to the scale factor given by RMT value 0.5 shown to the right. Via the tensor-product in getIdentity in line 37 each operator has Frobenius norm 1 which explains this factor, since $\|\frac{1}{2}\mathbb{1}^{(4)}\| = 1$.

- Record 3. (line 45) represents an irop that transforms according to $q_{op} = [2\ 0]$. This identifies it as the spin operator like in line 35 above up to a sign (note that the sign is irrelevant in bilinear products, as long as the *same* operator S is used consistently throughout). Incidentally, in the present setting the value of its RMT 0.866025= $\frac{\sqrt{3}}{2}$ is consistent already with both, the normalization of a spin-half operator ($\hat{S}^2 = \frac{3}{4}\mathbb{1}^{(4)}$), as well as the normalization out of the tensor-product in getIdentity in line 37, having $\|\hat{S}\|^2 = \text{tr}(\frac{3}{4}\mathbb{1}^{(4)}) = 3$ representing three normalized operators, indeed.
- Record 2. (line 44) represents an irop that transforms according to $q_{op} = [0\ 2]$. Therefore this entry describes the pseudo-spin operator in orbital space.
- Record 4. (line 46), finally, represents an irop that transforms according to $q_{op} = [2\ 2]$. As such this is a spin-orbit interaction ST up to a factor 2 when comparing to RMT value of 1.5 to the standard SU(2) spin normalization $\sqrt{(\hat{S}\hat{T})^2} = \sqrt{\hat{S}^2\hat{T}^2} = \frac{3}{4}\mathbb{1}^{(4)}$.

By construction, the above full operator basis in X permits one to extract all (other) irreducible operators within the present symmetry setting. Here this includes the pseudo-spin operator T in orbital space (choosing the same sign convention as in S),

```
47  >> T=-getsub(X,3)
48
49      Q:  3x { 1 x 2 }  having 'SU2,SU2'  { +-- }
50   data:  3-D double (112)      1 x 1 x 1 => 4 x 4 x 3  @ norm = √3
51
52      1.  1x1        |  2x2x3  2x2x1 [ 1 1 ; 1 1 ; 2 0 ]   -0.866025  {√4}
```

The remaining last operator within the present symmetry setting is the spin-orbit term

```
53  >> ST= getsub(X,4)/2;
54
55      Q:  3x { 1 x 2 }  having 'SU2,SU2'  { +-- }
56   data:  3-D double (112)      1 x 1 x 1 => 4 x 4 x 9  @ norm = 1.5
57
58      1.  1x1        |  2x2x3  2x2x3 [ 1 1 ; 1 1 ; 2 2 ]        0.75  {√4}
```

The combination of these four irops that describe spinors of dimension $1+3+3+3\times3 = 16 = 4^2$ thus, indeed, exhausts the operator basis of given local state space.

### 5.3.2 Appending additional symmetries

Alternative to the previous projective approach, one may tweak the output of getLocalSpace by adding other symmetries still. Starting with a plain SU(2) spin-half, one can manually add an additional SU(2) symmetry, here to the identity operator returned by getLocalSpace. For the example above, this starts here with the simpler spin model,

```
1    [S,IS]=getLocalSpace('Spin','SU2',1);   % initial spin-half (q=2S=1)
2    E=addSymmetry(IS.E,'SU2','q',1)
```

```
3  E =
4      Q:  2x { 1 x 2 }  having 'SU2,SU2'  { +- }
5   data:  2-D double (112)      1 x 1 => 4 x 4  @ norm = 2
6
7      1.  1x1        |   2x2    2x2 [ 1 1 ; 1 1 ]           1.  {√4}
```

By default, addSymmetry appends the specified symmetry in the scalar irep $q = 0$. By explicitly specifying $q = 1$ in line 2, this adds the second SU(2) symmetry in the form of an $S = 1/2$

multiplet. It is interpreted here as SU(2) symmetric orbital. The local state space therefore becomes $2 \times 2 = 4$ dimensional as seen in line 5. Since `addSymmetry` always applies a square-root factor of the multiplet dimension added, the identity `E` already has the correct normalization (which matter of fact, is the motivation as to why `addSymmetry` behaves this way).

Next, the spin operators are tweaked. First, this generates the orbital spin operator `T` in line 8 based on `S`, before adapting the spin operator `S` itself in line 9,

```
8    T=addSymmetry(S,'SU2','pos',1,'q',1);   % `spin' operator in orbital sector
9    S=addSymmetry(S,'SU2','q',1);            %  adapts existing spin operator
```

Line 8 uses the option `'pos',1` which inserts the new symmetry at location 1, i.e., *prepends* the symmetry rather than appending it (see `help addSymmetry` for more detailed usage information). The display of the operators above reads

```
10   T =
11       Q:  3x { 1 x 2 }  having 'SU2,SU2'  operator,  { +-- }
12     data:  3-D double (112)       1 x 1 x 1 => 4 x 4 x 3  @ norm = √3
13
14       1.  1x1         |  2x2x1  2x2x3 [ 1 1 ; 1 1 ; 0 2 ]    -0.866025  {√4}
15
16   S =
17       Q:  3x { 1 x 2 }  having 'SU2,SU2'  operator,  { +-- }
18     data:  3-D double (112)       1 x 1 x 1 => 4 x 4 x 3  @ norm = √3
19
20       1.  1x1         |  2x2x3  2x2x1 [ 1 1 ; 1 1 ; 2 0 ]    -0.866025  {√4}
```

which is consistent with the alternative approach earlier. With respect to the operator index (index 3) the above shows that $S$ ($T$) acts like a spin operator in the first (second) symmetry, respectively, thus confirming $\text{SU}(2)_{\text{spin}} \otimes \text{SU}(2)_{\text{orbital}}$ in this order. Similarly to `E` above, the operators `S` and `T` are again also correctly normalized ($\sqrt{3}$ in header lines 12 and 18), since, e.g., $\|S\|^2 = \text{tr}(\, 1^{(2)} \otimes \frac{3}{4} 1^{(2)} \,) = 3$.

## 5.4  Local operators beyond `getLocalSpace`

The routine `getLocalSpace` returns a set of elementary operators that are representative of the chosen site. Many operators can be derived in simple one-liners, as already shown earlier. However, there are exceptions as already partly discussed in the previous section, e.g., when generating the orbital pseudy spin operator `T` in line 47 or the spin-orbit operator `TS` in line 53 from a complete operator basis. This approach is reviewed once more in a more elaborate example for orbital spin operators below, together with an alternative strategy for obtaining local spin operators based on a Schrieffer-Wolff-like construction. Both represent instructive approaches for generating additional local operators.

The following discussion is based on the fermionic state space for `NC=2` orbitals in (56) with a local Hilbert space of $d_{\text{loc}} = 4^2 = 16$ states. The spin operator `S` for the entire fermionic state space as returned by (56) reads

```
21   >> S =
22       Q:  3x { 3 x 3 }  having 'A,SU2,SU2'  operator,  { +-- }
23     data:  3-D double (336)       3 x 3 x 1 => 11 x 11 x 3  @ norm = √12
24
25       1.  1x1         |  2x2x3  2x2x1 [ -1 1 1 ; -1 1 1 ;  0 2 0 ]    -0.866025  {√4}
26       2.  1x1         |  3x3x3  1x1x1 [  0 2 0 ;  0 2 0 ;  0 2 0 ]    -1.41421  {√3}
27       3.  1x1         |  2x2x3  2x2x1 [  1 1 1 ;  1 1 1 ;  0 2 0 ]    -0.866025  {√4}
```

As with the earlier symplectic example in the symplectic case, the norm of the individual RMTs already reflect the Casimir for the respective local spin, having values ($\sqrt{\frac{3}{4}}, \sqrt{2}, \sqrt{\frac{3}{4}}$), and thus $\sqrt{S(S+1)}$ for $q_1 = 2S \in \{1, 2, 1\}$. The normalization as well as the (relative) signs of the records in the irop S are important. While the signs here are all the same, this is not always the case, e.g., as seen in the earlier example. The global sign of any spin operator is fixed after defining the sign of the spin operator in the defining representation. Furthermore, since the spin operator is well-defined in the defining representation, it must be in any other many-body state space derived from it. Hence the spin operator, representing 'total spin', is well-defined and thus unique in any state space.

The spin operator above has irop labels $q_S =$[0 2 0]. Based on (56) with $q = (q_1, q_2, q_3)$, these symmetry labels are $q_1$ for abelian U(1) charge, $q_2 = 2S$ for the spin SU(2), and $q_3 = 2S$ for the orbital SU(2). As the irop labels $q_S$ show, the spin operator acts non-trivially only in the SU(2) spin sector. There it reflects the adjoint representation ($S = 1$) that incorporates the complete set of generators for that symmetry.

**Complete operator basis** Any local operator lives within the complete set of local operators. While constructing the latter may appear overkill for specific local operators, this approach is instructive, nevertheless. The complete operator basis Eop is built starting from the tensor product of the local state space with its dual via a $1j$-tensor (line 28), followed by a contraction of the same $1j$-tensor in the line below [using the local operators out of getLocalSpace as in (56); same construction as in lines 36–38]

```
28  >> U = getIdentity(IS.E,'-0');
29  >> Eop = contract(getIdentity(U,1,U,2),2,U,'2*',[1 3 2])
```

```
30      Q:  3x {60 x 3 }  having 'A,SU2,SU2'  { +-- }
31   data:  3-D double (7.734k)      6 x 6 x 60 => 16 x 16 x 256  @ norm = 16
32
33      1.  1x1x6    |  1x1x1  1x1x1 [ -2 0 0 ; -2 0 0 ;  0 0 0 ]      48 b
34      2.  1x1x6    |  2x1x2  2x1x2 [ -1 1 1 ; -2 0 0 ;  1 1 1 ]      48 b      {√4}
35      3.  1x1x3    |  1x1x1  3x1x3 [  0 0 2 ; -2 0 0 ;  2 0 2 ]      24 b      {√3}
36      4.  1x1x3    |  3x1x3  1x1x1 [  0 2 0 ; -2 0 0 ;  2 2 0 ]      24 b      {√3}
37      5.  1x1x2    |  2x1x2  2x1x2 [  1 1 1 ; -2 0 0 ;  3 1 1 ]      16 b      {√4}
38      6.  1x1      |  1x1x1  1x1x1 [  2 0 0 ; -2 0 0 ;  4 0 0 ]           1.
39      :   ...
40     59.  1x1x6    |  2x1x2  2x1x2 [  1 1 1 ;  2 0 0 ; -1 1 1 ]      48 b      {√4}
41     60.  1x1x6    |  1x1x1  1x1x1 [  2 0 0 ;  2 0 0 ;  0 0 0 ]      48 b
```

By construction, Eop encodes all $(d_{\text{loc}} = 16)^2 = 256$ local operators. It represents 60 irops that transform in a total of 24 different ireps, as seen from getIdentity(Eop,3). If one were to project the local state space to a particular single multiplet, this would generate an effective spin model. Only a global normalization of irops has to be fixed then.

For the full local state space, the spin operator as returned by getLocalSpace must also be contained in Eop. A search for all records in Eop that have the symmetry labels $q_S =$[0 2 0] on the operator index 3 via getsub results in,

```
42  >> S_ = getsub(Eop,[0 2 0],3)
43
44      Q:  3x { 3 x 3 }  having 'A,SU2,SU2'  { +-- }
45   data:  3-D double (384)      3 x 3 x 3 => 11 x 11 x 9  @ norm = 3
46
47      1.  1x1x3    |  2x2x3  2x2x1 [ -1 1 1 ; -1 1 1 ;  0 2 0 ]      24 b      {√4}
48      2.  1x1x3    |  3x3x3  1x1x1 [  0 2 0 ;  0 2 0 ;  0 2 0 ]      24 b      {√3}
49      3.  1x1x3    |  2x2x3  2x2x1 [  1 1 1 ;  1 1 1 ;  0 2 0 ]      24 b      {√4}
```

It describes an operator subspace of three entries (having dimensions `1x1x3` for all RMTs) which is solely due to having three QSpace records here. The spin operator can be built as some linear superposition within that subspace. However, a priori, it is unclear how to obtain the correct coefficients for this linear superposition as in lines 25–27 from the present setting.

The spin operator for the SU(2) spin symmetry is known, of course, because it is explicitly returned by `getLocalSpace`. This has the benefit that one has a reference to compare various approaches to. Suppose one needs the pseudo-spin operator in the local state space above for the orbital degrees of freedom or the intertwined spin-orbit operator. These are no longer available from `getLocalSpace`. For these, one needs to address the problem of how to make the spin operator unique, at least up to a global scale factor. For this purpose, one may target the spin operators more concretely, as discussed next.

**Spin operators from second-order fermionic hopping**    An alternative route to generalized 'spin operators' is physically motivated by Schrieff-Wolff transformations, where a second-order perturbation in the tunneling maps fermionic cotunneling to an effective 'spin model'. There with $\hat{f}_i$ the complete set of local fermionic operators with $i = 1, \ldots, N_f$ indexing all flavors or flavor combinations, one needs to decompose the pool of operators $\{\hat{f}_i^\dagger \hat{f}_{i'}\}$ into irops. Hence rather than building the full operator space for the local state space, one can constrain oneself to the operators obtained out of the bilinear `F`$^\dagger \otimes$`F`, here with a Kronecker rather than a dot-product on the irop indices.

Since `F`$^\dagger$ as a conjugate operator has the irop index ingoing, this procedure is equivalent to the construction of the complete operator basis in lines 28–29, except that one takes the irop space of `F`, instead. Replacing `IS.E` in line 28 by `F` on leg 3, one obtains,

```
50  >> U_ = getIdentity(F,3,'-0');
51     Eop_ = contract(getIdentity(U_,1,U_,2),2,U_,'2*',[1 3 2])

52        Q:  3x { 4 x 3 }  having 'A,SU2,SU2'  { +-- }
53     data:  3-D double (448)      1 x 1 x 4 => 4 x 4 x 16   @ norm = 4
54
55        1.  1x1        |  2x2x1  2x2x1 [ -1 1 1 ; -1 1 1 ;  0 0 0 ]         0.5  {√4}
56        2.  1x1        |  2x2x1  2x2x3 [ -1 1 1 ; -1 1 1 ;  0 0 2 ]    0.866025  {√4}
57        3.  1x1        |  2x2x3  2x2x1 [ -1 1 1 ; -1 1 1 ;  0 2 0 ]    0.866025  {√4}
58        4.  1x1        |  2x2x3  2x2x3 [ -1 1 1 ; -1 1 1 ;  0 2 2 ]         1.5  {√4}
```

The spinfull `NC=2` model here based on (56) has a combined total of $N_f = 2 \times 2 = 4$ flavors. Hence the pool of operators $\{\hat{f}_i^\dagger \hat{f}_{i'}\}$ has $4 \times 4 = 16$ entries. These are cast into three irops here that span the full space of the operator pool, as seen from the overall dimensions `4 x 4 x 16` in line 53. By having RMTs of dimension `1x1[x1]` only, each record now corresponds to an irop that is unique up to normalization.

The tensor `Eop_` still needs to be contracted onto the two irop indices in `F`$^\dagger$`F`. By choosing the SU(2) spin irop in line 56, one obtains the spin operator

```
59  >> x = getsub(Eop_,[0 2 0],3);    % select spin operator having q=[0 2 0]
60     Sop = contract({F,'1*',F,1},'24',x,'21')

61        Q:  3x { 3 x 3 }  having 'A,SU2,SU2'  { +-- }
62     data:  3-D double (336)      3 x 3 x 1 => 11 x 11 x 3   @ norm = √12
63
64        1.  1x1        |  2x2x3  2x2x1 [ -1 1 1 ; -1 1 1 ;  0 2 0 ]   -0.866025  {√4}
65        2.  1x1        |  3x3x3  1x1x1 [  0 2 0 ;  0 2 0 ;  0 2 0 ]    -1.41421  {√3}
66        3.  1x1        |  2x2x3  2x2x1 [  1 1 1 ;  1 1 1 ;  0 2 0 ]   -0.866025  {√4}
```

This is precisely the spin operator returned by `getLocalSpace` in lines 21–27, already also with the correct normalization, as confirmed by `norm(S-Sop)` $\leq 10^{-14}$. The normalization can be easily double-checked, bearing in mind that the spin operator returned by `getLocalSpace` describes the total spin of the local state space. For `NC=2` fermionic channels with $S_i$ the spin operator for channel (orbital) $i$, this leads to $\|S\|^2 = \|S_1 + S_2\|^2 = 2\,\|S_1\|^2\,\mathrm{tr}(\mathbb{1}_2^{(4)}) = 2\cdot(\tfrac{3}{4}2)\cdot 4 = 12$, in agreement with line 62.

The present approach now also allows the extension to other generalized spin operators. For example, one may choose the SU(2) pseudo-spin irop in the orbital sector (line 57),

```
67  >> x = getsub(Eop_,[0 0 2],3);    % select pseudo-spin operator
68     Top = contract({F,'1*',F,1},'24',x,'21')

69        Q: 3x { 3 x 3 }  having 'A,SU2,SU2'  { +-- }
70     data: 3-D double (336)      3 x 3 x 1 => 11 x 11 x 3  @ norm = √12
71
72        1.  1x1       | 2x2x1  2x2x3 [ -1 1 1 ; -1 1 1 ;  0 0 2 ]    -0.866025  {√4}
73        2.  1x1       | 1x1x1  3x3x3 [  0 0 2 ;  0 0 2 ;  0 0 2 ]    -1.41421   {√3}
74        3.  1x1       | 2x2x1  2x2x3 [  1 1 1 ;  1 1 1 ;  0 0 2 ]    -0.866025  {√4}
```

which shows the same RMT values as the previous spin operator `'Sop'`, or the spin-orbital sector in line 58,

```
75  >> x = getsub(Eop_,[0 2 2],3);    % select spin-orbit operator
76     STop = contract({F,'1*',F,1},'24',x,'21') / 2

77        Q: 3x { 4 x 3 }  having 'A,SU2,SU2'  { +-- }
78     data: 3-D double (448)      4 x 4 x 1 => 14 x 14 x 9  @ norm = 3
79
80        1.  1x1       | 2x2x3  2x2x3 [ -1 1 1 ; -1 1 1 ;  0 2 2 ]     0.75    {√4}
81        2.  1x1       | 1x3x3  3x1x3 [  0 0 2 ;  0 2 0 ;  0 2 2 ]     0.866025 {√3}
82        3.  1x1       | 3x1x3  1x3x3 [  0 2 0 ;  0 0 2 ;  0 2 2 ]     0.866025 {√3}
83        4.  1x1       | 2x2x3  2x2x3 [  1 1 1 ;  1 1 1 ;  0 2 2 ]    -0.75    {√4}
```

In the last case, the norm was adapted to yield the expected value $\|STop\|^2 = 2\cdot 2\cdot(\tfrac{3}{4}2)^2 = 9$, having $S = S_1 + S_2$, $T = T_1 + T_2$, where $S_i$ and $T_i$ are spin and orbital operators, respectively with $i = 1, 2$ indexing either orbital or spin. The same factor $1/2$ was already also encountered earlier in a similar context with the operator `ST` on p. 64.

## 5.5 Building rank-5 PEPS tensor

Projected entangled pair states (PEPS [30, 34, 53, 54]) can be used to tile a tensor network state in one or higher dimension. If a set of tensors is used to tile the lattice to infinity, this is referred to as iPEPS (or iTEBD in 1D [55]). As with any lattice model, one needs to (i) define the local state space of a site (in QSpace in terms of `getLocalSpace`), followed by (ii) an initialization of the tensor network state as a whole. Assuming a 2D square lattice, an $A$-tensor in iPEPS becomes a rank-5 tensor that combines the virtual indices left (L), top (T), right (R), and bottom (B) with the local state space of a site (s) [cf. Fig. 11].

For example, starting with a local state space described by an SU(3) spin in the defining representation $(10) \equiv \mathbf{3}$ [cf. Eq. (15)],

```
1  >> [S,IS]=getLocalSpace('Spin','SU3','10');
```

one may proceed to initialize the rank-5 tensor as sketched in Fig. 11(a). Assuming, for simplicity, that the state spaces on all virtual indices are the same, one nevertheless needs to

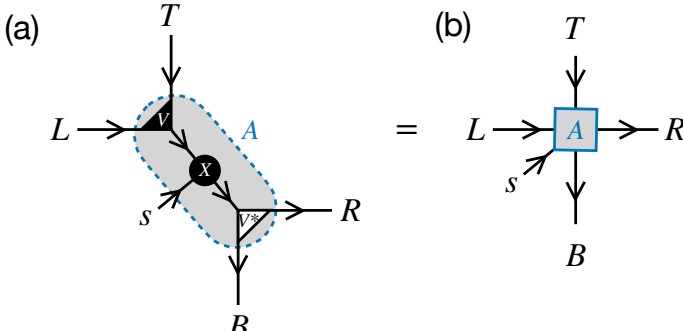

Figure 11: Example for building rank-5 tensor for 2D PEPS, which ties the virtual state space at the $L$(eft), $T$(op), $R$(ight), and $B$(ottom) with the local state space of a site ($s$). Contracting the tensor decomposition in (a) based on rank-3 tensors yields the rank-5 tensor in (b). Assuming periodic tiling of a 2D-PEPS based on a unit cell comprised of a single tensor $A$, the state spaces $L$ and $R$ must match, yet with reverse direction, and ditto for $T$ and $B$.

make an initial choice for this state space. To this end, one may build some arbitrary but fixed combination of symmetry sectors derived from the local state space E,

```
2  >> E=IS.E; U=getIdentity(E,'-0');
3     A2 = getIdentity(U,1,U,2);
4     E2 = getIdentity(U,1) + getIdentity(U,2) + getIdentity(A2,3);
5     E2.data=repmat({eye(2)},size(E2.data));  % 2 multiplets / symmetry sector
```

```
6  E2 =
7       Q:  2x { 4 x 2 }  having 'SU3'  { +- }
8     data:  2-D double (544)      8 x 8 => 30 x 30  @ norm = √30
9
10       1.  2x2       |    1x1 [ 00 ; 00 ]      32 b
11       2.  2x2       |    3x3 [ 01 ; 01 ]      32 b        {√3}
12       3.  2x2       |    3x3 [ 10 ; 10 ]      32 b        {√3}
13       4.  2x2       |    8x8 [ 11 ; 11 ]      32 b        {√8}
```

Line 4 combines the local state space, its dual via the $1j$ symbol U on index 2, and their fused state space via A2 into an identity tensor E2. Since there is no overlap of symmetry sectors of the terms added here, one can combine these spaces as a direct sum ($\oplus$) simply by adding them up. Line 5 then sets the dimension to two multiplets per symmetry sector by setting all RMTs in E2 to eye(2) = $\mathbb{1}^{(2)}$.

With the state space E2 for all virtual indices assumed the same, one can combine two of these, like $L$ and $T$, via the tensor $V$ as depicted in Fig. 11(a),

```
14  >> V=getIdentity(E2,E2);
15     X=getIdentity(V,3,E);    % combine with local state space E
```

Since the fused state space in X as in Fig. 11(a) shall be contracted to the conjugate of V (which then links to $B$ and $R$), their dimensions need to be matched,

```
16  >> [Q3,dd]=getQDimQS(V,3);
17     [I,J]=matchIndex(X.Q{3},Q3);
18     for i=1:numel(I)
```

```
19      s=size(X.data{I(i)});
20      X.data{I(i)}=randn([s(1:2), dd(J(i))]);
21    end
```

Here `getQDimQS` in line 16 returns the symmetry sectors (rows in `Q3`) and their respective number of multiplets (integer vector `dd`) for leg 3 in `V`. Line 17 finds all matching sectors in `X` on the third leg with the entries in `Q3`, returned as `I` and `J`, respectively. The loop then matches the dimension on the third leg of the respective RMT to the one in `V` via `dd`, while randomizing their content using `randn`. With this, all ingredients in Fig. 11(a) are defined and contractable. For transparency, one may add `itags` now,

```
22  >> LT=setitags(V,{'L','T','LT'});                    % L,T → LT
23     BR=setitags(V,{'B','R','BR'});                    % B,R → BR
24     X =setitags(getsub(X,I),{'LT','s','BR'});  % getsub is optional here
```

The QSpace `LT` describes the upper tensor `V` in Fig. 11(a) that combines $L$ and $T$. Similarly, `BR` describes the lower tensor `V` that combines $B$ and $R$ (up to tensor conjugation, which will reverse arrows in the contraction). Line 24 selects the subset of records in `X` that matched `BR` (this step is optional since non-matching symmetry sectors drop out in the contraction anyway), and also assigns `itags`. Based on the assigned `itags`, the contraction into the rank-5 tensor `A5` in Fig. 11(b) can be carried out making use of auto-contraction,

```
25  >> A5=contract({LT,X3},BR,'*',[1 2 4 5 3]); % final permution to LTBRs index order
26     d2=getDimQS(E2);
27     A5=A5*(d2(end)/norm(A5));
```

```
28  A5 =
29      Q: 5x {85 x 2 }  having 'SU3'  { L, T, B*, R*, s }
30   data: 5-D double (39.26k)   8 x 8 x 8 x 8 x 1 => 30 x 30 x 30 x 30 x 3  @ norm = 30
31
32      1.   2x2x2x2        | 1x1x1x3x3 [ 00 ; 00 ; 00 ; 10 ; 10 ]      128 b
33      2.   2x2x2x2        | 1x1x3x3x3 [ 00 ; 00 ; 01 ; 01 ; 10 ]      128 b
34      3.   2x2x2x2        | 1x1x3x1x3 [ 00 ; 00 ; 10 ; 00 ; 10 ]      128 b
35      4.   2x2x2x2        | 1x1x3x8x3 [ 00 ; 00 ; 10 ; 11 ; 10 ]      128 b
36      5.   2x2x2x2        | 1x1x8x3x3 [ 00 ; 00 ; 11 ; 10 ; 10 ]      128 b
37      6.   2x2x2x2        | 1x3x1x1x3 [ 00 ; 01 ; 00 ; 00 ; 10 ]      128 b
38      :   ...
39     84.   2x2x2x2x1 @10  | 8x8x3x8x3x10 [ 11 ; 11 ; 10 ; 11 ; 10 ]    1.2 k
40     85.   2x2x2x2x1 @10  | 8x8x8x3x3x10 [ 11 ; 11 ; 11 ; 10 ; 10 ]    1.2 k
```

The trailing permutation in line 25 sets the index order `LTBRs`, as also seen in the display in line 29. Lines 26–27 apply some normalization to `A5` given the earlier randomization in line 20 [`getDimQS` in line 26 returns the multiplet and state space dimensions for all legs in `E2`, with `d2(end)`= 30 the state space dimension of the last leg; this stands for the total number of states on the bond indices [`LTRB`] as described by `E2`; cf. line 8].

The tensor `A5` has multiplet dimensions `8 x 8 x 8 x 8 x 1` (line 30) that correspond to a respective `30 x 30 x 30 x 30 x 3` states. The `1` multiplet `q=(10)` on the last leg describes the local state space $\sigma \in s$ with `3` states. It is the same throughout all 85 records, which reflect permissible symmetry combinations across all five legs. By default, the full listing is truncated for readability (dots in line 38). The display also shows two entries with largest outer multiplicity [$M = 10$, indicated by the trailing `@10` with the RMT dimensions in lines 39–40, as well as the trailing 6-th dimension `x10` with the respective CGTs]. They also happen to be the largest and last two records. Otherwise, these would be shown as intermediate records.

## 5.6 fdm-NRG: Iterative diagonalization of Wilson chain and spectral functions

The following more elaborate example demonstrates how to use QSpace for the iterative diagonalization in the context of the numerical renormalization group (NRG [13,18]). It is included with this documentation since the NRG was one of the first applications of QSpace [8,19,50]. The example here is based on the standard single impurity Anderson model (SIAM).

The following code represents the script NRG/xnrgQS.m in the repository, with additional comments and a focus on auto-contractions based on itags. It reciprocates the hard-coded MEX routine NRGWilsonQS. For transparency and the sake of the discussion here, however, the iterative diagonalization can also be coded in Matlab itself, while nevertheless utilizing the QSpace toolbox via the MEX routines for tensor product, contraction, diagonalization, etc. It also uses QSpace Matlab utility routines in the lib/ directory of the repository, such as setdef, isset, add2struct, cto, etc., whose purpose is simple and intuitive, as will be indicated. A full explanation can be found in their respective help. Since the following source mimics the algorithm behind NRGWilsonQS, the format of the output data is also chosen in a compatible way. This then permits, for example, subsequent usage of the MEX routine fdmNRG_QS to compute correlation functions (see fdm-NRG below).

**NRG basics** The NRG is a long-standing powerful impurity solver that tackles $0 + 1$ dimensional physical problems [13,18]. For this, the impurity Hamiltonian is mapped onto the so-called Wilson chain. It is described by the general Hamiltonian

$$\hat{H} \;=\; \hat{H}_0(\hat{\mathbf{d}}, \hat{\mathbf{f}}_0) + \sum_{n=0}^{\infty} \big( t_n \hat{\mathbf{f}}_n^\dagger \cdot \hat{\mathbf{f}}_{n+1} + \text{H.c.} \big) \tag{57}$$

where $\hat{d}_\sigma \equiv (\hat{\mathbf{d}})_\sigma$ are the fermionic operators for the impurity ('$d$-level' for historical reasons), where $\sigma$ denotes spin. The fermionic operators for the bath $(\hat{\mathbf{f}}_n)_\sigma \equiv \hat{f}_{n\sigma}$ are organized into a semi-infinite chain where the impurity couples to the first Wilson site $\hat{\mathbf{f}}_0$ only. From a symmetry point of view, any term in the Hamiltonian needs to represent a scalar. Hence the hopping term on the right naturally includes the scalar dot product $\hat{\mathbf{f}}_n^\dagger \cdot \hat{\mathbf{f}}_{n+1} \equiv \sum_\sigma \hat{f}_{n\sigma}^\dagger \hat{f}_{n+1,\sigma}$ implemented via the contraction in line 155 below. The hopping amplitudes $t_n \sim \Lambda^{-n/2}$ in Eq. (57) derive from the NRG coarse-graining with its dimensionless discretization parameter $\Lambda \gtrsim 2$. The resulting exponential decay with increasing $n$ is essential to the NRG, as this justifies the iterative diagonalization. In this sense, the chain structure in NRG does not represent real space, but rather energy scales which, nevertheless, may be interpreted as an inverse length scale [56,57]. Due to the *effective* one-dimensional character of the Hamiltonian in Eq. (57), NRG is naturally suited to an MPS description. As a characteristic of NRG, the MPS is not obtained variationally but rather via iterative diagonalization [50,58]. Nevertheless, it is arguably closely related to a variational approach [24–26].

**Setup of model (SIAM)** The starting point of any NRG is the Hamiltonian H0 $[= \hat{H}_0$ in (57)$]$ that defines the impurity ($\hat{\mathbf{d}}$) together with its interaction with the bath site at the location of the impurity ($\hat{\mathbf{f}}_0$). The matrix elements of H0 are written in the basis encoded in the *A*-tensor A0. With index order LRs in mind, it fuses the impurity ($d \to$ L) with the first Wilson site ($f_0 \to$ s) where H0 is expressed in R. Together this sets the stage for the iterative diagonalization, fully defined by the fermionic operators F (with Z) for a Wilson site and the hopping amplitudes ff.

For the SIAM, the impurity is described by a single spinfull fermionic level that already shares the same structure as a Wilson site $\hat{\mathbf{f}}_n$. Moreover, for the SIAM all interaction is contained within the impurity itself via the Hubbard $U$, whereas the 'interaction' with the bath becomes a plain fermionic hopping via the impurity's hybridization function. Therefore for the case of the SIAM, the NRG setup can be simplified in that H0 describes the impurity ($\hat{\mathbf{d}}$) only. The bath-site $\hat{\mathbf{f}}_0$ can be already taken care of as an iterative NRG step. With this the sum in Eq. (57)

can be extended to start from $n = -1$, with the additional hopping term taking care of the hybridization of the impurity via $t_{-1} \equiv \sqrt{\frac{2D\Gamma}{\pi}}$, with the $D := 1$ the half-bandwidth of the bath. The latter sets the unit of energy, unless specified otherwise. In any case, the iterative NRG diagonalization only requires the initial `A0` and `H0`, the fermionic operators `F` (with `Z`), and the couplings `ff` in the same units as `H0`. This does not care about the precise starting point of the iterative diagonalization which is also in the spirit of the MEX routine `NRGWilsonQS`.

For the SIAM, therefore the 'left' state space of `A0` can be taken as the vacuum state $(|\rangle \to \mathrm{L})$, and the local state space as the impurity itself $(d \to s)$. Then `H0` describes the impurity Hamiltonian only. The hopping amplitude $\Gamma$ is prepended to the couplings as `ff(1)`$= t_{-1}$. Since `getNRGcoupling` takes $\Gamma$ as input parameter, the hoppings are already returned this way (line 29). The setup for the SIAM thus chooses the following adapted version of the Hamiltonian in Eq. (57),

$$
\hat{H} = \underbrace{\varepsilon_d \hat{n}_d + B\hat{S}_z + \frac{U}{2}\hat{n}_d(\hat{n}_d - 1)}_{\equiv H_{\mathrm{imp}} \to \text{H0 (lines 49–51)}} + \underbrace{\sum_{k=1}^{N}}_{\text{(lines 89–166)}} \underbrace{\left( \text{ff(k)}\, \hat{\mathbf{f}}_{k-2}^{\dagger} \cdot \hat{\mathbf{f}}_{k-1} + \text{H.c.} \right)}_{\text{(lines 153–157)}}
\tag{58}
$$

with `ff(k)`$\equiv t_{k-2}$ and $\mathbf{f}_{-1} \equiv \mathbf{d}$.

In the script `NRG/xnrgQS.m`, the initialization of the SIAM reads,

```
1   % SIAM impurity parameters (all energies in units of the half bandwidth D:=1)
2   %
3   %   U          onsite interaction
4   %   epsd       impurity level position
5   %   Gamma      hybridization strength of impurity
6   %   B          magnetic field acting on impurity (Zeeman splitting; applied as -BSz)
7
8     setdef('U',0.12,'Gamma',0.01,'B',0);   % set default values
9     setdef('epsd',-U/2);                    % half-filling by default
10
11  % NRG discretization parameters
12  %
13  %   Lambda    coarse graining strength (≳ 2, dimensionless)
14  %   N         length of Wilson chain
15  %   Nkeep     number of multiplets to keep (or states if all-abelian)
16  %   Etrunc    energy truncation threshold (complementary to Nkeep)
17  %   z         shift of logarithmic discretization, as in Λ^(z-n)/2 with z ∈ [0,1[
18  %
19  % By setting Nkeep large, by default, preference is given to
20  % truncation by energy; using Nkeep, nevertheless, as a safeguard.
21
22    setdef('Lambda',2,'N',55,'Nkeep',1024,'Etrunc',7,'z',0);
23
24  % Collect model and NRG parameters (global info structure for reference)
25    global param
26    param=add2struct('-',U,epsd,Gamma,B,N,Lambda,z);
27
28  % NRG hopping parameters
29    ff=getNRGcoupling(Gamma,Lambda,N,'z',z);   % default z=0 if not specified
30
31  % Local state space: single spinfull fermionic level
32  % For the SIAM this is the same for the impurity and the Wilson sites.
33  % The option '-v' enables more verbose output.
```

```
34    if ~isset('B')
35        SYM='Acharge,SU2spin';
36    else SYM='Acharge,Aspin';  % results in SS = [S₊,S₋,S_z] below => SS(end)=S_z
37    end
38    [FF,Z,SS,IS]=getLocalSpace('FermionS',SYM,'-v');
39
40  % Construct local occupation operator
41  % depending on SYM above, FF may contain a different number of operators (nF)
42    N0=QSpace(size(FF)); nF=numel(FF);
43        for i=1:nF                                    % i ≡ σ  for nF>1
44        N0(i)=contract(FF(i),'13*',FF(i),'13'); end   % f†_σ · f_σ
45    N0=sum(N0);                                       % Σ_σ n_σ
```

The command `setdef` in lines 8, 9, or 22 sets default values for the specified variables if not yet set, i.e., if they are not defined or empty. The command `add2struct` in line 26 adds specified variables as fields to the structure `param` where the leading input argument `'-'` indicates to start from an empty structure. This collects physical model and discretization parameters for general information purposes later. The call to `getNRGcoupling` in line 29 returns the Wilson chain hopping parameters `ff`, starting with `ff(1)`$= \sqrt{2\Gamma/\pi}$. The above also sets the stage for the definition of `A0` and `H0`,

```
46  % A-tensor for H0 (LRs index order convention)
47    A0 = getIdentity(getvac(IS.E),IS.E,[1 3 2]);
48
49  % Impurity Hamiltonian [in R-basis of A0, as in LRs]
50    H0 = epsd*N0 + (U/2)*N0*(N0-IS.E) + 0*IS.E;
51    if isset('B'), H0 = H0 - B*SS(end); end   % SS(3=end) contains S_z
```

The command `getvac` in line 47 returns the vacuum state space for the symmetry setting of the input. If a finite value for the magnetic field is set, the setup via `SYM` in line 36 switches to abelian U(1) spin symmetry. In this case, as can be checked by inspection of `SS`, the impurity $S_z$ operator is returned as `SS(3)` $\equiv$ `SS(end)` in line 51. The magnetic field is applied as $-BS_z^d$ at the impurity only [it is ignored for the bath, assuming $B \ll (D = 1)$], where the minus sign ensures that for $B > 0$ one also obtains a positive magnetization $\langle S_z^d \rangle$.

The above concludes the setup of the impurity. Based on `A0`, `H0`, and `ff`, this could proceed with the MEX routine `NRGWilsonQS` from here [hard-coded C++ code, e.g., called via the `rnrg` ('run NRG') script which for the SIAM calls `setup/setupSIAM_SU2x2.m` as a more elaborate version of the setup above]. For transparency, however, the iterative diagonalization is explicitly transcribed into the Matlab script `xnrgQS` discussed here.

**Iterative diagonalization**   To proceed with the iterative diagonalization, the setup above is further adapted. The couplings `ff` from line 29 above are in physical units of the half-bandwidth ($D = 1$). Since finite-size spectra should be compared in rescaled units of order 1, the exponentially decaying couplings `ff`$\sim \Lambda^{-k/2}$ are also rescaled to order 1,

```
52  % Rescale hoppings from global to iterative units of order 1 (see usage below)
53    rL=sqrt(Lambda);  % rL = `root Lambda'
54    ff=ff.* rL.^(1:N-1);
```

Since at the first iteration `ff(1)` incorporates the first Wilson site $\mathbf{f}_0$, this leaves the Hamiltonian `H0` at iteration '0' in its original units.

By default, the data from the iterative diagonalization is stored to files (one file for each iteration, starting from `*_00.mat` for iteration '0' which diagonalizes the input `H0`),

```
55   % Save data by default to files in the local Matlab data directory (LMA)
56   % as $LMA/NRG/myNRG_##.mat with ## the Wilson shell index k
57   % where LMA is expected to exist as environmental variable
58     setdef('sflag',1);  % set sflag=0 to disable
59     if sflag
60        fout={'NRG' '/myNRG'};
61        cto lma;  % `change to' directory $LMA
62        if ~isdir(['./' fout{1}]), system(['mkdir ' fout{1}]); end
63        fout=[fout{:}];
64        wblog('I/O','using %s_##.mat',fout);
65     end
```

Storing the data to files allows one to use the data later, e.g., for computing spectral functions via fdmNRG_QS further below. The code proceeds with initializing variables and data spaces,

```
66   % Initialize iterative diagonalization
67   %    K   kept space      (AK ≡ A_K)
68   %    D   discarded space (AD ≡ A_D)
69   % scalar operators only have KK → K, DD → D (like HK ≡ H_K, HD ≡ H_D)
70   % general operators also have off-diagonal blocks KD, DK (cf. FKK ≡ F_KK)
71     k=0;   % start at Wilson shell `0' (impurity only)
72     AK=setitags(setitags(A0,'-A',k),1,'Lvac');   % sets itags {'Lvac','K00','s00'}
73     HK=setitags(H0,'-op:K',k);                    % sets itags {'K00','K00*'}
74     AD=QSpace;                                     % inits to empty QSpace [= QSpace()]
75
76   % Fermionic operator F in KK space
77     FKK=QSpace(1,nF);
78
79   % Keep track of general data along iterations
80     EX=nan(N,2);      % 2 columns: highest kept and lowest discarded energy
81     E0=zeros(1,N);    % subtracted ground state energy for each iteration
82     N4=[];            % number of kept and discarded multiplets/states
83
84   % Initialize structure to collect iterative diagonalization
85     Inrg=struct('N',N,'Lambda',Lambda,'EK',[],'HK',QSpace(1,N));
86
87     fprintf(1,'\n'); disp(param);
88     fprintf(1,'\n');
```

By iteratively keeping all eigenstates below some energy (Etrunc) or count (Nkeep) threshold (cf. line 22), this partitions states into kept (AK≡ $A_K$) and discarded (AD≡ $A_D$) states for any iteration. Ultimately, the collection of AK for all iterations together with the lowest energy eigenstate for the last iteration $k = N$ constitutes the ground state MPS for given Wilons chain of length $N$. Expressing operator matrix elements in terms of kept and discarded (based on AK and AD, respectively), leads to sectors KK, KD, DK, and DD (e.g. see operator F → FKK ≡ $F_{KK}$). For scalar operators, there are no off-diagonal contributions, hence a single label K or D suffices, such as for HK ≡ $H_K$. By using itags as in lines 72–73 this will simplify the specification of contractions, but also characterize QSpace tensors more generally, while making the output more readable and identifiable. Basic NRG data will be collected into the info structure Inrg initialized in line 85. Line 87f summarizes the setup by displaying the parameters used.

The following then proceeds with the actual iterative diagonalization,

```
89   % Iterative diagonalization based on couplings ff(k)
90     for k=1:N  % Wilson shell index
```

```
91      if k<N || ~sflag
92          o={'Nkeep',Nkeep,'Etrunc',Etrunc};
93      else o={'Nkeep',0};   % by default, discard all states at last iteration
94      end
95
96   % Collect info  on total expanded dimension (K+D)
97   % this copies 2 entries for non-abelian: [multiplet, state space] dimension
98   % where q(1,*) are multiplet dimensions
99      q=getDimQS(HK); N4(k,:,2)=q(:,2); % total dimension K+D
100
101   % Exact diagonalization of expanded space (may also use QSpace/eig wrapper here)
102      [ee,I]=eigQS(HK,o{:});
103
104      ee=ee(:,1);       % first column (relevant for non-abelian symmetries only)
105      EK=QSpace(I.EK); % MEX files cannot return QSpace objects, only structures
106      ED=QSpace(I.ED); % hence the conversion to QSpace objects here
107
108   % Subtract ground state energy [except for k==1: H0 → keep E0(1)=0]
109      if k>1, E0(k)=min(ee); ee=ee-E0(k); end
110
111      if EK % i.e., non-empty
112       % EK is returned in compact diagonal format => take dimension on 2nd index
113         Dk=getDimQS(EK); N4(k,:,1)=Dk(:,2);
114         EK=EK-E0(k); q=EK.data;  % also subtract E0 in QSpace EK
115         EX(k,1)=max([q{:}]);      % largest kept energy
116      end
117      if ED % repeat for ED
118         ED=ED-E0(k); q=ED.data;
119         EX(k,2)=min([q{:}]);      % smallest discarded energy
120      end
121
122      if sflag % obtain AD before AK is changed right next
123      AD=contract(AK,I.AD,[1 3 2]); end
124      AK=contract(AK,I.AK,[1 3 2]); % LRs index order convention
125      if sflag
126         q=struct('AK',AK,'AD',AD,'HK',EK,'HD',ED,'E0',E0(k));
127         save(sprintf('%s_%02g.mat',fout,k-1),'-struct','q');
128      end
129
130   % Collect `finite size' energy spectra
131   % (note that nrg_plot.m prefers Inrg.HK if present, plots EE otherwise)
132      m=min(Nkeep,length(ee));
133      EE(1:m,k)=ee(1:m); % plain energies without symmetry resolution
134      Inrg.HK(k)=EK;     % same energies but with symmetry resolution
135
136      if k==N, fprintf(1,'\n\n'); break; end
137
138   % Wilson shall k starts here
139   % compute matrix elements for next iteration (propagate FKK)
140      for i=1:nF
141       % operator index order KK[op] here by having FF to the right
142         FKK(i)=contract(AK,'!2*',{AK,FF(i),'-op:^s'});
143      end
144
145   % Add new shell as described by `local space' in IS.E
146      AK=getIdentity(I.AK,2,IS.E,[1 3 2]); % LRs convention
```

```
147       AK=setitags(AK,'-A',k); % generates itags { K<k-1>, K<k>*, s<k>}
148
149     % Rescale and propagate Hamiltonian
150       HK=diag(EK*rL); % expand compact diagonal representation
151       HK=contract(AK,'!2*',{HK,AK});
152
153     % add hopping to newly added site
154       for i=1:nF, q=ff(k)*Z*FF(i);  % include fermionic parity Z here!
155         Q=contract(AK,'!2*',{FKK(i),'*',{AK,q,'-op:^s'}});
156         HK=HK+Q+Q';  % +Q' adds hermitian conjugate (" + H.c.")
157       end
158
159     % Make sure, zero-diagonal blocks are also included (important for k=1 only)
160       HK=HK+0*getIdentity(AK,2);
161
162     % Generic log output: current Wilson shell @ largest kept energy
163     % (to be compared to chosen truncation energy Etrunc)
164       fprintf(1,' %4d/%d (%g) EK=%.3g ... \r',k,N,max(Dk(1,:)),EX(k,1));
165
166   end % of iteration (Wilson shell) k
```

When adding a new Wilson site in lines 145–160 for the next iteration, the extended state space is exactly diagonalized in line 102. This also carries out the truncation into kept (K) and discarded (D) states [cf. lines 105 and 106]. Two parameters govern this truncation as provided with the options o{:}, namely Etrunc $\equiv E_{\text{trunc}}$ and Nkeep $\equiv N_{\text{keep}}$. This keeps at most $N_{\text{keep}}$ states (multiplets in the presence of non-abelian symmetries), or fewer if the energy threshold $E_{\text{trunc}}$ is reached earlier. Near-degenerate eigenstates are never cut across by the truncation but kept together (or truncated) in full to avoid artificial symmetry breaking of symmetries that are not explicitly enforced. As per lines 92–93, both truncation parameters, $N_{\text{keep}}$ and $E_{\text{trunc}}$, are used for all iterations except for the last one, where all states are discarded. Hence with no states kept, this way the Wilson chain is naturally terminated [59].

NRG converges exponentially with increasing $E_{\text{trunc}}$ [58]. On empirical grounds, typically a value of Etrunc $\equiv E_{\text{trunc}} \gtrsim 7$ (in rescaled units as in the source above) leads to good convergence. By truncating based on an energy threshold, this mimics truncation at about a comparable discarded weight and hence comparable accuracy along the Wilson chain [58]. Therefore it is typically recommended to truncate based on $E_{\text{trunc}}$, while keeping $N_{\text{keep}}$ sufficiently large, yet set nevertheless as a safeguard on what is affordable numerically. With this in mind, when increasing the complexity of the impurity or number of (bath) flavors, this quickly also increases the number of states within the energy interval $[0, E_{\text{trunc}}]$ relative to the ground state energy for any iteration. In this case, one needs to work (exponentially) harder by keeping more many-body states for comparable accuracy, i.e., the same $E_{\text{trunc}}$.

The iterative diagonalization is wrapped up again in the spirit of NRGWilsonQS,

```
167 % Finalize Inrg info structure (used by nrg_plot.m)
168   Inrg.E0=E0;
169   Inrg.EK=EX;
170   Inrg.EScale=Lambda.^(-(0:length(E0)-1)/2);
171   Inrg.Itr=add2struct('-',Etrunc,Nkeep);  % Itr = info on truncation
172
173 % Cumulative `physical' value for E0
174 % i.e., global ground state energy in units of bandwidth
175   Inrg.phE0=sum(E0.*Inrg.EScale);
176
```

```
177    if ndims(N4)==2
178        Inrg.NK=N4;    % abelian symmetries only
179    else Inrg.NK=reshape(N4,[],4); % non-abelian
180    end
181
182    if sflag
183        f=sprintf('%s_info.mat',fout);
184        save(f,'-struct','Inrg');
185    end
186
187 % Finalize numeric array EE for finite-size spectra
188 % collected data sets have variable length => replace trailing zeros
189 % for shorter records by nan except so that these are ignored by plot()
190    EE(~EE)=nan;            % same as EE(find(E==0))=nan;
191    EE(1,isnan(EE(1,:)))=0; % restore zero for first value (ground state)
```

The bare energy spectra collected in the numeric array `EE` as finalized above may be plotted as is or used for data analysis. The plot script `nrg_plot.m` (see below), however, prefers the QSpace array `Inrg.HK` collected in line 134, as this contains symmetry resolution. This is reflected then in the line colors of the NRG 'energy flow diagram' as generated by `nrg_plot`.

**NRG energy flow diagram**   The finite-size spectra collected from the iterative diagonalization can be combined and plotted as standard NRG energy flow diagrams [13, 18]. QSpace provides the general plot script `nrg_plot.m` for this purpose. These energy flow diagrams give a concise, detailed overview of the relevant energy scales of the impurity problem down to the lowest-energy fixed points. For this, it is important to be aware that the Wilson shell $k$ directly translates to an energy scale $\Lambda^{-k/2}$ in units of the half-bandwidth $D$: impurity models are 0+1 dimensional, with '0' the impurity, and '+1' the direction of the chain indicating energy scale. The following thus generates the NRG energy flow diagram which summarizes the collected finite-size spectra from the iterative diagonalization,

```
192 % Summarize result by plotting NRG energy flow diagram
193    if ~exist('plotflag','var') || plotflag  % disable by setting plotflag=0
194        param.D=Nkeep;
195        nrg_qdisp=10;  % turn on legend with symmetry labels (at most 10 entries)
196        nrg_plot       % main plot script for NRG energy flow diagram
197    end
```

The call to `nrg_plot` in line 196 generates a plot with five panels as shown in Fig. 12. All panels have the Wilson shell index on the horizontal axis, which starts from high energies at the left and progresses to exponentially small energy scales towards the right. The top two panels show standard NRG energy flow diagrams for even and odd iterations. Due to the open boundaries of the Wilson chain at any shell (iteration) $k$, intrinsic even-odd alternations arise in the energy spectra. Hence to avoid zig-zag lines in plots, a smooth energy flow is obtained by plotting even and odd iterations separately (see labels for vertical axis). While they provide qualitatively similar behavior in terms of flow across fixed points towards the low-energy stable fixed point, they differ significantly in their details. By convention, the impurity together with the first Wilson site $\hat{f}_0$, i.e., $H_0$ in Eq. (57), is considered an even iteration [13, 18].

For the default SIAM parameters in lines 8f, the convergence in the energy flow diagram in Fig. 12 for $k \gtrsim 35$ is attributed to having reached energies sufficiently below the Kondo scale. Generally, the NRG finite-size spectra converge in the low-energy fixed point to the corresponding problem-dependent stable fixed point spectrum. Therefore once the spectra

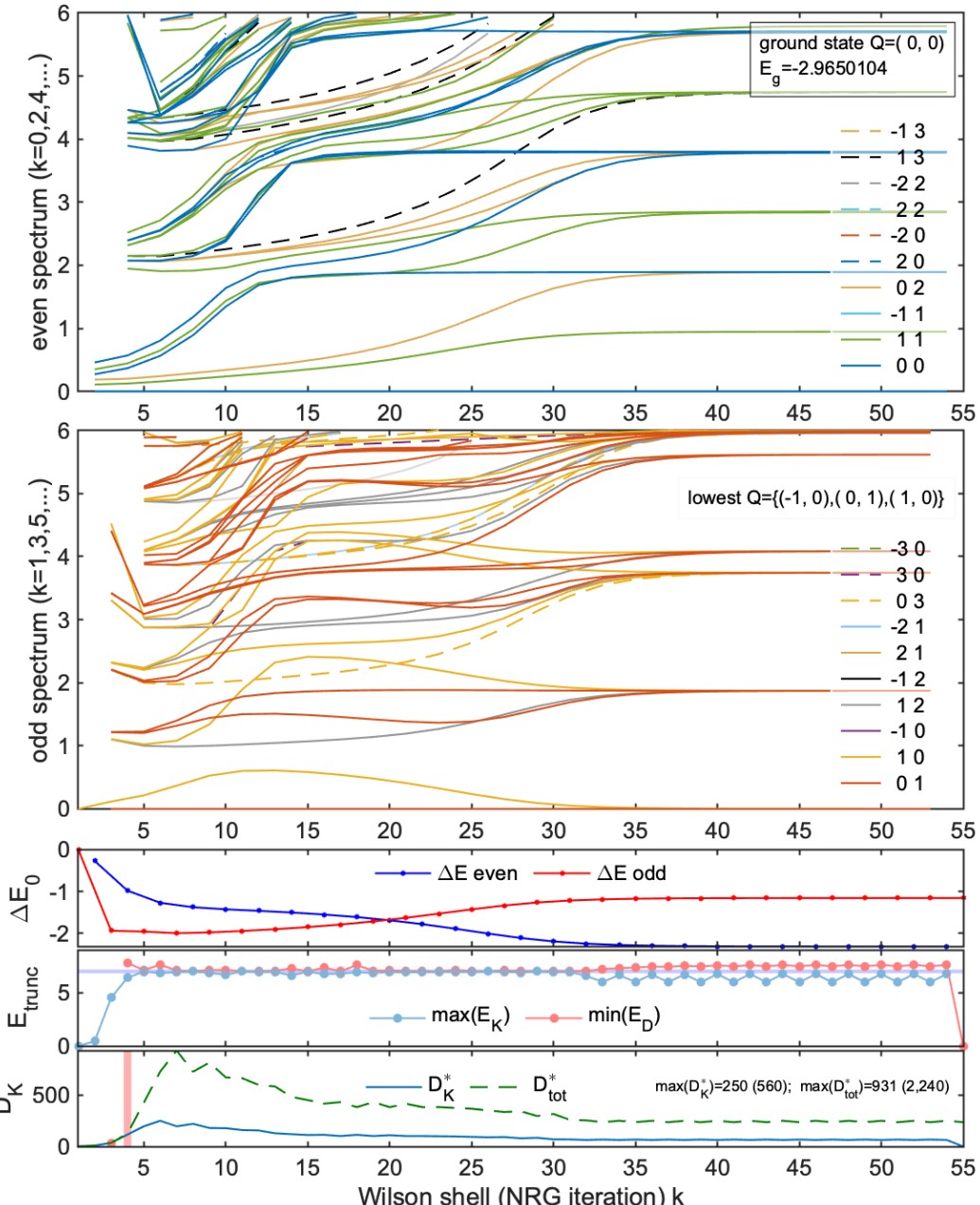

Figure 12: Standerd NRG energy flow diagram (top two panels) for the collected finite-size spectra from the iterative diagonalization of the Wilson chain. This plot is generated by `nrg_plot` (line 196), used with the default model parameters $U = 0.12$, $\varepsilon_d = -0.06$, $\Gamma = 0.01$, $B = 0$ and hence $U(1)_{charge} \otimes SU(2)_{spin}$, as well as the default NRG-specific parameters $N = 55$, $\Lambda = 2$, $z = 0$, `Etrunc` = 7, and `Nkeep` = 1024. The lower three panels show the subtracted ground state energy offset $\Delta E_0$ for even and odd iterations, the truncation energy bounded by $\max(E_K) < \min(E_D)$ relative to `Etrunc` (gray line), and the number of multiplets kept ($D_K^*$) together with the expanded Hilbert space dimension when adding a Wilson site ($D_{tot}^*$). The maximum of $D_K^* \leq 250$ kept SU(2) multiplets corresponds to $D_K \leq 560$ states (reflecting the data in `Inrg.NK`). The dimensions of the expanded state space that had to be diagonalized exactly in line 102 is also shown for reference, having $D_{tot}^* \leq 931$ multiplets corresponding to $D_{tot} \leq 2\,240$ states. With $D_K^* \mathrel{<=} 250 < (\text{Nkeep} = 1024)$, all truncation was performed based on energy, as also confirmed by the second to last panel.

become self-repetitive (straight lines for iterations $k \gtrsim 40$), the iterative NRG diagonalization can be stopped (here at iteration $k = N = 55$). The crossover around $k^* \sim 25$ towards the low-energy fixed point is in good agreement with the analytical value of the Kondo temperature [60,61], e.g., as returned by the function `TKondo.m` provided with the `QSpace` repository: based on the parameters in the global structure `param`, `TKondo` without any further input parameters returns $T_K^{\text{analytic}} = 2.20 \cdot 10^{-4}$ which compares well to $T_K^{\text{NRG}} \sim \Lambda^{-k^*/2} \sim 1.7 \cdot 10^{-4}$ (bearing in mind that the Kondo scale is a crossover scale, it is only determined up to a prefactor of order one, depending on the particular physical or numerical prescription used to define $T_K$).

The third panel out of `nrg_plot` in Fig. 12 shows the lowering of the ground state energy with each iteration by adding another Wilson site. This is based on the data in `Inrg.E0` which is again plotted separately for even and odd iterations. One observes that below the Kondo scale $k \gtrsim k^*$, even iterations lower the energy *more*. Hence the 'ground state' of the system is better reflected by even iterations. For this reason, `nrg_plot` adds the label 'ground state' to the upper panel for even iterations, together with its symmetry sector $Q$ and the cumulative 'ground state energy' $E_g$ taking all NRG iterations into account (`Inrg.phE0` in line 175). On physical grounds, the SIAM flows to strong coupling in which case one may think of $H_0$ as already forming a screened singlet, which is also an even iteration. Thus in the low-energy fixed point the impurity spin has been absorbed into, and in this sense, has become fully screened by the bath.

The fourth panel analyzes the truncation energy by plotting up to which energy states were kept along each iterative diagonalization. This also shows an additional line that corresponds to the lowest discarded energy. When truncating based on energy, the latter line lies above `Etrunc`, whereas the former line is upper bound by `Etrunc`, as seen in Fig. 12, having `Etrunc=7`.

The fifth (lowest) panel shows the number of states $D_K$ (or number of multiplets $D_K^*$ in the presence of non-abelian symmetries) that were actually kept along the iterative diagonalization (line 113), together with the expanded state space $D_{\text{tot}}$ (line 99) that had to be fully diagonalized (line 102) at each step before truncating, and thus splitting it into kept and discarded states (lines 105–106).

Certain degrees of freedom become frozen out when proceeding to lower energy scales. Generally, therefore, the entanglement and hence the number of required states along the iterative diagonalization show a monotonically decreasing behavior. This behavior agrees with Zamolodchikov's c-theorem [62] and is typically well reflected in the bottom panel of Fig. 12. It is also reflected in the upper two panels where levels tend to 'dilute' when moving to lower energy scales towards the right, i.e., levels have an average tendency to move upward. Of course, this excludes the start of the iterative diagonalization before truncation sets in (vertical marker line in the bottom panel) up until the maximum of states is reached soon after. The early iterations around when truncation sets in, are thus the most expensive ones in an NRG simulation. If these can be performed in a stable converged sense, e.g., by keeping sufficiently many states up to the required `Etrunc`, then arbitrary low-energy scales can be reliably reached by NRG at (typically considerably) lower numerical cost per iteration.

**fdm-NRG spectral functions**    The iterative diagonalization of the Wilson chain has the unique and powerful property that it builds an approximate, but complete many-body eigenbasis that is fully tractable, in practice [59, 63]. This permits one to obtain thermodynamical data in a text-book like fashion by directly evaluating Lehmann representations in the spirit of the *full density matrix* (fdm-)NRG [19,64]. This includes thermal quantities both in the static as well as the dynamical context. Since the underlying procedure of fdm-NRG requires significantly more book-keeping as compared to iterative diagonalization, e.g., when collecting spectral data into energy bins, etc., this algorithm only exists hardcoded as the MEX routine `fdmNRG_QS`. Typically, it is run after `NRGWilsonQS`, yet may also be run after the iterative diagonalization based

on xnrgQS above under the condition that the data was stored in a compatible file format by not disabling sflag in lines 58ff. Generally, dynamic correlation functions $\langle \mathcal{O}_1 \| \mathcal{O}_2^\dagger \rangle_\omega$ require the specification of a pair of operators $\mathcal{O}_1$ and $\mathcal{O}_2$ which in the context of the NRG are local operators. In QSpace this translates to operators that can be expressed within the state space described by A0. If $\mathcal{O}_1$ is not set, QSpace assumes $\mathcal{O}_1 = \mathcal{O}_2$, by default. Then by choosing the fermionic operator $\mathcal{O}_2 \equiv$ op2=F(1) below as the operator acting on the impurity, this computes the impurity spectral function $\langle \hat{d} \| \hat{d}^\dagger \rangle_\omega$ for the SIAM,

```
198  if sflag
199   % Compute the dynamical correlation function ⟨O₁|O₂†⟩ where O₁=op1 and O₂=op2
200   % are operators acting on the impurity (more precisely, acting within A0 here)
201     op1=[];    % operator O₁ (empty takes default: op1=op2)
202     op2=FF(1); % operator O₂: here fermionic annihilation operator
203     zflags=1;  % whether to use fermionic parity Z with op[12] (here: yes)
204
205     nostore=1; % do not store matrix elements of op[12] in NRG basis to files fout*
206     locRho=1;  % use local thermal density matrix, i.e., do not store Rho(T)
207                % to files fout* either, but keep it in memory
208
209   % Look for relevant options set in Matlab for fdmNRG_QS
210   % where a trailing '?' indicates a non-mandatory option
211     ofdm=setopts('-','T?','zflags?','-nostore?','-locRho?','nlog?','emin?','emax?');
212   % main fdm-NRG routine
213     [om,a0,Ifdm] = fdmNRG_QS(fout,op1,op2,Z, ofdm{:});
214     fdm_plot;   % plot script to summarize its output
215  end
```

One may specify an arbitrary temperature T in line 211 (by default, it is taken as the energy scale towards the end of, but still within given Wilson chain; in this sense, the default temperature is $T = 0^+$; the actual temperature used is returned in Ifdm.T). The result out of fdmNRG_QS in line 213 includes the discrete spectral data a0 [one column per requested operator pair in op1(i) and op2(i), where $i \le n$ correlation functions can computed in a single call to fdmNRG_QS]. The discrete spectral data a0 is collected on a uniform logarithmic frequency grid om $\equiv \omega$ with energy range [emin, emax] for both, positive and negative energies and nlog bins per decade (line 211). Any spectral weight outside this range is added to the closest extremal bin, so no spectral weight is lost. The returned frequency grid om already catenates negative and positive frequencies into a sorted vector that matches a0 in length. Again the spectral output is analyzed within Matlab based on the plot script fdm_plot in line 214. This also broadens the spectral data in a log-Gaussian fashion [19]. A detailed discussion of this is outside the scope of this documentation and hence is left for the future. In case of interest, please feel free to explore and, if needed, to contact support.

## 6   Conclusion

This documentation of the QSpace tensor library focuses on its basic structure in the ecosystem it was developed in, namely C++ MEX embedded into Matlab on Linux-like environments, including macOS. While the public repository with QSpace v4 nevertheless already also includes powerful applications in the context of NRG and DMRG, these are mostly only mentioned in passing here. Further documentation in that regard is left for the future.

The embedding into Matlab is due to historical reasons. Since the requirement of a standard Matlab license represents some barrier for the usage of QSpace in the community, how-

ever, the longer-term perspective is to replace the embedding into Matlab by other open environments eventually, with Julia a particularly attractive more recent candidate. Yet the QSpace core libraries will continue in the C++ code base in any case. The environment, however, that these are embedded into as an API library, may be subject to change in the future.

## 6.1  Acknowledgements

I thank the following people for carefully looking through this manuscript and providing valuable feedback: Jan von Delft (Ludwig Maximilians University (LMU), Munich, Germany) and his group members, as well as Seung-Sup Lee (Seoul National University, South Korea), Fabian Kugler (Flatiron Institute, New York, USA), Matan Lotem (Tel Aviv University, Israel), Anand Manaparambil (Adam Mickiewicz University, Poznan, Poland), Ji-Yao Chen (Sun Yat-sen University, China), and Adrien Florio (Brookhaven National Laboratory, New York, USA).

I acknowledge long-standing support from the condensed matter theory group of Prof. Jan von Delft (LMU) until 2017 and the office space still provided long after that during my frequent visits to Munich. I very much enjoyed the many inspiring discussions with students and postdocs who also provided valuable feedback on QSpace all along. Since 2006 the students in alphabetic order were: Arne Alex, Benedikt Bruognolo, Andreas Gleis, Cheng Guo, Markus Hanl, Theresa Hecht, Andreas Holzner, Jheng-Wei Li, Hamed Saberi, Frauke Schwarz, Katharina Stadler, Elias Walter. Former grad students outside or with a temporary link to the LMU: Bin-Bin Chen (LMU; Beihang University, Beijing; University of Hong Kong, China), Matan Lotem (Tel Aviv University, Israel), and Anand Manaparambil (Adam Mickiewicz University, Poznan, Poland). Former postdocs at the LMU, all of whom are faculty elsewhere by now: Wei Li (LMU; Institute of Theoretical Physics, CAS, Beijing, China), Seung-Sup Lee (LMU; Seoul National University, South Korea), Ji-Yao Chen (MPQ Garching; Sun Yat-sen University, China), Ireneusz Weymann (LMU; Adam Mickiewicz University, Poznan, Poland), Yilin Wang (BNL; Hefei National Laboratory, China). Since my relocation as a staff scientist to the Brookhaven National Lab (BNL, USA) in 2018, I am very grateful also to Alexei Tsvelik and Robert Konik to leave me with sufficient freedom to carry on the QSpace endeavor.

The QSpace project was initiated and developed over many years by myself while working at the LMU in Munich at the crossroads of NRG and DMRG. A major incentive thinking about a tensor library was an early DMRG Matlab code kindly provided by Frank Verstraete in early 2005 that demonstrated the power of simple transparent tensor routines, even if no symmetries were exploited then. The initial development (QSpace v1 for abelian symmetries) was funded under DFG-SFB631 and DFG-TR12. The later development of QSpace v2 onwards for non-abelian symmetries was funded by DFG WE-4819/1-1 (Independent researcher), DFG WE-4819/2-1 (Heisenberg fellowship), and DFG WE-4819/3-1 (PhD student with a focus on applications). Since 2018 the development of QSpace has been supported by the U.S. Department of Energy with a major focus, also after repeated inquiries from the physics community to push toward open-source as of QSpace v4.

**Funding information**    While developing QSpace v4 with the main incentive to prepare the code for open source as well as working on this documentation, A.W. was supported by the U.S. Department of Energy, Office of Science, Basic Energy Sciences, Materials Sciences and Engineering Division.

## 6.2  Feedback and support

While this documentation explains the more general setting of QSpace as a tensor library, by the very volume of the sources included in the public git repository, this documentation is necessarily far from complete. Therefore aside from this documentation, interested users

are encouraged to explore the git repository and specifically also the `Docu` folder there for additional updated resources.

**Feedback and bug reports**   General feedback, as well as bug reports are welcome. The primary contact in this regard should be via the public QSpace repository, e.g., by raising an issue there. The general contact will also be maintained and kept up to date with the wiki pages also supported by the public repository. As a last resort, an email may be sent directly to the author of this documentation as provided on the front page.

When a reproducible bug is encountered in QSpace, be it on the Matlab side or in the behavior of MEX files, a bug report is welcome. It will be checked and fixed in due time. Ideally, a bug report should include a minimal Matlab script that contains the problematic command that causes a particular error or unexpected behavior. This should permit one to reproduce the error without referencing other user-defined functions. This may be complimented with a small mat-file that contains particular instances of QSpace tensors. All of this can be uploaded with the git repository when raising an issue.

**Support**   The software in the public git repository for QSpace is open source under the Apache 2.0 license, hence free of charge, and provided as is without any guarantee. In this sense actual support beyond this documentation is subject to the specific circumstances, the availability of the author, or specific agreements.

# A   General Definitions

## A.1   Acronyms

The following list of acronyms is central to QSpace and this documentation. It is grouped by their meaning rather than sorted alphabetically, with more basic definitions coming first. The main text includes frequent hyperlinks into this listing.

**irep (multiplet)**   Irreducible representation – Non-abelian symmetries organize state spaces into multiplets that form irreducible units (ireps) when acted upon with the generators of the symmetry. That is, starting from any state in the multiplet, one can reach any other state by applying some sequence of generators. In this sense, all states within an irep are connected by symmetry. For a particular non-abelian symmetry, specific ireps include:

*Scalar representation*:   Trivial multiplet that consists of a single state invariant under all symmetry operations. It is also referred to as a singlet. In QSpace, the scalar representation is always referred to as $q = (0, \ldots, 0_{\mathfrak{r}}) \equiv 0$, for abelian and non-abelian symmetries alike, with $\mathfrak{r}$ defined in Eq. (59). The apparent exception of parity symmetry with $q \in \{+1, -1\}$ is equivalent to $\mathbb{Z}_2$ with $q \in \{0, 1\}$, which then again also has $q = 0$ as its scalar representation. In this sense, QSpace prefers $\mathbb{Z}_2$ to encode parity.

*Defining representation*:   The smallest non-trivial unit that by taking tensor-products with itself can generate any other multiplet for given symmetry. This is also reflected in ireps represented by Young tableaus for SU(N) where the defining irep represents a single box, and the tensor product of any irep (Young tableau) with the defining irep adds a box. This is also in the spirit of tensor networks where many-body Hilbert spaces are generated by iteratively adding sites that may be given, e.g., in the defining irep. For any non-abelian symmetry, similarly QSpace always also starts from the Lie algebra in the defining representation. Its symmetry labels are always $q_{\mathrm{def}} = (1, 0, \ldots, 0_{\mathfrak{r}})$ with $\mathfrak{r}$ the rank of the symmetry [see also Eq. (15)]. As such, it is one of the $\mathfrak{r}$ *fundamental representations* that have only one 1 it its $q$-labels and zero else.

*Adjoint representation and spin operator*: For any multiplet, there must exist a set of operators that can carry out the symmetry transformations of given non-abelian symmetry within the multiplet's state space. That is, there must exist an irreducible operator for any multiplet that transforms according to an irreducible representation that has as many operators as there are generators in the Lie algebra, i.e., has the dimension of the Lie algebra. This irreducible representation is called the *adjoint* ($q_{\text{adj}}$) which is always self-dual. The respective irop is also referred to as 'spin operator' with reference to SU(2). From the above argument together with 1$j$ symbols, the adjoint must always appear when fusing any multiplet $q$ with its dual $\bar{q}$ (see also construction of complete operators basis). This permits a generic way to generate the adjoint, namely fusing $q \otimes \bar{q}$ for any $q$. In the simplest case, when taking the defining irep $q_{\text{def}}$, the adjoint is guaranteed to occur without outer multiplicity. For example for SU(2), $\frac{1}{2} \otimes \frac{1}{2} = 0 \oplus 1$, or $\mathbf{1} + \mathbf{3}$ in terms of dimensionalities, or within QSpace having $q = 2S$, $1 \otimes 1 = 0 \oplus 2$. Here the spin operator transforms like an $S = 1$ multiplet, which is the adjoint representation for SU(2), indeed, also referred to as triplet then. For SU(3), this becomes $(10)\otimes(01) = (00)\oplus(11) \equiv \mathbf{1} + \mathbf{8}$ with now the octet $(11)$ the adjoint.

**irop** Irreducible operator – Similar to states, in the presence of non-abelian symmetries it also must be possible to organize all operators into irreducible units that are connected via commutators with the symmetry generators. Hence an irop directly also reflects the structure of a particular irep of a symmetry, and is thus said to transform according to that irep. A particularly simple example are scalar operators (like Hamiltonians or density matrices) which transform according to $q = 0$, and thus commute with all generators of the group.

**CGC** refers to a standard textbook Clebsch-Gordan coefficient space, i.e., a CGT of rank $r = 3$ with $q$-directions ++−. These arise out of tensor product decomposition (fusion rules) of two multiplets $(q_1, q_2 | q_{\text{tot}})$. Usually used with non-abelian symmetries, they can also be used to (trivially) represent the fusion rules for abelian symmetries or discrete symmetries.

**CGT** refers to a *Clebsch Gordon tensor* as a generalization of CGCs to tensors of arbitrary rank $r \geq 2$ via contractions. Like CGCs, CGTs are also always real. In QSpace, they are stored in generalized sparse column-major order and computed in high-precision format via MPFR using quad precision at 128 bits. For later retrieval, they are tabulated and stored with sorted $q$-labels in RC_STORE/CStore. This contains CGCs as a subset in CStore/++−/ and 1$j$ symbols in CStore/++/.

**CGR** CGT reference – QSpace tensors themselves no longer carry explicit CGTs since these can be tabulated and stored once and for all (as of QSpace v3). Therefore QSpace tensors only carry references to sorted CGTs, referred to as CGRs. These carry a permutation and the *w*-matrices as additional information. In a QSpace X, the CGRs are stored in X.info.cgr. The RC_STORE/CStore only stores sorted CGTs, meaning they all have their $q$-labels lexicographically sorted w.r.t. to their legs within the group of incoming and outgoing legs, with the incoming legs listed first and at least as many incoming as there are outgoing legs (for the reverse case, tensor conjugation is used). For this reason, a CGR also implicitly encodes an arbitrary permutation $p$ together with a potential conjugate flag in addition to the CGT reference. A CGR together with the stored (and thus *sorted*) CGT can thus describe an *unconstrained* CGT with respect to the index order of incoming or outgoing legs. This is central to the QSpace data structure.

**RMT** refers to generalized reduced matrix element tensor: as known from the Wigner-Eckart theorem [10], matrix elements of operators factorize into reduced matrix elements in a tensor

product with the respective CGT [cf. Eq. (16)] [9]. In QSpace, generalized column-major order is assumed for RMTs of any rank. The RMTs are always stored in double precision format, or interleaved double for complex-valued tensors for real and imaginary parts.

**MPS** Matrix product state [17, 30, 65] – Tensor network state with a one-dimensional structure, also referred to as *tensor train* [66], typically with open (or in the case of translational invariance, infinite) boundary conditions.

**PEPS** Projected entangled pair states – A way to describe physical systems by tensor networks in higher dimensions [53, 54, 67] with extension to fermionic systems [33].

**TNS** Tensor network state [29, 54, 68] – general terminology for arbitrary tensor network topographies, including MPS, PEPS, TTNs, etc. From a physical perspective, it typically refers to a single state, i.e, with no other external legs than the physical site indices.

**TTN** Tree tensor network state [69, 70] – arbitrary TNS in the absence of loops, which includes MPS with open boundaries as a trivial example.

**LRs** MPS index order convention $(l, r, \sigma)$ with $l \in$ L(eft) block, $r \in$ R(ight) block, $\sigma \in$ s for local state space. This keeps the typically large block dimensions for L and R to the front, like a matrix, that is stacked then with respect to the index for the small state space $\sigma$. Assuming column-major, the last index $\sigma$ is the slowest in terms of data layout in computer memory. See also operator index order convention.

**OC** Orthogonality center [71] – Tree tensor networks (TNSs), including the special case of an MPS with open boundary conditions, permit strict orthonormalization of all block state spaces w.r.t. any bond since cutting any bond bisects the TNS into disconnected tensor network blocks. This also permits a clean implementation of symmetries and well-conditioned local tensor environments [72, 73]. In this case, tensors in a TTN can have multiple indices that 'enter', but at most one index that 'leaves' it, in the sense that it acts as an isometry that maps all incoming state spaces onto one fused state space that is typically truncated. Such a TTN can only have one tensor with all indices incoming, referred to as the orthogonality center (OC). All arrows in a TTN flow toward the OC then. It ties together otherwise disconnected blocks of a TTN into the wave function for the full system. The OC may be moved iteratively throughout the tensor network, a process referred to as *sweeping* and related to automatic differentiation.

**OM** Outer multiplicity – only applies to CGTs, and refers to the fact, from a tensor and symmetry point of view that for fixed symmetry sectors on all legs, the fusion of incoming legs can result in *multiple orthogonal* combinations for the *same* fixed symmetry sectors of the outgoing legs [8, 15]. The presence of OM, $\mu = 1, \ldots, M$ with $M > 1$ thus translates to an additional trailing OM index $\mu$ in CGTs that links to an additional trailing OM index $\mu'$ in the respective RMTs via an auxiliary $w$-matrix [cf. Eq. (16)].

**MEX files** *Matlab executables* (MEX, [74]) are C++ binaries that are compiled via Matlab's mex interface, with file extensions *.mex*, like *.mexa64 (Linux) or *.mexmaci64 (macOS). They represent functions that can be called directly from within Matlab like regular Matlab functions, but are based on externally compiled binaries that are interfaced via Matlab's MEX API. This permits efficient complex data structures with access to C++ coding standards (C++11), highly optimized open-source libraries such as the standard template libraries (STL), as well

as C++ style garbage collection. The MEX compilation requires `mex setup C++`. QSpace uses the option `'-R2018a'` with `mex` to enable standard interleaved complex data format.

**NB!**    Nota bene – Latin for 'be aware' to indicate something important. Frequently used in source code comments and in `wblog` output.

**QS** or **QSP**    refers to QSpace, e.g., as trailing part of file names as a low-level way to differentiate namespaces like MEX functions `*QS`, or as prefix for QSpace specific environmental variables like `QS[P]_*`.

**Wb**    Alternative acronym for the initials of the author (AW), frequently used in help or usage information, and comments in the code.

## A.2    Terminology

The following glossary of terminology is central to QSpace and this documentation. Like with the acronyms above, the order of entries is grouped by meaning rather than alphabetically sorted. More elementary concepts come first, so reading through the entries from top to bottom should be meaningful. The main text includes frequent hyperlinks into this listing.

**Sites**    are the smallest physical units in a tensor network that are typically formulated on a lattice (a notable exception in the realm of physics are continuous MPS [75]). For a finite tensor network of $L$ sites then, sites are indexed $i = 1, \ldots, L$ in a well-defined order. Typically, sites are taken identical, while nevertheless the Hamiltonian may impose site-specific parameters.

**Dimension**    refers to the index range of a particular index (leg) of a tensor. It needs to be differentiated from the number of legs of a tensor referred to as its rank.

*Local dimension $d$*:    refers to the Hilbert space dimension $d$ of a site which typically is independent of site $i$. It is considered small and therefore is denoted by lowercase $d_{(i)}$.

*Bond dimension $D$*:    refers to the dimension of the auxiliary or virtual state space on a bond that connects two tensors within a TNS. Since it is typically significantly larger than the local dimension, it is denoted by a capital $D$, and typically varies for different bonds across a TNS.

*Effective dimension $D^*$*:    When dealing with non-abelian symmetries with multiplets that contain several states grouped as ireps, the total number of states $D$ on any Hilbert space can be *reduced* to an effective dimension $D^*$ which counts multiplets only. For larger-rank symmetries, one quickly encounters $D^* \ll D$. Empirically, one can gain a total dimensional reduction on average by a factor of about $\sim 3^{\mathfrak{r}}$ with $\mathfrak{r}$ the rank of the symmetry [8, 76], e.g., $\sim 3^{N-1}$ for SU($N$). This has dramatic effects on numerical efficiency. Since the RMTs carry effective dimensions, e.g., the typical cost for matrix operations of order $\mathcal{O}(D^3)$, when exploiting non-abelian symmetries can be effectively reduced to $\mathcal{O}((D^*)^3)$. For SU($N$), this implies a speedup of $\mathcal{O}(3^{3(N-1)} \sim \mathcal{O}(30^{N-1})$ which makes a *significant* difference for any step $N \to N + 1$. For SU(2) this amounts to more than one order, and for SU(3) about three orders of magnitude of speed up for comparable accuracy of the simulation.

*Singleton dimension*:    refers to a leg in a tensor whose dimension is 1, i.e., its index range is limited to a single value. Consider, for example, a rank-3 tensor $X$ with dimensions $D \times D \times 1$. It has a trailing *singleton* dimension, i.e., an index space that has a trivial range $X(i, j, 1)$ on that last index. If in the presence of symmetries, this index also carries the symmetry label $q = 0$, i.e., corresponds to the scalar irep like the vacuum state, then this additional dimension is also irrelevant from a symmetry point of view, and can be skipped altogether. In the case of trailing

scalar singleton dimensions beyond rank 2, the rank of a tensor thus can be trivially reduced. This is partly used in `QSpace` with irops that represent a scalar operator that reduces a rank-3 irop to a rank-2 scalar operator [e.g., see `@QSpace`/`fixScalarSop()`; for the reverse process of reattaching a singleton irop dimension, one may use `@QSpace`/`makeIrop()`]. Leading or intermediate singleton dimensions, on the other hand, are never skipped.

*Multiplet dimension*:  refers to the number of states in a specific multiplet. It is denoted by $|q|$ for irep $q$. See entry multiplet / irep dimension below for the scaling of typical dimensions depending on the non-abelian symmetry.

**State spaces**  Every index (leg) in a tensor network represents a state space. This can be either the physical state space of a site, or the virtual or auxiliary state space of a bond.

*Local state space*:  refers to the state space of a site $i$ in the lattice model analyzed, and is usually denoted by $\sigma_{(i)} \in s_{(i)}$ or $s$ itself, e.g., if $\sigma$ is already used to specify spin. Typically, sites are considered identical, thus skipping the subscript $i$ where unambiguous.

*Bond states*:  refer to the auxiliary or virtual indices that connect tensors. They can be interpreted as many-body block states if the tensor network has no loops, such as a tree tensor network or an MPS. In the presence of loops like in PEPS one may interpret the bond states as actual fictitious auxiliary state spaces that live on both sides of a bond and which thus describe an orthonormal state space consistent with symmetries *by fiat* in this ansatz. The pair of auxiliary states associated with a bond are then maximally entangled into a singlet, followed by local projection to the physical state space on every site. This PEPS interpretation directly generalizes the AKLT [77] construction.

**Operators with and without hats (carets)**  By definition, matrix or array coefficients represent operators or tensors in a particular basis of a state space. Therefore, general operators are shown with hats before casting them into a particular basis. This is generalized to arbitrary tensors, including *A-tensors* as in $A_r^{l\sigma} \equiv \langle l\sigma | \hat{A} | r \rangle$. The hat is removed once specific matrix (or array) elements are addressed. In this sense, operators with hats can only occur in the abstract, but not in the computational context. Tensors expressed within `QSpace` therefore never carry hats in discussions.

**(Generalized) matrix elements**  Matrix elements are coefficients of operators in a particular basis, such as $S_{\sigma'}^{\sigma} \equiv S_{\sigma'}^{\sigma} \equiv \langle \sigma | \hat{S} | \sigma' \rangle$ [cf. Eq. (6)]. This can be generalized to arbitrary tensors, e.g., $T_{\sigma_3 \sigma_4 \sigma_5}^{\sigma_1 \sigma_2} \equiv \langle \sigma_1 \sigma_2 | \hat{T} | \sigma_3 \sigma_4 \sigma_5 \rangle$, which are then referred to as coefficient spaces, array elements, *generalized matrix elements*, or simply still also *matrix elements*.

**Rank of a tensor**  The rank $r$ of a tensor, i.e., its *tensor rank*, refers to the number of legs or indices associated with a tensor. A tensor with $r$ legs is thus referred to as a tensor of rank $r$, or simply a rank-$r$ tensor. With the terms tensor and array used interchangeably, this also may be referred to as an array of rank $r$.

*Disambiguation:* The *tensor rank* is also known as order or degree of a tensor. It needs to be differentiated from the Schmidt or *matrix rank*.[7] The latter only applies to matrices and refers to the number of non-zero Schmidt coefficients. It is typically of lesser importance in tensor networks in practice, since the singular values from a singular value decomposition for

---

[7]This semantics is also consistent, for example, with Wolfram's tensor rank in Mathematica as obtained by the function `TensorRank[]` vs. `MatrixRank[]`. In Matlab, by contrast, the tensor rank of an array is obtained by `ndims()` which returns the *number of dimensions* while skipping trailing singletons beyond tensor rank 2.

some arbitrary but fixed bisection of a tensor are assumed to decay without abruptly becoming exactly zero numerically. A 'low-Schmidt-rank' *approximation* for tensors in this sense always translates to truncation on an auxiliary intermediate index. This comes with a truncation error, also referred to as the discarded weight, which is controlled by the number of states kept on that intermediate index. The bisection of a tensor by organizing all its indices into two non-empty groups is also unique for matrices only. This becomes ambiguous for tensors with three or more indices. The particular generalization of the matrix rank to tensors in terms of the *canonical polyadic decomposition* (CPD)[8] may be referred to as *CP rank* to distinguish it from *tensor rank*. Given a tensor of higher rank (many indices), a *low-rank approximation* can also be obtained by a *tensor decomposition* [79, 80] described by some tensor network of lower-rank tensors (i.e., tensors with fewer legs and auxiliary intermediate indices). A prototypical example is the decomposition of a many-body wave function into an MPS. In this sense, a low-rank approximation in the above sense has the same objective as a low CP-rank decomposition. Both refer to a tensor decomposition.

**Rank of a symmetry**   The rank $\mathfrak{r}_S$ of a continuous non-abelian symmetry $S$ refers to the number of symmetry labels required to specify an arbitrary multiplet. This is derived from the number of generators that can be simultaneously diagonalized. These are referred to as the Cartan subalgebra and generalize the $S_z$ operator for SU(2). For example, SU($N$) is a symmetry of rank $\mathfrak{r} = N - 1$, which implies a single label for SU(2), but two for SU(3), etc. An abelian symmetry like U(1) has $\mathfrak{r} = 0$. In that case the multiplet space becomes trivial, as the multiplet and weight label of its only state become identical. Therefore $\mathfrak{r} > 0$ distinguishes non-abelian from abelian symmetries.

On practical grounds, it is convenient to also define the adapted rank (note the marginally different font),

$$\mathfrak{r} \equiv \max(1, \mathfrak{r}) \tag{59}$$

which then specifies the number of symmetry labels required for any symmetry, abelian and non-abelian alike.

**Generators of Lie algebra**   The space of generators in any Lie algebra can be organized in a canonical way that generalizes the combination of the diagonal operator $S_z$ and raising and lowering operators $S_\pm$ known for SU(2). This corresponds to and can also be visualized by the root diagram for the adjoint representation [46].

*Cartan subalgebra*:   Maximal set of mutually commuting generators in a Lie algebra. This generalizes the single $S_z$ operator for SU(2) to a set of $\mathfrak{r}$ operators, with $\mathfrak{r}$ the symmetry rank [46]. Since mutually commuting, these 'z-operators' can be simultaneously diagonalized. Their combined diagonal entries then define the $q_z$ labels in Eq. (2) for any irep ($\mathfrak{r}$-tupel of numbers, also referred to as weights).

*Simple roots*:   The remainder of generators, aside from the Cartan subalgebra, can be organized into pairs of raising and lowering operators. For any Lie algebra, there exists a minimal set of so-called *simple* roots which permits one to traverse any multiplet (irep or irop) in the spirit of raising/lowering operators [46]. For a symmetry of rank $\mathfrak{r}$, it therefore suffices to constrain oneself to the $\mathfrak{r}$ *simple positive roots* ('raising' operators), together with their Hermitian conjugate, the $\mathfrak{r}$ *simple negative roots* ('lowering' operators).

---

[8]Also referred to as PARAFAC decomposition [78] which also may be seen as a generalization of a principal component analysis (PCA) to higher dimensions.

*Maximum-weight state and q-labels*:  The maximum weight state of a multiplet is destroyed by any raising operator. Then starting from the maximum-weight state, one can sequence the entire multiplet by applying the minimal set of lowering operators (simple negative roots) only. Eventually, one reaches the complimentary *minimal weight* state that is destroyed by any of the lowering operators. The $q_z$-labels for the maximum-weight state of an irep can be used to uniquely identify the irep. QSpace employs a linear map that transforms the (non-standard, but convenient) internal $q_z$ labels for the maximum weight state to standard multiplet labels consistent with the literature [8], e.g., consistent with labels for the Young tableau for SU($N$). They are referred to as $q$-labels [cf. Eq. (2)] and written in compact notation.

**Multiplet / irep dimension**   A continuous non-abelian symmetry of rank $\mathfrak{r}$ explores an $\mathfrak{r}$-dimensional label space. This consists of a non-negative dense set of integers $q_i \in \mathbb{N}_0$ that are collected into an $\mathfrak{r}$-tupel $q = (q_1, q_2, \ldots, q_{\mathfrak{r}} \geq 0)$. This not only holds for the multiplet labels, but also for the state space of each multiplet individually, since the generalization of spin $S_z \to q_z$ [8] also explores an $\mathfrak{r}$-dimensional weight space $q_z$. Empirically, this gives rise to a typical range of the multiplet dimension $|q|$ encountered in tensor network simulations

$$1 \leq |q| \lesssim 10^{\mathfrak{r}} \,. \tag{60}$$

This assumes that the entire physical state is globally 'non-magnetic', i.e., in or close to the singlet symmetry sector $q_{\text{tot}} = 0$. For example, abelian symmetries have $\mathfrak{r} = 0$. Therefore their multiplet dimension is $|q| = 1$ which is consistent with Eq. (60). In DMRG simulations with non-spontaneously broken SU(2) spin symmetry, the typical multiplet range reaches a dimension $q + 1 = 2S + 1 \lesssim 10$, i.e., $S \lesssim 5$. Since this range is there for every one of the $\mathfrak{r}$ dimensions in the label space, this motivates Eq. (60) as also confirmed in practice.

Equation (60) implies that the typical multiplet dimension grows *exponentially* with the rank of the symmetry. For example, for tensor network simulations that truncate dynamically on the bond state spaces while exploiting SU($N$) symmetry, the SU($N$) multiplets quickly explore dimensions from 1 ($q = 0$) up to $\sim 10^{N-1}$. Therefore increasing the symmetry rank by one, as in SU($N$) $\to$ SU($N + 1$), always makes a *significant* difference in numerical cost. On the other hand, symmetries of the same rank have comparable numerical costs. Hence when using QSpace it is important to be aware of the scaling of typical multiplet dimensions with symmetry rank as indicated with Eq. (60). This strongly affects the numerical cost when generating the RC_STORE. As compared to the cost of dealing with the RMTs, however, the numerical cost for RC_STORE is typically negligible for symmetries of rank $\mathfrak{r} \leq 2$, which thus includes SU(3).

Irep dimensions are known in QSpace from the RC_STORE. Generally, new ireps can only occur when decomposing tensor product spaces. This also automatically provides their dimension (the dimension of multiplets in any Lie algebra can also be determined analytically via the Weyl character formula [46, 81]). Other than that, multiplet dimensions are usually less important from the user's perspective. Nevertheless, the dimensions of the ireps are shown for information purposes. For any given QSpace X, irep dimensions for multiplets are explicitly stored with the CGRs in X.info.cgr.size. These are used when displaying a QSpace [e.g., see Fig. 7].

**Dual representation**   For any symmetry sector $q$ there exists exactly one unique dual representation denoted as $\bar{q}$. It is the only irep that, when fused with $q$, also generates the scalar irep, $q \otimes \bar{q} = 0 \oplus q_{\text{adj}} \oplus \ldots$ amongst other multiplets for $q \neq 0$ (based on the construction of a complete operator basis, this always also includes the adjoint $q_{\text{adj}}$ at least once). The scalar representation cannot result from any other fusion, i.e., $(qq'|0) = 0$ for any $q' \neq \bar{q}$. The

defining property $(q\bar{q}|0)$ is essential to $1j$ symbols. One way to think about dual representations from a physical point of view is that if one creates a particle with symmetry labels $q$, the respective hole must transform in $\bar{q}$, so they can annihilate each other. That is, if a particle creation operator transforms according to $q$, the annihilation operator must transform according to $\bar{q}$. In this sense, every irep $q$ has a unique dual $\bar{q}$ which represents its time-reversed partner. Therefore the dual representation must always have the same multiplet dimension, i.e., $|q| = |\bar{q}|$. For the case $q = \bar{q}$, the irep is called self-dual.

For SU($N$), the dual for a particular irep $q$ is simply given by $\bar{q} = \mathtt{flip}(q)$, i.e., by flipping the order of symmetry labels. This can be easily motivated pictorially with reference to Young tableaus: $\bar{q}$ is precisely the unique irep (Young tableau) that when 'flipped around' (rotated by $180°$) perfectly complements $q$ by filling up the gray shaded space in Eq. (12) below the Young tableau with boxes down to the dashed line at the bottom. This way $\bar{q}$ completes $q$ like two puzzle pieces that together result in a completely filled rectangular block of $(\mathfrak{r}+1) \times (\sum_i q_i)$ boxes. With the latter being equivalent to the scalar $q = 0$, $\bar{q}$ thus always permits the fusion $(q\bar{q}|0)$. It follows from the above argument that a rank-1 non-abelian symmetry such as SU(2) is necessarily self-dual, i.e., $q = \bar{q}$ for all $q$. The symplectic symmetries Sp($2N$) as well as SO($2n+1$) and SO($4n$) are self-dual as well, whereas for SO($4n+2$) the last two values in the $q$-labels of an irep need to be swapped to obtain the dual.

Many Abelian symmetries are also *not* self-dual. For example, U(1) with $q + \bar{q} = 0$ and hence $\bar{q} = -q$, this is not self-dual. It is instructive to compare to an SU(2) multiplet with spin $S$ and spin-projection $S_z$: the latter reflects the abelian behavior, having $S_z \to -S_z$ under time reversal, and thus '$\bar{S}_z = -S_z$' in case of a U(1) spin symmetry. By contrast, $S$ remains the same, $\bar{S} = S$, since the weight diagram is symmetric under inversion.

*Disambiguation:* In QSpace the operator $^*$ (raised asterisk) does not denote 'dual', but is reserved for (tensor) conjugation. The conjugation of a tensor, by definition, leaves symmetry labels invariant, thus having $q^* = q$, yet with reverted directions.

**Symmetry labels (individual symmetry)**     Any given index or state space is decomposed into symmetry sectors described by symmetry labels [cf. Eq. (2)]. For abelian symmetries, these can be chosen as plain integers $q \in \mathbb{Z}$, such as U(1) or $\mathbb{Z}_n$. For non-abelian symmetries, the symmetry labels specify ireps, and are tuples of non-negative integers, based on Dynkin labels which for SU($N$) directly specify the respective Young tableau.[9] The symmetry labels for SU(2) are chosen as $q = 2S$, and thus are also integers. The length of the tuple is equal to the rank of the symmetry. By having non-negative integers for non-abelian symmetries, this permits a compact notation of ireps. In QSpace therefore all symmetry labels are represented by signed integers, generally referred to as *q-labels*. For an individual symmetry $s$, they are given by $q \in \mathbb{Z}^{\mathfrak{r}_s}$, with $\mathfrak{r}_s \geq 1$ the number of labels [cf. Eq. (59)]. The scalar representation is also denoted by the shortcut notation $q = 0$.

*Note on multiplet specification by dimension*: The specification of an irep via the single label of its state space dimension only is generally insufficient for non-abelian symmetries of rank $\mathfrak{r} > 1$, because it is not unique. With reference to the literature, the bold notation in terms of multiplet dimension is used in parallel, nevertheless, if unambiguous, like $\mathbf{3} \equiv (10)$ for the defining representation of SU(3). However, note for example, that the ireps (40) and (21) of

---

[9]The emphasis on integers is also convenient within the QSpace core libraries since then there cannot be any issues with rounding errors. Initially, the irep labels for a non-abelian multiplet are derived internally from a linear map of the weights of the maximum weight state [8]. The weights are chosen in a most convenient, non-standard way in double precision format. The linear map on the maximum weight state also includes non-integers. Yet the resulting $q$-labels are standard and thus must correspond to integers up to rounding errors. This also serves as a simple internal consistency check.

SU(3) accidentally share the same multiplet dimension $d = 15$. Hence in that case the notation **15** to specify these multiplets becomes ambiguous [similarly so also for their respective duals (04) or (12), for which $\overline{\textbf{15}}$ becomes ambiguous].

**Symmetry labels (set of symmetries)**   In the presence of multiple ($n_{\text{sym}}$) symmetries, for any given index or state space the symmetry labels for all symmetries are catenated into a single tuple. For generality, also their combination is referred to as $q$-labels, $q \equiv (q_1, \ldots, q_{n_{\text{sym}}})$. Here $q_s$ with $s = 1, \ldots n_{\text{sym}}$ describes the symmetry labels for symmetry $s$, where $q_s$ itself may already contain multiple labels for non-abelian symmetries depending on the symmetry's rank. For a QSpace X, these combined $q$-labels are collected as rows in the matrices $X.Q\{j\}$ for leg $j$. The vacuum state has the scalar representation for all its symmetries, and is therefore also denoted by the shortcut notation $q = 0$, meaning zero labels across all symmetries present.

**Symmetry labels (multiple legs)**   For a tensor X of rank $r$ one may consider a particular symmetry sector with fixed $q$-labels for all legs. This corresponds to a QSpace record that via CGRs references the respective CGTs (one for each symmetry). Depending on the context, the combined set of these $q$-labels is also collectively referred to as $q$-labels for simplicity.

*Degenerate $q$-labels*:   A CGT has well-defined $q$-labels on all its legs. For a given CGT then, if multiple legs have the same $q$-labels as well as the same directions, this is referred to as *degenerate $q$-labels* in QSpace [15]. This is important for the interplay of CGRs with sorted CGTs.

**Scalar operator**   A scalar operator is an irop that transforms like the scalar representation $q = 0$. Therefore it consists of a single operator where the trailing singleton irop dimension on the third index can be trivially skipped. Examples include Hamiltonians or density matrices.

*A*-**tensor**   When fusing two (incoming) state spaces into the combined (outgoing) state space, this is described by a rank-3 tensor. The pairwise fusion of state spaces is a generic operation for any type of tensor network. When adhering to an LRs index order as with an MPS, this is frequently referred to as *A*-tensor in QSpace, and thus also in this documentation. A call to getIdentity() returns an *identity A*-tensor which describes the full, i.e., untruncated combined state space based on an identity matrix in multiplet space that is sliced up into RMTs [8]. As such, this represents a map from input multiplets to fused multiplets.

**QSpace record**   A QSpace consists of a listing of non-zero blocks (App. A.3), each of which has well-defined symmetry labels on all legs as discussed with Eq. (16). In a QSpace tensor X these entries are referred to as *records*, indexed by i = 1, ..., $n_X$ with $n_X$ the total number of blocks. Record $i$ then consists of the RMT X.data{i} together with symmetry labels X.Q{l}(i,:) on leg $l$. The QSpace display as in Fig. 7 shows one line per record.

**Vacuum state**   The vacuum state, like in the absence of any sites, always carries the scalar representation for all its symmetries. Hence it is described by the trivial symmetry labels $q = 0$. The identity operator for the vacuum state based on the symmetry configuration specified with reference to the symmetries of some existing QSpace tensor X can be obtained via getvac(X).

**Zero blocks and scalar operators**   When exploiting symmetries, tensors decompose into a block structure since many matrix elements are zero because of symmetry. If entire blocks are zero, because forbidden from a symmetry point of view, these are naturally absent in a

tensor library that exploits symmetries. However, there can also be blocks that are permitted by symmetry, but are zero nevertheless, in the sense that the RMT's Frobenius norm is below a threshold representing numerical noise. Such 'zero blocks' are generally also skipped, with one notable exception: scalar operators. Diagonal zero blocks in scalar operators are kept when present, by default, to avoid confusion. For example, a Hamiltonian may have zero eigenvalues for an entire block, more likely so for small state space such as a single site. Such blocks with all-zero eigenvalues contain relevant state space information that is generally of interest and, hence, is not skipped. This behavior is a safety measure against 'missing' state space. See also `skipZerosQS`.

### A.3   `QSpace` data structure

The data structure of `QSpace` within the MEX routines is represented by an extensive C++ class. On the Matlab interface, this is mapped onto a Matlab structure that, for convenience, is cast into a Matlab wrapper class `@QSpace`. Assuming a QSpace tensor `X` with $i = 1, \ldots, n_X$ records, within Matlab its data is organized as a structure object with the following fields :

- `X.data` contains the RMTs: it is a cell array where `X.data{i}` contains the RMT for record i. It has `double` floating point precision (interleaved double in case of complex matrix elements). With reference to symmetries, this corresponds to the block $\|X\|_q$ in Eq. (16) with $q \rightarrow$ i.

- `X.Q` contains the symmetry labels ($q$-labels): QSpace insists that all symmetry labels are plain integers ($q_i \in \mathbb{Z}$). In the QSpace MEX core routines, `X.Q` is represented as a C++ signed `int` array (4 bytes), thus $|q_i| \lesssim 10^9$. Like `X.data`, the field `X.Q` is also a cell array. In this case, however, `X.Q{l}(i,:)` contains the combined set of symmetry labels for record i on leg $l$.

- `X.info` represents a substructure with the following fields [e.g., see line 47 after Eq. (35)]:

  - `qtype` contains the symmetry setting in compact string notation (comma-separated list without white space, containing entries such as `'SU2'` for SU(2), or `'A'` for abelian which is taken synonymous for U(1), etc.). If `qtype` empty, the fully abelian setting of all-U(1) symmetries is assumed (see below).

  - `otype` contains 'operator type' (mostly a safeguard and this sense optional): values include `'operator'` to emphasize that given QSpace is not any rank $r \leq 3$ tensor, but represents an operator, either scalar rank-2 or an irop of rank 3 that adheres to index order conventions. In case a particular user-created tensor satisfies the definition of a standard operator of rank 2 or 3, the `operator` flag may be set manually via `X.info.otype='operator'`. This field accepts one other much lesser used value, `'A-matrix'`. This serves to emphasize that given rank-3 QSpace is an identity *A*-tensor with an MPS in mind.

  - `itags` (mandatory field) cell array of strings that contains the `itags` including *q*-directions via trailing asterisks `'*'`. As of QSpace v4, there must be one `itag` for each leg in the order of legs as present. Hence the number of `itags`, i.e., the length of the cell array, must match the number of entries in `X.Q`. This must hold even if the `itags` per se are empty, because of the additional information of the trailing markers such as *. Itags are restricted in length to at most 8 characters (excluding the trailing *) since within the QSpace core library they are directly mapped into numerical IDs of type `unsigned long` for convenience. The sign bits are used for other internal purposes. Hence the character set is restricted to the ASCII range

33-126 up to marker characters. As such this includes any case-sensitive alphanumeric strings with simple separators, yet without white space.

- – `cgr` is a structure array that contains the CGRs including the *w*-matrix as stored in the subfield `cgw` (always in `double` precision and real). The field `cgr` is only meaningful for non-abelian symmetries. Hence, if empty, the fully abelian setting of all-U(1) symmetries is assumed (see below). It is non-empty in the presence of (also) non-abelian symmetries. In this case the number of columns in `cgr` equals the number of symmetries used in the order specified during setup (cf. `getLocalSpace`), i.e., the entry `X.info.cgr(i,s)` represents the CGR for record i and symmetry *s*. This also includes any abelian symmetries present then, with trivial CGRs generated on the fly for the sake of overall code homogeneity.

- – `ctime` creation time stamp of `QSpace` object (for tracking purposes only).

*Default U(1) symmetries* – If the field `info.cgr` together with `info.qtype` is empty, the abelian setting of all-U(1) symmetries is assumed. In this case, the symmetry labels in the matrix `X.Q{l}` for leg *l* refer to a separate U(1) symmetry for each column then. This corresponded to the original setting in `QSpace` v1.[10]

To reveal the bare data structure of a `QSpace` `X`, one may call `struct(X)`, with an example shown with Eq. (35) and the subsequent lines. Since this data structure is technical and therefore rather unreadable in raw format, `QSpace` provides the `display` routine. Matlab invokes it automatically whenever an object is displayed, e.g., as part of an output or just by typing the name of a `QSpace` tensor. This conveniently extracts and displays the most relevant information in the `QSpace` data structure in a formatted output. See Fig. 7 for an example.

**Interplay of `QSpace` tensor and `RC_STORE`** A CGR references a CGT in the `RC_STORE`. However, since the latter only stores sorted CGTs, a CGR implicitly also stores a permutation (potentially including a conjugate flag) to be incorporated just in time when required with the sorted CGT. In this sense, the CGR, while referencing a sorted CGT, nevertheless can represent an unconstrained CGT that is consistent with any index order of given `QSpace` tensor. Note that this procedure also *defines* any unsorted CGT: it equals a sorted CGT with the given permutation (and possibly conjugation) applied. The presence of degenerate *q*-labels (i.e., when the same symmetry label is present at multiple legs) leads to subtleties that are properly dealt with in `QSpace` [15]. The setting above, based on 'reference plus permutation' is explicitly used by the CGRs within the `C++` core library. This becomes somewhat hidden and thus is implicit in the structure returned to Matlab: the *q*-labels and *q*-directions shown in `info.cgr` *already* have the permutation (and conjugation flag, if any) applied to them, while nevertheless also storing IDs and high-resolution time stamps to the sorted CGT in the field `cgr.cid`. By re-sorting the *q*-labels in the CGR, the `C++` code regenerates the relevant permutation and identifies the underlying sorted CGT. This permits `QSpace` to translate the Matlab CGRs in `info.cgr` into proper pointer references in the MEX core libraries. By also providing the IDs, this ensures and subsequently enforces that the correct underlying `RC_STORE` is present for consistency.

---

[10]The field `info.cgr` was added with `QSpace` v3, whereas `QSpace` v2 still had the CGTs stored with the `QSpace` tensors themselves without any CGRs [8].

# B  QSpace Installation and Environment

## B.1  System requirements

### B.1.1  Matlab and Unix-like environment

The QSpace core library is written in C++. This is embedded into Matlab via the MEX API which is included with the basic Matlab environment. All of the configurations assume a Unix-like environment, such as Linux (assumed, by default) or also macOS, both within a bash shell environment and standard access to Perl. To be specific, QSpace does not yet run on a Windows operating system, unless Linux is emulated.

Up to the current version QSpace v4, one needs access to a fully functional Matlab environment. A basic Matlab license suffices for this, i.e., there is no need for any particular Matlab toolbox. A potential exception is the Matlab compiler toolbox for MCC compilation if an application is intended to be run on a high-performance computing (HPC) cluster environment. If the Matlab license permits one to run arbitrarily many Matlab instances, there is no need for the MCC toolbox. By the setup above, QSpace requires an environmental setup for Matlab, e.g., see the `./system` folder provided with the git repository. In particular, the Matlab script `system/ml.m` shows how Matlab using QSpace can be started within a terminal without having to load the full Matlab desktop).

The QSpace functionality is fully packaged into hard-coded MEX routines as a wrapper for Matlab intended for tensor network applications. These binaries can be called directly from within Matlab like any other Matlab command. This was the philosophy of QSpace from its very inception: by embedding it into Matlab, this significantly enhances the user-friendliness of QSpace. This makes it efficient to test new tensor network algorithms right on the Matlab prompt, or run and debug Matlab scripts that use QSpace. This permits one to focus directly on tensor network algorithms that fully exploit complex symmetry settings, rather than having to worry about the underlying implementation. The inner workings of the C++ core tensor library are mostly hidden from the point of view of a user application within Matlab.

**Matlab terminal mode**  Matlab can be started either with the full-fledged graphical Matlab desktop, or within an existing terminal (referred to as 'terminal mode'). The latter still permits all graphical capabilities, such as plotting figures which respond to the environmental system variable `DISPLAY`. The embedding of QSpace into Matlab was developed in this slim mode. Hence this is also the recommended way of using QSpace with Matlab. For one, this is light-weight and hence beneficial when working remotely. On the other hand, by directly using a regular terminal for the Matlab prompt, this also permits color coded log output for the sake of readability (the latter is based on printed escape sequences which are supported by terminals, but not the Matlab desktop).

**Matlab scoping**  The git repository provides a full Matlab environment for QSpace, including many convenient Matlab utility routines in `./lib`. As this may interfere with other user-specific environments within Matlab, the QSpace Matlab environment may be wrapped into a package as this supports namespaces to isolate project environments (not included with the QSpace git repository itself at the current stage). Using `import` statements should allow one to access to model class or function names defined in other scopes (packages) without having to use fully qualified references throughout. At a lower level, including or excluding the QSpace directories into the Matlab `path` as in `./startup.m` has a similar effect.

### B.1.2  C++ libraries required for MEX compilation

Since MEX files are compiled for usage in Matlab, it is advised to link against external libraries as provided with the Matlab install where present. The main reason is that these libraries are modified such that they can properly deal with error handling consistent with the Matlab ecosystem (e.g., an error should not terminate an entire Matlab session, but rather return to the Matlab prompt). The required dependencies are specified in `Source/Makefile`. It consists of the following three standard libraries:

**GMP/MPFR multi-precision libraries**   QSpace uses a higher-precision data format for all its CGTs based on the GNU GMP/MPFR multi-precision library [44,45]. All CGTs in the `RC_STORE` are stored in this format, using quad precision at 128 bits. The MPFR library is linked from the location of the Matlab install under `MATLAB_ROOT`. See `Source/Makefile` for more details. By contrast, the GMP library is linked from the operating system for technical reasons, hence must be installed if not present (while the GMP is also present under `MATLAB_ROOT`, it does not come with a header file there, besides that it also appears to have non-standard mathworks/Matlab adapted `'mw_*'` naming conventions).

**LAPACK/BLAS libraries**   QSpace uses the standard LAPACK/BLAS library [82] as provided with Matlab (`-lmwblas`, `-lmwlapack`; see `Source/Makefile`). From a coding point of view, they behave identically to standard LAPACK/BLAS routines. It is *a priori* unclear, though, how Matlab itself uses these libraries internally. Therefore at times minor performance differences may be observed when benchmarking with native linear algebra routines at the Matlab prompt. For example, there exist multiple LAPACK routines with and without divide-and-conquer schemes, such as `dgesvd` vs. `dgesdd` for singular value decomposition.

**OpenMP library**   QSpace permits parallelization on top of standard parallelization in library routines such as LAPACK/BLAS, based on the *open multi-processing* (openMP [83]) framework to parallelize loops. Together with the parallelization within LAPACK/BLAS this results in nested parallelization. The library `(lib)iomp5` is provided with Matlab and hence also linked from there. See `Source/Makefile` for more details.

## B.2   Download and install

The public git repository for QSpace includes the C++ core sources, as well as the extensive source for the Matlab wrapper environment. As such there is no automated installation process per se for QSpace. The git repository may be pulled into a dedicated QSpace directory in one's Matlab home directory, as specified by the environmental variable `MYMATLAB`. References to directories or files in the git repository therefore, by default, assume `$MYMATLAB` as the root directory. After setup of the system environment, only the MEX binaries need to be compiled in `./Source` ≡ `$MYMATLAB/Source/` as discussed below.

Currently, there are two (nearly identical) public git repos available under the QSpace project [1] https://bitbucket.org/qspace4u/

- `qspace-v4-pub/` – main repository developed on and intended for a Unix-like environment. It contains the most recent and up-to-date QSpace version, yet without any MEX binaries. Thus, The MEX files must be (re)compiled by the user. This is also the setting associated with this documentation.

- `qspace-v4-osx12-monterey/` – duplicate of the above intended for macOS, which also includes compiled MEX binaries for Intel CPUs. These `*.mexmaci64` binaries are compiled with OSX12/Monterey on Intel-based Macs. Hence as of QSpace v4.0, the ARM

architectures (M1, M2, etc.) are not yet supported. For updates, please consult the documentation with the repository. The advantage of this repository is that, assuming that all required libraries are present, `QSpace` should work *as is* without the need for any additional compilation. Yet because this git repository also contains binaries, it is not updated as frequently. Because macOS is unix-like, one can always clone the repo `qspace-v4-pub` above instead, and (re)compile for macOS.

### B.2.1 System environment setup

The setup of the `QSpace` system environment is centralized around `system/matlab_setup.sh` (bash shell script). As this file is part of the git repository, it should not be edited. By default, it expects a bash script `system/matlab_setup_user.sh` which is not part of the git repository, to avoid that it gets overwritten during git updates. Hence this file needs to be generated. It is the only file that should be edited to configure the environment (see environmental variables for more). A template is provided in `system/matlab_setup_user.sh-template`. In summary, the following is the recommended procedure to set up the `QSpace` system environment (command displays are shown with blue background here to indicate `bash` script),

```
1    cd $MYMATLAB/system/
2    cp matlab_setup_user.sh-template matlab_setup_user.sh
3    edit matlab_setup_user.sh   # use your favorite plain-text editor
```

where the path `$MYMATLAB` specifies the location of the `QSpace` repository. This reflects the environmental variable of the same set in `matlab_setup_user.sh` later. The edit in line 3 only requires very few lines to change when using the specified template. It can be done with any plain text editor (in the case `edit → vim`, the repository also provides the optional `system/.vim` and `.vimrc` to configure `vim`; for these files to take effect, one may `rsync` them to one's home directory via `rsync -irp ./system/.vim* $HOME/`). If one wants to use a different file other than `matlab_setup_user.sh`, then the script `system/matlab_setup.sh` permits one to define the environmental variable `QS_CONFIG_ML_SH`, instead. This then needs to specify the alternative user file name with a fully qualified path if not located within the `system/` folder of the repository.

The setup is finalized by the Perl script `system/matlab_setup.pl` which is automatically called at the end of `system/matlab_setup.sh`. Again since it is part of the repository, the user should not edit it. This script cleans up the setup, in that double-checks and completes the setup where possible. For convenience, one may define an alias (feel free to rename),

```
4    alias mlsetup="source $MYMATLAB/system/matlab_setup.sh"
```

This may also be added to one's `$HOME/.bashrc` setup for later usage. Once the edit in line 3 is complete and the alias above is defined, then the simple command

```
5    mlsetup
```

should suffice to configure the required `QSpace` Matlab environment in the current shell. Because the environmental setup is centralized around `system/matlab_setup.sh`, the call to `mlsetup` is sufficient to setup the Matlab environment whenever one deals with `QSpace`. This includes the regular usage of `QSpace` (e.g., see `system/ml`), but also compilation (e.g., see `Source/Makefile` or also `MCC/Makefile.template`).

**Potential hickups**   The environmental setup of `QSpace` makes certain assumptions that may lead to potential hiccups. These concern the system `PATH`:

- *Include current directoy* – The environmental setup of QSpace assumes that the current directory `'.'` is included in the system PATH. This concerns, for example, the usage of auxiliary perl scripts located in `./system` or `./Source`. Hence the system PATH should include the current directory `'.'`, e.g., with lower priority towards the end of PATH.

- *Matlab programs* – It is recommended to have `$MATLAB_ROOT/bin` in the system's PATH with higher priority, i.e., by listing it to the front of `$PATH` to avoid mixups with programs that accidentally share the same command name, like mex, etc.

### B.2.2   MEX compilation

Once the git repo is cloned and the environment set up, the last step is to (re)compile the MEX files. The recommended way for this is via `./Source/Makefile` at the level of the operating system, which calls Matlab's mex command from there (in principle, mex also could be called from within Matlab). In either case, this first requires the Matlab environment to be configured as described above, since the Makefile makes use of this.

The MEX compilation based on the mex command requires a C++ compiler installed on the system. Matlab has constraints on which C++ compilers are supported and compatible with a given Matlab release. Hence the recommended way to choose a compatible C++ compiler is via Matlab's setup in this regard,

```
6   mex setup C++      # required once for a given Matlab release
```

which may be run either from a terminal, e.g., after a call to mlsetup, or from the Matlab prompt. Note that the QSpace Makefiles assume gcc on Linux and Xcode/clang on macOS. The mex setup above looks for compatible C++ compilers on the system. If more than one compiler is found, this becomes an interactive process where one needs to choose. The information on the chosen compiler is then stored within one's $HOME directory. Hence this only needs to be run once for a given Matlab version. However, since the selection of a compatible C++ compiler is Matlab release specific, the setup in line 6 needs to be repeated whenever one switches to a new Matlab version.

Once a compatible C++ compiler has been chosen, one can proceed to compile the MEX files in a terminal. A test compile (based on `helloworld.cc`) can be run first to double-check whether the Matlab setup proceeds without error,

```
7   cd ./Source
8   mlsetup && make test    # compiles helloworld.cc into MEX file
```

When the above commands run successfully, this can be followed by the full compilation

```
9   make -B all             # may add option -j $np to parallelize compilation
```

This compiles a total of about 40 MEX files. About half of these are QSpace core routines that are compiled into the `./bin` folder with file names ending in QS, thus also referred to as QS-routines. The remaining utility routines are compiled into the `./util` folder. The full (re-)compilation can be rather time consuming when run sequentially. However, since there are no dependencies across the MEX files, the compilation of the MEX files can be trivially parallelized to speed up the process. This is achieved by specifying the option `-j $np` to make in line 9 where np specifies the number of compilation targets to run in parallel. A sensible choice for np will depend on the number of available cores on one's system.

**Safeguards and assertions**   The `C++` core library of QSpace includes many internal safeguards to ensure consistency. These safeguards usually correspond to simple and thus fast checks. In this sense, the overhead should be minimal. Some of the more severe checks can be turned off, however, by defining `__WB_SKIP_ASSERT` in `wblib.h`.

### B.2.3   Matlab startup

A typical Matlab startup proceeds as in the bash script `system/ml` provided with the repository. With the QSpace repository located at `$MYMATLAB`, starting Matlab from that directory (`cd $MYMATLAB`) automatically sources `./startup.m` which is important to set up the QSpace environment *within* Matlab, e.g., by properly extending the Matlab path. Since this script is part of the git repository, it should not be altered. For that purpose, at the end of the script `./startup.m`, it looks for another script `./startup_loc.m` in the same directory, and if present, also runs it. This is intended to add automated user-specific configuration to one's Matlab environment, by generating and editing that file (using one's favorite text editor),

```
10 │   edit ./startup_loc.m        # optional Matlab specific user configuration
```

As seen in `system/ml`, it first sources `system/matlab_setup.sh` (equivalent to `mlsetup` in lines 4-5 above), switches to the directory `$MYMATLAB`, and then starts Matlab from there. In the case of `system/ml`, this starts Matlab into a terminal without the full desktop environment (by specifying the option `-nodesktop`) and without the Matlab splash window during startup (`-nosplash`). With the `DISPLAY` variable active, i.e., not disabled via `-nodisplay`, the full graphical interface for figures is available, nevertheless. Running `system/ml` without any errors or warnings is an essential first assurance that QSpace is set up properly.

### B.3   QSpace Environment within Matlab

The extensive `C++` code base provides the QSpace core functionality. This is packaged into elementary MEX routines to run tensor network algorithms within Matlab. By convention, all MEX routines that deal with elementary QSpace tensor operations have a trailing `QS` to their name, e.g., as in `eigQS()`. They are referred to as `QS`-routines, and are located in the `bin/` directory of the QSpace repository. Other MEX utility routines are collected in the `util/` directory. These have no trailing `QS` extension to their name. Both directories are included in the Matlab path via the `startup` script in `$MYMATLAB`.

**@QSpace wrapper class**   The QSpace class in `Class/@QSpace` wraps the `C++` QSpace class object after being cast into a Matlab data structure by the MEX routines. For the sake of the discussion in this documentation, to differentiate it from the `C++` counterpart, this Matlab wrapper class is frequently referred to as `@QSpace` or simply also `QSpace` with differentiated color-coding to further emphasize the Matlab class context where important.

By also overloading a range of Matlab functions, the @QSpace class simplifies operations in Matlab and therefore makes QSpace considerably more user-friendly. Simple operations on QSpace objects are directly coded within Matlab in @QSpace, while it leaves the heavy-duty weight lifting to the MEX routines. The class constructor `QSpace()` together with the more than a hundred other class routines in `Class/@QSpace` [cf. Sec. F.1] ensure that the proper QSpace tensor structure is maintained consistent with the MEX core routines. Last but not least, this wrapper class also provides a compact formatted display of QSpace objects that summarizes the most important information about a QSpace tensor (e.g. see Fig. 7 in Sec. 3.2).

For a QSpace `X`, the command `X=struct(X)` strips it down to the underlying plain structure object in Matlab [see (35) for an example]. Conversely, given such a structure, it can be cast

back to a class object via `X=QSpace(X)`. By construction, @QSpace methods only apply to, and hence can only be called on QSpace objects. By Matlab convention, this implies that an @QSpace member function is called if at least one QSpace object is present in the input argument list. It does not necessarily have to be the first argument, though.

*MEX files return structures*: Matlab does not permit MEX routines to return a user-defined Matlab class object like `QSpace`. Hence QSpace objects out of MEX returns can only be returned as a structure object.[11] For that reason, very simple wrapper routines exist for `QS`-routines, that effectively behave like `foo(X)` ≡ `QSpace(fooQS(X))`, with `contractQS` an example. The name for these routines is the same as for the respective MEX routine up to a trailing `'QS'`, i.e., `fooQS()` becomes `foo()`. To locate uniquely which function gets called, one may use Matlab's command `which` in any context.

More generally, when an @QSpace routine `foo()` is called, it operates on an input that includes at least one QSpace object. It may include calls to multiple MEX routines or none at all. If QSpace tensors are returned, they are usually cast into `QSpace` objects if they exist as plain data structures, e.g., out of MEX routines. The @QSpace routines typically perform additional tasks coded within Matlab. Therefore the type and order of the return arguments are not necessarily the same as compared to a respective MEX routine called. For example, `eig()` mimics the output of Matlab's native `eig()` routine rather than that of `eigQS()`.

## B.4 MEX environment

`MEX` routines are C++ compiled binaries that can be called from within Matlab like any other Matlab function. From a C++ point of view, the standard entry point

```
0    int main(int nargin, char **argin) { ... }
```

is replaced by

```
1  void mexFunction(
2      int nargout, mxArray *argout[],
3      int nargin, const mxArray *argin[]) {
4      ...
5  }
```

where the Matlab arguments via the generic data type `mxArray*` are handled via the Matlab API. The input in line 3 is by `const` pointer, and hence *by reference* and read-only. This way QSpace can avoid copies for input QSpaces where read-only suffices. All MEX routines in QSpace are also effectively wrapper routines that package a particular functionality of the C++ QSpace sources. This way, all MEX files are typically rather simple C++ files themselves.

**Memory management** Once a function is called in Matlab, it stays resident in memory so that later calls gain performance. The same also holds for MEX routines, with the effect that global variables in MEX routines are *persistent* in memory, i.e., they keep their data from one MEX call to the next. QSpace widely exploits this when dealing with the `RC_STORE`, because data can be read on demand once and for all. Nevertheless, this data cannot be shared across different MEX files. Functions and their associated memory spaces can be cleared in Matlab via `clear functions`. This triggers standard C++ garbage collection, which ensures that the global data with MEX routines is also properly released. It is important to note, though, that

---

[11]Effectively, the casting into a `QSpace` Matlab class and the corresponding error handling needs to occur within the Matlab session, and cannot be outsourced to the MEX API. The situation is different when MEX routines store data into files, since Matlab does permit one to save a QSpace data structure as a designated `QSpace` object into a `mat`-file. The casting into the Matlab run-time object then occurs when the `mat`-file is loaded.

QSpace *does not* use the memory allocation routine provided by the Matlab API, but rather uses standard C++ memory management. Hence while `clear functions` does release all memory allocated by the MEX functions via the underlying C++ class destructors, this is not necessarily visible in the memory consumption of the Matlab process as a whole from the perspective of the operating system. Subsequent MEX calls may reuse freed-up memory. The detailed behavior here, however, very much depends on the particular operating system.

**QSpace avoids copies of arguments at MEX interface**   The MEX API hands over input variables by constant reference (line 3 above). This is used by QSpace to avoid generating explicit copies of input QSpace tensors where possible. That is, in Matlab spirit, QSpace copies are generated *just in time* (JIT) only where required. The data format of generalized column-major for tensors is consistent across all Matlab and QSpace arrays. With Matlab's switch to interleaved complex data since 2018, this is also consistent with the data format for complex numbers in standard LAPACK/BLAS routines. Similarly, also the output in line 2 above is handed over as a pointer. Therefore to avoid copying output data back to Matlab, output QSpace tensors are created in MEX-compliant data spaces as soon as possible within the C++ core routines.

**Error handling**   MEX routines temporarily switch the error handling from Matlab to standard C++ which also concerns handling interrupts such as `Ctrl-C`. Errors are therefore caught within C++. Before returning to Matlab, however, the Matlab-specific signal handlers are restored. This is ensured by an outer try/catch block in every MEX routine.

When interrupting Matlab while inside a MEX routine, the behavior is typically as follows: When `Ctrl-C` is pressed the first time, (1) this instantaneously prints a message that confirms that the interrupt was received. However, the MEX routine will continue and still try to finish its current step until the next point that is natural for the MEX routine to terminate, e.g., after some finished iteration in a loop. When pressing `Ctrl-C` additional times, the MEX routine gives (2) a warning that the MEX routine will be terminating immediately, and (3) actually terminate immediately. Enforcing the termination is considered unsafe, though, with potentially erroneous behavior. This motivates the above behavior.

**Warnings**   Warnings in MEX routines are typically general `wblog` messages to `stdout` both within Matlab scripts, as well as in MEX files with the tag `'WRN'` in them. In MEX routines, this reflects standard output, though. Hence this will typically not be recognized by the Matlab debugger as a warning (this is in contrast to MEX errors `'ERR'` which always are dealt with as an error also within Matlab by throwing an exception). Only in very few instances, does a warning within a MEX function also escalate to the Matlab environment as a proper Matlab warning that can be caught via `dbstop if warning` with the Matlab debugger. So far this only concerns `contractQS`. This MEX routine issues a proper warning when it tries to automatically fix missing `conj` flags when contracting tensors, or when regular expressions operating on `itags` within `contractQS` have no effect because they do not match. Since this should be checked and fixed in the Matlab code right where it is called, it is useful to be able to stop the Matlab execution there.

## B.5   HPC environment and parellization

**MCC compilation**   The Matlab compiler (MCC) toolbox can be acquired with Matlab. It permits one to decouple an arbitrary number of cluster jobs from Matlab license requirements since the Matlab license is required only for the MCC compilation. This collects all relevant code from the Matlab path, including MEX binaries. It builds a standalone application that can

eventually be deployed without a Matlab license. While there is no runtime speedup in this process, the MCC compilation output contains everything required for a cluster job to run.

**Parallelization (multiple jobs)**   When running multiple jobs, it is highly recommended to have a mostly complete RC_STORE that is used in a centralized fashion, e.g., over a network drive, in combination with a differential setup with job-specific local storage (see more on this with the environmental variable RC_STORE). If an RC_STORE needs to be built from scratch, a good practice is to run a test job upfront which generates most of the symmetry-related data. When post-analyzing job data where one still also non-trivially operates on QSpace tensors, e.g., via contractions, then the original RC_STORE must be accessible.

**Parallelization (single job)**   QSpace parallelizes at the level of MEX routines which is motivated by the following: QSpace tensors intrinsically consist of lists of blocks (RMTs) each of which is stored in full, generalized column-major format. When working through such lists, e.g., in the context of contraction, eigen- or singular-value decomposition, it is desirable to parallelize the respective loops. For the RMTs, this amounts to mostly trivial parallelization. For example, in the context of contractions there are frequently a large number of pairwise contractions of RMTs to be performed which have matching symmetry sectors in the contracted indices. In the presence of multiple symmetries and rank $r > 2$ tensors, this can quickly reach hundreds or many thousands of individual pairwise contractions [34].

In this sense, *outer loops* are parallelized in QSpace using openMP. This responds to the environmental variable QSP_NUM_THREADS. At a lower level, also the elementary steps of operating on individual (pairs of) RMTs are parallelized simply by linking the standard LAPACK/BLAS libraries. These parallelize themselves, typically responding to OMP_NUM_THREADS (since Matlab has a tendency to overwrite or ignore those, one may rather use setNumThreads.m, instead, as provided with the QSpace repository). Together with the openMP parallelization, this gives rise to *nested* parallelization, in that every worker in the loop itself can parallelize.

Parallelizing loops, ideally, relates to trivial and thus efficient parallelization. This is mostly the case for the RMTs, and hence also in the presence of only abelian symmetries, where all symmetry-related constraints can be quickly obtained on the fly. For non-abelian symmetries, however, QSpace makes use of an RC_STORE that needs to be kept consistent across all workers. Therefore, critical symmetry-related operations that may lead to updates in the RC_STORE need to be forced into serial mode via `critical` sections or object-specific openMP locks (mutex and semaphores) for synchronization. Since symmetry-related operations can trigger a range of *a priori* unexpected additional actions, this slows down the parallelization of loops in QSpace for larger QSP_NUM_THREADS. Hence typically, one may use QSP_NUM_THREADS $\lesssim 8$.

From the above outset, the parallelization is optimized by combining openMP with native parallelization of LAPACK/BLAS routines. Much of this parallelization is CPU-centered still. Hence as of now, QSpace does not support GPU parallelization [84]. If in the future openMP, together with LAPACK/BLAS, seamlessly supports GPUs, this also should enable access to GPUs for QSpace.

**Subtleties and potential issues with parallelization**   Tensor operations can trigger an *a priori* unexpected set of additional actions that need to be properly synchronized across different threads when parallelizing. The following, for example, shows how contractions can trigger tensor product decompositions which may result in proliferating requests for CGT-specific locks, and worst case, in deadlocks.

To be specific, consider the example of contracting two tensors. The contraction needs to be performed both, at the level of RMTs, as well as at the level of CGTs for each non-abelian symmetry involved. Because the latter are tabulated, most CGTs may be present already from

earlier simulations. But there is a chance that new symmetry-related data is created. Therefore when a particular CGT resulting out of the contraction is requested as a reference (CGR) from the internal buffer in memory, there needs to be a lock on this particular object in the buffer: after all, it may not be in memory yet e.g., because a simulation just started. The thread thus (i) issues an object-specific openMP lock and then (ii) reads the object from the buffer. If present, it can release the lock. If not yet present, the thread keeps the lock active, and (iii) tries to read the CGR from the `RC_STORE`. If not present there either, the thread (iv) will generate the requested CGT, e.g., by performing the contraction explicitly and tabulating the result for the future, before releasing the lock.

With the further assumption that the contraction results in a rank-3 tensor that has two incoming and one outgoing index or vice versa, this reflects a standard CGC that can also be obtained from tensor product decomposition, instead. If at least one input multiplet is of smaller dimension, in the sense that it may reasonably occur in a local state space of a site, `QSpace` does not obtain the CGT by contraction, but (v) rather gives preference to the full tensor product decomposition, as this automatically also generates full outer multiplicity where present. This is carried out by the present thread in addition to (iv) above. The thread then (vi) projects the contracted CGT in (iv) onto the CGT out of the tensor product decomposition in (v), before it finally releases the lock.

Now to further complicate things, a tensor product decomposition not only generates the particular CGT of interest, but typically also additional other CGTs. Accidentally, some of these may coincide with CGTs that occur in the present tensor contraction as encountered by *other* threads which simultaneously also try to acquire locks themselves. Worst case, such scenarios can result in deadlocks if `QSP_NUM_THREADS` is chosen too large. This situation, however, can only occur if a sufficiently large number of new objects needs to be generated for the existing `RC_STORE`. Hence a pragmatic way to deal with competing locks as described above, is to test run a particular simulation first in serial mode by unsetting `QSP_NUM_THREADS`. This will generate most of the required entries in the `RC_STORE`, which then can be made accessible globally in the spirit of a differential storage setup via the environmental variable `RC_STORE`.

## B.6 Environmental variables

The following environmental variables are specific to `QSpace`. They are defined when setting up the Matlab environment for `QSpace` via `system/matlab_setup.sh`. The mandatory ones are indicated, otherwise default values will be auto-determined by `matlab_setup.pl` which finalizes the setup. As these variables are used in various places either during MEX compilation, startup of Matlab, or while using `QSpace` within Matlab, they are expected to be exported to sub-shells (e.g., using bash `export`).

**ARCH** system architecture (auto-determined based on system call `uname -sm`, by default) – Used in `Source/Makefile` to distinguish between system environments as specified with Matlab in `$MATLAB_ROOT/sys/os/$ARCH/`, such as `glnxa64` for Linux or `maci64` for macOS.

**LMA** local Matlab data directory (path, mostly optional) – In some applications provided with `QSpace`, this expects the designation of a (local) Matlab directory intended for data storage. For example, used with `'cto lma'` to change directories ('change to') which then looks for the environmental variable in upper case letters `LMA`. E.g., see line 61 with Sec. 5.6 above.

**MATLAB_ROOT** Matlab root directory (path; mandatory) – The location of the Matlab install: if multiple Matlab versions are present, this typically includes the Matlab version string. The Matlab binary is expected to be located in `$(MATLAB_ROOT)/bin/[matlab]`. The variable

`MATLAB_ROOT` is essential for the compilation via the `Makefile`s since these need to link to headers and libraries in the Matlab install.

`MCC_TAG`   system dependent subdirectory in `./MCC` used with MCC compilation (string which defaults to `'bin*'` based on the MEX file extension `[mex]*`).

`MYMATLAB`   location of the cloned QSpace git repository (path; mandatory) – Since the repository also includes an extended Matlab environment with wrapper and utility routines from within which applications and simulations are run, this is referred to as 'my Matlab', typically located somewhere within the HOME directory like ~/Matlab/.

`MEX`   location of MEX / C++ source directory (path; defaults to `./Source`).

`QSP_NUM_THREADS`   (non-negative integer) – Number of threads to be used with QSpace on top of standard LAPACK/BLAS parallelization (which itself parallelizes based on `OMP_NUM_THREADS`). This gives rise to nested parallelization. If `QSP_NUM_THREADS` is not set, no such parallelization is performed.

`QS_LOG_COLOR`   Enables/disables color coding (intended for Matlab terminal mode only, assuming a dark terminal background) – By default, the color coding is turned on in interactive terminal mode where it may be disabled by `export QS_LOG_COLOR=0`. The color setting affects both, the output of MEX routines, as well as log entries purely within Matlab, e.g., based on the `lib/wblog.m` logging routine.

`QS_CONFIG_ML_SH`   Optional specification of bash configuration file for Matlab system environment (used by `system/matlab_setup.sh`; defaults to `system/matlab_setup_user.sh`).

`RC_STORE`   The environmental variable

$$RC\_STORE=path1:...:pathN$$

points to the location where symmetry-related data is written or, subsequently, read. The variable `RC_STORE` contains one or multiple paths separated by a colon `':'` for the sake of a *differential* storage setup, where the last path is assumed to contain the newest symmetry-related data if any. To be specific, the lookup of symmetry-related data proceeds as follows,

$$\underbrace{pathN \rightarrow path1 \rightarrow ... \rightarrow pathN\text{-}1}_{\text{read until found}} \overset{\text{not found}}{\longrightarrow} \underbrace{pathN}_{\text{write}}$$

When QSpace requires a particular symmetry related object, this starts by looking into the *last* (because newest) path, `pathN`. If the object looked for is found there, it is read and returned. If the object is not found, QSpace proceeds to the *first* and then the subsequent directories until the object is found. If nowhere found, then the desired symmetry-related object is computed, where by the iterative nature of QSpace, all required input for this is assumed to be present from the earlier RC_STORE history. The result is then stored in the *last* path, `pathN`, in `RC_STORE` and, finally, also returned.

   The above procedure permits a differential setup where all paths except the last one (`pathN`) are read-only. New or updated data is written to the last path only. For an interactive Matlab session, a single path is recommended. For cluster jobs, the recommended setting is two paths: `RC_STORE=(global_path):(local_path)`, where `global_path` points to an `RC_STORE` that is

mostly complete in the required symmetry-related data based on earlier calculations. This may be available from a network drive, where required entries are typically read once and for all, and stored in memory (larger CGT objects are frequently purged from memory, though, since CGRs are typically all that is needed once no more new OM components are generated for the CGTs under consideration). By contrast, the `local_path` is meant to be a local directory with the cluster job. The major reason for this differential setup above is to avoid race conditions and thus clashes across different cluster jobs operating on the same RC_STORE [newly generated CGTs get assigned IDs which are referenced elsewhere, such as in X-symbols; therefore these must be consistent across the entire RC_STORE; they may get corrupted in case of race conditions from parallel executions of different simulations, e.g., cluster jobs; by contrast, the parallel execution within threads of the *same* simulation uses OMP critical sections, as well as other OMP locks based on mutex and semaphores for synchronization].

Once all jobs are finished, their RC_STORE have diverged from each other if `local_path`, is non-empty. This way they have become incompatible with each other. However, for future purposes, one may choose one particular job, e.g., one that generated the most new symmetry-related data in terms of the number of file entries or disk usage in `local_path`, and simply `rsync` all content in its `local_path` on top of `global_path` (assuming there are no more active jobs that still access `global_path`).

**RC_SYNC**   (optional path) The environmental variable RC_SYNC is complimentary to RC_STORE. For historical reasons, it is intended for file locks only to synchronize symmetry-related operations across parallel jobs. However, since file locks are not entirely reliable over network drives, if configured at all due to their impact on performance, the initial concept of file locks has been mostly abandoned in QSpace, such that setting the variable RC_SYNC is optional. For cluster jobs the recommended setting, instead, is to use a differential RC_STORE.

**Relevant system environmental variables**

**DISPLAY**   handles the graphical display of graphical output from Matlab when run in terminal mode (assuming Matlab is not run with the `-nodisplay` option). The variable DISPLAY belongs to the X11 window system (e.g., `XQuartz` on macOS) and is usually automatically set if X11 is present on one's system.

**PATH**   system path – Matlab and the C++ compiler must be visible on the system PATH. Therefore it needs to be properly adapted. Furthermore, the current directory `'.'` is expected to be included in the system PATH, such that QSpace specific local bash or Perl scripts, e.g., in `./system` or `./Source` are found.

## B.7   `RC_STORE` **database**

The RC_STORE is a file-based database that stores all symmetry-related content for non-abelian symmetries [8,15]. All data is computed on demand once and for all, and then stored for later usage. The root path of this database is specified via the environmental variable RC_STORE. To avoid race conditions across parallel jobs, this supports a differential setup via the specification of multiple paths. Nevertheless, for the sake of the argument here, RC_STORE is assumed to specify a single path.

For each symmetry `$sym`, QSpace generates a sub-directory `$RC_STORE/$sym/`, e.g., with `sym=SU2`, etc. Referring to this as the current directory `'.'` in what follows, QSpace maintains a detailed log file `./$sym.log` for all operations carried out for symmetry `$sym`. Furthermore, QSpace stores and maintains three database directories, R-, C-, and X-stores (for historical reasons, this is only partly reflected in the name 'RC'_STORE, since X was added with QSpace

v3). All data files are stored in Matlab binary format (`mat`-files), even if all file extensions have been altered by QSpace-specific conventions to reflect their content.

- `./RStore` – stores all irreducible representations $R_q$ for the ireps $q$ encountered during given RC_STORE history for symmetry `sym`. The respective data is stored in files `RStore/(q).rep` where the extension indicates 'representation'. Here $R_q$ includes a full basis decomposition in terms of weight labels that may include degeneracies, i.e., inner multiplicity [38], as well as a sparse representation of the diagonal Cartan subalgebra (generalized '$z$-operators' `Sz` $\equiv S_z$) and of the simple roots (generalized minimal set of raising operators `Sp` $\equiv S_+$) of the Lie algebra [8, 46].

- `./CStore` – stores all sorted CGTs $C_q$ as files `CStore/$qdir/(q).cgd` where the extension indicates generalized *Clebsch-Gordon data*. All CGTs are stored in sparse format generalized to arbitrary rank and in roughly quad-precision (GMP/MPFR). The latter ensures that all CGTs are exact when converted to double precision. The CGTs are grouped in subdirectories based on their sorted $q$-directions `$qdir` [incoming (+) before outgoing (-)]. Their combination thus represents the tensor rank. For example, standard CGCs are stored in the subdirectory `CStore/++-/`. All $1j$ symbols are stored in `CStore/++/`. These duplicate the entries of the type $(q\bar{q}|0)$ in `CStore/++-/` if present [15]. The latter directory also stores the fusion rules as obtained from full tensor-product decompositions. As these represent *maps* of 2 in- into 1 out-going leg, i.e., operates on a total of *3* legs, this is reflected in the file extension `*.mp3`).

- `./XStore` – stores the X-symbols [15] derived from all encountered pairwise CGT contractions $C \equiv A * B$ that are not trivially zero due to non-permissible[12] combinations of symmetry labels in the uncontracted indices present in $C$. With this the `XStore` typically contains by far the largest number of files. Most of these are small because X-symbols are small objects in general. The X-symbols are stored in files like `XStore/$qdir/(A)_ia (B)_ib.x3d` where the extension indicates X-symbol *data* which always operates on *3* outer multiplicity indices. The file name itself shows the contraction performed with reference to the input tensors `A` and `B` (specified by their compact $q$-labels), as well as the respective contracted indices `ia` and `ib` (also in compact format if multiple indices are contracted simultaneously). The files are grouped into subdirectories based on the $q$-direction `$qdir` of the *resulting* tensor $C$.

**Sorted CGTs** The `./CStore` stores *sorted* CGTs only: For one, this includes the more trivial ordering that all incoming (+) legs are listed before all outgoing (-) ones. Yet this also includes the considerably more subtle ordering within each group of directions where the legs are sorted w.r.t. their $q$-labels in lexicographic order. If there are more outgoing than incoming indices, then the conjugate CGT is stored. Therefore all CGTs in the RC_STORE have at least as many incoming indices as there are outgoing ones. These $q$-directions are then used as subfolder names to organize the CGTs in the RC_STORE, like `CStore/++-/` for standard rank-3 CGCs. By only storing sorted CGTs in the above sense, this avoids the proliferation of closely related entries in the RC_STORE. A general non-sorted, and in this sense unconstrained CGT can be obtained, nevertheless, in combination with a CGR.

RC_STORE **initialization** Whenever a symmetry `$sym` is requested for the first time, QSpace creates and initializes the above directory structure in `$RC_STORE/$sym/`. It creates the representation $R_{def}$ for the defining irep in the `RStore`, as well as it's dual if different. QSpace

---

[12]The check on zero-contraction of a pair of CGTs is based on simple checks concerning the resulting rank-2 or rank-3 CGTs $C$ only. Therefore many contractions that resulted in zero for rank $r > 3$ are stored, nevertheless. While such contractions may be zero during the initial build of the RC_STORE depending on the particular contraction performed, the resulting CGT can be permissible, nevertheless, and hence non-zero eventually.

then loops several times by generating tensor-product decompositions of the existing content in the `RStore` with itself while putting an upper threshold on the product dimension of the input for the case of larger-rank symmetries. This is intended at the very least to generate all smaller ireps that typically are required in the description of a local site before performing any contraction.

The initialization process is speedy for small symmetries such as seconds for SU(2), but becomes quickly more elaborate for larger symmetries [the cost grows exponentially with the rank of the symmetry; cf. Eq. (60)]. For rank-2 symmetries like SU(3), the initialization process should be completed within minutes. But for SU(4) one should be already prepared to wait several hours while the `RC_STORE` is built the first time. The initial process of generating lower-dimensional ireps may be interrupted, though, worst case by killing the process (Ctrl-C while still waiting for the present generation cycle to finish). The next time the `RC_STORE` is accessed for a given symmetry, it already exists. Still, subsequent computations for larger-rank symmetries may take considerable time in any case, since the `RC_STORE` still needs to be extended in the course of building a many-body Hilbert space.

Once a tensor network simulation is finished, though, by construction all relevant symmetry-related data is present. That is, repeating the same or similar simulations will have access to a (near) complete `RC_STORE`. At this point the full speedup of exploiting non-abelian symmetries is achieved. This is also a major reason why a centrally maintained `RC_STORE` is important for larger symmetries, like beyond SU(3), as this can make use of existing symmetry-related data. Besides, later access to an `RC_STORE` is also important if one wants to analyze and still operate non-trivially on QSpace tensors at a later time in a new Matlab session or with memory cleared, since QSpace tensors are tied to a particular `RC_STORE`.

### B.8  Logging

QSpace uses a streamlined format when logging data to standard output (`stdout`). This is based on the routine `wblog()` that behaves similarly in Matlab scripts (which call `lib/wblog.m`), as well as with a `C++` analog in MEX routines. It represents a tweaked formatted output to the `stdout` stream, like `fprintf(stdout,fmt,arg1,arg2,...)` with `fmt` some format string. This translates to `wblog(tag,fmt,arg1,arg2,...)`. For example,

```
1    wblog('TAG','some info (value of pi=\%g)\nand more\N-> hint: ...',pi)
```

generates output with the format

```
2    [file-name:lineno]   HH:MM:SS  TAG some info (value of pi=3.14159)
3    [file-name:lineno]   HH:MM:SS  TAG and more
4    -> hint: ...
```

It automatically prepends a header portion of about 36 characters that includes an automatically determined pointer into the source code at the location of the `wblog` command for reference [file name (possibly abbreviated) and line number (lineno)], a time stamp in 24H format, and a 3-character tag (the first input argument) as described below. The remainder of the input to `wblog` is mostly forwarded to `printf()` and hence accepts the same syntax with minor additional tweaks concerning the format string. Newlines are permitted, as in the example above, which then repeat the header portion. Tweaks include, e.g., the non-standard `\N` which inserts a newline *without* the header portion. Every `wblog` entry is terminated with a newline. This can be prevented by a trailing `'\\'` in the format string.

The 3-character TAG is used for a rough categorization of log entries via tags. It can be chosen arbitrarily, but is typically within the following set that has special meanings assigned:

| | |
|---|---|
| `'ERR'` | for errors[13] |
| `'WRN'` | for warnings[14] |
| `'<i>'` | for more important infos |
| `'I/O'` | for I/O related messages |
| `'NB!'` | for critical things to be aware of ('nota bene'), |
| `'SUC'` | succeeded with a certain operation |
| `'ok.'` | some successful check performed |
| `'* '` | log-level 1 |
| `' * '` | log-level 2 |
| `'  *'` | log-level 3 |
| `...` | etc. |

When using `QSpace` in interactive Matlab terminal mode, some log output is shown in color for emphasis depending on their TAG (like `ERR` messages in dark red, `WRN`s in lighter red, `NB!` or `OK!` in dark green, `ok.` in lighter green, etc.; the particular color coding may change in future versions, though).

The leading pointer into the source code facilitates to track down where a particular log message originated from, be it a `C++` reference from within a MEX file, or reference to a Matlab script. The file name always reflects the bare file name, i.e., skips any path. It may include program tags, current Matlab subroutine, etc. To align log entries with respect to the subsequent time stamp, 20 characters are reserved for `file-name:lineno`. If it does not fit within that space, various schemes for abbreviating the file name are adopted depending on the context, thus giving preference to the readability of the log output by aligning entries. Skipped string portions in the file name are indicated by a prime (`'`).

## C  `QSpace` applications in the repository

The `QSpace` repository also includes state-of-the-art implementations of fdm-NRG [19] and DMRG. The standard iterative diagonalization of the NRG was already discussed in more detail in Sec. 5.6. However, the full implementation of fdm-NRG is considerably more involved, with its documentation beyond the scope of this documentation. Similar so for the DMRG. Please consult the `./Docu` folder in the git repository for future updates. Nevertheless, it is emphasized here in passing, that these implementations already exist in the public `QSpace` repository. The interested reader may find it rewarding to explore. In case of more detailed interest, please also feel free to contact the author in this regard.

### C.1  DMRG simulations

By default, `runDMRG` located in the `DMRG/` folder launches a DMRG simulation for the ground state of the isotropic Heisenberg model (`wsys='Heisenberg'`) with nearest-neighbor interaction $J = 1$ and next-nearest neighbor interactions set to $J_2 = 0.25$ exploiting SU(2) spin symmetry,

```
1    runDMRG
```

---

[13]The `'ERR'` tag throws an error in the MEX files, but continues in Matlab code. To also throw an error in the latter case, QSpace uses `wbdie()`. For the debugger to also stop with `ERR` in Matlab code, QSpace provides the command `dberr`.

[14]The `'WRN'` tag in the MEX files and also Matlab code usually does not raise any signal or issue but continues. It leaves it up to the user to check the log output for warnings. Nevertheless, to stop the debugger in an interactive session with `'WRN'` in Matlab code, QSpace provides the command `dbwrn`.

The Hamiltonian is encoded in `HAM` which represents the QSpace Matlab `Class/@Hamilton1D`. Simply typing `HAM` will show essential information concerning the model Hamiltonian. The model parameters are stored in the structure `HAM.info.param`. The lattice structure can be inspected via `plot(HAM)`. A summary of the DMRG sweeps can be generated by the plot script `plot_Hamilton1D`.

### C.2   NRG impurity solver

A typical NRG run involves two parts [13,18]: (i) Iterative diagonalization (line 1 below which by default selects the single impurity Anderson model). This collects finite-size spectra in kept and discarded resolution [19,59] as already explicitly demonstrated in Sec. 5.6. This then permits one to (ii) define a thermal state via the full density matrix (fdm) for statistical properties but also spectral properties (line 4 below) that can be evaluated efficiently in textbook-like fashion in Lehman representation [19,50],

```
1    rnrg;      % NRG iterative diagonalization
2  % nrg_plot  % generates NRG energy flow diagram
3
4    rfdm       % computes fdm-NRG correlation functions
5  % fdm_plot  % summarizes computed fdm-NRG spectral data
```

A graphical summary of the results can be generated by the respective plot scripts in lines 2 and 5 (these scripts are automatically called at the end of the respective previous commands in lines 1 and 4, hence they are commented out). Further hard-coded QSpace MEX applications exist in the fdm-NRG context in the repository, as listed in Sec. E.2 together with the MEX routines underlying `rnrg` and `rfdm` above.

## D   Additional information and documentation

### D.1   Additional usage and help info

All QSpace routines, MEX as well as Matlab functions, come with their own help and usage information (like 'man pages'). It can be looked up with the standard Matlab `help` command.

**Help and usage of MEX functions**   For MEX routines denoted by `fooQS` here, the following are all equivalent

- `help fooQS`

- `fooQS -h`

- `fooQS -?`

where the last two are simple redirects to the first call. That is, all help files to MEX files are stored as Matlab m-files. For this reason, Matlab finds *two* entries for every MEX routine in its respective directory: the actual MEX binary `fooQS.mex*`, but also `fooQS.m`. The latter contains usage information based on an extended header comment without any source. It is recommended that both files are located in the same folder in the Matlab path. Then when calling the function, this finds the `fooQS.mex*` file first. When calling the help, Matlab ignores the MEX file and thus finds the `fooQS.m` first, instead. Further flags supported by the QSpace MEX files are

- `fooQS --version`   shows version information of `QSpace` and linked libraries

- `fooQS --ping`   dummy call of a MEX routine with the effect
  that it becomes resident in the Matlab workspace

**Help and usage of Matlab (class) routines**   Similar to the help on MEX routines above, also all Matlab routines have a help that can be looked up in standard Matlab syntax. For example, help on the Matlab QSpace class routines (located under `Class/@QSpace/`) based on the placeholder routine `foo` can be obtained as follows:

- `help foo`   display help for function `foo`

- `help QSpace/foo`   display help on a particular class method

- `methods QSpace`   listing of all existing methods in `Class/@QSpace`

**Identifying and locating functions**   Given the many functions and methods in the `QSpace` environment within Matlab, one can identify and thus locate the relevant functions at the Matlab prompt but also at any debug breakpoint, with the native Matlab command

```
which [path/]foo[QS] [-all]
```

where terms in square brackets are optional. The first line in the output then shows the routine that has the highest precedence, and hence would get called in given context. The option `-all` permits one to see all overloaded routines, as well as class routines that share the same name. With this, all subsequent output lines, if any, show routines or functions that are shadowed. A subsequent help on a particular function or MEX routine from that list will provide more detailed usage information.

**Matlab debugger**   The Matlab debugger offers a useful low-level tool that permits one to gain better insight into the behavior of particular Matlab sources. Matlab simulations can be stopped at any location in a Matlab function (at specified line #) by setting a breakpoint,

```
dbstop in [class/]foo [at line #]
```

This permits one to inspect and alter variables in a particular context. Also in the absence of any error whatsoever, this can be useful for analyzing or a better understanding of the behavior of a code. The commands `dbup` and `dbdown` traverse the caller stack. See also the QSpace related `dberr` and `dbwrn` that respond to wblog.

## D.2   `Docu/` folder

The QSpace repository also contains the documentation folder `./Docu` that is maintained with the repository. It may be looked up for comments on most recent changes, such as `readme` files, or additional more recent documentation. Other main entries are:

**QSdocs.pdf**   This contains a complete listing of the help to all MEX routines as well as QSpace methods (member functions). It has been auto-compiled into the single PDF file `Docu/QSdocs.pdf` (70+ pages). This file may be consulted in parallel to this documentation.

**Doxygen**   Some general documentation of the `C++` code base that was auto-generated via `doxygen`. This provides an overview of the general outline of the `C++` code structure. Its output is located in `Docu/html/`, which can be viewed in a web browser starting from the `Docu/html/index.html`. It also compiled into a single PDF `Docu/doxygen.pdf` (900+ pages).

# E  Listing of MEX Routines

## E.1  `QSpace` core MEX routines

The following short description of the most relevant MEX routines is sorted in alphabetic order. For a detailed description of all MEX routines, please see their respective help. Note that the help output for all MEX routines has also been automatically compiled into the single PDF file `Docu/QSdocs.pdf` for reference. The MEX routines are all coded in C++, with the sources located in `./Source`. With about 100,000 lines of code, this far exceeds the present documentation which rather focuses on `QSpace` usage and applications. Nevertheless, an auto-generated doxygen documentation is included in the repository.

**compactQS()**  Used by `getLocalSpace()` to bootstrap `QSpace` tensors from Fock space. This routine obtains reduced matrix elements based on the Wigner-Eckart theorem and existing rank-3 CGTs in the `RC_STORE`. It ensures that all non-zero matrix elements of the input were accounted for in magnitude and sign. It is essential to `getLocalSpace`, but is typically never called explicitly or needed elsewhere in an application.

**contractQS()**  This routine performs (nested sets of) pairwise contractions of `QSpace` tensors. It supports two modes: (i) An explicit mode for a single pairwise contraction where contraction indices are fully specified. Here, $q$-directions together with `itags`, if set, need to match nevertheless. (ii) Auto-contraction based on `itags` with options to set `itags` on the fly: this mode supports contractions of a set of tensors where the order of the contraction in terms of pair-wise contractions is explicitly specified based on a nested cell input data structure. In case of errors in the input, the auto-contraction of nested cell structures can be debugged by adding the verbose option `'-v'`.

**diagQS()**  Returns all diagonal matrix elements of a scalar tensor as a single column vector (hence for rank-2 tensors only); In the presence of non-abelian symmetries, this returns a second column that specifies the degeneracies when converting multiplets to states. See also `QSpace/diag.m`.

**eigQS()**  Performs eigendecomposition of a scalar tensor (hence for rank-2 tensors only). The first output argument returns the complete set of eigenvalues as a column vector. In the presence of non-abelian symmetries, this returns a second column that specifies the degeneracies when converting multiplets to states. These degeneracy factors are important as weight factors, e.g., for a normalized density matrix spectrum. This routine also implements various truncation schemes with truncation based on combinations of (i) maximal number of multiplets, (ii) an energy threshold (assuming the input is a Hamiltonian), or (iii) weight threshold (assuming the input is a density matrix). In this case the second output argument returns a structure with `QSpace` tensors $\mathsf{AK} \equiv A_K$, $\mathsf{AD} \equiv A_D$, $\mathsf{EK} \equiv E_K$, and $\mathsf{ED} \equiv E_D$ for eigenstates ($\mathsf{A*}$) and eigenvalues ($\mathsf{E*}$) split into kept ($\mathsf{K}$) and discarded ($\mathsf{D}$) state spaces, respectively. The eigenvectors in $\mathsf{A*}$ are stored as columns, and the eigenvalues in $\mathsf{E*}$ in compact diagonal `QSpace` representation, meaning that this only stores the diagonals as a vector in the RMTs. It may be expanded using `@QSpace/diag()`.

**getDimQS()**  (integer vector or matrix) Get overall dimensions of a `QSpace` tensor for all legs based on all QSpace records. For a tensor of rank $r$, a $1 \times r$ vector is returned for all-abelian symmetries. In the presence of non-abelian symmetries, a second row is returned. In this case, the first row contains the effective dimension (in terms of multiplets), and the second row the

corresponding full dimension based on states. See also getQDimQS for detailed dimensions of symmetry sectors on a particular leg.

**getIdentityQS()**  Get identity tensor either (i) for the state space with respect to a single QSpace, with an optional flag `'-0'` to obtain the $1j$-tensor, instead, or (ii) get the identity *A*-tensor for the tensor product space of two state spaces with reference to two input QSpace tensors. By default, as of QSpace v4, the $1j$ symbol in case (i) marks the dual space on the second index with a trailing prime (') [for backward compatibility, the same itag for the dual space is returned with the option `'-z'`, standing for 'zero' as an alternative to `'-0'`. However, the recommended way is to use markers as with the updated behavior of `'-0'`].

**getQDimQS()**  Get detailed dimensions for all symmetry sectors on a specified leg of a QSpace tensor, returning *q*-label resolved dimensions both for the reduced multiplet level (RMTs), as well as the combined multiplet dimensions. See also getDimQS for total dimensions.

**getRC()**  Load RC_STORE specific data (for inspection, experimental, or debugging purposes).

**getSymStates()**  Group Fock space decomposition into symmetry spaces before call to compactQS in getLocalSpace. Like compactQS, this routine is never explicitly called or needed elsewhere in an application.

**isIdentityCG()**  (boolean) Whether given QSpace tensor has identity CGRs; returns 1 for scalar operators or rank-2 tensors with fully abelian symmetries. By contrast, $1j$ tensors with non-abelian symmetries return 0.

**isIdentityQS()**  (boolean) Whether QSpace represents an identity operator as a whole, which thus also checks the RMTs to be identities within numerical (double precision) noise.

**isHConjQS()**  (boolean) Whether rank-2 QSpace is Hermitian conjugate within numerical noise. In the case of all-abelian symmetries, the input tensor $H$ may represent a scalar operator decomposed as a tensor of even rank $r = 2n$, where a return value of 1 requires, aside from additional checks on the RMTS, that the first $n$ indices are incoming, and the remaining $n$ indices outgoing.

**normQS()**  (number) Computes Frobenius norm $\|X\| \equiv \sqrt{\text{tr}(X^\dagger X)}$ for QSpace X as in Eq. (22).

**orthoQS()**  Computes QR-like decomposition based on prior SVD for given QSpace tensor, also allowing for truncation [same as svdQS, except that singular values are already contracted onto one of the isometries out of the SVD as specified by the input].

**permuteQS()**  Permute legs of QSpace tensor together with optional tensor conjugation.

**plusQS()**  Add QSpace tensors, as in `C = A+bfac*B` where, by default, `bfac=1` results in plain addition. By contrast, specifying `bfac=-1` results in subtraction, etc.

**setupRCStore()**  Triggers initial setup and generation of RC_STORE for specified symmetry (this is automatically triggered anyways when calling compactQS with a new symmetry in getLocalSpace, so there is usually no need to explicitly call setupRCStore).

**skipZerosQS()**  Skip QSpace records that have all matrix elements in their RMT below some numerical threshold (default $10^{-14}$ for double precision noise). These zero blocks are allowed by symmetry, but are zero due to application-specific reasons. By default, diagonal zero blocks for rank-2, i.e., scalar operators are kept, nevertheless. Zero-blocks for rank-2 operators can also be forced to be skipped with the option `'--all'`.

**svdQS()**  Singular value decomposition (SVD) of some tensor typically representing an orthogonality center (OC). For non-abelian symmetries, the SVD is constrained w.r.t. a particular leg since no automated fusion of legs is performed in this routine.

**traceQS()**  (number) Full trace of rank-2 scalar QSpace tensor, properly including weights by degeneracies within multiplets.

### E.2  NRG related hard-coded MEX applications

The following MEX routines are also located in the `./bin` folder. Most of these have already been applied in the past, as indicated by the references below. However, this documentation mentions these in passing for completeness only. In any case, a help is provided with each of these routines.

**NRGWilsonQS()**  Iterative diagonalization for Wilson's Numerical Renormalization Group (NRG, [18]) for quantum impurity models [19, 50].

**fdmNRG_QS()**  Full density matrix (fdm-NRG) approach to computing impurity spectral that conserve spectral sum rules to within numerical precision for arbitrary temperatures [19, 50]. Together with `NRGWilsonQS` this is a powerful black-box quantum impurity solver.

**fgrNRG()**  Fermi Golden rule adaptation of `fdmNRG_QS` to compute absorption and emission spectra on quantum impurities [21, 50, 85].

**tdmNRG()**  Time-dependent quench adaptation of fdm-NRG starting from thermal equilibrium in the initial Hamiltonian (closely related to fgr-NRG transformed to real-time domain).

**getSmoothSpec()**  Log-Gauss broadening of spectral data, e.g., as obtained from fdm-NRG [13, 19], with smooth transition to linear binning across zero energy.

**getSmoothTDM()**  Broadening of spectral data for quantum quenches to smoothen data in the real-time domain [50]. Closely related to `getSmoothSpec()`.

### E.3  QSpace MEX utility routines

The `./util` folder contains hardcoded C++ utility MEX routines. The listing below only mentions the more important ones. Additional MEX files in the public repository but not listed here, were partly for testing purposes and hence can be safely ignored. Yet as always, the help provided with them explains their purpose.

**helloworld()**  Hello-world routine to test MEX file setup.

**kramers()**  Kramers-Kronig transformation of spectral data (back and forth from real to imaginary and vice versa; e.g., used with NRG spectral data).

**matchIndex()**     matches two sets of indices that are stored as rows in a matrix (typically containing integer values, such as symmetry labels, to avoid issues with double precision noise). It returns the indices of the matching rows, similar to Matlab's `intersect(A,B,'rows')`, yet with crucial practical differences. If rows are not unique, in the sense that certain index sets appear multiple times identically across different rows, then all-to-all matches are included within such 'degenerate' subspaces as in a tensor product. An optional additional info structure is returned as third argument. This contains additional data such as indices of non-matching entries, etc. See also uniquerows.

**cgs2double()**     converts an MPFR-encoded Clebsch-Gordan data space, like the multidimensional arrays as stored in the RC_STORE, to regular double-precision decimal format (mostly for inspection and testing purposes only). See also lower-level routine mpfr2dec or @SymStore.

**mpfr2dec()**     converts MPFR-encoded numerical arrays as stored in the RC_STORE to regular double-precision decimal format (mostly for inspection and testing purposes only). See also higher-level routine cgs2double.

**uniquerows()**     group rows in a matrix (typically assuming integer values to avoid issues with double precision noise), returning a matrix with unique rows. Additional return arguments provide detailed information on the dimension of grouped blocks with identical row entries together with indices to the original data. See also matchIndex.

**wbrat()**     Acquire a rational approximation for float numbers based on continued fractions (similar in functionality to Matlab's `rat.m` function, but this routine automatically also checks for possible representations in terms of the square root of rationals, as they frequently occur in CGTs. It picks the closest representation (with or without square root) in terms of numerical accuracy. This routine returns a string with syntax that can be reevaluated within Matlab. An additional info structure returned as second argument provides the integers `P` and `Q` as obtained for the rational approximation `P/Q`. If this routine fails a find a rational approximation with the requested accuracy, the original value is returned instead.

## F   `QSpace` Related Matlab Environment

The following gives a listing and brief description of the more important Matlab routines and additional class constructions. Since there are plenty of helper routines, including some 250+ m-files in `./lib`, these cannot be all described in detail here. Yet as always, all routines are provided with a help that explains their purpose.

### F.1   Class/`@QSpace`

The @QSpace class is a Matlab wrapper environment for the C++ QSpace class used in MEX files. Since a tensor `X` thus becomes QSpace object `X`, it also simly referred to as QSpace `X` then. The @QSpace class contains more than a hundred class methods which cannot be all discussed in detail here. Only the most noteworthy ones are listed in the following. A complete listing including their help output has also been compiled and included in the PDF file Docu/QSdocs.pdf in the QSpace repository. Rather than alphabetically sorted, the following entries are grouped mainly by topic.

**QSpace()**     constructor of the @QSpace wrapper class.

**display()**   prints a convenient formatted summary of a given QSpace tensor. Matlab automatically invokes it when displaying a variable by simply typing its name at the Matlab prompt. It shows a 2-line header generated by **info()**, followed by a listing of records. See Fig. 7 for an example, together with a detailed explanation of the printed output. For larger QSpaces with more than about 12 non-zero blocks (records), `display` truncates the listing, by default: it shows the first 5 entries, up to two intermediate entries if largest in RMT size followed by the last two entries. The listing of all entries can be enforced, nevertheless, by explicitly invoking the `display` routine with the equivalent options `'-f'` or `'-a'`.

Like logging with wblog(), the display of QSpace tensors also supports color-highlighted output for readability when Matlab is run in interactive Matlab terminal mode. In this case, `itags` are color-coded by interpreting trailing marker characters.

**plus(), minus(), sum()**   Overload of simple algebraic operators + (`plus`) or − (`minus`). These are extended wrappers of the MEX routine plusQS. The routine `sum(A,dim)` permits to sum up a QSpace array *A* along dimension `dim` (by default, `dim=1`; this follows similar semantics as Matlab's native `sum` routine). This adds up QSpaces and hence returns a QSpace (vector).

Simpler cases where for a rank-2 tensor a plain numerical value *x* is added or subtracted, are handled within the Matlab wrapper routine itself. For example, in `H+x` with `H` some rank-2 tensor, the scalar *x* is interpreted as proportional to the identity matrix for all symmetry sectors that are present in `H` (be aware, though, that `H` may have missing diagonal all-zero blocks; hence to ensure a full state space, one may rather use `H+x` → `H+x*Id`, instead, with `Id` representing a complete identity operator within a given state space).

**oplus()**   The routine `oplus(A,B,d12)` permits the direct sum of two tensors along dimensions `d12` (default: `[1 2]`) while respecting the symmetry structure. Both QSpaces *A* and *B* must have the same tensor structure (same rank, same *q*-directions, etc). If `d12` specifies two dimensions (indices), this catenates blocks in a 'block-diagonal' fashion along specified dimensions `d12` [the Matlab analogon for matrices with `d12=[1 2]` is `blkdiag(A,B)`, except that `oplus` only combines blocks that also share the same symmetry sectors; other blocks unique to either `A` or `B` are included as is in the output]. If `d12` specifies a single dimension (index), symmetry matching blocks are catenated along that dimension only [the Matlab analogon in this case would be `cat(d12,A,B)`].

**uplus(), uminus()**   are unary operators that implement the unary plus and minus, i.e., syntax such as `X=+Y` or `X=-Y`.

**transpose(), ctranspose()**   overload the operators `.'` and `'` for plain transpose and Hermitian conjugate, respectively.

**times(), mtimes(), mrdivide()**   overload the respective algebraic operators `.*`, `*`, or `/` (for division by number only). The routine `mtimes` is an extended wrapper routine to the MEX function contractQS. Simple multiplications with scalar numeric values are handled within the wrapper routine itself.

**comm(), acomm()**   computes commutator or anticommutator, respectively, for a pair of operators. For irops, the irop index will necessarily also be contracted in this process. These routines are based on contractQS.

**real(), imag(), isreal()**   extract real and imaginary parts of a QSpace (simple Matlab functions that operate on the RMTs (i.e., on `X.data` for a QSpace `X`), or check whether QSpace is complex [`~isreal()`].

**sqrt()**   takes square root of rank-2 QSpace that are diagonal (and diagonal only; this is enforced). Warnings are issued if input data is negative or complex.

**logical(), not(), isempty()**   boolean operators that check whether a QSpace is empty. The method `not()` is the inverse of `logical()`. For example, for QSpaces `X` and `Y`, this permits simple syntax such as `if X && ~Y`, `(do something); end`. The method `isempty()` is somewhat more elaborate than `logical()`, as it includes further structural checks.

**eq(), ne(), isequal(), sameas()**   Overload of Matlab's `==` and `~=` operators (wrappers to `@QSpace/isequal`). This permits the syntax `X==Y` for QSpaces `X` and `Y`. The method `sameas()` permits differences in data up to some epsilon (default: $10^{-10}$ on a relative scale).

**getsub()**   selects sub-list of QSpace records while maintaining the QSpace structure, otherwise. The index of records to select may be specified explicitly, as in `getsub(X,I)` with record index `I`, or implicitly by selecting a particular symmetry sector, as in `getsub(X,[qlabels],dim)` which only selects the records that have specified `qlabels` (set specified as rows in a matrix) on leg `dim`.

**subsref(), subsasgn()**   overloads indexing and access to fields in a QSpace object. As a technical subtlety concerning Matlab, note that for a QSpace `X`, typing `X.Q{:}` on the Matlab prompt does not work as expected for a plain Matlab structure. Since the operation `X.Q` needs to be rerouted via the overloaded method `subsref` here, this operates like a function call. Therefore typing `X.Q{:}` on the Matlab prompt only displays `X.Q{1}`. To display all entries in `X.Q`, this requires, e.g., a temporary assignment followed by its display, like `Q=X.Q; Q{:}`. Similarly, catenating symmetry labels across all legs as in `Q=X.Q; Q=[Q{:}]` is not equivalent to `Q=[X.Q{:}]`. The latter rather corresponds to `X.Q{1}` only. This is a Matlab peculiarity, which returns a comma-separated list for `Q{:}`. For a function call, this would require the rather awkward syntax `Q=cell(1,numel(X.Q)); [Q{:}]=X.Q{:}` which is simply equivalent to `Q=X.Q` in the first place.

**diag()**   toggle rank-2 QSpace tensor between diagonal and non-diagonal representation. This mimics the behavior of Matlab's `diag()` function. This function deals with the compact diagonal behavior, e.g., as returned by eigQS or svdQS. With the option `'-d'`, this routine returns a plain numerical vector of all diagonal entries (see also diagQS in this regard).

**getitags()**   gets the complete set of `itags` for given QSpace tensor.

**setitags()**   Set `itags` of QSpace while preserving *q*-directions (trailing conjugate flags `'*'`). Any trailing `'*'` specified with the input strings will be ignored. This routine accepts a range of specialized settings. For example, for *A*-tensors, the following are equivalent,

```
1    A=setitags(A,'-A',4)
2    setitags(A,{'K03','K04', 's04'})
3    setitags(A,{'K03','K04*','s04'})
```

as may be used, e.g., for site 4 in an MPS (e.g., see Fig. 2). Conjugate flags, as in line 3, are ignored. Since line 2-3 specify `A` by name, they will also assign the changes to `A` in the workspace same as in line 1. For operators, `S=setitags(S,'-op:s',4)` sets `itags` `{'s04','s04' [,'op']}`, with the last 'operator' index skipped if `S` is of rank 2. See help on `setitags` for more details.

**itagrep()** Change `itags` based on regular expressions (uses Matlab's `regexprep`).

**untag()** Remove `itags` for specified QSpace tensors. For convenience, if no return argument is requested and all tensors are specified by name, as in `untag(A,B,...)`, all objects will get updated in the caller workspace, nevertheless, using Matlab's `assignin('caller',...)`. Therefore `[A,B,...] = untag(A,B,...)`; and `untag(A,B,...)` are equivalent.

**isscalarop()** (boolean) whether input represents scalar operator, i.e., is of rank 2 and is block-diagonal for all symmetries.

**fixScalarOp()** for scalar operators with trailing irop index having $q_{\mathrm{irop}} = 0$, this trailing singleton irop index (third leg) is skipped. See also reverse operation `makeIrop()`.

**makeIrop()** transform scalar rank-2 operator back to an irop by explicitly adding a trivial irop index (third leg) with scalar symmetry labels $q = 0$. This is the reverse operation to `fixScalarOp()`.

**appendSingletons()** Append indices in the scalar symmetry sector $q = 0$ for all symmetries and thus of dimensions 1 (singletons) to given QSpace tensor.

**addSymmetry()** Add new symmetry to specified QSpace tensor. By default, this appends the symmetry in the irep of the scalar representation $q = 0$ as an additional symmetry. This routine is intended for specialized model setups to tweak an existing output of `getLocalSpace()`. Like `getLocalSpace`, there should be no need to call this routine after the model setup.

   If the symmetry label $q$ for the new symmetry is explicitly specified and non-scalar with multiplet dimension $|q| > 1$, then a $\sqrt{|q|}$ factor is applied to given QSpace `X`. This mimics having added $|q|$ copies, such that the norm $\|X\|^2 =$ `norm(X)^2` will change by a factor $|q|$.

**getvac()** Get identity operator ('state space') in the vacuum state for the symmetry setting of the input QSpace. Hence this has dimensions $1 \times 1$ and all-zero $q$-labels, yet is aware of all symmetries and their order. With the option `'-1d'`, this returns a vector of length $d = 1$, with an example shown in (41).

**SEntropy()** compute von-Neumann entropy assuming that the input QSpace represents a scalar density matrix.

**plotQSpectra()** Plot eigenspectrum of scalar input operator in a symmetry-resolved manner. With option `'-ES'`, this plots the entanglement spectrum instead (the latter is intended for normalized density matrices only).

**contract()** simple wrapper for `contractQS`.

**dim()** simple wrapper for `getDimQS`.

**eig()**    simple wrapper for `eigQS`.

**getIdentity()**    simple wrapper for `getIdentityQS`.

**permute()**    simple wrapper for `permuteQS`.

**skipzeros()**    simple wrapper for `skipzerosQS`.

## F.2   Class/`@SymOp`

Utility class to generate a minimal set of generators (symmetry operators, or '`SymOp`' for short) in terms of raising and $z$-operators (simple positive roots Sp and Cartan subalgebra Sz, respectively) when setting up symmetries in models. It is used in `getLocalSpace` only, e.g., with objects of this type returned in `IS.SOP` → Sp, Sz together with the MEX routine `getSymStates` to bootstrap symmetry spaces. This class should not be relevant otherwise, hence can be safely ignored from an application point of view.

## F.3   Class/`@SymStore`

The `@SymStore` class offers basic utility routines to access the `RC_STORE`. It is intended mainly for information, testing, and debugging purposes. It does not try to generate any symmetry-related data. Therefore if the requested data is not yet present in the `RC_STORE`, this class returns an error.

**Load generators**    The minimal set of generators (simple roots and Cartan subalgebra, or conversely, raising and $z$-operators) for SU(2) for the defining representation $q = 2S = 1$ can be loaded from the `RC_STORE` via `I = SymStore('SU2','-R','(1)')` with the option `'-R'` which thus looks up the folder `$RC_STORE/SU2/RStore/`. The round brackets with the string of the $q$-labels in the last input argument are optional. The returned generators are encoded as sparse MPFR arrays. These can be converted to regular double-precision arrays via the MEX utility routine `cgs2double(I.Sz)`. Since the output is rank-2 in the present context, this permits and thus also returns a regular sparse double array.

**Load CGTs**    CGTs can be loaded with the option `'-C'` which looks up the `CStore` folder. For example, the CGC for the tensor product $(S_1 = 1) \otimes (S_2 = 1) \to (S_{\text{tot}} = 1)$ for `'SU2'` spins can be loaded via `I = SymStore('SU2','-C','(2,2;2)','-f')`, bearing in mind the labeling convention with Eq. (7) and that within QSpace, $q = 2S$, thus resulting in `'(2,2;2)'`. Since CGTs may get large, the data returned with the `-C` option effectively contains a reference by default. The option `'-f'` then enforces that the full CGT is loaded. With this then, `I.cdata` contains the CGT in MPFR format, while `I.CData ≡ cgs2double(I.cdata)` contains the same data but already converted to a double-precision multidimensional array.

**Load fusion map**    Lastly, the fusion map out of two multiplets can be accessed via the option `'-M'` which looks up the currently generated `CStore/++-/*.mp3` files. For example, with reference to the previous example, the command `q = SymStore('SU2','-M','(2,2)')`

```
1      J12: [2 2]              % (re)confirms input multiplets [q₁,q₂] ("[J₁,J₂]")
2        J: [3x1 double]       % complete set of fused multiplets (stored as rows)
3    omult: [1 1 1]            % respective OM (one value for each row in J)
4      cgr: [3x1 struct]       % respective CGRs
5    c2eps: 0                  % floating point error estimate
```

```
6
7  >> q.J'
8
9      0    2    4
```

returns all outcomes of the fusion of the two SU(2) multiplets $(S_1 = 1) \otimes (S_2 = 1)$ with `q.J` (having `J*` $\equiv$ `q*` for historical reasons here). With $q = 2S$, therefore $S \in \{0, 1, 2\}$ as expected.

### F.4  Class/`@Hamilton1D`

An application class that underlies `DMRG/runDMRG.m`. This permits to run DMRG simulations on simple pre-setup models as defined in `runDMRG` with the option to adapt to one's needs. Since this class is rather extensive on its own, it will be described in more detail elsewhere. Yet please feel free to explore. In case of issues or more detailed questions, please also feel free to contact the author of this documentation in this regard.

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
