# Peer review of "QSpace - An Open-Source Tensor Library for Abelian and non-Abelian Symmetries"

_SciPost Physics Codebases, doi:SciPost Phys. Codebases 40 (2024) , SciPost Phys. Codebases 40-r4.0 (2024)_

## Round 1 · Referee Report · Anonymous (Referee 1) · 2024-6-28

Report

The paper documents QSpace, a versatile tensor library designed to handle both Abelian and non-Abelian symmetries in quantum many-body systems. The library aims to simplify the implementation of tensor network algorithms by abstracting the complexity associated with handling symmetries. QSpace takes a bottom-up approach to the management of ( non-)Abelian symmetries, using generalised Clebsch-Gordan coefficients to facilitate operations across different symmetries. It includes a comprehensive set of essential tools and elementary operations on tensors, along with tutorials and practical applications that demonstrate its capabilities in complex quantum problems.

The manuscript is well and precisely written. It serves as a useful handbook to help users get started with the QSpace library. The underlying concepts of handling (non)-Abelian symmetries are outlined, yet the focus is on user interfaces and practical use. As such, the paper is a self-contained reference to the QSpace library.

I suggest publication in its present form in SciPost Physics Codebases, with possible incorporation of the minor comments listed below.

  • It would be helpful to write out abbreviations the first time they appear, e.g. for irops and ireps.
  • In the discussion of the growing number of individual symmetry blocks, the alternative of storing the symmetric tensors as a matrix from the incoming to the outgoing indices could be discussed. This reduces the number of symmetry blocks considerably.
  • Given the quantum many-body context for which the library is designed and used, an example of a one-dimensional TN algorithm such as DMRG could be appreciated by users starting with the library.
  • In two dimensions, using PEPS, iterative power methods are often used to contract the infinite tensor network. A comment on the reliability of the numerical implementation of the CGTs would be helpful, e.g. can errors proliferate in such algorithms?
  • Trailing singleton dimensions are not shown in the RMT, but are shown for the CGTs.
  • Inconsistent use of "j symbols" and "j-symbols".
  • In Eq. (12) and surrounding text, one could use q_{\mathfrak{r}} instead of q_r, as in Eq. (14).
  • On page 56, "The second symmetry label describes the SU(2) spin" should be q_2

Recommendation

Publish (easily meets expectations and criteria for this Journal; among top 50%)

  • validity: -
  • significance: -
  • originality: -
  • clarity: -
  • formatting: -
  • grammar: -

Author:  Andreas Weichselbaum  on 2024-07-04  [id 4600]

(in reply to Report 1 on 2024-06-28)

**Reply on DMRG application:**

The focus of this documentation is on QSpace as a tensor library, i.e., on the description and handling of tensors. Matter of fact, a full fledged DMRG exists in the repository, [cf. App. C.1, DMRG simulations]. Its detailed documentation, however, would far exceed the scope of the present paper. Hence this is left for the future.

**Reply on PEPS and iterative power methods:**

The CGTs are computed once and for all in high-precision format and thus exact in double precision accuracy [cf. Sec. 2.13: CGT data storage and precision].

In iterative power methods then, it is reasonable to expect that the required pool of symmetry-related objects quickly becomes exhausted. The necessary objects in the RC_STORE, if not yet present, would be generated once and for all. Hence there should be no issue whatsoever with respect to proliferating errors from the CGT perspective in this regard.

I'd expect that the dominant numerical error in iterative power methods stems from the RMTs. However, because the RMTs can be considerably smaller in size in the presence of non-abelian symmetries, the numerically accumulated error should be less, e.g., compared to simulations that do not exploit symmetries or just U(1) symmetries.

Moreover, any symmetry explicitly handled in a tensor library is exact by construction. Hence it cannot be broken due to numerical noise. The ability then to manually reduce symmetry permits one to study effects related to (spontaneous) symmetry breaking where relevant.

---

## Round 2 · Author Response

I thank the referee for carefully reading the manuscript and his/her very supportive report.

I posted the reply to the questions concerning DMRG and PEPS with iterative power methods online with the report. The remaining reply and changes are listed below.

---

## Round 2 · List of Changes

The acronyms irops and ireps are now introduced the first time they are used.

The referee suggested fusion of indices for higher-rank tensors
to avoid a growing number of individual symmetry block.
I added a paragraph addressing this point at the end of
Sec. 2.11 (page 22).

On showing trailing singleton dimensions with RMTs only:
To motivate the shown behavior, I extended the second
paragraph in Sec. 3.2 QSpace display on page 35.

The inconsistent use of "j symbols" and "j-symbols"
is fixed (hyphens removed throughout).

The format of the symmetry rank from r -> mathfrak{r}
around Eq. (12) is fixed: this needed to read mathfrak{r}, indeed,
for consistency.

The typo on page 56 concerning q_2 is also fixed.

---

## Editorial Decision

published